# UNIFIED CONTINUOUS GENERATIVE MODELS FOR DENOISING-BASED DIFFUSION

## ABSTRACT

Recent advances in continuous generative models, encompassing multi-step processes such as diffusion and flow matching (typically requiring 8-1000 steps) and few-step methods such as consistency models (typically 1-8 steps), have yielded impressive generative performance. However, existing work often treats these approaches as distinct paradigms, leading to disparate training and sampling methodologies. We propose a unified framework for the training, sampling, and analysis of diffusion, flow matching, and consistency models. Within this framework, we derive a surrogate unified objective that, for the first time, theoretically shows that the few-step objective can be viewed as the multi-step objective plus a regularization term. Building on this framework, we introduce the **U**nified **C**ontinuous **G**enerative **M**odels **T**rainer and **S**ampler (UCGM-$\{T, S\}$), which enables efficient and stable training of both multi-step and few-step models. Empirically, our framework achieves state-of-the-art results. On ImageNet $256 \times 256$ with a 675M diffusion transformer, UCGM-T trains a multi-step model achieving 1.30 FID in 20 steps, and a few-step model achieving 1.42 FID in only 2 steps. Moreover, applying UCGM-S to REPA-E (Leng et al., 2025) improves its FID from 1.26 (at 250 steps) to 1.06 in only 40 steps, without additional cost.

## 1 INTRODUCTION

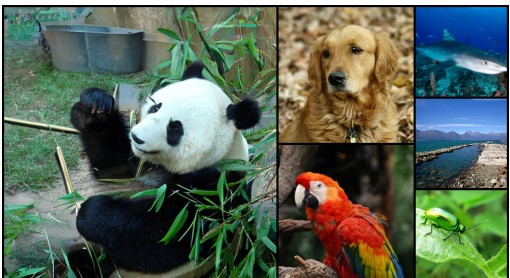 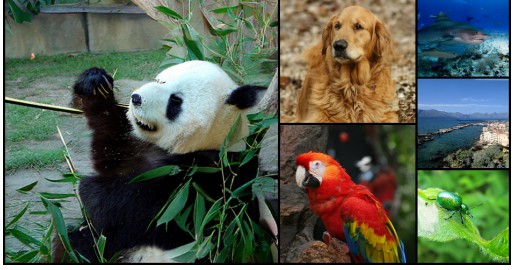

(a) **NFE** $= 40$, **FID** $= 1.48$.  (b) **NFE** $= 2$, **FID** $= 1.75$.

Figure 1: **Generated samples from two** $675$M **diffusion transformers trained with our UCGM on ImageNet-1K** $512 \times 512$**.** The figure showcases generated samples illustrating the flexibility of Number of Function Evaluation (NFE) and superior performance achieved by our UCGM. The left subfigure presents results with NFE $= 40$ (multi-step), while the right subfigure shows results with NFE $= 2$ (few-step). Note that the samples are sampled ***without** classifier-free guidance (CFG) or other guidance* techniques.

Continuous generative models, encompassing diffusion models (Ho et al., 2020; Song et al., 2020a), flow-matching models (Lipman et al., 2022; Ma et al., 2024), and consistency models (Song et al., 2023; Lu & Song, 2024), have demonstrated remarkable success in synthesizing high-fidelity data across diverse applications, including image and video generation (Peebles & Xie, 2023; Chen et al., 2024c; Ma et al., 2024; Xie et al., 2024a; Ho et al., 2022; Chen et al., 2025c).

Training and sampling of these models necessitate substantial computational resources (Karras et al., 2022; 2024b). Moreover, current research treats distinct model paradigms (diffusion models/flow matching (Karras et al., 2022) v.s. consistency models (Song et al., 2023)) independently, leading to paradigm-specific training and sampling methodologies. This fragmentation introduces two primary challenges: (a) **a deficit in unified theoretical and empirical understanding**, which constrains the

Table 1: **Existing continuous generative paradigms as special cases of our UCGM.** Prominent continuous generative models, such as Diffusion, Flow Matching, and Consistency models, can be formulated as specific parameterizations of our UCGM. The columns detail the required parameterizations for the transport coefficients $\alpha(\cdot), \gamma(\cdot), \hat{\alpha}(\cdot), \hat{\gamma}(\cdot)$ and parameters $\lambda, \rho, \nu$ of UCGM. Note that $\sigma(t)$ is defined as $e^{4(2.68t - 1.59)}$ in this table.

| Paradigm | | UCGM-based Parameterization | | | | | | |
|---|---|---|---|---|---|---|---|---|
| Type | e.g., | $\alpha(t) =$ | $\gamma(t) =$ | $\hat{\alpha}(t) =$ | $\hat{\gamma}(t) =$ | $\lambda \in [0,1]$ | $\rho \in [0,1]$ | $\nu \in \{1,2\}$ |
| Diffusion | EDM (Karras et al., 2022) | $\frac{\sigma(t)}{\sqrt{\sigma^2(t)+\frac{1}{4}}}$ | $\frac{1}{\sqrt{\sigma^2(t)+\frac{1}{4}}}$ | $\frac{-0.5}{\sqrt{\sigma^2(t)+\frac{1}{4}}}$ | $\frac{2\sigma(t)}{\sqrt{\sigma^2(t)+\frac{1}{4}}}$ | 0 | $\geq 0$ | 2 |
| Flow Matching | FM (Lipman et al., 2022) | $t$ | $1-t$ | $1$ | $-1$ | 0 | $\geq 0$ | 1 |
| Consistency | sCM (Lu & Song, 2024) | $\sin(t \cdot \frac{\pi}{2})$ | $\cos(t \cdot \frac{\pi}{2})$ | $\cos(t \cdot \frac{\pi}{2})$ | $\sin(t \cdot \frac{-\pi}{2})$ | 1 | 1 | 1 |

transfer of advancements across different paradigms; and (b) **limited cross-paradigm generalization**, as algorithms optimized for one paradigm (e.g., diffusion models) are often incompatible with others.

To address these limitations, we introduce UCGM, a novel framework that establishes a unified foundation for the theoretical understanding, training and sampling of continuous generative models (diffusion, flow matching, and consistency models). Within this framework, we derive a surrogate unified objective, which not only offers a formulation equivalent to the unified objective, but also, for the first time, shows that the few-step objective can be viewed as the multi-step objective plus a self-consistency term. Within this formulation, we link the instability of few-step model training to the self-alignment term that dominates the training dynamics as $\lambda \to 1$.

The unified trainer UCGM-T is built upon a unified objective, parameterized by a consistency ratio $\lambda \in [0,1]$. This allows a single training paradigm to flexibly produce models tailored for different inference regimes: models behave akin to multi-step diffusion or flow-matching approaches when $\lambda$ is close to 0, and transition towards few-step consistency-like models as $\lambda$ approaches 1. Furthermore, our unified framework supports compatibility with diverse noise schedules (e.g., linear, triangular, quadratic) without requiring algorithm-specific modifications.

Complementing UCGM-T, we propose a unified sampler UCGM-S that operates seamlessly with models trained under our objective. UCGM-S is designed to enhance and accelerate sampling from pre-trained models—including those from previous paradigms as well as ones trained via UCGM-T. The unifying power of UCGM is further demonstrated by its ability to encapsulate several major continuous generative paradigms as special instances, as summarized in Tab. 1. Moreover, as shown in Fig. 1, models trained with UCGM achieve high sample quality across a wide range of Number of Function Evaluations (NFEs).

**In summary, our contributions are:**

(a) We propose a unified framework that provides a theoretical foundation for the training and sampling of continuous generative models–including diffusion models, flow matching models, and consistency models–and derive a surrogate unified objective that, for the first time, theoretically shows that the few-step objective can be viewed as the multi-step objective plus a self-alignment term.

(b) We introduce a unified trainer UCGM-T , that seamlessly bridges few-step (e.g., consistency models) and multi-step (e.g., diffusion, flow matching) generative paradigms, accommodating diverse model architectures, latent autoencoders, and noise schedules. We also propose a unified sampler UCGM-S, which is compatible with our trained models and further accelerate and improve pre-trained models from existing yet distinct paradigms.

(c) We empirically validate the effectiveness and efficiency of UCGM. Our approach consistently matches or surpasses SOTA methods across various datasets, architectures, and resolutions, for both few-step and multi-step generation tasks (cf., the experimental results in Sec. 4).

## 2 PRELIMINARIES

Given a training dataset $\mathcal{D}$, let $p_{\text{data}}(\mathbf{x})$ represent its underlying data distribution, or $p_{\text{data}}(\mathbf{x}|\mathbf{c})$ under a condition $\mathbf{c}$. Continuous generative models seek to learn an estimator that gradually transforms a simple source distribution $p_{\mathbf{z}}(\mathbf{z})$ into a complex target distribution $p_{\text{data}}(\mathbf{x})$ within a continuous space. Typically, $p_{\mathbf{z}}(\mathbf{z})$ is represented by the standard Gaussian distribution $\mathcal{N}(\mathbf{0}, \mathbf{I})$. For brevity, we hereafter omit subscripts when the context is clear, and assume independence, i.e., $p(\mathbf{x}, \mathbf{z}) = p(\mathbf{x})p(\mathbf{z})$. For instance, diffusion models reverse a noising process that gradually perturbs a data sample $\mathbf{x} \sim p(\mathbf{x})$ into a noisy version $\mathbf{x}_t = \alpha(t)\mathbf{x} + \sigma(t)\mathbf{z}$, where $\mathbf{z} \sim \mathcal{N}(\mathbf{0}, \mathbf{I})$. Over the range

$t \in [0, T]$, the perturbation intensifies with increasing $t$, where higher $t$ values indicate more noise. Below, we introduce three learning paradigms for continuous generative models.

**Diffusion models (Ho et al., 2020; Song et al., 2020b; Karras et al., 2022).** In the widely adopted EDM method (Karras et al., 2022), the noising process is defined by setting $\alpha(t) = 1$, $\sigma(t) = t$. The training objective is given by $\mathbb{E}_{\mathbf{x},\mathbf{z},t}\left[\omega(t)\left\|\boldsymbol{f}_{\boldsymbol{\theta}}(\mathbf{x}_t, t) - \mathbf{x}\right\|_2^2\right]$ where $\omega(t)$ is a weighting function. The diffusion model is parameterized by $\boldsymbol{f}_{\boldsymbol{\theta}}(\mathbf{x}_t, t) = c_{\text{skip}}(t)\mathbf{x}_t + c_{\text{out}}(t)\boldsymbol{F}_{\boldsymbol{\theta}}(c_{\text{in}}(t)\mathbf{x}_t, c_{\text{noise}}(t))$ where $\boldsymbol{F}_{\theta}$ is a neural network, and the coefficients $c_{\text{skip}}$, $c_{\text{out}}$, $c_{\text{in}}$, and $c_{\text{noise}}$ are manually designed. During sampling, EDM solves the Probability Flow Ordinary Differential Equation (PF-ODE) (Song et al., 2020b): $\frac{d\mathbf{x}_t}{dt} = [\mathbf{x}_t - \boldsymbol{f}_{\boldsymbol{\theta}}(\mathbf{x}_t, t)]/t$, integrated from $t = T$ to $t = 0$.

**Flow matching (Lipman et al., 2022).** Flow matching models are similar to diffusion models but differ in the transport process from the source to the target distribution and in the neural network training objective. The forward transport process utilizes differentiable coefficients $\alpha(t)$ and $\gamma(t)$, such that $\mathbf{x}_t = \alpha(t)\mathbf{z} + \gamma(t)\mathbf{x}$. Typically, the coefficients satisfy the boundary conditions $\alpha(1) = \gamma(0) = 1$ and $\alpha(0) = \gamma(1) = 0$. The training objective is given by $\mathbb{E}_{\mathbf{x},\mathbf{z},t}\left[\omega(t)\left\|\boldsymbol{F}_{\boldsymbol{\theta}}(\mathbf{x}_t, t) - (\frac{d\alpha_t}{dt}\mathbf{z} + \frac{d\gamma_t}{dt}\mathbf{x})\right\|_2^2\right]$. Similar to diffusion models, the reverse transport process (i.e., sampling process) begins at $t = 1$ with $\mathbf{x}_1 \sim \mathcal{N}(\mathbf{0}, \mathbf{I})$ and solves the PF-ODE: $\frac{d\mathbf{x}_t}{dt} = \boldsymbol{F}_{\boldsymbol{\theta}}(\mathbf{x}_t, t)$, integrated from $t = 1$ to $t = 0$.

**Consistency models (Song et al., 2023; Lu & Song, 2024).** A consistency model $\boldsymbol{f}_{\boldsymbol{\theta}}(\mathbf{x}_t, t)$ is trained to map the noisy input $\mathbf{x}_t$ directly to the corresponding clean data $\mathbf{x}$ in one or few steps by following the sampling trajectory of the PF-ODE starting from $\mathbf{x}_t$. To be valid, $\boldsymbol{f}_{\boldsymbol{\theta}}$ must satisfy the boundary condition $\boldsymbol{f}_{\boldsymbol{\theta}}(\mathbf{x}, 0) \equiv \mathbf{x}$. Inspired by EDM (Karras et al., 2022), one approach to enforce this condition is to parameterize the consistency model as $\boldsymbol{f}_{\boldsymbol{\theta}}(\mathbf{x}_t, t) = c_{\text{skip}}(t)\mathbf{x}_t + c_{\text{out}}(t)\boldsymbol{F}_{\boldsymbol{\theta}}(c_{\text{in}}(t)\mathbf{x}_t, c_{\text{noise}}(t))$ with $c_{\text{skip}}(0) = 1$ and $c_{\text{out}}(0) = 0$. The training objective is defined between two adjacent time steps with a finite distance: $\mathbb{E}_{\mathbf{x}_t,t}[\omega(t)d(\boldsymbol{f}_{\boldsymbol{\theta}}(\mathbf{x}_t, t), \boldsymbol{f}_{\boldsymbol{\theta}^-}(\mathbf{x}_{t-\Delta t}, t - \Delta t))]$, where $\boldsymbol{\theta}^-$ denotes $\text{stopgrad}(\boldsymbol{\theta})$, $\Delta t > 0$ is the distance between adjacent time steps, and $d(\cdot, \cdot)$ is a metric function. Discrete-time consistency models are sensitive to the choice of $\Delta t$, necessitating manually designed annealing schedules (Song & Dhariwal, 2023; Geng et al., 2024) for rapid convergence. This limitation is addressed by proposing a training objective for continuous consistency models (Lu & Song, 2024), derived by taking the limit as $\Delta t \to 0$.

## 3 METHODOLOGY

This section elaborates on our two primary contributions: (1) the unified framework for continuous generation models and a surrogate loss function that affords a theoretical interpretation of model behavior. (2) the concrete instantiation of the unified framework through UCGM-T (for training) and UCGM-S (for sampling).

### 3.1 UNIFIED FRAMEWORK FOR CONTINUOUS GENERATIVE MODELS

We first propose a unified multi-step objective for diffusion and flow-matching models, which constitute all multi-step continuous generative models. Furthermore, we extend this unified multi-step objective to encompass both few-step models and multi-step models.

**Unified objective for multi-step continuous generative models.** We introduce a generalized training objective below that effectively trains generative models while encompassing the formulations presented in existing studies (Karras et al., 2022; Lipman et al., 2022; Liu et al., 2022; Ho et al., 2020; Song et al., 2020a):

$$\mathcal{L}(\boldsymbol{\theta}) := \mathbb{E}_{(\mathbf{z},\mathbf{x})\sim p(\mathbf{z},\mathbf{x}),t}\left[\frac{1}{\omega(t)}\left\|\boldsymbol{F}_{\boldsymbol{\theta}}(\mathbf{x}_t, t) - \mathbf{z}_t\right\|_2^2\right], \tag{1}$$

where time $t \in [0, 1]$, $\omega(t)$ is the weighting function for the loss, $\boldsymbol{F}_{\boldsymbol{\theta}}$ is a neural network[1] with parameters $\boldsymbol{\theta}$, $\mathbf{x}_t = \alpha(t)\mathbf{z} + \gamma(t)\mathbf{x}$, and $\mathbf{z}_t = \hat{\alpha}(t)\mathbf{z} + \hat{\gamma}(t)\mathbf{x}$. Here, $\alpha(t)$, $\gamma(t)$, $\hat{\alpha}(t)$, and $\hat{\gamma}(t)$ are the unified transport coefficients defined for UCGM. In this paper, we refer to equation (1) as **the multi-step objective**. Additionally, to efficiently and robustly train multi-step continuous generative models using (1), we propose the following *necessary assumption*:

---

[1]For simplicity, unless otherwise specified, we assume that any conditioning information $\mathbf{c}$ is incorporated into the network input. Thus, $\boldsymbol{F}_{\boldsymbol{\theta}}(\mathbf{x}_t, t)$ should be understood as $\boldsymbol{F}_{\boldsymbol{\theta}}(\mathbf{x}_t, t, \mathbf{c})$ when $\mathbf{c}$ is applicable.

> **Assumption 1 .** *The coefficients function $\alpha(t), \gamma(t), \hat{\alpha}(t), \hat{\gamma}(t)$ satisfy the following constraints:*
> *(a) $\alpha(t) \in C^1[0,1]$ and is non-decreasing, with $\alpha(0) = 0$, $\alpha(1) = 1$.*
> *(b) $\gamma(t) \in C^1[0,1]$ and is non-increasing, with $\gamma(0) = 1$, $\gamma(1) = 0$.*
> *(c) $\forall t \in [0,1], |\alpha(t) \cdot \hat{\gamma}(t) - \hat{\alpha}(t) \cdot \gamma(t)| > 0$.*

Under the Assump. 1, diffusion and flow matching are special cases of multi-step objective (1):

(a) **Diffusion**: following EDM (Karras et al., 2022; 2024b), by setting $\alpha(t) = 1$ and $\sigma(t) = t$, diffusion models based on EDM can be derived from (1) provided that the constraint $\gamma(t)/\alpha(t) = t$ is satisfied[2].

(b) **Flow Matching**: Similarly, flow matching can be derived only when $\hat{\alpha}(t) = \frac{\mathrm{d}\alpha(t)}{\mathrm{d}t}$ and $\hat{\gamma}(t) = \frac{\mathrm{d}\gamma(t)}{\mathrm{d}t}$ (see Sec. 2 for more technical details about EDM-based and flow-based models).

**Unified objective for both multi-step and few-step models.**   To facilitate the interpretation of our framework, we define two prediction functions based on model $\boldsymbol{F_\theta}$ as:

$$\boldsymbol{f^x}(\boldsymbol{F}_t^{\boldsymbol{\theta}}, \mathbf{x}_t, t) := \frac{\alpha(t) \cdot \boldsymbol{F}_t^{\boldsymbol{\theta}} - \hat{\alpha}(t) \cdot \mathbf{x}_t}{\alpha(t) \cdot \hat{\gamma}(t) - \hat{\alpha}(t) \cdot \gamma(t)} \quad \& \quad \boldsymbol{f^z}(\boldsymbol{F}_t^{\boldsymbol{\theta}}, \mathbf{x}_t, t) := \frac{\hat{\gamma}(t) \cdot \mathbf{x}_t - \gamma(t) \cdot \boldsymbol{F}_t^{\boldsymbol{\theta}}}{\alpha(t) \cdot \hat{\gamma}(t) - \hat{\alpha}(t) \cdot \gamma(t)} , \quad (2)$$

where we define $\boldsymbol{F}_t^{\boldsymbol{\theta}} := \boldsymbol{F_\theta}(\mathbf{x}_t, t)$. The training objective (1) thus becomes (cf., App. F.1.1):

$$\mathcal{L}(\boldsymbol{\theta}) = \mathbb{E}_{(\mathbf{z},\mathbf{x}) \sim p(\mathbf{z},\mathbf{x}),t} \left[ \frac{1}{\hat{\omega}(t)} \|\boldsymbol{f^x}(\boldsymbol{F_\theta}(\mathbf{x}_t, t), \mathbf{x}_t, t) - \mathbf{x}\|_2^2 \right] . \quad (3)$$

To align with the gradient of multi-step objective (1), we define a new weighting function $\hat{\omega}(t)$ in (3) as $\hat{\omega}(t) := \frac{\alpha(t) \cdot \alpha(t) \cdot \omega(t)}{(\alpha(t) \cdot \hat{\gamma}(t) - \hat{\alpha}(t) \cdot \gamma(t))^2}$ . To unify few-step models (such as consistency models) with multi-step models, we adopt a modified version of (3) by incorporating a consistency ratio $\lambda \in [0,1]$:

$$\mathcal{L}(\boldsymbol{\theta}) = \mathbb{E}_{(\mathbf{z},\mathbf{x}) \sim p(\mathbf{z},\mathbf{x}),t} \left[ \frac{1}{\hat{\omega}(t)} \|\boldsymbol{f^x}(\boldsymbol{F_\theta}(\mathbf{x}_t, t), \mathbf{x}_t, t) - \boldsymbol{f^x}(\boldsymbol{F_{\theta^-}}(\mathbf{x}_{\lambda t}, \lambda t), \mathbf{x}_{\lambda t}, \lambda t)\|_2^2 \right] , \quad (4)$$

where consistency models and conventional multi-steps models are special cases within the context of (4) (cf., App. F.1.1 and App. F.1.3):

(a) **Diffusion / Flow Matching:** setting $\lambda = 0$ yields diffusion and flow matching, and our unified objective (4) degrades to the objective (3), which is equivalent to the multi-step objective (1).

(b) **Consistency Model:** setting $\lambda = 1 - \frac{\Delta t}{t}$ with $\Delta t \to 0$ recovers consistency models.

**Equivalent surrogate objective for unified objective** (4).   Building on the unified objective (4), we derive an equivalent surrogate objective. Importantly, this surrogate not only provides an equivalent reformulation of the unified objective but also sheds light on the theoretical origin of instability in few-step models, like consistency model.

> **Theorem 1 (Surrogate objective for unified objective of linear case ($\alpha(t) = t$, $\gamma(t) = 1 - t$)) .** *Under Assump. 1, let's consider a surrogate objective*
>
> $$\mathcal{G}(\boldsymbol{\theta}) = \mathbb{E}_{\mathbf{z},\mathbf{x},t} \left[ \underbrace{\left\| \boldsymbol{F_\theta}(\mathbf{x}_t, t) - \mathbf{z}_t \right\|_2^2}_{\text{Flow Matching Objective}} + \frac{\lambda}{1-\lambda} \underbrace{\left\| \boldsymbol{F_\theta}(\mathbf{x}_t, t) - \boldsymbol{F_{\theta^-}}(\mathbf{x}_{\lambda t}, \lambda t) \right\|_2^2}_{\text{Self-Alignment Term}} \right], \quad (5)$$
>
> *where $\mathbf{x}_t = t \cdot \mathbf{z} + (1 - t) \cdot \mathbf{x}, \mathbf{z}_t = \mathbf{z} - \mathbf{x}, \hat{\omega}(t) = t^2 \cdot (1 - \lambda), 0 < \lambda < 1$. The following equation holds: $\nabla_{\boldsymbol{\theta}} \mathcal{L}(\boldsymbol{\theta}) = \nabla_{\boldsymbol{\theta}} \mathcal{G}(\boldsymbol{\theta}), \forall \boldsymbol{\theta}$. See App. F.1.5 for proof and general case.*

Thm. 1 establishes that optimizing the unified objective in (4) is equivalent to optimizing the surrogate objective in (5). This equivalence is useful for analysis because the surrogate, $\mathcal{G}(\boldsymbol{\theta})$, can be decomposed into two distinct components: a multi-step objective term and a self-alignment term. We can offer a physical interpretation for each component by considering the underlying function $\boldsymbol{F_\theta}(\mathbf{x}_t, t)$ as a learned velocity field:

- **Flow matching objective:** This term corresponds to the learning objective of multi-step models (1). It learns the mean velocity $\mathbf{z}_t = \mathbf{z} - \mathbf{x} = \frac{\mathbf{x}_1 - \mathbf{x}_0}{1 - 0}$ of a flow trajectory.

---

[2]In EDM, with $\sigma(t) = t$, the input of neural network $\boldsymbol{F_\theta}$ is $c_{\text{in}}(t)\mathbf{x}_t = c_{\text{in}}(t) \cdot (\mathbf{x} + t \cdot \mathbf{z})$. Although $c_{\text{in}}(t)$ can be manually adjusted, the coefficient before $\mathbf{z}$ remains $t$ times that of $\mathbf{x}$.

- **Self-alignment term:** This term can be considered as a regularization term, which enforces consistency of the velocity of any points within a flow trajectory, ultimately helping to straighten the learned trajectories.

> **Remark 1 (Analysis of instability of few-step objective (i.e. $\lambda \to 1$)) .** *According to Thm. 1, as $\lambda \to 1$, the self-alignment term dominates the loss function. This term only requires the velocity to be consistent in each flow trajectory, without constraining it to match the mean velocity. Thus, while a pre-trained velocity field may initially be straightened under this objective, prolonged training with few-step objective ultimately degrades the quality of the velocity field.*

## 3.2 Instantiating the Unified Framework for Training (UCGM-T)

Applying the gradient identity from Lu & Song (2024)[3], we derive the unified objective:

$$\mathbb{E}_{(\mathbf{z},\mathbf{x}) \sim p(\mathbf{z},\mathbf{x}),t} \left[ \left\| \boldsymbol{F}_{\boldsymbol{\theta}}(\mathbf{x}_t, t) - \boldsymbol{F}_{\boldsymbol{\theta}^-}(\mathbf{x}_t, t) + 2 \cdot \frac{\Delta \boldsymbol{f}^{\mathbf{x}}}{B(t) - B(\lambda t)} \right\|_2^2 \right], \tag{6}$$

where the detailed derivation from (4) to (6) is provided in App. F.1.7, and

$$\Delta \boldsymbol{f}^{\mathbf{x}} := \boldsymbol{f}^{\mathbf{x}}(\boldsymbol{F}_t^{\boldsymbol{\theta}^-}, \mathbf{x}_t, t) - \boldsymbol{f}^{\mathbf{x}}(\boldsymbol{F}_{\lambda t}^{\boldsymbol{\theta}^-}, \mathbf{x}_{\lambda t}, \lambda t), \ B(t) := \alpha(t)/(\alpha(t)\hat{\gamma}(t) - \hat{\alpha}(t)\gamma(t)) .$$

**Second-order estimator as $\lambda \to 1$.** We identify that the direct estimation of the difference quotient in objective (6) is only a first-order approximation, which is susceptible to numerical precision errors. To mitigate this issue, we propose a second-order estimator:

$$\frac{\Delta \boldsymbol{f}^{\mathbf{x}}}{B(t) - B(\lambda t)} \approx \frac{\boldsymbol{f}^{\mathbf{x}}(\boldsymbol{F}_{\boldsymbol{\theta}^-}(\mathbf{x}_{t+\epsilon}, t+\epsilon), \mathbf{x}_{t+\epsilon}, t+\epsilon) - \boldsymbol{f}^{\mathbf{x}}(\boldsymbol{F}_{\boldsymbol{\theta}^-}(\mathbf{x}_{t-\epsilon}, t-\epsilon), \mathbf{x}_{t-\epsilon}, t-\epsilon)}{B(t+\epsilon) - B(t-\epsilon)} .$$

See App. F.2.3 for further analysis of this second-order estimator. To stabilize the training, we implement two strategies for the second-order estimation: (1) We adopt a distributive reformulation of the second-difference term to prevent direct subtraction $\Delta \boldsymbol{f}_t^{\mathbf{x}} = \boldsymbol{f}^{\mathbf{x}}(\boldsymbol{F}_{\boldsymbol{\theta}^-}(\mathbf{x}_{t+\epsilon}, t+\epsilon), \mathbf{x}_{t+\epsilon}, t+\epsilon) \cdot \frac{1}{2\epsilon} - \boldsymbol{f}^{\mathbf{x}}(\boldsymbol{F}_{\boldsymbol{\theta}^-}(\mathbf{x}_{t-\epsilon}, t-\epsilon), \mathbf{x}_{t-\epsilon}, t-\epsilon) \cdot \frac{1}{2\epsilon}$. (2) we also observe that applying numerical truncation $\text{clip}(\cdot, -1, 1)$ to the second-order estimator enhances training stability (Lu & Song, 2024).

**Generalized time distribution (GTD)** $\text{Beta}(\theta_1, \theta_2)$**.** Previous studies (Yao et al., 2025; Esser et al., 2024; Song et al., 2023; Lu & Song, 2024; Karras et al., 2022; 2024b) employ non-linear functions to transform the time variable $t$, initially sampled from a uniform distribution $t \sim \mathcal{U}(0, 1)$. This transformation shifts the distribution of sampled times, effectively performing importance sampling and thereby accelerating the training convergence rate. For example, the lognorm function $f_{\text{lognorm}}(t; \mu, \sigma) = 1/1+\exp(-\mu-\sigma\cdot\Phi^{-1}(t))$ is widely used (Yao et al., 2025; Esser et al., 2024), where $\Phi^{-1}(\cdot)$ denotes the inverse Cumulative Distribution Function of the standard normal distribution. In this work, we demonstrate that commonly used time distribution after non-linear time transformation can be well-approximated by the Beta distribution (a detailed analysis is provided in App. F.2.1). Consequently, we simplify the process by directly sampling time from a Beta distribution, i.e., $t \sim \text{Beta}(\theta_1, \theta_2)$, where $\theta_1$ and $\theta_2$ are parameters that control the shape of time distribution (see App. D.1.3 for their settings).

**Learning enhanced target score function.** We additionally incorporate the enhanced target score function proposed in recent work (Tang et al., 2025) into our unified training objective in (6). This technique is not our main contribution but can be seamlessly integrated into our framework. For completeness, we provide the formulation and further analysis in App. F.1.8.

An ablation study for our proposed techniques is shown in Tab. 13, and the pseudocode is in Alg. 1.

## 3.3 Instantiating the Unified Framework for Sampling (UCGM-S)

For classical iterative sampling models, such as a trained flow-matching model $\boldsymbol{f}_{\boldsymbol{\theta}}$, sampling from the learned distribution $p(\mathbf{x})$ involves solving the PF-ODE (Song et al., 2020b). This process typically uses numerical ODE solvers, such as the Euler or Runge-Kutta methods (Ma et al., 2024), to iteratively transform the initial Gaussian noise $\tilde{\mathbf{x}}$ into a sample from $p(\mathbf{x})$ by solving the ODE (i.e., $\frac{d\tilde{\mathbf{x}}_t}{dt} = \boldsymbol{f}_{\boldsymbol{\theta}}(\tilde{\mathbf{x}}_t, t)$), Similarly, sampling processes in models like EDM (Karras et al., 2022; 2024b) and

---

[3]$\nabla_{\theta}\mathbb{E}[\boldsymbol{F}_{\theta}^{\top}\boldsymbol{y}] = \frac{1}{2}\nabla_{\theta}\mathbb{E}[\|\boldsymbol{F}_{\theta} - \boldsymbol{F}_{\theta^-} + \boldsymbol{y}\|_2^2]$.

consistency models (Song et al., 2023) involve a comparable gradual denoising procedure. Building on these observations and our unified trainer UCGM-T, we first propose a general iterative sampling process with two stages below:

(a) **Decomposition:** At time $t$, the current input $\tilde{\mathbf{x}}_t$ is decomposed into two components: $\tilde{\mathbf{x}}_t = \alpha(t) \cdot \hat{\mathbf{z}}_t + \gamma(t) \cdot \hat{\mathbf{x}}_t$. This decomposition uses the estimation model $\boldsymbol{F}_{\boldsymbol{\theta}}$. Specifically, the model output $\boldsymbol{F}_t = \boldsymbol{F}_{\boldsymbol{\theta}^-}(\tilde{\mathbf{x}}_t, t)$ is computed, yielding the estimated clean component $\hat{\mathbf{x}}_t = \boldsymbol{f}^{\mathbf{x}}(\boldsymbol{F}_t, \tilde{\mathbf{x}}_t, t)$ and the estimated noise component $\hat{\mathbf{z}}_t = \boldsymbol{f}^{\mathbf{z}}(\boldsymbol{F}_t, \tilde{\mathbf{x}}_t, t)$.

(b) **Reconstruction:** The next time step's input, $t'$, is generated by combining the estimated components: $\tilde{\mathbf{x}}_{t'} = \alpha(t') \cdot \hat{\mathbf{z}}_t + \gamma(t') \cdot \hat{\mathbf{x}}_t$. The process then iterates to stage (a).

We then introduce two enhancement techniques below to optimize the sampling process:

(i) **Extrapolating the estimation.** Directly utilizing the estimated $\hat{\mathbf{x}}_t$ and $\hat{\mathbf{z}}_t$ to reconstruct the subsequent input $\tilde{\mathbf{x}}_{t'}$ can result in significant estimation errors, as the estimation model $\boldsymbol{F}_{\boldsymbol{\theta}}$ does not perfectly align with the target function $\boldsymbol{F}^{\text{target}}$ for solving the PF-ODE.
Note that CFG guides a conditional model using an unconditional model, i.e., $\boldsymbol{f}_{\boldsymbol{\theta}}(\tilde{\mathbf{x}}, t) = \boldsymbol{f}_{\boldsymbol{\theta}}(\tilde{\mathbf{x}}, t) + \kappa \cdot \left( \boldsymbol{f}_{\boldsymbol{\theta}}^{\varnothing}(\tilde{\mathbf{x}}, t) - \boldsymbol{f}_{\boldsymbol{\theta}}(\tilde{\mathbf{x}}, t) \right)$ where $\kappa$ is the guidance ratio. This approach can be interpreted as leveraging a less accurate estimation to guide a more accurate one (Karras et al., 2024a). Extending this insight, we propose to extrapolate the next time-step estimates $\hat{\mathbf{x}}_{t'}$ and $\hat{\mathbf{z}}_{t'}$ using the previous estimates $\hat{\mathbf{x}}_t$ and $\hat{\mathbf{z}}_t$, formulated as: $\hat{\mathbf{x}}_{t'} \leftarrow \hat{\mathbf{x}}_{t'} + \kappa \cdot (\hat{\mathbf{x}}_{t'} - \hat{\mathbf{x}}_t)$ and $\hat{\mathbf{z}}_{t'} \leftarrow \hat{\mathbf{z}}_{t'} + \kappa \cdot (\hat{\mathbf{z}}_{t'} - \hat{\mathbf{z}}_t)$, where $\kappa \in [0, 1]$ is the extrapolation ratio. This extrapolation process can significantly enhance sampling quality and reduce the number of sampling steps. Notably, this technique is compatible with CFG and does not introduce additional computational overhead (see Sec. 4.2 for experimental details and App. F.1.10 for theoretical analysis).

(ii) **Incorporating stochasticity.** During the aforementioned sampling process, the input $\tilde{\mathbf{x}}_t$ is deterministic, potentially limiting the diversity of generated samples. To mitigate this, we introduce a stochastic term $\rho$ to $\tilde{\mathbf{x}}_t$, defined as: $\tilde{\mathbf{x}}_{t'} = \alpha(t') \cdot \left( \sqrt{1 - \rho} \cdot \hat{\mathbf{z}}_t + \sqrt{\rho} \cdot \mathbf{z} \right) + \gamma(t') \cdot \hat{\mathbf{x}}_t$, where $\mathbf{z} \sim \mathcal{N}(\mathbf{0}, \mathbf{I})$ is a random noise vector, and $\rho$ is the stochasticity ratio. This stochastic term acts as a random perturbation to $\tilde{\mathbf{x}}_t$, thereby enhancing the diversity of generated samples. We adopt $\rho = \lambda$ as the default configuration, with further analysis provided in App. F.1.11.

**Unified sampling algorithm UCGM-S.** Putting all these factors together, here we introduce a unified sampling algorithm applicable to consistency models and diffusion/flow-based models, as presented in Alg. 2. An ablation study for our proposed techniques is in Tab. 14. Extensive experiments (cf., Sec. 4) demonstrate two key features of this algorithm:

(a) *Reduced computational resources:* It decreases the number of sampling steps required by existing models while maintaining or enhancing performance.

(b) *High compatibility:* It is compatible with existing models, irrespective of their training objectives or noise schedules, without necessitating modifications to model architectures or tuning.

## 4 EXPERIMENT

This section details the experimental setup and evaluation of our proposed methodology, UCGM-$\{T, S\}$. Note that our approach relies on specific parameterizations of the transport coefficients $\alpha(\cdot)$, $\gamma(\cdot)$, $\hat{\alpha}(\cdot)$, and $\hat{\gamma}(\cdot)$, as detailed in Alg. 1 and Alg. 2. Therefore, Tab. 7 summarizes the parameterizations used in experiments, including configurations for compatibility with prior methods.

### 4.1 EXPERIMENTAL SETTING

**Datasets.** We utilize ImageNet-1K (Deng et al., 2009) at resolutions of $512 \times 512$ and $256 \times 256$ as our primary datasets, following prior studies (Karras et al., 2024b; Song et al., 2023) and adhering to ADM's data preprocessing protocols (Dhariwal & Nichol, 2021). Additionally, CIFAR-10 (Krizhevsky et al., 2009b) at a resolution of $32 \times 32$ is employed for ablation studies.

For both $512 \times 512$ and $256 \times 256$ images, experiments are conducted using latent space generative modeling in line with previous works. Specifically: (a) For $256 \times 256$ images, we employ multiple widely-used autoencoders, including SD-VAE (Rombach et al., 2022), VA-VAE (Yao et al., 2025), and E2E-VAE (Leng et al., 2025). (b) For $512 \times 512$ images, a DC-AE (*f32c32*) (Chen et al., 2024c) with a higher compression rate is used to conserve computational resources. When utilizing SD-VAE for $512 \times 512$ images, a $2\times$ larger patch size is applied to maintain computational parity with the $256 \times 256$ setting. Consequently, the computational burden for generating images at both $512 \times 512$

Table 2: **System-level quality comparison for multi-step generation task on class-conditional ImageNet-1K.** Notation A⊕B denotes the result obtained by combining methods A and B. $\downarrow$/$\uparrow$ indicate a decrease/increase, respectively, in the metric compared to the baseline performance of the pre-trained models.

| 512 × 512 | | | | | 256 × 256 | | | | |
|---|---|---|---|---|---|---|---|---|---|
| METHOD | NFE ($\downarrow$) | FID ($\downarrow$) | #Params | #Epochs | METHOD | NFE ($\downarrow$) | FID ($\downarrow$) | #Params | #Epochs |
| Diffusion & flow-matching Models | | | | | | | | | |
| ADM-G (Dhariwal & Nichol, 2021) | 250×2 | 7.72 | 559M | 388 | ADM-G (Dhariwal & Nichol, 2021) | 250×2 | 4.59 | 559M | 396 |
| U-ViT-H/4 (Bao et al., 2023) | 50×2 | 4.05 | 501M | 400 | U-ViT-H/2 (Bao et al., 2023) | 50×2 | 2.29 | 501M | 400 |
| DiT-XL/2 (Peebles & Xie, 2023) | 250×2 | 3.04 | 675M | 600 | DiT-XL/2 (Peebles & Xie, 2023) | 250×2 | 2.27 | 675M | 1400 |
| SiT-XL/2 (Ma et al., 2024) | 250×2 | 2.62 | 675M | 600 | SiT-XL/2 (Ma et al., 2024) | 250×2 | 2.06 | 675M | 1400 |
| MaskDiT (Zheng et al., 2023) | 79×2 | 2.50 | 736M | - | MDT (Gao et al., 2023) | 250×2 | 1.79 | 675M | 1300 |
| EDM2-S (Karras et al., 2024b) | 63 | 2.56 | 280M | 1678 | REPA-XL/2 (Yu et al., 2024) | 250×2 | 1.96 | 675M | 200 |
| EDM2-L (Karras et al., 2024b) | 63 | 2.06 | 778M | 1476 | REPA-XL/2 (Yu et al., 2024) | 250×2 | 1.42 | 675M | 800 |
| EDM2-XXL (Karras et al., 2024b) | 63 | 1.91 | 1.5B | 734 | Light.DiT (Yao et al., 2025) | 250×2 | 2.11 | 675M | 64 |
| DiT-XL/1⊕Chen et al. (2024c) | 250×2 | 2.41 | 675M | 400 | Light.DiT (Yao et al., 2025) | 250×2 | 1.35 | 675M | 800 |
| U-ViT-H/1⊕Chen et al. (2024c) | 30×2 | 2.53 | 501M | 400 | DDT-XL/2 (Wang et al., 2025) | 250×2 | 1.31 | 675M | 256 |
| REPA-XL/2 (Yu et al., 2024) | 250×2 | 2.08 | 675M | 200 | DDT-XL/2 (Wang et al., 2025) | 250×2 | 1.26 | 675M | 400 |
| DDT-XL/2 (Wang et al., 2025) | 250×2 | **1.28** | 675M | - | REPA-E-XL (Leng et al., 2025) | 250×2 | **1.26** | 675M | 800 |
| GANs & masked & autoregressive models | | | | | | | | | |
| VQGAN⊕Esser et al. (2021) | 256 | 18.65 | 227M | - | VQGAN⊕Sun et al. (2024) | - | 2.18 | 3.1B | 300 |
| MAGVIT-v2 (Yu et al., 2023) | 64×2 | 1.91 | 307M | 1080 | MAR-L (Li et al., 2024) | 256×2 | 1.78 | 479M | 800 |
| MAR-L (Li et al., 2024) | 256×2 | **1.73** | 479M | 800 | MAR-H (Li et al., 2024) | 256×2 | **1.55** | 943M | 800 |
| VAR-$d36$-s (Tian et al., 2024) | 10×2 | 2.63 | 2.3B | 350 | VAR-$d30$-re (Tian et al., 2024) | 10×2 | 1.73 | 2.0B | 350 |
| Ours: UCGM-S sampling with models trained by prior works | | | | | | | | | |
| UCGM-S⊕Karras et al. (2024b) | $40^{\downarrow 23}$ | $2.53^{\downarrow 0.03}$ | 280M | - | UCGM-S⊕Wang et al. (2025) | $100^{\downarrow 400}$ | $1.27^{\uparrow 0.01}$ | 675M | - |
| UCGM-S⊕Karras et al. (2024b) | $50^{\downarrow 13}$ | $2.04^{\downarrow 0.02}$ | 778M | - | UCGM-S⊕Yao et al. (2025) | $100^{\downarrow 400}$ | $1.21^{\downarrow 0.14}$ | 675M | - |
| UCGM-S⊕Karras et al. (2024b) | $40^{\downarrow 23}$ | $1.88^{\downarrow 0.03}$ | 1.5B | - | UCGM-S⊕Leng et al. (2025) | $80^{\downarrow 420}$ | $\mathbf{1.06}^{\downarrow 0.20}$ | 675M | - |
| UCGM-S⊕Wang et al. (2025) | $200^{\downarrow 300}$ | $\mathbf{1.25}^{\downarrow 0.03}$ | 675M | - | UCGM-S⊕Leng et al. (2025) | $20^{\downarrow 480}$ | $2.00^{\uparrow 0.74}$ | 675M | - |
| Ours: models trained and sampled using UCGM-{T, S} (setting $\lambda = 0$) | | | | | | | | | |
| ⊕DC-AE (Chen et al., 2024c) | 40 | **1.48** | 675M | 800 | ⊕SD-VAE (Rombach et al., 2022) | 60 | 1.41 | 675M | 400 |
| ⊕DC-AE (Chen et al., 2024c) | 20 | 1.68 | 675M | 800 | ⊕VA-VAE (Yao et al., 2025) | 60 | 1.21 | 675M | 400 |
| ⊕SD-VAE (Rombach et al., 2022) | 40 | 1.67 | 675M | 320 | ⊕E2E-VAE (Leng et al., 2025) | 40 | **1.21** | 675M | 800 |
| ⊕SD-VAE (Rombach et al., 2022) | 20 | 1.80 | 675M | 320 | ⊕E2E-VAE (Leng et al., 2025) | 20 | 1.30 | 675M | 800 |

and $256 \times 256$ resolutions remains comparable across our trained models[4]. Further details on datasets and autoencoders are provided in App. D.1.1.

**Neural network architectures.** We evaluate UCGM-S sampling using models trained with established methodologies. These models employ various architectures from two prevalent families commonly used in continuous generative models: (a) Diffusion Transformers, including variants such as DiT (Peebles & Xie, 2023), UViT (Bao et al., 2023), SiT (Ma et al., 2024), Lightening-DiT (Yao et al., 2025), and DDT (Wang et al., 2025). (b) UNet-based convolutional networks, including improved UNets (Karras et al., 2022; Song et al., 2020b) and EDM2-UNets (Karras et al., 2024b). For training models specifically for UCGM-T, we consistently utilize DiT as the backbone architecture. We train models of various sizes (B: 130M, L: 458M, XL: 675M parameters) and patch sizes. Notation such as XL/2 denotes the XL model with a patch size of 2. Following prior work (Yao et al., 2025; Wang et al., 2025), minor architectural modifications are applied to enhance training stability (details in App. D.1.2).

**Implementation details.** Our implementation is developed in PyTorch (Paszke, 2019). Training employs AdamW (Loshchilov & Hutter, 2017) for multi-step sampling models. For few-step sampling models, RAdam (Liu et al., 2019) is used to improve training stability. Consistent with standard practice in generative modeling (Yu et al., 2024; Ma et al., 2024), an exponential moving average (EMA) of model weights is maintained throughout training using a decay rate of 0.9999. All reported results utilize the EMA model. Comprehensive hyperparameters and additional implementation details are provided in App. D.1.3. Consistent with prior work (Song et al., 2020b; Ho et al., 2020; Lipman et al., 2022; Brock et al., 2018), we adopt standard evaluation protocols. The primary metric for assessing image quality is the Fréchet Inception Distance (FID) (Heusel et al., 2017), calculated on $50,000$ images (FID-50K).

### 4.2 COMPARISON WITH SOTA METHODS FOR MULTI-STEP GENERATION

Our experiments on ImageNet-1K at $512 \times 512$ and $256 \times 256$ resolutions systematically validate the three key advantages of UCGM: (1) sampling acceleration via UCGM-S on pre-trained models, (2) ultra-efficient generation with joint UCGM-T + UCGM-S, and (3) broad compatibility.

**UCGM-S: Plug-and-play sampling acceleration without additional cost.** UCGM-S provides free sampling acceleration for pre-trained generative models. It reduces the required Number of

---

[4]Previous works often employed the same autoencoders and patch sizes for both resolutions, resulting in higher computational costs for generating $512 \times 512$ images. For example, the DiT-XL/2 model requires $524.60$ GFLOPs for $512 \times 512$ generation, in contrast to $118.64$ GFLOPs for $256 \times 256$.

Table 3: **System-level quality comparison for few-step generation task on class-conditional ImageNet-1K.**

| | 512 × 512 | | | | 256 × 256 | | | |
|---|---|---|---|---|---|---|---|---|
| METHOD | NFE (↓) | FID (↓) | #Params | #Epochs | METHOD | NFE (↓) | FID (↓) | #Params | #Epochs |
| **Consistency training & distillation** | | | | | | | | | |
| sCT-M (Lu & Song, 2024) | 1 | 5.84 | 498M | 1837 | iCT (Song & Dhariwal, 2023) | 2 | 20.3 | 675M | - |
| | 2 | 5.53 | 498M | 1837 | Shortcut-XL/2 (Frans et al., 2024) | 1 | 10.6 | 676M | 250 |
| sCT-L (Lu & Song, 2024) | 1 | 5.15 | 778M | 1274 | | 4 | 7.80 | 676M | 250 |
| | 2 | 4.65 | 778M | 1274 | | 128 | 3.80 | 676M | 250 |
| sCT-XXL (Lu & Song, 2024) | 1 | 4.29 | 1.5B | 762 | IMM-XL/2 (Zhou et al., 2025) | 1×2 | 7.77 | 675M | 3840 |
| | 2 | 3.76 | 1.5B | 762 | | 2×2 | 5.33 | 675M | 3840 |
| sCD-M (Lu & Song, 2024) | 1 | 2.75 | 498M | 1997 | | 4×2 | 3.66 | 675M | 3840 |
| | 2 | 2.26 | 498M | 1997 | | 8×2 | 2.77 | 675M | 3840 |
| sCD-L (Lu & Song, 2024) | 1 | 2.55 | 778M | 1434 | IMM ($\omega = 1.5$) | 1×2 | 8.05 | 675M | 3840 |
| | 2 | 2.04 | 778M | 1434 | | 2×2 | 3.99 | 675M | 3840 |
| sCD-XXL (Lu & Song, 2024) | 1 | 2.28 | 1.5B | 921 | | 4×2 | 2.51 | 675M | 3840 |
| | 2 | **1.88** | 1.5B | 921 | | 8×2 | **1.99** | 675M | 3840 |
| **GANs & masked & autoregressive models** | | | | | | | | | |
| BigGAN (Brock et al., 2018) | 1 | 8.43 | 160M | - | BigGAN (Brock et al., 2018) | 1 | 6.95 | 112M | - |
| StyleGAN (Sauer et al., 2022) | 1×2 | **2.41** | 168M | - | GigaGAN (Kang et al., 2023) | 1 | 3.45 | 569M | - |
| MAGVIT-v2 (Yu et al., 2023) | 64×2 | 1.91 | 307M | 1080 | StyleGAN (Sauer et al., 2022) | 1×2 | **2.30** | 166M | - |
| VAR-$d$36-s (Tian et al., 2024) | 10×2 | 2.63 | 2.3B | 350 | VAR-$d$30-re (Tian et al., 2024) | 10×2 | 1.73 | 2.0B | 350 |
| **Ours: models trained and sampled using UCGM-{T,S} (setting $\lambda = 0$)** | | | | | | | | | |
| ⊕DC-AE (Chen et al., 2024c) | 32 | **1.55** | 675M | 800 | ⊕VA-VAE (Yao et al., 2025) | 16 | 2.11 | 675M | 400 |
| ⊕DC-AE (Chen et al., 2024c) | 16 | 1.81 | 675M | 800 | ⊕VA-VAE (Yao et al., 2025) | 8 | 6.09 | 675M | 400 |
| ⊕DC-AE (Chen et al., 2024c) | 8 | 3.07 | 675M | 800 | ⊕E2E-VAE (Leng et al., 2025) | 16 | **1.40** | 675M | 800 |
| ⊕DC-AE (Chen et al., 2024c) | 4 | 74.0 | 675M | 800 | ⊕E2E-VAE (Leng et al., 2025) | 8 | 2.68 | 675M | 800 |
| **Ours: models trained and sampled using UCGM-{T,S} (setting $\lambda = 1$)** | | | | | | | | | |
| ⊕DC-AE (Chen et al., 2024c) | 1 | 2.42 | 675M | 840 | ⊕VA-VAE (Yao et al., 2025) | 2 | **1.42** | 675M | 432 |
| ⊕DC-AE (Chen et al., 2024c) | 2 | **1.75** | 675M | 840 | ⊕VA-VAE (Yao et al., 2025) | 1 | 2.19 | 675M | 432 |
| ⊕SD-VAE (Rombach et al., 2022) | 1 | 2.63 | 675M | 360 | ⊕SD-VAE (Rombach et al., 2022) | 1 | 2.10 | 675M | 424 |
| ⊕SD-VAE (Rombach et al., 2022) | 2 | 2.11 | 675M | 360 | ⊕E2E-VAE (Leng et al., 2025) | 1 | 2.29 | 675M | 264 |

Function Evaluations (NFEs) while preserving or improving generation quality, as measured by FID. Applied to $512 \times 512$ image generation, the approach demonstrates notable efficiency gains:

(a) For the diffusion-based models, such as a pre-trained EDM2-XXL model, UCGM-S reduced NFEs from 63 to 40 (a 36.5% reduction), concurrently improving FID from 1.91 to 1.88.

(b) When applied to the flow-based models, such as a pre-trained DDT-XL/2 model, UCGM-S achieved an FID of 1.25 with 200 NFEs, compared to the original 1.28 FID requiring 500 NFEs. This demonstrates a performance improvement achieved alongside enhanced efficiency.

This approach generalizes across different generative model frameworks and resolutions. For instance, on $256 \times 256$ resolution using the flow-based REPA-E-XL model, UCGM-S attained 1.06 FID at 80 NFEs, which surpasses the baseline performance of 1.26 FID achieved at 500 NFEs.

In summary, UCGM-*S acts as a broadly applicable technique for efficient sampling, demonstrating cases where performance (FID) improves despite a reduction in sampling steps*.

**UCGM-T + UCGM-S: Synergistic efficiency.** The combination of UCGM-T training and UCGM-S sampling yields highly competitive generative performance with minimal NFEs:

(a) $512 \times 512$: With a DC-AE autoencoder, our framework achieved 1.48 FID at 40 NFEs. This outperforms DiT-XL/1⊕DC-AE (2.41 FID, 500 NFEs) and EDM2-XXL (1.91 FID, 63 NFEs), with comparable or reduced model size.

(b) $256 \times 256$: With an E2E-VAE autoencoder, we attained 1.21 FID at 40 NFEs. This result exceeds prior SOTA models like MAR-H (1.55 FID, 512 NFEs) and REPA-E-XL (1.26 FID, 500 NFEs).

Importantly, models trained with UCGM-T maintain robustness under extremely low-step sampling regimes. At 20 NFEs, the $256 \times 256$ performance degrades gracefully to 1.30 FID, a result that still exceeds the performance of several baseline models sampling with significantly higher NFEs.

In summary, *the demonstrated robustness and efficiency of* UCGM-*{T, S} across various scenarios underscore the high potential of our* UCGM *for multi-step continuous generative modeling*.

### 4.3 COMPARISON WITH SOTA METHODS FOR FEW-STEP GENERATION

As evidenced by the results in Tab. 3, our UCGM-{T, S} framework exhibits superior performance across two key settings: $\lambda = 0$, characteristic of a multi-step regime akin to diffusion and flow-matching models, and $\lambda = 1$, indicative of a few-step regime resembling consistency models.

**Few-step regime ($\lambda = 1$).** Configured for few-step generation, UCGM-{T, S} achieves SOTA sample quality with minimal NFEs, surpassing existing specialized consistency models and GANs:

(a) $512 \times 512$: Using a DC-AE autoencoder, our model achieves an FID of 1.75 with 2 NFEs and 675M parameters. This outperforms sCD-XXL, a leading consistency distillation model, which reports 1.88 FID with 2 NFEs and 1.5B parameters.

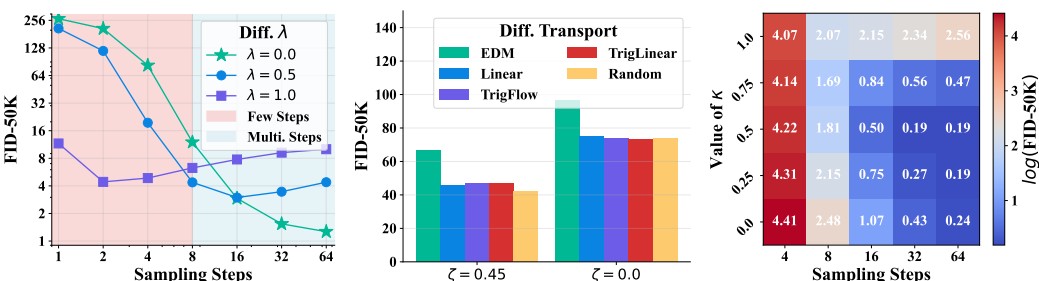

(a) **Various $\lambda$ and sampling steps.** (b) **Different $\zeta$ and transport types.** (c) **Various $\kappa$ and sampling steps.**

Figure 2: **Ablation studies of UCGM on ImageNet-1K** $256 \times 256$. These studies evaluate key factors of the proposed UCGM. Ablations presented in (a) and (c) utilize XL/1 models with the VA-VAE autoencoder. For the results shown in (b), B/2 models with the SD-VAE autoencoder are used to facilitate more efficient training.

(b) $256 \times 256$: Using a VA-VAE autoencoder, our model achieves an FID of $1.42$ with 2 NFEs. This is a notable improvement over IMM-XL/2, which obtains $1.99$ FID with $8 \times 2 = 16$ NFEs, demonstrating higher sample quality while requiring $8 \times$ fewer sampling steps.

In summary, *these results demonstrate the capability of* UCGM-*{T, S} to deliver high-quality generation with minimal sampling cost, which is advantageous for practical applications.*

**Multi-step regime ($\lambda = 0$).** Even when models are trained for multi-step generation, it nonetheless demonstrates competitive performance even when utilizing a moderate number of sampling steps.

(a) $512 \times 512$: Using a DC-AE autoencoder, our model obtains an FID of $1.81$ with 16 NFEs and 675M parameters. This result is competitive with or superior to existing methods such as VAR-$d$30-s, which reports $2.63$ FID with $10 \times 2 = 20$ NFEs and 2.3B parameters.

(b) $256 \times 256$: Using an E2E-VAE autoencoder, our model achieves an FID of $1.40$ with 16 NFEs. This surpasses IMM-XL/2, which obtains $1.99$ FID with $8 \times 2 = 16$ NFEs, demonstrating improved quality at the same sampling cost.

In summary, *our UCGM-{T, S} framework demonstrates versatility and high performance across both few-step ($\lambda = 1$) and multi-step ($\lambda = 0$) sampling regimes. As shown, it consistently achieves SOTA or competitive sample quality relative to existing methods, often requiring fewer sampling steps or parameters, which are important factors for efficient high-resolution image synthesis.*

### 4.4 ABLATION STUDY OVER THE KEY FACTORS OF UCGM

Unless otherwise specified, experiments in this section are conducted with $\kappa = 0.0$ and $\lambda = 0.0$.

**Effect of $\lambda$ in UCGM-T.** Fig. 2a demonstrates that varying $\lambda$ influences the range of effective sampling steps for trained models. For instance, with $\lambda = 1$ [5], optimal performance is attained at 2 sampling steps. In contrast, with $\lambda = 0.5$, optimal performance is observed at 16 steps.

**Impact of $\zeta$ and transport type in UCGM.** The results in Fig. 2b demonstrates that UCGM-{T, S} is applicable with various transport types, albeit with some performance variation. Investigating these performance differences constitutes future work. The results also illustrate that the enhanced training objective (achieved with $\zeta = 0.45$ compared to $\zeta = 0.0$, per Sec. 3) consistently improves performance across all tested transport types, underscoring the efficacy of this technique.

**Setting different $\kappa$ in UCGM-S.** Experimental results, depicted in Fig. 2c, illustrate the impact of $\kappa$ on the trade-off between sampling steps and generation quality: (a) High $\kappa$ values (e.g., $1.0$ and $0.75$) prove beneficial for extreme few-step sampling scenarios (e.g., 4 steps); (b) Moreover, mid-range $\kappa$ values ($0.25$ to $0.5$) achieve superior performance with fewer steps compared to $\kappa = 0.0$.

## 5 ADDITIONAL EXPERIMENTS ON LARGE-SCALE MODELS AND DATASETS

### 5.1 COMPARISON WITH TEXT-TO-IMAGE MODELS

We evaluate the practical efficacy of UCGM on text-to-image synthesis, with comprehensive benchmarks detailed in Tab. 4. The training efficiency is notable: fine-tuning the SANA-0.6B and SANA-1.6B backbones (batch sizes 128 and 64, respectively) for $40,000$ steps required only 60 NVIDIA H800 GPU hours.

---

[5]For the purpose of a fair ablation study, additional stabilizing techniques were omitted for this $\lambda = 1$ case.

Table 4: **System-level comparison of UCGM against few-step text-to-image baselines.** Throughput (batch size 10) and latency (batch size 1) are evaluated on a single NVIDIA A100 GPU (BF16).

| Method | NFE ↓ | Throughput ↑ (samples/s) | Latency (s) ↓ | #Params | GenEval ↑ | DPG-Bench ↑ |
|---|---|---|---|---|---|---|
| SDXL-DMD2 (Yin et al., 2024a) | 2 | 2.89 | 0.40 | 0.9B | 0.58 | - |
| FLUX-Schnell (Labs, 2024) | 2 | 0.92 | 1.15 | 12.0B | 0.71 | - |
| SANA-Sprint-0.6B (Chen et al., 2025c) | 2 | 6.46 | 0.25 | 0.6B | 0.76 | 81.5 |
| SANA-Sprint-1.6B (Chen et al., 2025c) | 2 | 5.68 | 0.24 | 1.6B | 0.77 | 82.1 |
| SDXL-LCM (Luo et al., 2023) | 2 | 2.89 | 0.40 | 0.9B | 0.44 | - |
| PixArt-LCM (Chen et al., 2024b) | 2 | 3.52 | 0.31 | 0.6B | 0.42 | - |
| PCM (Wang et al., 2024) | 2 | 2.62 | 0.56 | 0.9B | 0.55 | - |
| SD3.5-Turbo (Esser et al., 2024) | 2 | 1.61 | 0.68 | 8.0B | 0.53 | - |
| PixArt-DMD (Chen et al., 2024a) | 1 | 4.26 | 0.25 | 0.6B | 0.45 | - |
| SDXL-DMD2 (Yin et al., 2024a) | 1 | 3.36 | 0.32 | 0.9B | 0.59 | - |
| FLUX-Schnell (Labs, 2024) | 1 | 1.58 | 0.68 | 12.0B | 0.69 | - |
| SANA-Sprint-0.6B (Chen et al., 2025c) | 1 | 7.22 | 0.21 | 0.6B | 0.72 | 78.6 |
| SANA-Sprint-1.6B (Chen et al., 2025c) | 1 | 6.71 | 0.21 | 1.6B | 0.76 | 80.1 |
| SDXL-LCM (Luo et al., 2023) | 1 | 3.36 | 0.32 | 0.9B | 0.28 | - |
| PixArt-LCM (Chen et al., 2024b) | 1 | 4.26 | 0.25 | 0.6B | 0.41 | - |
| PCM (Wang et al., 2024) | 1 | 3.16 | 0.40 | 0.9B | 0.42 | - |
| SD3.5-Turbo (Esser et al., 2024) | 1 | 2.48 | 0.45 | 8.0B | 0.51 | - |
| **UCGM-0.6B (Ours, $\lambda = 1$)** | 2 | 6.50 | 0.26 | 0.6B | **0.84** | **81.0** |
| **UCGM-1.6B (Ours, $\lambda = 1$)** | 2 | 5.71 | 0.25 | 1.6B | **0.82** | **82.4** |
| **UCGM-0.6B (Ours, $\lambda = 1$)** | 1 | 7.30 | 0.23 | 0.6B | **0.79** | **78.2** |
| **UCGM-1.6B (Ours, $\lambda = 1$)** | 1 | 6.75 | 0.22 | 1.6B | **0.80** | **80.7** |

The tuning settings follow those outlined in App. D.3.

Empirical results establish a new Pareto frontier for generation quality and speed. At 2 NFE, UCGM sets a high performance standard, with our 0.6B model achieving a GenEval score of **0.84**. This significantly outperforms the 12B-parameter FLUX-Schnell (0.71) and SANA-Sprint-1.6B (0.77), demonstrating that massive parameter counts are not a prerequisite for high fidelity. This advantage persists in the challenging single-step (1 NFE) regime, where UCGM-0.6B attains a score of **0.79**—surpassing SANA-Sprint-1.6B (0.76)—while delivering the highest throughput in the benchmark at 7.30 samples/s.

Beyond quantitative metrics, UCGM proves that objective formulation outweighs system complexity. Unlike baselines like SANA-Sprint that rely on composite adversarial losses, we achieve superior fidelity using solely the singular objective in (6).

## 5.2 Extension to Unified Multimodal Models

We push the scalability limits of UCGM by applying it to the **20B-parameter Multi-Modal Diffusion Transformer (MM-DiT)**. The experimental outcomes, detailed in App. D.3, highlight our engineering success and underscore two distinct advantages:

(a) **Successful training of high-capacity models:** We successfully scaled UCGM to the 20B regime using **12,976 H800 GPU hours**. Relying exclusively on public datasets (e.g., LAION-5B), our model achieves performance parity with state-of-the-art backbones on **GenEval, DPG-Bench and WISE** benchmarks, demonstrating that it handles large-scale UMMs.

(b) **Robust distillation in few-step regimes:** Distilling 20B-parameter models presents severe stability hurdles. Our benchmarks reveal that Consistency Models (Song et al., 2023) suffer from **catastrophic collapse**, and MeanFlow (Geng et al., 2025) encounters **Out-of-Memory (OOM)** errors. In contrast, UCGM successfully distills the 20B UMM, maintaining superior stability and quality where prior arts fail.

## 6 Conclusion

We present UCGM, a unified and efficient framework for training and sampling both few-step and multi-step continuous generative models. Within this framework, we derive a surrogate unified objective that theoretically decomposes the few-step objective into the multi-step objective plus a self-alignment term. Building on this foundation, we introduce UCGM-T, which seamlessly bridges few-step (e.g., consistency models) and multi-step (e.g., diffusion, flow matching) generative paradigms, supporting diverse model architectures, latent autoencoders, and noise schedules. We further propose UCGM-S, a unified sampler compatible with our trained models, which can also accelerate and enhance pre-trained models from existing paradigms.

## 7 ETHICS STATEMENT

This research adheres to the *ICLR Code of Ethics* and is committed to the principles of responsible and transparent scientific inquiry. The study involves no human participants, personal or sensitive data, or any activities requiring approval from an institutional ethics review board. All datasets used are publicly accessible under appropriate licenses, with proper attribution given to their original sources. To promote openness and reproducibility, we provide our implementation code and experimental settings for verification and further development by the research community. We also declare that no conflicts of interest or external funding have influenced the design, execution, or presentation of this work.

## 8 REPRODUCIBILITY STATEMENT

Comprehensive details regarding the datasets, model architectures, optimization settings, and training procedures are provided in Sec. 4.1 of the main paper and in App. D. These materials are designed to facilitate the reliable and transparent reproduction of our results. Additionally, our source code will be made publicly available upon acceptance of the paper and **is included in the supplementary material**.

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

CONTENTS

## A    USE OF LLMs

During the preparation of this paper, we used OpenAI's ChatGPT to assist with language refinement, including grammar correction, style polishing, and improving readability. The model was not used for generating ideas, experimental design, analysis, or writing substantive technical content. All scientific contributions, including theoretical derivations, method development, and experimental results, are entirely the work of the authors.

## B    BROADER IMPACTS

This paper proposes a unified implementation and theoretical framework for recent popular continuous generative models, such as diffusion models, flow matching models, and consistency models. This work should provide positive impacts for the generative modeling community.

## C    LIMITATIONS

**Integration of training acceleration techniques.**    This work does not explore the integration of advanced training acceleration methods for diffusion models, such as REPA (Yu et al., 2024).

**Exploration of downstream applications.**    The current study focuses on establishing the foundational framework. Comprehensive exploration of its application to complex downstream generative tasks, including text-to-image and text-to-video generation, is reserved for future research.

**Stabilization of few-step objectives.**    While we theoretically decompose the few-step objective into the multi-step objective and a self-alignment term, and identify the self-alignment term as the source of potential instability, methods for stabilizing the few-step objective are not investigated in this work. We leave this as an important direction for future research.

## D    DETAILED EXPERIMENT

### D.1    DETAILED EXPERIMENTAL SETTING

#### D.1.1    DETAILED DATASETS

**Image datasets.**    We conduct experiments on two datasets: CIFAR-10 (Krizhevsky et al., 2009a), ImageNet-1K (Deng et al., 2009):

(a) CIFAR-10 is a widely used benchmark dataset for image classification and generation tasks. It consists of $60,000$ color images, each with a resolution of $32 \times 32$ pixels, categorized into $10$ distinct classes. The dataset is divided into $50,000$ training images and $10,000$ test images.

(b) ImageNet-1K is a large-scale dataset containing over 1.2 million high-resolution images across $1,000$ categories.

**Latent space datasets.**    However, directly training diffusion transformers in the pixel space is computationally expensive and inefficient. Therefore, following previous studies (Yu et al., 2024; Ma et al., 2024), we train our diffusion transformers in latent space instead. Tab. 5 presents a comparative analysis of various Variational Autoencoder (VAE) architectures. SD-VAE is characterized by a higher spatial resolution in its latent representation (e.g., $H/8 \times W/8$) combined with a lower channel capacity (4 channels). Conversely, alternative models such as VA-VAE, E2E-VAE, and DC-AE achieve more significant spatial compression (e.g., $H/16 \times W/16$ or $H/32 \times W/32$) at the expense of an increased channel depth (typically 32 channels).

A key consideration is that the computational cost of a diffusion transformer subsequently processing these latent representations is primarily dictated by their spatial dimensions, rather than their channel capacity (Chen et al., 2024c). Specifically, if the latent map is processed by a transformer by dividing it into non-overlapping patches, the cost is proportional to the number of these patches. This quantity is given by (H/Compression Ratio/Patch Size) $\times$ (W/Compression Ratio/Patch Size). Here, H and W are the input image dimensions, Compression Ratio refers to the spatial compression factor of the VAE (e.g., 8, 16, 32 as detailed in Tab. 5), and Patch Size denotes the side length of the patches processed by the transformer.

#### D.1.2    DETAILED NEURAL ARCHITECTURE

Diffusion Transformers (DiTs) represent a paradigm shift in generative modeling by replacing the traditional U-Net backbone with a Transformer-based architecture. Proposed by *Scalable Diffusion Models with Transformers* (Peebles & Xie, 2023), DiTs exhibit superior scalability and performance in image generation tasks. In this paper, we utilize three key variants—DiT-B (130M parameters), DiT-L (458M parameters), and DiT-XL (675M parameters).

Table 5: **Comparison of different VAE architectures in terms of latent space dimensions and channel capacity.** The table contrasts four variational autoencoder variants (SD-VAE, VA-VAE, E2E-VAE, and DC-AE) by their spatial compression ratios (latent size) and feature channel dimensions. Here, H and W denote input image height and width (e.g., $256 \times 256$ or $512 \times 512$), respectively.

|  | SD-VAE (both `ema` and `mse` versions) (Rombach et al., 2022) | VA-VAE (Yao et al., 2025) | E2E-VAE (Leng et al., 2025) | DC-AE (*f32c32*) (Chen et al., 2024c) |
|---|---|---|---|---|
| Latent Size | $(H/8) \times (W/8)$ | $(H/16) \times (W/16)$ | $(H/16) \times (W/16)$ | $(H/32) \times (W/32)$ |
| Channels | 4 | 32 | 32 | 32 |

To improve training stability, informed by recent studies (Yao et al., 2025; Wang et al., 2025), we incorporate several architectural modifications into the DiT model: (a) SwiGLU feed-forward networks (FFN) (Shazeer, 2020); (b) RMSNorm (Zhang & Sennrich, 2019) without learnable affine parameters; (c) Rotary Positional Embeddings (RoPE) (Su et al., 2024); and (d) parameter-free RMSNorm applied to Key (K) and Query (Q) projections in self-attention layers (Vaswani et al., 2017).

### D.1.3 DETAILED IMPLEMENTATION DETAILS

Experiments were conducted on a cluster equipped with 8 H800 GPUs, each with 80 GB of VRAM.

**Hyperparameter configuration.** Detailed hyperparameter configurations are provided in Tab. 6 to ensure reproducibility. The design of time schedules for sampling processes varies in complexity. For few-step models, typically employing 1 or 2 sampling steps, manual schedule design is straightforward. However, the time schedule $\mathcal{T}$ utilized by our UCGM-S often comprises a large number of time points, particularly for a large number of sampling steps $N$. Manual design of such dense schedules is challenging and can limit the achievable performance of our UCGM-{T, S}, as prior work (Yao et al., 2025; Wang et al., 2025) has established that carefully designed schedules significantly enhance multi-step models, including flow-matching variants. To address this, we propose transforming each time point $t \in \mathcal{T}$ using a generalized Kumaraswamy transformation: $f_{\text{Kuma}}(t; a, b, c) = (1 - (1 - t^a)^b)^c$. This choice is motivated by the common practice in prior studies of applying non-linear transformations to individual time points to construct effective schedules. A specific instance of such a transformation is the `timeshift` function $f_{\text{shift}}(t; s) = \frac{st}{1+(s-1)t}$, where $s > 0$ (Yao et al., 2025). We find that the Kumaraswamy transformation, by appropriate selection of parameters $a, b, c$, can effectively approximate $f_{\text{shift}}$ and other widely-used functions (cf., App. F.2.2), including the identity function $f(t) = t$ (Yu et al., 2024; Leng et al., 2025). Empirical evaluations suggest that the parameter configuration $(a, b, c) = (1.17, 0.8, 1.1)$ yields robust performance across diverse scenarios, corresponding to the "Auto" setting in Tab. 6.

**Detailed implementation techniques of enhancing target score function.** We enhance the target score function for conditional diffusion models by modifying the standard score $\nabla_{\mathbf{x}_t} \log p_t(\mathbf{x}_t|\mathbf{c})$ (Song et al., 2020b) to an enhanced version derived from the density $p_t(\mathbf{x}_t|\mathbf{c}) \left( \frac{p_{t,\boldsymbol{\theta}}(\mathbf{x}_t|\mathbf{c})}{p_{t,\boldsymbol{\theta}}(\mathbf{x}_t)} \right)^\zeta$. This corresponds to a target score of $\nabla_{\mathbf{x}_t} \log p_t(\mathbf{x}_t|\mathbf{c}) + \zeta \left( \nabla_{\mathbf{x}_t} \log p_{t,\boldsymbol{\theta}}(\mathbf{x}_t|\mathbf{c}) - \nabla_{\mathbf{x}_t} \log p_{t,\boldsymbol{\theta}}(\mathbf{x}_t) \right)$. The objective is to guide the learning process towards distributions that yield higher quality conditional samples.

Accurate estimation of the model probabilities $p_{t,\boldsymbol{\theta}}$ is crucial for the effectiveness of this enhancement. We find that using parameters from an Exponential Moving Average (EMA) of the model during training improves the stability and quality of these estimates, resulting better $\mathbf{x}^\star$ and $\mathbf{z}^\star$ in Alg. 1.

When training few-step models, direct computation of the enhanced target score gradient typically requires evaluating the model with and without conditioning (for the $p_{t,\boldsymbol{\theta}}$ terms), incurring additional computational cost. To address this, we propose an efficient approximation that leverages a well-pre-trained multi-step model, denoted by parameters $\boldsymbol{\theta}^\star$. Instead of computing the score gradient explicitly, the updates for the variables $\mathbf{x}^\star$ and $\mathbf{z}^\star$ (as used in Alg. 1) are calculated based on features or outputs derived from a single forward pass of the pre-trained model $\boldsymbol{\theta}^\star$.

Specifically, we compute $\boldsymbol{F}_t = \boldsymbol{F}_{\boldsymbol{\theta}^\star}(\mathbf{x}_t, t)$, representing features extracted by the pre-trained model $\boldsymbol{\theta}^\star$ at time $t$ given input $\mathbf{x}_t$. The enhanced updates $\mathbf{x}^\star$ and $\mathbf{z}^\star$ are then computed as follows:

(a) For $t \in [0, s]$, the updates are: $\mathbf{x}^\star \leftarrow \mathbf{x} + \zeta \cdot (\boldsymbol{f}^{\mathbf{x}}(\boldsymbol{F}_t, \mathbf{x}_t, t) - \mathbf{x})$, $\mathbf{z}^\star \leftarrow \mathbf{z} + \zeta \cdot (\boldsymbol{f}^{\mathbf{z}}(\boldsymbol{F}_t, \mathbf{x}_t, t) - \mathbf{z})$.

(b) For $t \in (s, 1]$, the updates are: $\mathbf{x}^\star \leftarrow \mathbf{x} + \frac{1}{2}(\boldsymbol{f}^{\mathbf{x}}(\boldsymbol{F}_t, \mathbf{x}_t, t) - \mathbf{x})$ and $\mathbf{z}^\star \leftarrow \mathbf{z} + \frac{1}{2}(\boldsymbol{f}^{\mathbf{z}}(\boldsymbol{F}_t, \mathbf{x}_t, t) - \mathbf{z})$.

Table 6: **Hyperparameter configurations for UCGM-{T, S} training and sampling on ImageNet-1K.** We maintain a consistent batch size of 1024 across all experiments. Training durations (epoch counts) are provided in other tables throughout the paper. The table specifies optimizer choices, learning rates, and key parameters for both UCGM-T and UCGM-S variants across different model architectures and datasets.

| Task | | | Optimizer | | | UCGM-T | | | | UCGM-S | | | |
|---|---|---|---|---|---|---|---|---|---|---|---|---|---|
| Resolution | VAE/AE | Model | Type | lr | $(\beta_1,\beta_2)$ | Transport | $(\theta_1,\theta_2)$ | $\lambda$ | $\zeta$ | $\rho$ | $\kappa$ | $\mathcal{T}$ | $\nu$ |
| *Multi-step model training and sampling* | | | | | | | | | | | | | |
| | E2E-VAE | XL/1 | AdamW | 0.0002 | (0.9,0.95) | Linear | (1.0,1.0) | 0 | 0.67 | 0 | 0.5 | Auto | 1 |
| 256 | SD-VAE | XL/2 | AdamW | 0.0002 | (0.9,0.95) | Linear | (2.4,2.4) | 0 | 0.44 | 0 | 0.21 | Auto | 1 |
| | VA-VAE | XL/1 | AdamW | 0.0002 | (0.9,0.95) | Linear | (1.0,1.0) | 0 | 0.47 | 0 | 0.5 | Auto | 1 |
| 512 | DC-AE | XL/1 | AdamW | 0.0002 | (0.9,0.95) | Linear | (1.0,1.0) | 0 | 0.57 | 0 | 0.46 | Auto | 1 |
| | SD-VAE | XL/4 | AdamW | 0.0002 | (0.9,0.95) | Linear | (2.4,2.4) | 0 | 0.60 | 0 | 0.4 | Auto | 1 |
| *Few-step model training and sampling* | | | | | | | | | | | | | |
| | E2E-VAE | XL/1 | RAdam | 0.0001 | (0.9,0.999) | Linear | (0.8,1.0) | 1 | 1.3 | 1 | 0 | {1,0.5} | 1 |
| 256 | SD-VAE | XL/2 | RAdam | 0.0001 | (0.9,0.999) | Linear | (0.8,1.0) | 1 | 2.0 | 1 | 0 | {1,0.3} | 1 |
| | VA-VAE | XL/2 | RAdam | 0.0001 | (0.9,0.999) | Linear | (0.8,1.0) | 1 | 2.0 | 1 | 0 | {1,0.3} | 1 |
| 512 | DC-AE | XL/1 | RAdam | 0.0001 | (0.9,0.999) | Linear | (0.8,1.0) | 1 | 1.5 | 1 | 0 | {1,0.6} | 1 |
| | SD-VAE | XL/4 | RAdam | 0.0001 | (0.9,0.999) | Linear | (0.8,1.0) | 1 | 1.5 | 1 | 0 | {1,0.5} | 1 |

Table 7: **Comparison of different transport types employed during the sampling and training phases of our UCGM-{T, S}.** "TrigLinear" and "Random" are introduced herein specifically for ablation studies. "TrigLinear" is constructed by combining the transport coefficients of "Linear" and "TrigFlow". "Random" represents a randomly designed transport type used to demonstrate the generality of our UCGM. Other transport types are adapted from existing methods and transformed into the transport coefficient representation used by UCGM.

| | Linear | ReLinear | TrigFlow | EDM ($\sigma(t) = e^{4\cdot(2.68t-1.59)}$) | TrigLinear | Random |
|---|---|---|---|---|---|---|
| $\alpha(t)$ | $t$ | $1-t$ | $\sin(t \cdot \frac{\pi}{2})$ | $\sigma(t)/\sqrt{\sigma^2(t)+0.25}$ | $\sin(t \cdot \frac{\pi}{2})$ | $\sin(t \cdot \frac{\pi}{2})$ |
| $\gamma(t)$ | $1-t$ | $t$ | $\cos(t \cdot \frac{\pi}{2})$ | $1/\sqrt{\sigma^2(t)+0.25}$ | $\cos(t \cdot \frac{\pi}{2})$ | $1-t$ |
| $\hat{\alpha}(t)$ | $1$ | $-1$ | $\cos(t \cdot \frac{\pi}{2})$ | $-0.5/\sqrt{\sigma^2(t)+0.25}$ | $1$ | $1$ |
| $\hat{\gamma}(t)$ | $-1$ | $1$ | $-\sin(t \cdot \frac{\pi}{2})$ | $2\sigma(t)/\sqrt{\sigma^2(t)+0.25}$ | $-1$ | $-1-e^{-5t}$ |
| e.g., | (Ma et al., 2024) | (Yao et al., 2025) | (Chen et al., 2025c) | (Karras et al., 2022) | N/A | N/A |

We consistently set the time threshold $s = 0.75$. This approach allows us to incorporate the guidance from the enhanced target signal with the computational cost equivalent to a single forward evaluation of the pre-trained model $\theta^\star$ per step. The enhancement ratio $\zeta$ is constrained to $[0, \infty)$ in this case.

**Baselines.** We compare our approach against several SOTA continuous and discrete generative models. We broadly categorize these baselines by their generation process:

(a) Multi-step models. These methods typically synthesize data through a sequence of steps. We include various diffusion models, encompassing classical formulations like DDPM and score-based models (Song et al., 2020a; Ho et al., 2020), and advanced variants focusing on improved sampling or performance in latent spaces (Dhariwal & Nichol, 2021; Karras et al., 2022; Peebles & Xie, 2023; Zheng et al., 2023; Bao et al., 2023). We also consider flow-matching models (Lipman et al., 2022), which leverage continuous normalizing flows and demonstrate favorable training properties, along with subsequent scaling efforts (Ma et al., 2024; Yu et al., 2024; Yao et al., 2025). Additionally, we also include autoregressive models (Li et al., 2024; Tian et al., 2024; Yu et al., 2023) as the baselines, which generate data sequentially, often in discrete domains.

(b) Few-step models. These models are designed for efficient, often single-step or few-step, generation. This category includes generative adversarial networks (Goodfellow et al., 2020), which achieve efficient one-step synthesis through adversarial training, and their large-scale variants (Brock et al., 2018; Sauer et al., 2022; Kang et al., 2023). We also evaluate consistency models (Song et al., 2023), proposed for high-quality generation adaptable to few sampling steps, and subsequent techniques aimed at improving their stability and scalability (Song & Dhariwal, 2023; Lu & Song, 2024; Zhou et al., 2025).

Crucially, we demonstrate the compatibility of UCGM-S with models pre-trained using these methods. We show how these models can be represented within the UCGM framework by defining the functions $\alpha(\cdot)$, $\gamma(\cdot)$, $\hat{\alpha}(\cdot)$, and $\hat{\gamma}(\cdot)$. Detailed parameterizations are provided in Tab. 7, with guidance for their specification presented in App. F.2.4.

## D.2 EXPERIMENTAL RESULTS ON SMALL DATASETS

Since most existing few-step generation methods (Song et al., 2023; Geng et al., 2024) are limited to training models on low-resolution, small-scale datasets like CIFAR-10 (Krizhevsky et al., 2009a), we conduct our comparative experiments on CIFAR-10 to ensure fair comparison. To demonstrate the versatility of our UCGM, we employ both the "EDM" transport (see Tab. 7 for definition) and the standard 56M-parameter UNet architecture, following established practices in prior work Song et al. (2023); Geng et al. (2024).

Table 8: **System-level quality comparison for few-step generation task on unconditional CIFAR-10** ($32 \times 32$).

| Metric | PD (Salimans & Ho, 2022) | 2-RF (Liu et al., 2022) | DMD (Yin et al., 2024b) | CD (Song et al., 2023) | sCD (Lu & Song, 2024) |
|---|---|---|---|---|---|
| FID ($\downarrow$) | 4.51 | 4.85 | 3.77 | 2.93 | 2.52 |
| NFE ($\downarrow$) | 2 | 1 | 1 | 2 | 2 |

| Metric | iCT (Song & Dhariwal, 2023) | ECT (Geng et al., 2024) | sCT (Lu & Song, 2024) | IMM (Zhou et al., 2025) | **UCGM** |
|---|---|---|---|---|---|
| FID ($\downarrow$) | 2.83 / 2.46 | 3.60 / 2.11 | 2.97 / 2.06 | 3.20 / 1.98 | 2.82 / 2.17 |
| NFE ($\downarrow$) | 1 / 2 | 1 / 2 | 1 / 2 | 1 / 2 | 1 / 2 |

As shown in Tab. 8, our UCGM achieves SOTA performance with just 1 NFE (Neural Function Evaluation) while maintaining competitive results for 2 NFEs. These results underscore UCGM's robust compatibility across diverse datasets, network architectures, and transport types.

## D.3 EXPERIMENTAL RESULTS ON LARGE-SCALE UNIFIED MULTIMODAL MODELS

To evaluate the scalability and efficacy of UCGM on Unified Multimodal Models (UMMs), we employ the widely adopted Multi-Modal Diffusion Transformer (MM-DiT) architecture (Esser et al., 2024; Wu et al., 2025a) as our primary backbone. Tab. 9 summarizes the performance across three benchmarks. Crucially, UCGM-S demonstrates superior efficiency, significantly outperforming the standard Euler sampler while maintaining identical NFE budgets.

For evaluation, we employ GenEval (Ghosh et al., 2023) and DPG-Bench (Hu et al., 2024) for text-to-image generation, and WISE (Niu et al., 2025) for world knowledge assessment.

**Multi-step Regime ($\lambda = 0$).** In the multi-step setting, our trained UCGM-20B achieves performance parity with state-of-the-art generative models. **Remarkably, our model achieves these results relying exclusively on publicly available datasets**, whereas many SOTA baselines depend on large-scale proprietary data. Our training corpus includes Megalith-10M (Matsubara & Team, 2024), BLIP3o-Pretrain (Chen et al., 2025a), LAION-5B (LAION, 2024), Conceptual 12M (Changpinyo et al., 2021), and text-to-image-2M (He & contributors, 2024) for pre-training, followed by fine-tuning on high-quality instruction-following datasets (BLIP3-o-60K (Chen et al., 2025a), Echo-4o-Image (Ye et al., 2025), and ShareGPT-4o-Image (Chen et al., 2025d)). We adhere to a rigorous training protocol: pre-training for 60k steps (batch size 8,192) and fine-tuning for 3k steps (batch size 1,024) on NVIDIA H800 GPUs (12,976 GPU hours in total).

The training details for UCGM-20B are as follows: we utilize the "Linear" transport type (as defined in Tab. 7), with learning rates of $1 \times 10^{-4}$ for pre-training and $1 \times 10^{-4}$ for fine-tuning. The model is trained using the AdamW optimizer with a cosine learning rate schedule.

**Few-step Regime ($\lambda = 1$).** Scaling distillation to large UMMs presents significant challenges for existing methods. As shown in Tab. 9, standard few-step techniques such as Consistency Models (CM) (Song et al., 2023) suffer from catastrophic model collapse, while MeanFlow (Geng et al., 2025) encounters prohibitive memory costs (OOM). Even when stabilized with our proposed techniques (denoted as CM* and MeanFlow*), these baselines fail to produce competitive results. In contrast, UCGM exhibits exceptional robustness, successfully distilling the 20B-parameter UMM into a few-step generator without compromising stability or requiring excessive memory overhead.

The training configuration for few-step UCGM-20B mirrors that of the multi-step variant, with the exception of a reduced learning rate of $1 \times 10^{-5}$.

**Dynamic $\lambda$ Strategy.** To bridge the gap between generation quality and inference latency, we propose a dynamic $\lambda$ training strategy. By conditioning the architecture on a scalar $\lambda \in [0, 1]$ sampled randomly during training, the model learns a continuous spectrum of generation behaviors. This design empowers users to navigate the trade-off between fidelity and speed at inference time. Empirical results confirm that this dynamic approach is highly effective: it not only matches the

Table 9: **System-level comparison of UCGM against SOTA unified multimodal models.** We report inference efficiency (NFE) and generation performance across three benchmarks. **Bold** and underline denote the best and second-best results, respectively. $^\dagger$ indicates evaluation using LLM-rewritten prompts on GenEval. CM$^*$ and MeanFlow$^*$ denote baselines re-implemented with our stabilizing techniques or memory-efficient approximations (finite difference) to enable training on large-scale UMMs. All experiments were conducted on NVIDIA H800 GPUs.

| Method | NFE ↓ | Image Generation | | |
|---|---|---|---|---|
| | | GenEval ↑ | DPG-Bench ↑ | WISE ↑ |
| **Multi-step models** | | | | |
| Show-o (Xie et al., 2024b) | 50×2 | 0.68 | 67.27 | 0.35 |
| Show-o2-7B (Xie et al., 2025) | 50×2 | 0.76 | 86.14 | 0.39 |
| OmniGen (Xiao et al., 2024) | 50×2 | 0.70 | 81.16 | - |
| OmniGen2 (Wu et al., 2025b) | 50×2 | 0.80 / 0.86$^\dagger$ | 83.57 | - |
| Janus-Pro (Chen et al., 2025e) | - | 0.80 | 84.19 | 0.35 |
| MetaQuery-XL (Pan et al., 2025) | 30×2 | 0.78 / 0.80$^\dagger$ | 81.10 | 0.55 |
| BLIP3-o-8B (Chen et al., 2025b) | 30×2 + 50×2 | 0.84 | 81.60 | **0.62** |
| UniWorld-V1 (Lin et al., 2025) | 28×2 | 0.80 / 0.84$^\dagger$ | - | 0.55 |
| OpenUni-L-512 (Wu et al., 2025c) | 20×2 | 0.85 | 81.54 | 0.52 |
| Bagel (Deng et al., 2025) | 50×2 | 0.82 / 0.88$^\dagger$ | | 0.52 |
| Qwen-Image-20B (Wu et al., 2025a) | 50×2 | **0.87** | **88.32** | **0.62** |
| **UCGM-20B (ours, $\lambda = 0$)** | 20×2 | **0.87** | 86.27 | 0.58 |
| Qwen-Image-20B + 20-step Euler Sampler | 20×2 | 0.85 | 82.51 | 0.53 |
| **Qwen-Image-20B + 20-step UCGM-S** | 20×2 | **0.87** | 88.28 | 0.61 |
| **Few-step models** | | | | |
| OpenUni-L-512⊕CM (Song et al., 2023) (model collapse) | 1 | 0.0 | - | - |
| Qwen-Image-20B⊕CM (Song et al., 2023) (model collapse) | 1 | 0.0 | - | - |
| Qwen-Image-20B⊕CM$^*$ | 8 | 0.51 | 72.17 | 0.24 |
| Qwen-Image-20B⊕CM$^*$ | 4 | 0.51 | 71.27 | 0.22 |
| Qwen-Image-20B⊕CM$^*$ | 2 | 0.44 | 66.39 | 0.19 |
| Qwen-Image-20B⊕CM$^*$ | 1 | 0.01 | 15.41 | 0.04 |
| Qwen-Image-20B⊕MeanFlow (Geng et al., 2025) (out of memory) | - | - | - | - |
| Qwen-Image-20B⊕MeanFlow$^*$ | 8 | 0.49 | 83.81 | 0.37 |
| Qwen-Image-20B⊕MeanFlow$^*$ | 4 | 0.44 | 83.28 | 0.34 |
| Qwen-Image-20B⊕MeanFlow$^*$ | 2 | 0.31 | 80.39 | 0.22 |
| Qwen-Image-20B⊕MeanFlow$^*$ | 1 | 0.05 | 62.19 | 0.10 |
| OpenUni-L-512⊕UCGM (ours, $\lambda = 1$) | 2 | 0.83 | 81.05 | 0.48 |
| OpenUni-L-512⊕UCGM (ours, $\lambda = 1$) | 1 | 0.76 | 73.27 | 0.42 |
| **Qwen-Image-20B⊕UCGM (ours, $\lambda = 1$)** | 8 | 0.60 | 84.68 | 0.43 |
| **Qwen-Image-20B⊕UCGM (ours, $\lambda = 1$)** | 4 | 0.62 | 84.23 | 0.41 |
| **Qwen-Image-20B⊕UCGM (ours, $\lambda = 1$)** | 2 | 0.55 | 67.54 | 0.23 |
| **Qwen-Image-20B⊕UCGM (ours, $\lambda = 1$)** | 1 | 0.32 | 56.43 | 0.22 |
| **Qwen-Image-20B⊕UCGM (ours, dynamic $\lambda$)** | 8 | **0.86** | **86.49** | **0.55** |
| **Qwen-Image-20B⊕UCGM (ours, dynamic $\lambda$)** | 4 | 0.85 | 83.92 | 0.52 |
| **Qwen-Image-20B⊕UCGM (ours, dynamic $\lambda$)** | 2 | 0.82 | 81.34 | 0.47 |
| **Qwen-Image-20B⊕UCGM (ours, dynamic $\lambda$)** | 1 | 0.47 | 57.39 | 0.28 |

flexibility of multi-step models but also secures superior performance across varying NFE budgets, highlighting the versatility of UCGM for unified multimodal generation.

We have included additional qualitative results in Fig. 3, Fig. 4, and Fig. 5 for our trained UCGM-20B.

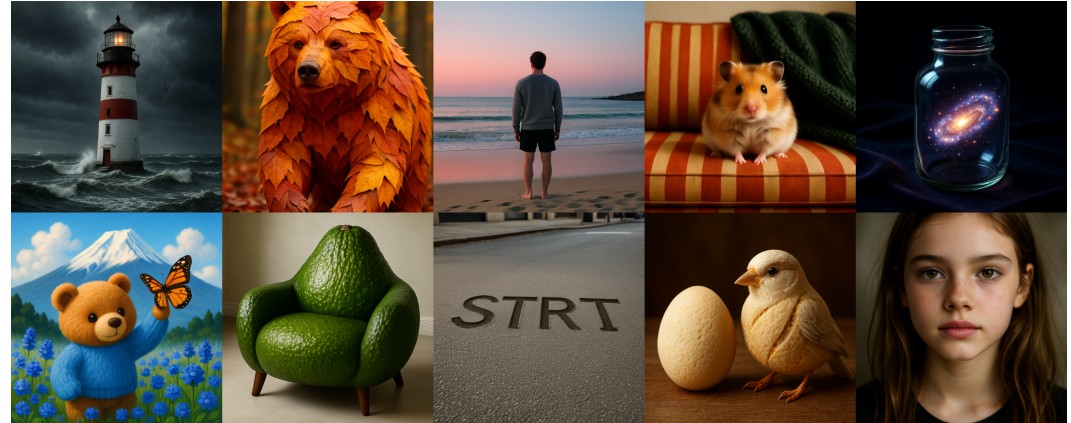

Figure 3: **Image generation results from UCGM-20B ($\kappa = 0.0$). Note that the model failed to generate the "START" image.**

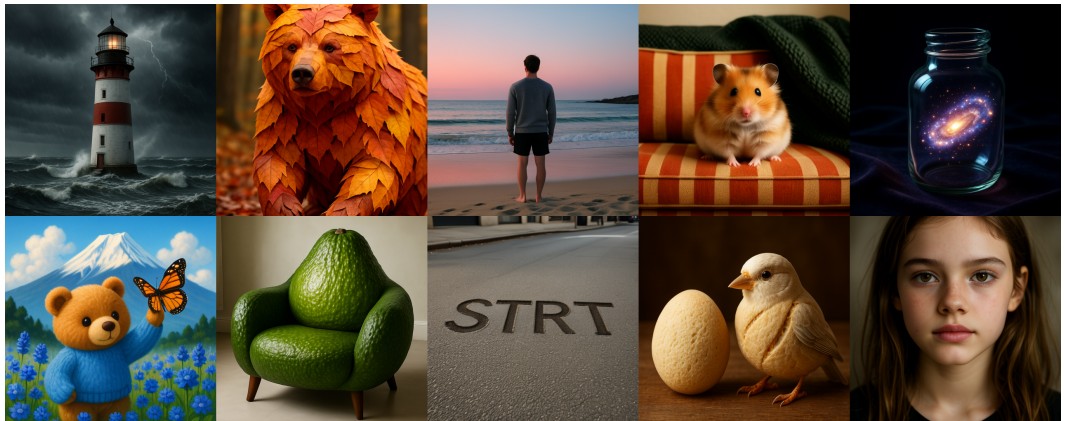

Figure 4: **Image generation results from UCGM-20B ($\kappa = 0.5$). The model failed to generate the "START" image.**

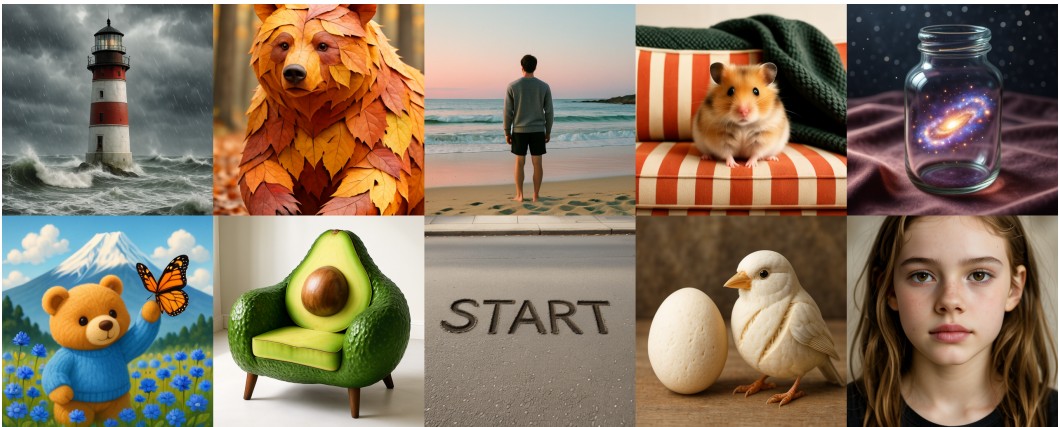

Figure 5: **Image generation results from UCGM-20B ($\kappa = 0.9$). The "START" image was generated successfully.**

## D.4 DETAILED COMPARISON WITH SOTA METHODS FOR MULTI-STEP GENERATION

Table 10: **System-level quality comparison for multi-step generation task on class-conditional ImageNet-1K.** Notation A⊕B denotes the result obtained by combining methods A and B. $\downarrow$/$\uparrow$ indicate a decrease/increase, respectively, in the metric compared to the baseline performance of the pre-trained models.

| METHOD | VAE/AE | Patch Size | Activation Size | NFE ($\downarrow$) | FID ($\downarrow$) | IS ($\uparrow$) | #Params | #Epochs |
|---|---|---|---|---|---|---|---|---|
| **512 × 512** | | | | | | | | |
| **Diffusion & flow-matching models** | | | | | | | | |
| ADM-G (Dhariwal & Nichol, 2021) | - | - | - | 250×2 | 7.72 | 172.71 | 559M | 388 |
| U-ViT-H/4 (Bao et al., 2023) | SD-VAE (Rombach et al., 2022) | 4 | 16×16 | 50×2 | 4.05 | 263.79 | 501M | 400 |
| DiT-XL/2 (Peebles & Xie, 2023) | SD-VAE (Rombach et al., 2022) | 2 | 32×32 | 250×2 | 3.04 | 240.82 | 675M | 600 |
| SiT-XL/2 (Ma et al., 2024) | SD-VAE (Rombach et al., 2022) | 2 | 32×32 | 250×2 | 2.62 | 252.21 | 675M | 600 |
| MaskDiT (Zheng et al., 2023) | SD-VAE (Rombach et al., 2022) | 2 | 32×32 | 79×2 | 2.50 | 256.27 | 736M | - |
| EDM2-S (Karras et al., 2024b) | SD-VAE (Rombach et al., 2022) | - | - | 63 | 2.56 | - | 280M | 1678 |
| EDM2-L (Karras et al., 2024b) | SD-VAE (Rombach et al., 2022) | - | - | 63 | 2.06 | - | 778M | 1476 |
| EDM2-XXL (Karras et al., 2024b) | SD-VAE (Rombach et al., 2022) | - | - | 63 | 1.91 | - | 1.5B | 734 |
| DiT-XL/1⊕(Chen et al., 2024c) | DC-AE (Chen et al., 2024c) | 1 | 16×16 | 250×2 | 2.41 | 263.56 | 675M | 400 |
| U-ViT-H/1⊕(Chen et al., 2024c) | DC-AE (Chen et al., 2024c) | 1 | 16×16 | 30×2 | 2.53 | 255.07 | 501M | 400 |
| REPA-XL/2 (Yu et al., 2024) | SD-VAE (Rombach et al., 2022) | 2 | 32×32 | 250×2 | 2.08 | 274.6 | 675M | 200 |
| DDT-XL/2 (Wang et al., 2025) | SD-VAE (Rombach et al., 2022) | 2 | 32×32 | 250×2 | **1.28** | **305.1** | 675M | - |
| **GANs & masked & autoregressive models** | | | | | | | | |
| VQGAN⊕(Esser et al., 2021) | - | - | - | 256 | 18.65 | - | 227M | - |
| MAGVIT-v2 (Yu et al., 2023) | - | - | - | 64×2 | 1.91 | 324.3 | 307M | 1080 |
| MAR-L (Li et al., 2024) | - | - | - | 256×2 | **1.73** | 279.9 | 479M | 800 |
| VAR-$d$36-s (Tian et al., 2024) | - | - | - | 10×2 | 2.63 | 303.2 | 2.3B | 350 |
| **Ours: UCGM-S sampling with models trained by prior works** | | | | | | | | |
| EDM2-S (Karras et al., 2024b) | SD-VAE (Rombach et al., 2022) | - | - | 40$^{\downarrow23}$ | 2.53$^{\downarrow0.03}$ | - | 280M | - |
| EDM2-L (Karras et al., 2024b) | SD-VAE (Rombach et al., 2022) | - | - | 50$^{\downarrow13}$ | 2.04$^{\downarrow0.02}$ | - | 778M | - |
| EDM2-XXL (Karras et al., 2024b) | SD-VAE (Rombach et al., 2022) | - | - | 40$^{\downarrow23}$ | 1.88$^{\downarrow0.03}$ | - | 1.5B | - |
| DDT-XL/2 (Wang et al., 2025) | SD-VAE (Rombach et al., 2022) | 2 | 32×32 | 200$^{\downarrow300}$ | **1.25**$^{\downarrow0.03}$ | - | 675M | - |
| **Ours: models trained and sampled using UCGM-{T,S} (setting $\lambda = 0$)** | | | | | | | | |
| Ours-XL/1 | DC-AE (Chen et al., 2024c) | 1 | 16×16 | 40 | **1.48** | - | 675M | 800 |
| Ours-XL/1 | DC-AE (Chen et al., 2024c) | 1 | 16×16 | 20 | 1.68 | - | 675M | 800 |
| Ours-XL/4 | SD-VAE (Rombach et al., 2022) | 4 | 16×16 | 40 | 1.67 | - | 675M | 320 |
| Ours-XL/4 | SD-VAE (Rombach et al., 2022) | 4 | 16×16 | 20 | 1.80 | - | 675M | 320 |
| **256 × 256** | | | | | | | | |
| **Diffusion & flow-matching models** | | | | | | | | |
| ADM-G (Dhariwal & Nichol, 2021) | - | - | - | 250×2 | 4.59 | 186.70 | 559M | 396 |
| U-ViT-H/2 (Bao et al., 2023) | SD-VAE (Rombach et al., 2022) | 2 | 16 × 16 | 50×2 | 2.29 | 263.88 | 501M | 400 |
| DiT-XL/2 (Peebles & Xie, 2023) | SD-VAE (Rombach et al., 2022) | 2 | 16 × 16 | 250×2 | 2.27 | 278.24 | 675M | 1400 |
| SiT-XL/2 (Ma et al., 2024) | SD-VAE (Rombach et al., 2022) | 2 | 16 × 16 | 250×2 | 2.06 | 277.50 | 675M | 1400 |
| MDT (Gao et al., 2023) | SD-VAE (Rombach et al., 2022) | 2 | 16 × 16 | 250×2 | 1.79 | 283.01 | 675M | 1300 |
| REPA-XL/2 (Yu et al., 2024) | SD-VAE (Rombach et al., 2022) | 2 | 16 × 16 | 250×2 | 1.96 | 264.0 | 675M | 200 |
| REPA-XL/2 (Yu et al., 2024) | SD-VAE (Rombach et al., 2022) | 2 | 16 × 16 | 250×2 | 1.42 | 305.7 | 675M | 800 |
| Light.DiT (Yao et al., 2025) | VA-VAE (Yao et al., 2025) | 1 | 16 × 16 | 250×2 | 2.11 | - | 675M | 64 |
| Light.DiT (Yao et al., 2025) | VA-VAE (Yao et al., 2025) | 1 | 16 × 16 | 250×2 | 1.35 | - | 675M | 800 |
| DDT-XL/2 (Wang et al., 2025) | SD-VAE (Rombach et al., 2022) | 2 | 16 × 16 | 250×2 | 1.31 | 308.1 | 675M | 256 |
| DDT-XL/2 (Wang et al., 2025) | SD-VAE (Rombach et al., 2022) | 2 | 16 × 16 | 250×2 | 1.26 | 310.6 | 675M | 400 |
| REPA-E-XL (Leng et al., 2025) | E2E-VAE (Leng et al., 2025) | 1 | 16 × 16 | 250×2 | **1.26** | 314.9 | 675M | 800 |
| **GANs & masked & autoregressive models** | | | | | | | | |
| VQGAN⊕(Sun et al., 2024) | - | - | - | - | 2.18 | - | 3.1B | 300 |
| MAR-L (Li et al., 2024) | - | - | - | 256×2 | 1.78 | 296.0 | 479M | 800 |
| MAR-H (Li et al., 2024) | - | - | - | 256×2 | **1.55** | 303.7 | 943M | 800 |
| VAR-$d$30-re (Tian et al., 2024) | - | - | - | 10×2 | 1.73 | 350.2 | 2.0B | 350 |
| **Ours: UCGM-S sampling with models trained by prior works** | | | | | | | | |
| DDT-XL/2 (Wang et al., 2025) | SD-VAE (Rombach et al., 2022) | 2 | 16 × 16 | 100$^{\downarrow400}$ | 1.27$^{\uparrow0.01}$ | - | 675M | - |
| Light.DiT (Yao et al., 2025) | VA-VAE (Yao et al., 2025) | 1 | 16 × 16 | 100$^{\downarrow400}$ | 1.21$^{\downarrow0.14}$ | - | 675M | - |
| REPA-E-XL (Leng et al., 2025) | E2E-VAE (Leng et al., 2025) | 1 | 16 × 16 | 80$^{\downarrow420}$ | **1.06**$^{\downarrow0.20}$ | - | 675M | - |
| REPA-E-XL (Leng et al., 2025) | E2E-VAE (Leng et al., 2025) | 1 | 16 × 16 | 20$^{\downarrow480}$ | 2.00$^{\uparrow0.74}$ | - | 675M | - |
| **Ours: models trained and sampled using UCGM-{T,S} (setting $\lambda = 0$)** | | | | | | | | |
| Ours-XL/2 | SD-VAE (Rombach et al., 2022) | 2 | 16 × 16 | 60 | 1.41 | - | 675M | 400 |
| Ours-XL/1 | VA-VAE (Yao et al., 2025) | 1 | 16 × 16 | 60 | 1.21 | - | 675M | 400 |
| Ours-XL/1 | E2E-VAE (Leng et al., 2025) | 1 | 16 × 16 | 40 | **1.21** | - | 675M | 800 |
| Ours-XL/1 | E2E-VAE (Leng et al., 2025) | 1 | 16 × 16 | 20 | 1.30 | - | 675M | 800 |

## D.5 DETAILED COMPARISON WITH SOTA METHODS FOR FEW-STEP GENERATION

Table 11: **System-level quality comparison for few-step generation task on class-conditional ImageNet-1K** ($512 \times 512$)**.**

| METHOD | VAE/AE | Patch Size | Activation Size | NFE ($\downarrow$) | FID ($\downarrow$) | IS | #Params | #Epochs |
|---|---|---|---|---|---|---|---|---|
| | | | $512 \times 512$ | | | | | |
| | | Consistency training & distillation | | | | | | |
| sCT-M (Lu & Song, 2024) | - | - | - | 1 | 5.84 | - | 498M | 1837 |
| sCT-M (Lu & Song, 2024) | - | - | - | 2 | 5.53 | - | 498M | 1837 |
| sCT-L (Lu & Song, 2024) | - | - | - | 1 | 5.15 | - | 778M | 1274 |
| sCT-L (Lu & Song, 2024) | - | - | - | 2 | 4.65 | - | 778M | 1274 |
| sCT-XXL (Lu & Song, 2024) | - | - | - | 1 | 4.29 | - | 1.5B | 762 |
| sCT-XXL (Lu & Song, 2024) | - | - | - | 2 | 3.76 | - | 1.5B | 762 |
| sCD-M (Lu & Song, 2024) | - | - | - | 1 | 2.75 | - | 498M | 1997 |
| sCD-M (Lu & Song, 2024) | - | - | - | 2 | 2.26 | - | 498M | 1997 |
| sCD-L (Lu & Song, 2024) | - | - | - | 1 | 2.55 | - | 778M | 1434 |
| sCD-L (Lu & Song, 2024) | - | - | - | 2 | 2.04 | - | 778M | 1434 |
| sCD-XXL (Lu & Song, 2024) | - | - | - | 1 | 2.28 | - | 1.5B | 921 |
| sCD-XXL (Lu & Song, 2024) | - | - | - | 2 | **1.88** | - | 1.5B | 921 |
| | | GANs & masked & autoregressive models | | | | | | |
| BigGAN (Brock et al., 2018) | - | - | - | 1 | 8.43 | - | 160M | - |
| StyleGAN (Sauer et al., 2022) | - | - | - | $1\times2$ | **2.41** | 267.75 | 168M | - |
| MAGVIT-v2 (Yu et al., 2023) | - | - | - | $64\times2$ | 1.91 | 324.3 | 307M | 1080 |
| VAR-$d$36-s (Tian et al., 2024) | - | - | - | $10\times2$ | 2.63 | 303.2 | 2.3B | 350 |
| | | Ours: models trained and sampled using UCGM-{T,S} (setting $\lambda = 0$) | | | | | | |
| Ours-XL/1 | DC-AE (Chen et al., 2024c) | 1 | $16\times16$ | 32 | **1.55** | - | 675M | 800 |
| Ours-XL/1 | DC-AE (Chen et al., 2024c) | 1 | $16\times16$ | 16 | 1.81 | - | 675M | 800 |
| Ours-XL/1 | DC-AE (Chen et al., 2024c) | 1 | $16\times16$ | 8 | 3.07 | - | 675M | 800 |
| Ours-XL/1 | DC-AE (Chen et al., 2024c) | 1 | $16\times16$ | 4 | 74.0 | - | 675M | 800 |
| | | Ours: models trained and sampled using UCGM-{T,S} (setting $\lambda = 1$) | | | | | | |
| Ours-XL/1 | DC-AE (Chen et al., 2024c) | 1 | $16\times16$ | 1 | 2.42 | - | 675M | 840 |
| Ours-XL/1 | DC-AE (Chen et al., 2024c) | 1 | $16\times16$ | 2 | **1.75** | - | 675M | 840 |
| Ours-XL/4 | SD-VAE (Rombach et al., 2022) | 4 | $16\times16$ | 1 | 2.63 | - | 675M | 360 |
| Ours-XL/4 | SD-VAE (Rombach et al., 2022) | 4 | $16\times16$ | 2 | 2.11 | - | 675M | 360 |
| | | | $256 \times 256$ | | | | | |
| | | Consistency training & distillation | | | | | | |
| iCT (Song & Dhariwal, 2023) | - | - | - | 2 | 20.3 | - | 675M | - |
| Shortcut-XL/2 (Frans et al., 2024) | SD-VAE (Rombach et al., 2022) | 2 | $16\times16$ | 1 | 10.6 | - | 676M | 250 |
| Shortcut-XL/2 (Frans et al., 2024) | SD-VAE (Rombach et al., 2022) | 2 | $16\times16$ | 4 | 7.80 | - | 676M | 250 |
| Shortcut-XL/2 (Frans et al., 2024) | SD-VAE (Rombach et al., 2022) | 2 | $16\times16$ | 128 | 3.80 | - | 676M | 250 |
| IMM-XL/2 (Zhou et al., 2025) | SD-VAE (Rombach et al., 2022) | 2 | $16\times16$ | $1\times2$ | 7.77 | - | 675M | 3840 |
| IMM-XL/2 (Zhou et al., 2025) | SD-VAE (Rombach et al., 2022) | 2 | $16\times16$ | $2\times2$ | 5.33 | - | 675M | 3840 |
| IMM-XL/2 (Zhou et al., 2025) | SD-VAE (Rombach et al., 2022) | 2 | $16\times16$ | $4\times2$ | 3.66 | - | 675M | 3840 |
| IMM-XL/2 (Zhou et al., 2025) | SD-VAE (Rombach et al., 2022) | 2 | $16\times16$ | $8\times2$ | 2.77 | - | 675M | 3840 |
| IMM ($\omega = 1.5$) | SD-VAE (Rombach et al., 2022) | 2 | $16\times16$ | $1\times2$ | 8.05 | - | 675M | 3840 |
| IMM ($\omega = 1.5$) | SD-VAE (Rombach et al., 2022) | 2 | $16\times16$ | $2\times2$ | 3.99 | - | 675M | 3840 |
| IMM ($\omega = 1.5$) | SD-VAE (Rombach et al., 2022) | 2 | $16\times16$ | $4\times2$ | 2.51 | - | 675M | 3840 |
| IMM ($\omega = 1.5$) | SD-VAE (Rombach et al., 2022) | 2 | $16\times16$ | $8\times2$ | **1.99** | - | 675M | 3840 |
| | | GANs & masked & autoregressive models | | | | | | |
| BigGAN (Brock et al., 2018) | - | - | - | 1 | 6.95 | - | 112M | - |
| GigaGAN (Kang et al., 2023) | - | - | - | 1 | 3.45 | 225.52 | 569M | - |
| StyleGAN (Sauer et al., 2022) | - | - | - | $1\times2$ | **2.30** | 265.12 | 166M | - |
| VAR-$d$30-re (Tian et al., 2024) | - | - | - | $10\times2$ | 1.73 | 350.2 | 2.0B | 350 |
| | | Ours: models trained and sampled using UCGM-{T,S} (setting $\lambda = 0$) | | | | | | |
| Ours-XL/1 | VA-VAE (Yao et al., 2025) | 1 | $16\times16$ | 16 | 2.11 | - | 675M | 400 |
| Ours-XL/1 | VA-VAE (Yao et al., 2025) | 1 | $16\times16$ | 8 | 6.09 | - | 675M | 400 |
| Ours-XL/1 | E2E-VAE (Leng et al., 2025) | 1 | $16\times16$ | 16 | **1.40** | - | 675M | 800 |
| Ours-XL/1 | E2E-VAE (Leng et al., 2025) | 1 | $16\times16$ | 8 | 2.68 | - | 675M | 800 |
| | | Ours: models trained and sampled using UCGM-{T,S} (setting $\lambda = 1$) | | | | | | |
| Ours-XL/1 | VA-VAE (Yao et al., 2025) | 1 | $16\times16$ | 2 | **1.42** | - | 675M | 432 |
| Ours-XL/1 | VA-VAE (Yao et al., 2025) | 1 | $16\times16$ | 1 | 2.19 | - | 675M | 432 |
| Ours-XL/2 | SD-VAE (Rombach et al., 2022) | 2 | $16\times16$ | 1 | 2.10 | - | 675M | 424 |
| Ours-XL/1 | E2E-VAE (Leng et al., 2025) | 1 | $16\times16$ | 1 | 2.29 | - | 675M | 264 |

## D.6 Case Studies

In this section, we provide several case studies to intuitively illustrate the technical components proposed in this paper.

### D.6.1 Analysis of Consistency Ratio $\lambda$

We evaluate our approach on three synthetic benchmark datasets from `scikit-learn` (Pedregosa et al., 2011): the Two Moons (non-linear separation, see Fig. 6a), S-Curve (manifold structure, see Fig. 6b), and Swiss Roll (non-linear dimensionality reduction, see Fig. 6c). These studies yield two primary observations:

(a) Our UCGM successfully captures the structure of the data distribution and maps initial points sampled from a Gaussian distribution to the target distribution, regardless of whether the task is few-step ($\lambda = 1$) or multi-step ($\lambda = 0$) generation.

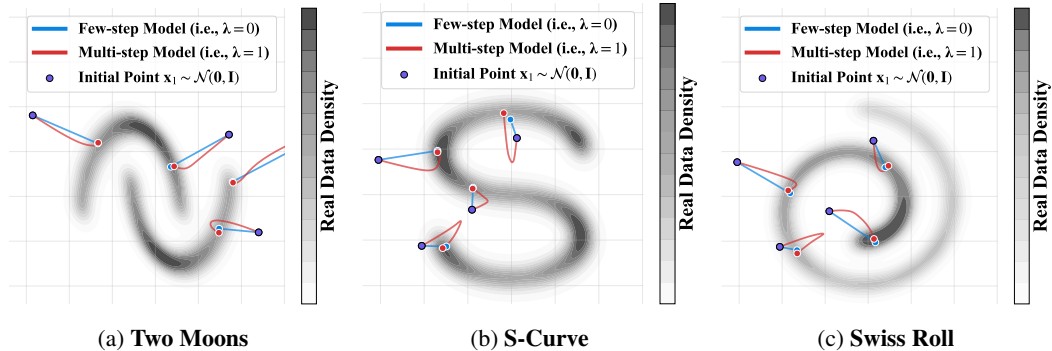

Figure 6: **Case studies of UCGM on three synthetic datasets.** These intuitive studies evaluate the ability of our UCGM to capture the latent data structure for both few-step generation ($\lambda = 1$) and multi-step generation ($\lambda = 0$) tasks.

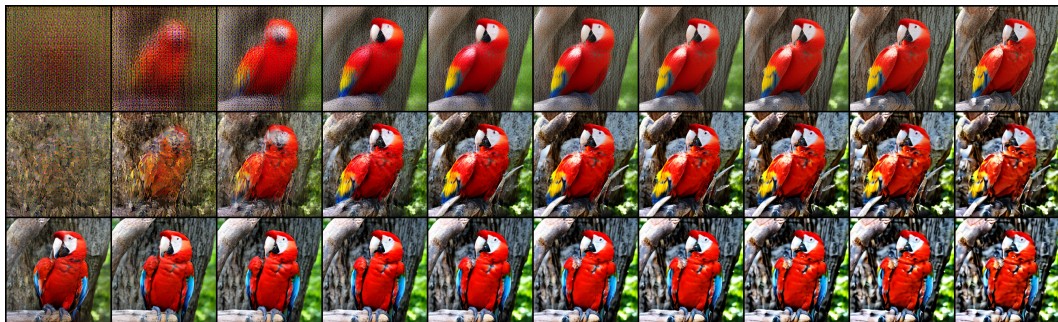

Figure 7: **Intermediate images generated during** $60$**-step sampling from UCGM-S.** Columns display intermediate images $\hat{\mathbf{x}}_t$ produced at different timesteps $t$ during a single sampling trajectory, ordered from left to right by decreasing $t$. Rows correspond to models trained with $\lambda \in \{0.0, 0.5, 1.0\}$, ordered from top to bottom. Note that the initial noise for generating these images is the same.

(b) Models trained for multi-step ($\lambda = 0$) and few-step ($\lambda = 1$) generation map the same initial Gaussian noise to nearly identical target data points.

To further validate these findings and explore additional properties of the consistency ratio $\lambda$, we conduct experiments on a real-world dataset (ImageNet-1K). Specifically, we trained three models with three different settings of $\lambda \in \{0.0, 0.5, 1.0\}$.

The experimental results presented in Fig. 7 demonstrate the following:

(a) For $\lambda = 1.0$, high visual fidelity is achieved early in the sampling process. In contrast, for $\lambda = 0.0$, high visual fidelity emerges in the mid to late stages. For $\lambda = 0.5$, high-quality images appear in the mid-stage of sampling.

(b) Despite being trained with different settings of $\lambda$ values, the models produce remarkably similar generated images.

In summary, we posit that while the setting of $\lambda$ affects the dynamics of the generation process, it does not substantially impact the final generated image quality. Detailed analysis of these phenomena is provided in App. F.1.1, App. F.1.3 and App. F.1.4.

### D.6.2 ANALYSIS OF TRANSPORT TYPES

Generated samples, obtained using UCGM-S with two distinct pre-trained models from prior works, are presented in Fig. 9 and Fig. 8. When using the identical initial Gaussian noise for both models, the generated images exhibit notable visual similarity. This observation is unexpected, considering the models were trained independently (Karras et al., 2024b; Wang et al., 2025) using distinct algorithms, transport formulations, network architectures, and data augmentation strategies. The similarity suggests that despite these differences, the learned probability flow ODEs may be converging to similar solutions. See App. F.1.2 for a comprehensive analysis of this phenomenon.

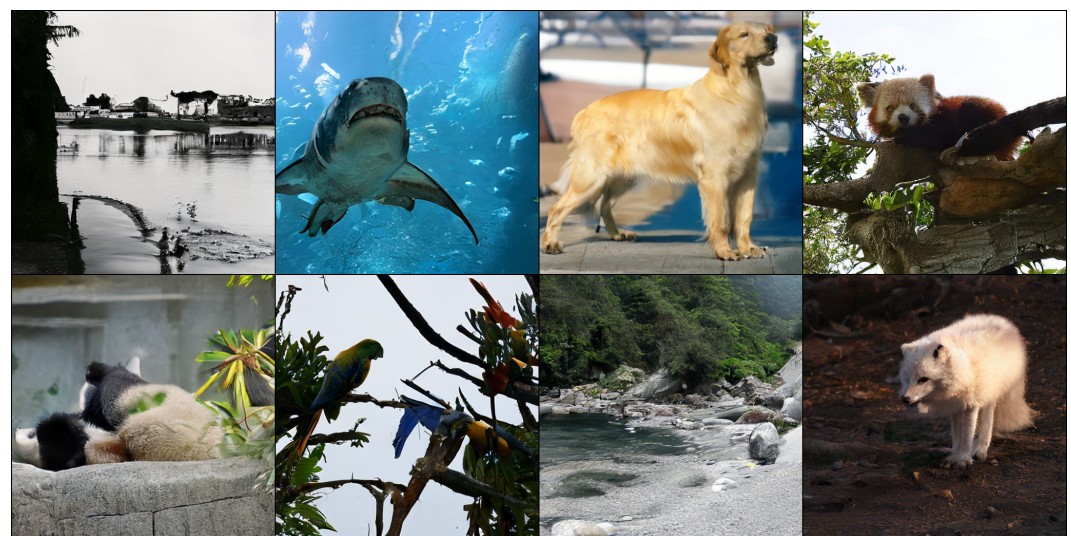

Figure 8: **Visualization of generated images ($512 \times 512$) from pre-trained EDM2-S (Karras et al., 2024b).**

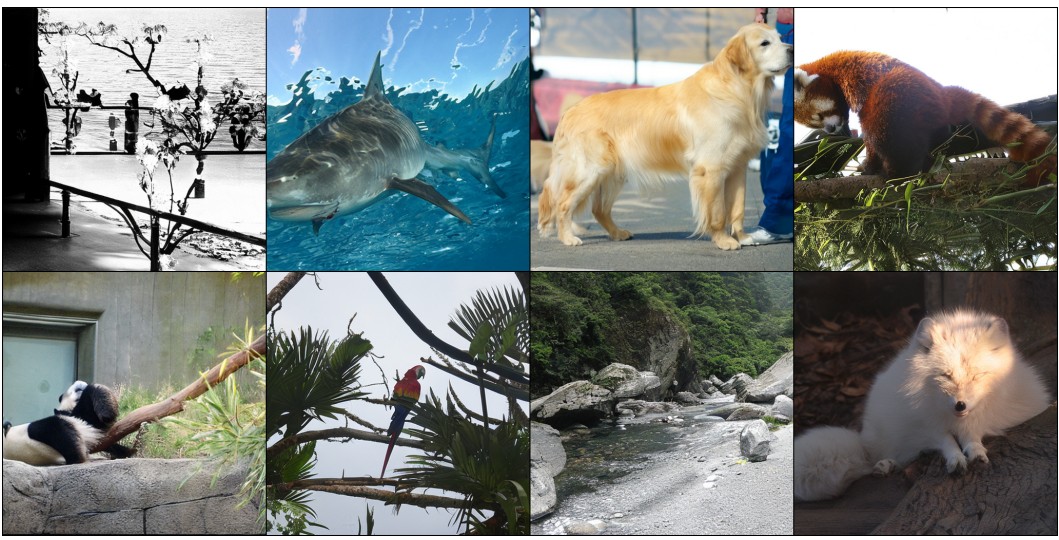

Figure 9: **Visualization of generated images ($512 \times 512$) from pre-trained DDT-XL/2 (Wang et al., 2025).**

### D.7 ANALYSIS OF PRE-TRAINED MODEL TUNING

Table 12: **System-level quality comparison for few-step generation on class-conditional ImageNet-1K after tuning.** Notation $\downarrow$/$\uparrow$ indicate performance decrease/increase relative to the baseline "Generation (Gen.)" performance of the "Original (Orig.)" pre-trained models at the respective NFE. Tuning time is evaluated on a cluster with 8 NVIDIA H800 GPUs.

| METHOD | #Params | Orig. Few-step Gen. | | Orig. Multi-step Gen. | | Tuning Efficiency | | Tuned Few-step Gen. | |
|---|---|---|---|---|---|---|---|---|---|
| | | NFE ($\downarrow$) | FID ($\downarrow$) | NFE ($\downarrow$) | FID ($\downarrow$) | #Epochs | Time | NFE ($\downarrow$) | FID ($\downarrow$) |
| REPA (Yu et al., 2024) | 675M | 2 | 177 | 80 | 1.86 | 0.64 | $\approx$ 13 minutes | 2 | $1.95^{\downarrow 175}$ |
| Lightning-DiT (Yao et al., 2025) | 675M | 2 | 217 | 80 | 1.49 | 0.64 | $\approx$ 10 minutes | 2 | $2.06^{\downarrow 215}$ |
| REPA-E (Leng et al., 2025) | 675M | 2 | 193 | 80 | 1.54 | 0.40 | $\approx$ 8 minutes | 2 | $1.39^{\downarrow 192}$ |
| DDT (Wang et al., 2025) | 675M | 2 | 191 | 80 | 1.46 | 0.32 | $\approx$ 11 minutes | 2 | $1.90^{\downarrow 189}$ |

In addition to our previous studies and experiments, where we demonstrated that our UCGM-S is a plug-and-play, training-free method for accelerating the sampling process of given pre-trained models from prior works (Yu et al., 2024; Yao et al., 2025; Leng et al., 2025; Wang et al., 2025) (cf.,

App. D.4), we have also proven that our UCGM-T is an efficient and effective unified framework for training both few-step and multi-step continuous generative models (cf., App. D.5 and App. D.4).

In this section, we evaluate the effectiveness of UCGM for tuning existing pre-trained generative models to enhance few-step generation performance. Tab. 12 presents the experimental results.

Specifically, the results demonstrate that UCGM-T facilitates the efficient conversion of continuous multi-step generative models (including diffusion and flow matching models) into high-performance few-step variants through minimal fine-tuning. For instance, the pre-trained REPA-E model (Leng et al., 2025), exhibiting 1.54 FID at 80 NFEs and 193 FID at 2 NFEs, can be efficiently tuned using UCGM-T in *only approximately* 8 *minutes (*0.4 *epoch)*. This tuning process yields a model *achieving* 1.39 *FID at* 2 *NFEs*, representing a substantial improvement in few-step generation quality with negligible tuning cost.

## D.8 Ablation study on UCGM techniques

Tab. 13 and Tab. 14 present the ablation studies on the proposed techniques in UCGM-T and UCGM-S, conducted under the same experimental setup as Tab. 12.

For UCGM-T, we observe that removing the generalized time distribution (GTD) does not affect the performance. This is expected, since GTD is designed to generalize beyond specific time distributions, whereas our experiments are conducted under a uniform distribution. In contrast, removing both GTD and the stabilizing technique leads to a significant degradation in FID scores across all backbones, demonstrating the importance of our proposed training stabilization method.

For UCGM-S, the stochastic sampling technique, which unifies ODE and SDE samplers in a generalized formulation, does not change the quantitative performance under our setting. However, removing both the stochastic component and the extrapolation strategy results in a substantial increase in NFE, indicating that the extrapolation-based acceleration is highly effective for efficient sampling.

Table 13: **Ablation study on UCGM-T techniques.**

| UCGM-T ($\lambda = 1$) | DDT (Wang et al., 2025) | | Light.DiT (Yao et al., 2025) | | REPA-E (Leng et al., 2025) | |
|---|---|---|---|---|---|---|
| | FID ↓ | NFE ↓ | FID ↓ | NFE ↓ | FID ↓ | NFE ↓ |
| original | 1.90 | 2 | 2.06 | 2 | 1.39 | 2 |
| w/o GTD | 1.90 | 2 | 2.06 | 2 | 1.39 | 2 |
| w/o GTD & stab. | 4.75 | 2 | 13.87 | 2 | 2.45 | 2 |

Table 14: **Ablation study on UCGM-S techniques.**

| UCGM-S | DDT (Wang et al., 2025) | | Light.DiT (Yao et al., 2025) | | REPA-E (Leng et al., 2025) | |
|---|---|---|---|---|---|---|
| | FID ↓ | NFE ↓ | FID ↓ | NFE ↓ | FID ↓ | NFE ↓ |
| original | 1.27 | 100 | 1.21 | 100 | 1.06 | 80 |
| w/o stoch. | 1.27 | 100 | 1.21 | 100 | 1.06 | 80 |
| w/o stoch. & extr. | 1.26 | 500 | 1.35 | 500 | 1.26 | 500 |

# E    PSEUDOCODE

## E.1    TRAINING ALGORITHM FOR UCGM-T

---

**Algorithm 1** (**UCGM-T**). A Unified and Efficient Trainer for Few-step and Multi-step Continuous Generative Models (including Diffusion, Flow Matching, and Consistency Models)

---

**Require:** Dataset $D$, transport coefficients $\{\alpha(\cdot), \gamma(\cdot), \hat{\alpha}(\cdot), \hat{\gamma}(\cdot)\}$, neural network $\boldsymbol{F}_{\boldsymbol{\theta}}$, enhancement ratio $\zeta$, Beta distribution parameters $(\theta_1, \theta_2)$, learning rate $\eta$, $\boldsymbol{\theta}^- = \boldsymbol{\theta}$ only in value.

**Ensure:** Trained neural network $\boldsymbol{F}_{\boldsymbol{\theta}}$ for generating samples from $p(\mathbf{x})$.

1: **repeat**
2:    Sample $\mathbf{z} \sim \mathcal{N}(\mathbf{0}, \mathbf{I})$, $\mathbf{x} \sim D$, $t \sim \phi(t) := \text{Beta}(\theta_1, \theta_2)$
3:    Compute input data, such as $\mathbf{x}_t = \alpha(t) \cdot \mathbf{z} + \gamma(t) \cdot \mathbf{x}$ and $\mathbf{x}_{\lambda t} = \alpha(\lambda t) \cdot \mathbf{z} + \gamma(\lambda t) \cdot \mathbf{x}$
4:    Compute model output $\boldsymbol{F}_t = \boldsymbol{F}_{\boldsymbol{\theta}}(\mathbf{x}_t, t)$ and set $\mathbf{z}^\star = \mathbf{z}$ and $\mathbf{x}^\star = \mathbf{x}$
5:    **if** $\zeta \in (0, 1)$ **then**
6:        Get enhanced $\mathbf{x}^\star = \boldsymbol{\xi}(\mathbf{x}, t, \boldsymbol{f}^{\mathbf{x}}(\boldsymbol{F}_{\boldsymbol{\theta}^-}(\mathbf{x}_t, t), \mathbf{x}_t, t), \boldsymbol{f}^{\mathbf{x}}(\boldsymbol{F}_{\boldsymbol{\theta}^-}(\mathbf{x}_t, t, \varnothing), \mathbf{x}_t, t))$ and $\mathbf{z}^\star = \boldsymbol{\xi}(\mathbf{z}, t, \boldsymbol{f}^{\mathbf{z}}(\boldsymbol{F}_{\boldsymbol{\theta}^-}(\mathbf{x}_t, t), \mathbf{x}_t, t), \boldsymbol{f}^{\mathbf{z}}(\boldsymbol{F}_{\boldsymbol{\theta}^-}(\mathbf{x}_t, t, \varnothing), \mathbf{x}_t, t))$ {Note that $\boldsymbol{\xi}(\mathbf{a}, t, \mathbf{b}, \mathbf{d}) := \mathbf{a} + (\zeta + \mathbf{1}_{t>s}(\frac{1}{2} - \zeta)) \cdot (\mathbf{b} - \mathbf{1}_{t>s} \cdot \mathbf{a} - \mathbf{d}(1 - \mathbf{1}_{t>s}))$, where $\mathbf{1}(\cdot)$ is the indicator function}
7:    **end if**
8:    **if** $\lambda \in [0, 1)$ **then**
9:        Compute $\mathbf{z}_t^\star = \hat{\alpha}(t) \cdot \mathbf{z}^\star + \hat{\gamma}(t) \cdot \mathbf{x}^\star$ and $\mathbf{z}_{\lambda t}^\star = \hat{\alpha}(\lambda t) \cdot \mathbf{z}^\star + \hat{\gamma}(\lambda t) \cdot \mathbf{x}^\star$
10:       Compute $D(t) = \alpha(t)\hat{\gamma}(t) - \hat{\alpha}(t)\gamma(t)$, $B(t) = \frac{\alpha(t)}{D(t)}$
11:       Let $C(t) = \frac{\alpha(t)}{2D(t)}$, $A(t) = B(t) - B(\lambda t)$ and $\hat{\omega}(t) = C(t) \cdot A(t)$
12:       Let $\Delta\mathbf{z}_t = \mathbf{z}_t^\star - \mathbf{z}_{\lambda t}^\star$
13:       Compute loss $\mathcal{L}_t(\boldsymbol{\theta}) = \|\boldsymbol{F}_{\boldsymbol{\theta}}(\mathbf{x}_t, t) - \mathbf{z}_t^\star\|_2^2 + \frac{B(\lambda t)}{\hat{\omega}(t)} \|\boldsymbol{F}_{\boldsymbol{\theta}}(\mathbf{x}_t, t) - \boldsymbol{F}_{\boldsymbol{\theta}^-}(\mathbf{x}_{\lambda t}, \lambda t) - \Delta\mathbf{z}_t\|_2^2$
14:    **else if** $\lambda = 1$ **then**
15:       Comupte $\mathbf{x}_{t+\epsilon}^\star = \alpha(t+\epsilon) \cdot \mathbf{z}^\star + \gamma(t+\epsilon) \cdot \mathbf{x}^\star$ and $\mathbf{x}_{t-\epsilon} = \alpha(t-\epsilon) \cdot \mathbf{z}^\star + \gamma(t-\epsilon) \cdot \mathbf{x}^\star$
16:       Let $\Delta\boldsymbol{f}_t^{\mathbf{x}} = \boldsymbol{f}^{\mathbf{x}}(\boldsymbol{F}_{\boldsymbol{\theta}^-}(\mathbf{x}_{t+\epsilon}, t+\epsilon), \mathbf{x}_{t+\epsilon}^\star, t+\epsilon) - \boldsymbol{f}^{\mathbf{x}}(\boldsymbol{F}_{\boldsymbol{\theta}^-}(\mathbf{x}_{t-\epsilon}, t-\epsilon), \mathbf{x}_{t-\epsilon}^\star, t-\epsilon)$
17:       Let $\Delta B(t) = \frac{\alpha(t+\epsilon)}{\alpha(t+\epsilon)\hat{\gamma}(t+\epsilon) - \hat{\alpha}(t+\epsilon)\gamma(t+\epsilon)} - \frac{\alpha(t-\epsilon)}{\alpha(t-\epsilon)\hat{\gamma}(t-\epsilon) - \hat{\alpha}(t-\epsilon)\gamma(t-\epsilon)}$
18:       Compute $\boldsymbol{F}_t^{\text{target}} = \boldsymbol{F}_{\boldsymbol{\theta}^-}(\mathbf{x}_t, t) - 2 \cdot \text{clip}(\frac{\Delta\boldsymbol{f}_t^{\mathbf{x}}}{\Delta B(t)}, -1, 1)$
19:       Compute loss $\mathcal{L}_t(\boldsymbol{\theta}) = \|\boldsymbol{F}_t - \boldsymbol{F}_t^{\text{target}}\|_2^2$
20:    **end if**
21:    Update $\boldsymbol{\theta} \leftarrow \boldsymbol{\theta} - \eta\nabla_{\boldsymbol{\theta}} \int_0^1 \phi(t)\mathcal{L}_t(\boldsymbol{\theta})\mathrm{d}t$
22: **until** Convergence

---

## E.2 Sampling Algorithm for UCGM-S

---

**Algorithm 2 (UCGM-S).** A Unified and Efficient Sampler for Few-step and Multi-step Continuous Generative Models (including Diffusion, Flow Matching, and Consistency Models)

---

**Require:** Initial $\tilde{\mathbf{x}} \sim \mathcal{N}(\mathbf{0}, \mathbf{I})$, transport coefficients $\{\alpha(\cdot), \gamma(\cdot), \hat{\alpha}(\cdot), \hat{\gamma}(\cdot)\}$, trained model $\boldsymbol{F}_{\boldsymbol{\theta}}$, sampling steps $N$, order $\nu \in \{1, 2\}$, time schedule $\mathcal{T}$, extrapolation ratio $\kappa$, stochastic ratio $\rho$.

**Ensure:** Final generated sample $\tilde{\mathbf{x}} \sim p(\mathbf{x})$ and history samples $\{\hat{\mathbf{x}}_i\}_{i=0}^{N}$ over generation process.

1: Let $N \leftarrow \lfloor (N+1)/2 \rfloor$ if using second order sampling ($\nu = 2$) {Adjusts total steps to match first-order evaluation count}

2: **for** $i = 0$ to $N - 1$ **do**

3:     Compute model output $\boldsymbol{F} = \boldsymbol{F}_{\boldsymbol{\theta}^-}(\tilde{\mathbf{x}}, t_i)$, and then $\hat{\mathbf{x}}_i = \boldsymbol{f}^{\mathbf{x}}(\boldsymbol{F}, \tilde{\mathbf{x}}, t_i)$ and $\hat{\mathbf{z}}_i = \boldsymbol{f}^{\mathbf{z}}(\boldsymbol{F}, \tilde{\mathbf{x}}, t_i)$

4:     **if** $i \geq 1$ **then**

5:         Compute extrapolated estimation $\hat{\mathbf{z}} = \hat{\mathbf{z}}_i + \kappa \cdot (\hat{\mathbf{z}}_i - \hat{\mathbf{z}}_{i-1})$ and $\hat{\mathbf{x}} = \hat{\mathbf{x}}_i + \kappa \cdot (\hat{\mathbf{x}}_i - \hat{\mathbf{x}}_{i-1})$

6:     **end if**

7:     Sample $\mathbf{z} \sim \mathcal{N}(\mathbf{0}, \mathbf{I})$ {An example choice of $\rho$ for performing SDE-similar sampling is: $\rho = \text{clip}(\frac{|t_i - t_{i+1}| \cdot 2\alpha(t_i)}{\alpha(t_{i+1})}, 0, 1)$}

8:     Compute estimated next time sample $\mathbf{x}' = \alpha(t_{i+1}) \cdot (\sqrt{1 - \rho} \cdot \hat{\mathbf{z}} + \sqrt{\rho} \cdot \mathbf{z}) + \gamma(t_{i+1}) \cdot \hat{\mathbf{x}}$

9:     **if** order $\nu = 2$ **and** $i < N - 1$ **then**

10:        Compute prediction $\boldsymbol{F}' = \boldsymbol{F}_{\boldsymbol{\theta}}(\mathbf{x}', t_{i+1})$, $\hat{\mathbf{x}}' = \boldsymbol{f}^{\mathbf{x}}(\boldsymbol{F}', \mathbf{x}', t_{i+1})$ and $\hat{\mathbf{z}}' = \boldsymbol{f}^{\mathbf{z}}(\boldsymbol{F}', \mathbf{x}', t_{i+1})$

11:        Compute corrected next time sample $\mathbf{x}' = \tilde{\mathbf{x}} \cdot \frac{\gamma(t_{i+1})}{\gamma(t_i)} + \left( \alpha(t_{i+1}) - \frac{\gamma(t_{i+1})\alpha(t_i)}{\gamma(t_i)} \right) \cdot \frac{\hat{\mathbf{x}} + \hat{\mathbf{x}}'}{2}$

12:     **end if**

13:     Reset $\tilde{\mathbf{x}} \leftarrow \mathbf{x}'$

14: **end for**

---

# F  THEORETICAL ANALYSIS

## F.1  MAIN RESULTS

### F.1.1  LEARNING OBJECTIVE WHEN $\lambda = 0$

Recall that $(\mathbf{z}, \mathbf{x}) \sim p(\mathbf{z}, \mathbf{x})$ is a pair of latent and data variables (typically independent), and let $t \in [0, 1]$. We have four differentiable scalar functions $\alpha, \gamma, \hat{\alpha}, \hat{\gamma} \colon [0, 1] \to \mathbb{R}$, the *noisy interpolant* $\mathbf{x}_t = \alpha(t)\,\mathbf{z} + \gamma(t)\,\mathbf{x}$ and $\boldsymbol{F}_t = \boldsymbol{F}_{\boldsymbol{\theta}}(\mathbf{x}_t, t)$. We define the $\mathbf{x}$- and $\mathbf{z}$-prediction functions by

$$\boldsymbol{f}^{\mathbf{x}}(\boldsymbol{F}_t, \mathbf{x}_t, t) = \frac{\alpha(t)\,\boldsymbol{F}_t \;-\; \hat{\alpha}(t)\,\mathbf{x}_t}{\alpha(t)\,\hat{\gamma}(t) \;-\; \hat{\alpha}(t)\,\gamma(t)}, \quad \text{and} \quad \boldsymbol{f}^{\mathbf{z}}(\boldsymbol{F}_t, \mathbf{x}_t, t) = \frac{\hat{\gamma}(t)\,\mathbf{x}_t \;-\; \gamma(t)\,\boldsymbol{F}_t}{\alpha(t)\,\hat{\gamma}(t) \;-\; \hat{\alpha}(t)\,\gamma(t)}.$$

Finally, let $\hat{\omega}(t) > 0$ be a weight function. We consider the $\mathbf{x}$- and $\mathbf{z}$-prediction losses

$$\mathcal{L}_{\mathbf{x}}(\boldsymbol{\theta}) = \mathbb{E}_{(\mathbf{z}, \mathbf{x}) \sim p(\mathbf{z}, \mathbf{x}),\, t}\Big[\frac{1}{\hat{\omega}(t)}\,\big\|\boldsymbol{f}^{\mathbf{x}}(\boldsymbol{F}_t, \mathbf{x}_t, t) - \mathbf{x}\big\|_2^2\Big],$$

$$\mathcal{L}_{\mathbf{z}}(\boldsymbol{\theta}) = \mathbb{E}_{(\mathbf{z}, \mathbf{x}) \sim p(\mathbf{z}, \mathbf{x}),\, t}\Big[\frac{1}{\hat{\omega}(t)}\,\big\|\boldsymbol{f}^{\mathbf{z}}(\boldsymbol{F}_t, \mathbf{x}_t, t) - \mathbf{z}\big\|_2^2\Big].$$

Recall that our unified loss function is defined by:

$$\mathcal{L}(\boldsymbol{\theta}) = \mathbb{E}_{(\mathbf{z}, \mathbf{x}) \sim p(\mathbf{z}, \mathbf{x}),\, t}\,\frac{1}{\hat{\omega}(t)}\,\big\|\boldsymbol{f}^{\mathbf{x}}(\boldsymbol{F}_{\boldsymbol{\theta}}(\mathbf{x}_t, t), \mathbf{x}_t, t) - \boldsymbol{f}^{\mathbf{x}}(\boldsymbol{F}_{\boldsymbol{\theta}^-}(\mathbf{x}_{\lambda t}, \lambda t), \mathbf{x}_{\lambda t}, \lambda t)\big\|_2^2.$$

We have $\mathcal{L}(\boldsymbol{\theta}) = \mathcal{L}_{\mathbf{x}}(\boldsymbol{\theta})$ when $\lambda = 0$, since $\boldsymbol{f}^{\mathbf{x}}(\boldsymbol{F}_0, \mathbf{x}_0, 0) = \mathbf{x}$. Then, we define the direct-field loss

$$\mathcal{L}_{\boldsymbol{F}}(\boldsymbol{\theta}) = \mathbb{E}_{(\mathbf{z}, \mathbf{x}),\, t}\Big[w_{\boldsymbol{F}}(t)\,\big\|\boldsymbol{F}_t - (\hat{\alpha}(t)\,\mathbf{z} + \hat{\gamma}(t)\,\mathbf{x})\big\|_2^2\Big], \quad w(t) > 0.$$

---

**Lemma 1 (Equivalence of x-prediction and direct-field loss) .** *For all $\boldsymbol{\theta}$,*

$$\boldsymbol{f}^{\mathbf{x}}(\boldsymbol{F}_t, \mathbf{x}_t, t) - \mathbf{x} = \frac{\alpha(t)}{\alpha(t)\,\hat{\gamma}(t) - \hat{\alpha}(t)\,\gamma(t)}\,\big[\boldsymbol{F}_t - (\hat{\alpha}(t)\,\mathbf{z} + \hat{\gamma}(t)\,\mathbf{x})\big].$$

*Hence*

$$\mathcal{L}_{\mathbf{x}}(\boldsymbol{\theta}) = \mathbb{E}_{(\mathbf{z}, \mathbf{x}),\, t}\Big[\frac{\alpha(t)^2}{\hat{\omega}(t)\,\big(\alpha(t)\,\hat{\gamma}(t) - \hat{\alpha}(t)\,\gamma(t)\big)^2}\,\big\|\boldsymbol{F}_t - (\hat{\alpha}(t)\,\mathbf{z} + \hat{\gamma}(t)\,\mathbf{x})\big\|_2^2\Big],$$

*so $\mathcal{L}_{\mathbf{x}}$ is equivalent to $\mathcal{L}_{\boldsymbol{F}}$ with*

$$w_{\boldsymbol{F}}(t) = \frac{\alpha(t)^2}{\hat{\omega}(t)\,\big(\alpha(t)\,\hat{\gamma}(t) - \hat{\alpha}(t)\,\gamma(t)\big)^2}.$$

---

*Proof.* Compute

$$\boldsymbol{f}^{\mathbf{x}}(\boldsymbol{F}_t, \mathbf{x}_t, t) - \mathbf{x} = \frac{\alpha(t)\,\boldsymbol{F}_t - \hat{\alpha}(t)\,\mathbf{x}_t}{\alpha(t)\,\hat{\gamma}(t) - \hat{\alpha}(t)\,\gamma(t)} - \mathbf{x}.$$

Since $\mathbf{x}_t = \alpha(t)\mathbf{z} + \gamma(t)\mathbf{x}$, the numerator becomes

$$\alpha\,\boldsymbol{F}_t - \hat{\alpha}\big(\alpha\mathbf{z} + \gamma\mathbf{x}\big) - \big(\alpha\,\hat{\gamma} - \hat{\alpha}\,\gamma\big)\mathbf{x} = \alpha(t)\Big[\boldsymbol{F}_t - \big(\hat{\alpha}(t)\mathbf{z} + \hat{\gamma}(t)\mathbf{x}\big)\Big].$$

Dividing by $\alpha\,\hat{\gamma} - \hat{\alpha}\,\gamma$ yields the desired factorization. Substituting into $\mathcal{L}_{\mathbf{x}}$ gives the weight $w(t)$ as above. $\qquad\square$

---

**Lemma 2 (Equivalence of z-Prediction and Direct-Field Loss) .** *For all $\boldsymbol{\theta}$,*

$$\boldsymbol{f}^{\mathbf{z}}(\boldsymbol{F}_t, \mathbf{x}_t, t) - \mathbf{z} = \frac{\gamma(t)}{\alpha(t)\,\hat{\gamma}(t) - \hat{\alpha}(t)\,\gamma(t)}\,\big[(\hat{\alpha}(t)\,\mathbf{z} + \hat{\gamma}(t)\,\mathbf{x}) - \boldsymbol{F}_t\big].$$

*Hence*

$$\mathcal{L}_{\mathbf{z}}(\boldsymbol{\theta}) = \mathbb{E}_{(\mathbf{z}, \mathbf{x}),\, t}\Big[\frac{\gamma(t)^2}{\hat{\omega}(t)\,\big(\alpha(t)\,\hat{\gamma}(t) - \hat{\alpha}(t)\,\gamma(t)\big)^2}\,\big\|\boldsymbol{F}_t - (\hat{\alpha}(t)\,\mathbf{z} + \hat{\gamma}(t)\,\mathbf{x})\big\|_2^2\Big],$$

*so $\mathcal{L}_{\mathbf{z}}$ is equivalent to $\mathcal{L}_{\boldsymbol{F}}$ with*

$$w_{\boldsymbol{F}}(t) = \frac{\gamma(t)^2}{\hat{\omega}(t)\left(\alpha(t)\,\hat{\gamma}(t) - \hat{\alpha}(t)\,\gamma(t)\right)^2}.$$

*Proof.* Compute

$$\boldsymbol{f}^{\mathbf{z}}(\boldsymbol{F}_t, \mathbf{x}_t, t) - \mathbf{z} = \frac{\hat{\gamma}(t)\,\mathbf{x}_t - \gamma(t)\,\boldsymbol{F}_t}{\alpha(t)\,\hat{\gamma}(t) - \hat{\alpha}(t)\,\gamma(t)} - \mathbf{z}.$$

Using $\mathbf{x}_t = \alpha\mathbf{z} + \gamma\mathbf{x}$, the numerator is

$$\hat{\gamma}(\alpha\mathbf{z} + \gamma\mathbf{x}) - \gamma\,\boldsymbol{F}_t - \left(\alpha\,\hat{\gamma} - \hat{\alpha}\,\gamma\right)\mathbf{z} = \gamma(t)\Big[\hat{\alpha}(t)\mathbf{z} + \hat{\gamma}(t)\mathbf{x} - \boldsymbol{F}_t\Big].$$

Dividing by $\alpha\,\hat{\gamma} - \hat{\alpha}\,\gamma$ gives the factorization. Substitution into $\mathcal{L}_{\mathbf{z}}$ yields the stated equivalence. $\qquad\square$

### F.1.2 CLOSED-FORM SOLUTION ANALYSIS WHEN $\lambda = 0$

when $\lambda = 0$, we aim to derive the Probability Flow Ordinary Differential Equation (PF-ODE) (Song et al., 2020b) corresponding to a defined forward process from time 0 to 1.

**Lemma 3 (Probability Flow ODE for the linear Gaussian forward process) .** *Let $p(\mathbf{x})$ be a data distribution on $\mathbb{R}^d$, and let $\mathbf{z} \sim \mathcal{N}(\mathbf{0}, \mathbf{I}_d)$ be independent of $\mathbf{x}$. Let $\alpha, \gamma : [0,1] \to \mathbb{R}$ be continuously differentiable scalar functions satisfying*

$$\alpha(0) = 0, \quad \alpha(1) = 1, \qquad \gamma(0) = 1, \quad \gamma(1) = 0,$$

*and assume $\gamma(t) \neq 0$ for $t \in (0,1)$. Define the* forward process

$$\mathbf{x}_t = \alpha(t)\,\mathbf{z} + \gamma(t)\,\mathbf{x}, \qquad t \in [0,1],$$

*so that $\mathbf{x}_0 = \mathbf{x} \sim p(\mathbf{x})$ and $\mathbf{x}_1 = \mathbf{z} \sim \mathcal{N}(0, I)$. Let $p_t(\mathbf{x}_t)$ denote the marginal density of $\mathbf{x}_t$. Then the* Probability Flow ODE *for this process,*

$$\frac{\mathrm{d}\,\mathbf{x}_t}{\mathrm{d}t} = \mathbf{f}(\mathbf{x}_t, t) - \tfrac{1}{2}\,g(t)^2\,\nabla_{\mathbf{x}_t} \log p_t(\mathbf{x}_t),$$

*takes the explicit form*

$$\frac{\mathrm{d}\,\mathbf{x}_t}{\mathrm{d}t} = \frac{\gamma'(t)}{\gamma(t)}\,\mathbf{x}_t - \left[\alpha(t)\,\alpha'(t) - \frac{\gamma'(t)}{\gamma(t)}\,\alpha(t)^2\right]\nabla_{\mathbf{x}_t} \log p_t(\mathbf{x}_t)\,. \tag{7}$$

*Proof.* We first represent the forward process $\mathbf{x}_t$ as the solution of a SDE (Song et al., 2020b):

$$\mathrm{d}\,\mathbf{x}_t = \mathbf{f}(\mathbf{x}_t, t)\,\mathrm{d}t + g(t)\,\mathrm{d}\mathbf{w}_t,$$

where $\mathbf{w}_t$ is a standard $d$-dimensional Wiener process, and where $\mathbf{f}(\cdot, t)$ and $g(t)$ are to be determined so that $\mathbf{x}_t = \alpha(t)\,\mathbf{z} + \gamma(t)\,\mathbf{x}$ in law.

1. Drift term via the conditional mean. Since $\mathbf{z}$ and $\mathbf{x}$ are independent,

$$\mathbb{E}[\mathbf{x}_t \mid \mathbf{x}] = \gamma(t)\,\mathbf{x}.$$

Differentiating in $t$ gives

$$\frac{\mathrm{d}}{\mathrm{d}t}\mathbb{E}[\mathbf{x}_t \mid \mathbf{x}] = \gamma'(t)\,\mathbf{x}. \tag{8}$$

On the other hand, we use the method of separation of variables, which is a classical method in solving PDEs, and we set the drift term as $\mathbf{f}(\mathbf{x}_t, t) = H(t)\,\mathbf{x}_t$ for some matrix $H(t)$, then

$$\frac{\mathrm{d}}{\mathrm{d}t}\mathbb{E}[\mathbf{x}_t \mid \mathbf{x}] = H(t)\,\mathbb{E}[\mathbf{x}_t \mid \mathbf{x}] = H(t)\,\gamma(t)\,\mathbf{x}. \tag{9}$$

Comparing (8) and (9) yields $H(t) = \gamma'(t)/\gamma(t)\,\mathbf{I}_d$, so

$$\mathbf{f}(\mathbf{x}_t, t) = \frac{\gamma'(t)}{\gamma(t)}\,\mathbf{x}_t.$$

2. Diffusion term via the conditional variance. The covariance of $\mathbf{x}_t$ given $\mathbf{x}$ is

$$\text{Var}(\mathbf{x}_t \mid \mathbf{x}) = \alpha(t)^2 \, \mathbf{I}_d.$$

For a linear SDE with drift matrix $H(t)$ and scalar diffusion $g(t)$, the covariance $\Sigma(t)$ satisfies the following Lyapunov equation (Jiménez, 2015):

$$\frac{\mathrm{d}\,\Sigma(t)}{\mathrm{d}t} = H(t)\,\Sigma(t) + \Sigma(t)\,H(t)^\top + g(t)^2\,\mathbf{I}_d.$$

Substitute $\Sigma(t) = \alpha(t)^2 \mathbf{I}_d$ and $H(t) = \frac{\gamma'(t)}{\gamma(t)}\mathbf{I}_d$. Since $\frac{\mathrm{d}}{\mathrm{d}t}\big(\alpha(t)^2\big) = 2\,\alpha(t)\,\alpha'(t)$, we get

$$2\,\alpha(t)\,\alpha'(t)\,\mathbf{I}_d = 2\,\frac{\gamma'(t)}{\gamma(t)}\,\alpha(t)^2\,\mathbf{I}_d \; + \; g(t)^2\,\mathbf{I}_d.$$

Rearranging yields

$$g(t)^2 = 2\,\alpha(t)\,\alpha'(t) \; - \; 2\,\frac{\gamma'(t)}{\gamma(t)}\,\alpha(t)^2.$$

3. Probability Flow ODE. By general theory (see, e.g., *de Bortoli et al.*), the probability flow ODE associated with the SDE $\mathrm{d}\mathbf{x}_t = \mathbf{f}(\mathbf{x}_t,t)\,\mathrm{d}t + g(t)\,\mathrm{d}\mathbf{w}_t$ is

$$\frac{\mathrm{d}\,\mathbf{x}_t}{\mathrm{d}t} = \mathbf{f}(\mathbf{x}_t,t) \; - \; \tfrac{1}{2}\,g(t)^2\,\nabla_{\mathbf{x}_t}\log p_t(\mathbf{x}_t).$$

Substituting the expressions for $\mathbf{f}$ and $g^2$ above gives

$$\frac{\mathrm{d}\,\mathbf{x}_t}{\mathrm{d}t} = \frac{\gamma'(t)}{\gamma(t)}\,\mathbf{x}_t \; - \; \Big[\alpha(t)\,\alpha'(t) \; - \; \tfrac{\gamma'(t)}{\gamma(t)}\,\alpha(t)^2\Big]\,\nabla_{\mathbf{x}_t}\log p_t(\mathbf{x}_t),$$

i.e.,

$$\mathbf{f}(\mathbf{x}_t,t) = \frac{\gamma'(t)}{\gamma(t)}\,\mathbf{x}_t, \qquad g(t)^2 = 2\,\alpha(t)\,\alpha'(t) \; - \; 2\,\frac{\gamma'(t)}{\gamma(t)}\,\alpha(t)^2.$$

which is exactly the claimed formula (7). This result is also proved with another method in (Holderrieth & Erives, 2025) (see Proposition 1 in their section 4.2). □

> **Lemma 4 (Tweedie formula (Song et al., 2020b) for the linear Gaussian model) .** *Under the linear Gaussian interpolation model $\mathbf{x}_t \mid \mathbf{x} \sim \mathcal{N}\big(\gamma(t)\,\mathbf{x},\ \alpha^2(t)\,\mathbf{I}\big)$, the conditional expectation of $\mathbf{x}$ given $\mathbf{x}_t$ is*
>
> $$\mathbb{E}[\mathbf{x} \mid \mathbf{x}_t] \;=\; \frac{\mathbf{x}_t + \alpha^2(t)\,\nabla_{\mathbf{x}_t}\log p_t(\mathbf{x}_t)}{\gamma(t)}\,.$$

*Proof.* We write the conditional expectation by Bayes' rule:

$$\mathbb{E}[\mathbf{x} \mid \mathbf{x}_t] = \int \mathbf{x}\,p(\mathbf{x} \mid \mathbf{x}_t)\,d\mathbf{x} = \frac{1}{p_t(\mathbf{x}_t)}\int \mathbf{x}\,p_t(\mathbf{x}_t \mid \mathbf{x})\,p(\mathbf{x})\,d\mathbf{x},$$

where $p_t(\mathbf{x}_t) = \int p_t(\mathbf{x}_t \mid \mathbf{x})\,p(\mathbf{x})\,d\mathbf{x}$.

Since $p_t(\mathbf{x}_t \mid \mathbf{x}) = (2\pi\alpha^2(t))^{-d/2}\exp\big(-\frac{1}{2\alpha^2(t)}\|\mathbf{x}_t - \gamma(t)\mathbf{x}\|^2\big)$, we have

$$\nabla_{\mathbf{x}_t} p_t(\mathbf{x}_t \mid \mathbf{x}) = -\frac{1}{\alpha^2(t)}\,(\mathbf{x}_t - \gamma(t)\mathbf{x})\,p_t(\mathbf{x}_t \mid \mathbf{x}).$$

Differentiating the marginal,

$$\nabla_{\mathbf{x}_t} p_t(\mathbf{x}_t) = \int \nabla_{\mathbf{x}_t} p_t(\mathbf{x}_t \mid \mathbf{x})\,p(\mathbf{x})\,d\mathbf{x} = -\frac{1}{\alpha^2(t)}\int (\mathbf{x}_t - \gamma(t)\mathbf{x})\,p_t(\mathbf{x}_t \mid \mathbf{x})\,p(\mathbf{x})\,d\mathbf{x}.$$

Multiply by $-\alpha^2(t)$ and split:

$$-\alpha^2(t)\,\nabla_{\mathbf{x}_t} p_t(\mathbf{x}_t) = \mathbf{x}_t\,p_t(\mathbf{x}_t) - \gamma(t)\int \mathbf{x}\,p_t(\mathbf{x}_t \mid \mathbf{x})\,p(\mathbf{x})\,d\mathbf{x}.$$

Rearrange and divide by $\gamma(t)p_t(\mathbf{x}_t)$:

$$\frac{1}{p_t(\mathbf{x}_t)}\int \mathbf{x}\,p_t(\mathbf{x}_t \mid \mathbf{x})\,p(\mathbf{x})\,d\mathbf{x} = \frac{\mathbf{x}_t + \alpha^2(t)\,\nabla_{\mathbf{x}_t} p_t(\mathbf{x}_t)/p_t(\mathbf{x}_t)}{\gamma(t)} = \frac{\mathbf{x}_t + \alpha^2(t)\,\nabla_{\mathbf{x}_t}\log p_t(\mathbf{x}_t)}{\gamma(t)}.$$

Hence $\mathbb{E}[\mathbf{x} \mid \mathbf{x}_t] = (\mathbf{x}_t + \alpha^2(t)\,\nabla_{\mathbf{x}_t}\log p_t(\mathbf{x}_t))\,/\,\gamma(t)$, as claimed. □

**Lemma 5 (Optimal predictors as conditional expectations) .** *For each fixed $t$ and observed $\mathbf{x}_t$, the pointwise minimizers $\boldsymbol{f}_\star^{\mathbf{x}}$ and $\boldsymbol{f}_\star^{\mathbf{z}}$ for the objective function $\mathcal{L}(\boldsymbol{\theta})$ satisfy*

$$\boldsymbol{f}_\star^{\mathbf{x}}(\boldsymbol{F}_t, \mathbf{x}_t, t) = \mathbb{E}[\mathbf{x} \mid \mathbf{x}_t], \quad \boldsymbol{f}_\star^{\mathbf{z}}(\boldsymbol{F}_t, \mathbf{x}_t, t) = \mathbb{E}[\mathbf{z} \mid \mathbf{x}_t].$$

*Proof.* Fix $t$ and $\mathbf{x}_t$. By Lem. 1 and Lem. 2, we conclude that the minimizers of $\mathcal{L}(\boldsymbol{\theta})$ are equivalent to those of $\mathcal{L}_{\mathbf{x}}$ and $\mathcal{L}_{\mathbf{z}}$.

Then, up to an additive constant independent of $\boldsymbol{f}^{\mathbf{x}}$, the contribution of $(t, \mathbf{x}_t)$ to $\mathcal{L}_{\mathbf{x}}$ is

$$\mathcal{J}_{\mathbf{x}}\big(\boldsymbol{f}^{\mathbf{x}}(\boldsymbol{F}_t, \mathbf{x}_t, t)\big) = \mathbb{E}\big[\|\boldsymbol{f}^{\mathbf{x}}(\boldsymbol{F}_t, \mathbf{x}_t, t) - \mathbf{x}\|_2^2 \mid \mathbf{x}_t\big].$$

For any random vector $X$, the function $w \mapsto \mathbb{E}\|w - X\|^2$ is uniquely minimized at $w = \mathbb{E}[X]$. Therefore

$$\boldsymbol{f}_\star^{\mathbf{x}}(\boldsymbol{F}_t, \mathbf{x}_t, t) = \arg\min_w \mathbb{E}\big[\|w - \mathbf{x}\|^2 \mid \mathbf{x}_t\big] = \mathbb{E}[\mathbf{x} \mid \mathbf{x}_t].$$

The same argument applies to

$$\mathcal{J}_{\mathbf{z}}\big(\boldsymbol{f}^{\mathbf{z}}(\boldsymbol{F}_t, \mathbf{x}_t, t)\big) = \mathbb{E}\big[\|\boldsymbol{f}^{\mathbf{z}}(\boldsymbol{F}_t, \mathbf{x}_t, t) - \mathbf{z}\|_2^2 \mid \mathbf{x}_t\big],$$

yielding

$$\boldsymbol{f}_\star^{\mathbf{z}}(\boldsymbol{F}_t, \mathbf{x}_t, t) = \mathbb{E}[\mathbf{z} \mid \mathbf{x}_t].$$

$\square$

**Theorem 2 .** *Under the linear Gaussian interpolation model $\mathbf{x}_t = \alpha(t)\,\mathbf{z} + \gamma(t)\,\mathbf{x}$, with $\mathbf{z} \sim \mathcal{N}(0, \mathbf{I})$ independent of $\mathbf{x}$, we have*

$$\boldsymbol{f}_\star^{\mathbf{x}}(\boldsymbol{F}_t, \mathbf{x}_t, t) = \frac{\mathbf{x}_t + \alpha^2(t)\,\nabla_{\mathbf{x}_t} \log p_t(\mathbf{x}_t)}{\gamma(t)}, \quad \boldsymbol{f}_\star^{\mathbf{z}}(\boldsymbol{F}_t, \mathbf{x}_t, t) = -\alpha(t)\,\nabla_{\mathbf{x}_t} \log p_t(\mathbf{x}_t).$$

*Then for every $t$,*

$$\alpha'(t)\,\boldsymbol{f}_\star^{\mathbf{z}}(\boldsymbol{F}_t, \mathbf{x}_t, t) + \gamma'(t)\,\boldsymbol{f}_\star^{\mathbf{x}}(\boldsymbol{F}_t, \mathbf{x}_t, t) = \frac{\gamma'(t)}{\gamma(t)}\,\mathbf{x}_t - \Big[\alpha(t)\,\alpha'(t) - \frac{\gamma'(t)}{\gamma(t)}\,\alpha^2(t)\Big]\nabla_{\mathbf{x}_t} \log p_t(\mathbf{x}_t).$$

*As a result, by Lem. 3, we conclude:*

$$\frac{\mathrm{d}\,\mathbf{x}_t}{\mathrm{d}t} = \alpha'(t)\,\boldsymbol{f}_\star^{\mathbf{z}}(\boldsymbol{F}_t, \mathbf{x}_t, t) + \gamma'(t)\,\boldsymbol{f}_\star^{\mathbf{x}}(\boldsymbol{F}_t, \mathbf{x}_t, t)$$

*Proof.* **Tweedie formula for $\boldsymbol{f}_\star^{\mathbf{x}}(\boldsymbol{F}_t, \mathbf{x}_t, t)$.** According to Lem. 5 and Lem. 4, we have

$$\boldsymbol{f}_\star^{\mathbf{x}}(\boldsymbol{F}_t, \mathbf{x}_t, t) = \mathbb{E}[\mathbf{x} \mid \mathbf{x}_t] = \frac{\mathbf{x}_t + \alpha^2(t)\,\nabla_{\mathbf{x}_t} \log p_t(\mathbf{x}_t)}{\gamma(t)}.$$

**Derivation of $\mathbb{E}[\mathbf{z} \mid \mathbf{x}_t]$ for $\boldsymbol{f}_\star^{\mathbf{z}}(\boldsymbol{F}_t, \mathbf{x}_t, t)$.** From $\mathbf{x}_t = \alpha(t)\,\mathbf{z} + \gamma(t)\,\mathbf{x}$ we solve $\mathbf{z} = (\mathbf{x}_t - \gamma(t)\,\mathbf{x})/\alpha(t)$. Taking conditional expectation and substituting the above,

$$\mathbb{E}[\mathbf{z} \mid \mathbf{x}_t] = \frac{1}{\alpha(t)}\Big(\mathbf{x}_t - \gamma(t)\,\mathbb{E}[\mathbf{x} \mid \mathbf{x}_t]\Big)$$

$$= \frac{1}{\alpha(t)}\Big(\mathbf{x}_t - \gamma(t)\,\frac{\mathbf{x}_t + \alpha^2(t)\,\nabla_{\mathbf{x}_t} \log p_t(\mathbf{x}_t)}{\gamma(t)}\Big) = -\alpha(t)\,\nabla_{\mathbf{x}_t} \log p_t(\mathbf{x}_t).$$

Thus, according to Lem. 5, we can obtain

$$\boldsymbol{f}_\star^{\mathbf{z}}(\boldsymbol{F}_t, \mathbf{x}_t, t) = -\alpha(t)\,\nabla_{\mathbf{x}_t} \log p_t(\mathbf{x}_t).$$

**Combine to obtain the claimed identity.**

$$\alpha'(t)\,\boldsymbol{f}_\star^{\mathbf{z}}(\boldsymbol{F}_t, \mathbf{x}_t, t) + \gamma'(t)\,\boldsymbol{f}_\star^{\mathbf{x}}(\boldsymbol{F}_t, \mathbf{x}_t, t)$$

$$= \alpha'(t)\big[-\alpha(t)\nabla_{\mathbf{x}_t} \log p_t(\mathbf{x}_t)\big] + \gamma'(t)\,\frac{\mathbf{x}_t + \alpha^2(t)\,\nabla_{\mathbf{x}_t} \log p_t(\mathbf{x}_t)}{\gamma(t)}$$

$$= -\alpha(t)\,\alpha'(t)\,\nabla_{\mathbf{x}_t} \log p_t(\mathbf{x}_t) + \frac{\gamma'(t)}{\gamma(t)}\,\mathbf{x}_t + \frac{\gamma'(t)}{\gamma(t)}\,\alpha^2(t)\,\nabla_{\mathbf{x}_t} \log p_t(\mathbf{x}_t)$$

$$= \frac{\gamma'(t)}{\gamma(t)}\,\mathbf{x}_t - \Big[\alpha(t)\,\alpha'(t) - \frac{\gamma'(t)}{\gamma(t)}\,\alpha^2(t)\Big]\nabla_{\mathbf{x}_t} \log p_t(\mathbf{x}_t).$$

This matches the claimed formula. $\qquad\square$

**Remark 2 (Velocity field of the flow ODE) .** *Given* $\mathbf{x}$ *and* $\mathbf{z}$*, the field* $\mathbf{v}^{(\mathbf{z},\mathbf{x})}(\mathbf{y}, t) = \alpha'(t)\mathbf{z} + \gamma'(t)\mathbf{x}$ *could transport* $\mathbf{z}$ *to* $\mathbf{x}$*, so the velocity field of the flow ODE can be computed as*

$$
\begin{aligned}
\mathbf{v}^*(\mathbf{x}_t, t) &= \mathbb{E}_{(\mathbf{z},\mathbf{x})|\mathbf{x}_t}\Big[\mathbf{v}^{(\mathbf{z},\mathbf{x})}(\mathbf{x}_t, t)|\mathbf{x}_t\Big] \\
&= \mathbb{E}_{(\mathbf{z},\mathbf{x})|\mathbf{x}_t}[\alpha'(t)\mathbf{z} + \gamma'(t)\mathbf{x}|\mathbf{x}_t] \\
&= \alpha'(t) \cdot \mathbb{E}[\mathbf{z}|\mathbf{x}_t] + \gamma'(t) \cdot \mathbb{E}[\mathbf{x}|\mathbf{x}_t] \\
&= \alpha'(t) \cdot \boldsymbol{f}_\star^{\mathbf{z}}(\boldsymbol{F}_t, \mathbf{x}_t, t) + \gamma'(t) \cdot \boldsymbol{f}_\star^{\mathbf{x}}(\boldsymbol{F}_t, \mathbf{x}_t, t) .
\end{aligned}
$$

**Corollary 1 (Closed-form PF–ODE for an arbitrary Gaussian mixture in $\mathbb{R}^d$) .** *Let*

$$
p(\mathbf{x}) = \sum_{j=1}^{K} w_j\, p_j\big(\mathbf{x}; \boldsymbol{m}_j, \boldsymbol{\Sigma}_j\big), \quad w_j > 0,\ \sum_j w_j = 1,
$$

*be a Gaussian-mixture density on $\mathbb{R}^d$, where $p_j(\mathbf{x})$ is the density of the $j$-th component, and $m_j$ is the mean and $\Sigma_j$ is the covariance matrix of the $j$-th component. In addition, let $\alpha, \gamma$ satisfy the hypotheses of Lem. 3, and define the forward map*

$$
\mathbf{x}_t = \alpha(t)\,\mathbf{z} + \gamma(t)\,\mathbf{x}, \quad \mathbf{x} \sim p(\mathbf{x}),\ \mathbf{z} \sim \mathcal{N}(\mathbf{0}, \mathbf{I}).
$$

*For each component $j$ set*

$$
\boldsymbol{\mu}_j(t) = \gamma(t)\,\boldsymbol{m}_j, \qquad \boldsymbol{\Sigma}_j(t) = \gamma(t)^2\,\boldsymbol{\Sigma}_j + \alpha(t)^2\,\mathbf{I}, \qquad \phi_j(\mathbf{x}_t) = \mathcal{N}\big(\mathbf{x}_t; \boldsymbol{\mu}_j(t), \boldsymbol{\Sigma}_j(t)\big)
$$

*so that*

$$
p_t(\mathbf{x}_t) = \sum_{j=1}^{K} w_j\, \mathcal{N}\big(\mathbf{x}_t; \boldsymbol{\mu}_j(t), \boldsymbol{\Sigma}_j(t)\big).
$$

*Then the Probability-Flow ODE (7) admits the closed-form drift*

$$
\frac{\mathrm{d}\mathbf{x}_t}{\mathrm{d}t} = \frac{\gamma'(t)}{\gamma(t)}\,\mathbf{x}_t + \left[\alpha(t)\,\alpha'(t) - \frac{\gamma'(t)}{\gamma(t)}\,\alpha(t)^2\right] \sum_{j=1}^{K} \frac{w_j\,\phi_j(\mathbf{x}_t)}{p_t(\mathbf{x}_t)}\,\boldsymbol{\Sigma}_j(t)^{-1}\big(\mathbf{x}_t - \boldsymbol{\mu}_j(t)\big).
$$

*Proof.* Step 1. *Affine transform of a Gaussian mixture.* Conditioned on the $j$-th component, $\mathbf{x} \sim \mathcal{N}(\boldsymbol{m}_j, \boldsymbol{\Sigma}_j)$, and hence

$$
\mathbf{x}_t = \alpha(t)\,\mathbf{z} + \gamma(t)\,\mathbf{x}\,\Big|\,(j) \sim \mathcal{N}\big(\gamma(t)\,\boldsymbol{m}_j,\ \alpha(t)^2\mathbf{I} + \gamma(t)^2\boldsymbol{\Sigma}_j\big).
$$

Defining

$$
\boldsymbol{\mu}_j(t) = \gamma(t)\,\boldsymbol{m}_j, \quad \boldsymbol{\Sigma}_j(t) = \gamma(t)^2\,\boldsymbol{\Sigma}_j + \alpha(t)^2\,\mathbf{I},
$$

we conclude that the marginal of $\mathbf{x}_t$ is

$$
\begin{aligned}
p_t(\mathbf{x}_t) &= \sum_{j=1}^{K} p_t(\mathbf{x}_t, N = j) \\
&= \sum_{j=1}^{K} p(N = j)p_t(\mathbf{x}_t|N = j) \\
&= \sum_{j=1}^{K} w_j\, p_t(\alpha\mathbf{z} + \gamma\mathbf{x}|N = j) \\
&= \sum_{j=1}^{K} w_j\, \mathcal{N}\big(\mathbf{x}_t; \boldsymbol{\mu}_j(t), \boldsymbol{\Sigma}_j(t)\big).
\end{aligned}
$$

Step 2. *Score of the mixture.* Set

$$\phi_j(\mathbf{x}_t) = \mathcal{N}\big(\mathbf{x}_t;\, \boldsymbol{\mu}_j(t), \boldsymbol{\Sigma}_j(t)\big), \quad p_t(\mathbf{x}_t) = \sum_{j=1}^{K} w_j\, \phi_j(\mathbf{x}_t).$$

Then by the usual mixture-rule,

$$\nabla_{\mathbf{x}_t} \log p_t = \frac{1}{p_t(\mathbf{x}_t)} \sum_{j=1}^{K} w_j\, \phi_j(\mathbf{x}_t)\, \nabla_{\mathbf{x}_t} \log \phi_j(\mathbf{x}_t).$$

Since for each Gaussian component

$$\nabla_{\mathbf{x}_t} \log \phi_j(\mathbf{x}_t) = -\, \boldsymbol{\Sigma}_j(t)^{-1}\big(\mathbf{x}_t - \boldsymbol{\mu}_j(t)\big),$$

we obtain the closed-form score

$$\nabla_{\mathbf{x}_t} \log p_t(\mathbf{x}_t) = -\frac{1}{p_t(\mathbf{x}_t)} \sum_{j=1}^{K} w_j\, \mathcal{N}\big(\mathbf{x}_t;\, \boldsymbol{\mu}_j(t), \boldsymbol{\Sigma}_j(t)\big)\, \boldsymbol{\Sigma}_j(t)^{-1}\big(\mathbf{x}_t - \boldsymbol{\mu}_j(t)\big).$$

Step 3. *Substitution into the PF–ODE.* By Lem. 3, the Probability–Flow ODE reads

$$\frac{\mathrm{d}\mathbf{x}_t}{\mathrm{d}t} = \frac{\gamma'(t)}{\gamma(t)}\, \mathbf{x}_t \;-\; \Big[\alpha(t)\,\alpha'(t) \;-\; \frac{\gamma'(t)}{\gamma(t)}\,\alpha(t)^2\Big]\, \nabla_{\mathbf{x}_t} \log p_t(\mathbf{x}_t).$$

Substituting the expression for $\nabla \log p_t$ above (and observing that the two '$-$' signs cancel) yields

$$\frac{\mathrm{d}\mathbf{x}_t}{\mathrm{d}t} = \frac{\gamma'(t)}{\gamma(t)}\, \mathbf{x}_t \;+\; \Big[\alpha(t)\,\alpha'(t) - \frac{\gamma'(t)}{\gamma(t)}\,\alpha(t)^2\Big] \sum_{j=1}^{K} \frac{w_j\, \mathcal{N}\big(\mathbf{x}_t;\, \boldsymbol{\mu}_j(t), \boldsymbol{\Sigma}_j(t)\big)}{p_t(\mathbf{x}_t)}\, \boldsymbol{\Sigma}_j(t)^{-1}\big(\mathbf{x}_t - \boldsymbol{\mu}_j(t)\big),$$

which is exactly the claimed closed-form drift. $\qquad\square$

---

**Corollary 2 (Closed-form PF–ODE for a symmetric two-peak Gaussian mixture) .** *Let $p(x)$ be the one-dimensional, symmetric, two-peak Gaussian mixture*

$$p(x) = \tfrac{1}{2}\,\mathcal{N}\big(x; -m, \sigma^2\big) + \tfrac{1}{2}\,\mathcal{N}\big(x; +m, \sigma^2\big),$$

*and let $\alpha, \gamma$ be as in Lem. 3. Define*

$$x_t = \alpha(t)\, z + \gamma(t)\, x, \qquad \Sigma_t = \gamma(t)^2\, \sigma^2 + \alpha(t)^2, \qquad \mu_\pm(t) = \pm\gamma(t)\, m.$$

*Then the marginal density of $x_t$ is*

$$p_t(x_t) = \tfrac{1}{2}\,\mathcal{N}\big(x_t; \mu_-(t), \Sigma_t\big) + \tfrac{1}{2}\,\mathcal{N}\big(x_t; \mu_+(t), \Sigma_t\big),$$

*and the Probability-Flow ODE (7) admits the closed-form drift*

$$\frac{\mathrm{d}x_t}{\mathrm{d}t} = \frac{\gamma'(t)}{\gamma(t)}\, x_t + \Big[\alpha(t)\,\alpha'(t) \;-\; \frac{\gamma'(t)}{\gamma(t)}\,\alpha(t)^2\Big] \frac{1}{\Sigma_t}\Big[x_t \;-\; \gamma(t)\, m\, \tanh\Big(\frac{\gamma(t)\, m}{\Sigma_t}\, x_t\Big)\Big].$$

---

*Proof.* Step 1. *Marginal law under the affine map.* Conditional on $x = \pm m$, one has

$$x_t = \alpha z + \gamma x \,\Big|\, (x = \pm m) \;\sim\; \mathcal{N}\big(\pm\gamma m,\; \alpha^2 + \gamma^2 \sigma^2\big) = \mathcal{N}\big(\mu_\pm(t), \Sigma_t\big).$$

Since each peak has weight $\tfrac{1}{2}$, the marginal of $x_t$ is $\tfrac{1}{2}\mathcal{N}(\mu_-, \Sigma_t) + \tfrac{1}{2}\mathcal{N}(\mu_+, \Sigma_t)$.

Step 2. *Score of the bimodal mixture.* Write $\phi_\pm(x_t) = \mathcal{N}(x_t; \mu_\pm(t), \Sigma_t)$, so $p_t = \tfrac{1}{2}(\phi_- + \phi_+)$. Then

$$\frac{\mathrm{d}}{\mathrm{d}x_t} \log p_t = \frac{1}{p_t}\, \tfrac{1}{2}\big(\phi_-\, \nabla \log \phi_- + \phi_+\, \nabla \log \phi_+\big), \qquad \nabla \log \phi_\pm = -\,\frac{x_t - \mu_\pm(t)}{\Sigma_t}.$$

Hence

$$\frac{\mathrm{d}}{\mathrm{d}x_t} \log p_t = -\frac{1}{2\, p_t\, \Sigma_t}\Big[\phi_-(x_t - \mu_-) + \phi_+(x_t - \mu_+)\Big].$$

Define

$$r_\pm(x_t) = \frac{\phi_\pm(x_t)}{\phi_-(x_t) + \phi_+(x_t)}, \quad \phi_- + \phi_+ = 2\,p_t.$$

Then

$$\frac{\mathrm{d}}{\mathrm{d}x_t} \log p_t = -\frac{1}{\Sigma_t}\Big[ r_-(x_t - \mu_-) + r_+(x_t - \mu_+)\Big].$$

A direct computation shows

$$r_+ - r_- = \tanh\Big(\frac{\gamma m}{\Sigma_t}\,x_t\Big), \quad r_-(x_t + \gamma m) + r_+(x_t - \gamma m) = x_t - \gamma m\,\tanh\Big(\frac{\gamma m}{\Sigma_t}\,x_t\Big).$$

Therefore

$$\frac{\mathrm{d}}{\mathrm{d}x_t} \log p_t = -\frac{1}{\Sigma_t}\Big[ x_t - \gamma m\,\tanh\big(\tfrac{\gamma m}{\Sigma_t}\,x_t\big)\Big].$$

Step 3. *Substitution into the PF–ODE.* By Lem. 3,

$$\frac{\mathrm{d}x_t}{\mathrm{d}t} = \frac{\gamma'}{\gamma}\,x_t - \Big[\alpha\,\alpha' - \frac{\gamma'}{\gamma}\,\alpha^2\Big]\frac{\mathrm{d}}{\mathrm{d}x_t}\log p_t.$$

Since $\frac{\mathrm{d}}{\mathrm{d}x_t}\log p_t$ carries a "$-$" sign, the two negatives cancel, yielding exactly

$$\frac{\mathrm{d}x_t}{\mathrm{d}t} = \frac{\gamma'}{\gamma}\,x_t + \Big[\alpha\,\alpha' - \frac{\gamma'}{\gamma}\,\alpha^2\Big]\frac{1}{\Sigma_t}\Big[ x_t - \gamma m\,\tanh\big(\tfrac{\gamma m}{\Sigma_t}\,x_t\big)\Big],$$

as claimed. $\qquad\square$

> **Remark 3 (OU-type schedule for the symmetric bimodal case) .** *Specialize Cor. 2 to the Ornstein–Uhlenbeck-type schedule with*
>
> $$\gamma(t) = e^{-st}, \qquad \alpha(t) = \sqrt{1 - e^{-2st}},$$
>
> *and noise variance $\sigma^2$ in each mixture component. Then the marginal variance is*
>
> $$\Sigma_t = \gamma(t)^2\,\sigma^2 \,+\, \alpha(t)^2 = \sigma^2 e^{-2st} + (1 - e^{-2st}),$$
>
> *and one obtains the closed-form drift of the Probability-Flow ODE:*
>
> $$\boxed{\frac{\mathrm{d}x_t}{\mathrm{d}t} = -\,s\,x_t + \frac{s}{\Sigma_t}\Big[ x_t - m\,e^{-st}\,\tanh\Big(\frac{m\,e^{-st}}{\Sigma_t}\,x_t\Big)\Big].}$$

*Proof.* We start from the general drift in Cor. 2:

$$\frac{\mathrm{d}x_t}{\mathrm{d}t} = \frac{\gamma'}{\gamma}\,x_t + \Big[\alpha\,\alpha' - \frac{\gamma'}{\gamma}\,\alpha^2\Big]\frac{1}{\Sigma_t}\Big[ x_t - \gamma\,m\,\tanh\big(\frac{\gamma\,m}{\Sigma_t}\,x_t\big)\Big].$$

We now substitute $\gamma(t) = e^{-st}$, $\alpha(t) = \sqrt{1 - e^{-2st}}$ and compute each piece in detail:
Derivative of $\gamma$:

$$\gamma'(t) = -s\,e^{-st}, \quad \Longrightarrow \quad \frac{\gamma'(t)}{\gamma(t)} = -s.$$

Marginal variance $\Sigma_t$:

$$\Sigma_t = \gamma(t)^2\,\sigma^2 + \alpha(t)^2 = \sigma^2\,e^{-2st} + (1 - e^{-2st}).$$

Square of $\alpha$ and its derivative:

$$\alpha(t)^2 = 1 - e^{-2st}, \qquad \frac{\mathrm{d}}{\mathrm{d}t}\big[\alpha(t)^2\big] = 2s\,e^{-2st} \implies 2\,\alpha\,\alpha' = 2s\,e^{-2st} \implies \alpha\,\alpha' = s\,e^{-2st}.$$

Combination term

$$\alpha\,\alpha' - \frac{\gamma'}{\gamma}\,\alpha^2 = s\,e^{-2st} - (-s)\,(1 - e^{-2st}) = s\big[e^{-2st} + 1 - e^{-2st}\big] = s.$$

Substitution into the general drift formula gives

$$\frac{dx_t}{dt} = -s\,x_t + s\,\frac{1}{\Sigma_t}\Big[x_t - e^{-st}\,m\,\tanh\Big(\frac{e^{-st}\,m}{\Sigma_t}\,x_t\Big)\Big].$$

Hence the final, closed-form Probability-Flow ODE is

$$\frac{dx_t}{dt} = -\,s\,x_t + \frac{s}{\Sigma_t}\Big[x_t - m\,e^{-st}\,\tanh\Big(\frac{m\,e^{-st}}{\Sigma_t}\,x_t\Big)\Big],$$

where $\Sigma_t = \sigma^2 e^{-2st} + (1 - e^{-2st})$. □

---

**Remark 4 (Triangular schedule for the symmetric bimodal case).** *Specialize Cor. 2 to the trigonometric schedule*

$$\gamma(t) = \cos\Big(\frac{\pi}{2}\,t\Big), \qquad \alpha(t) = \sin\Big(\frac{\pi}{2}\,t\Big),$$

*with noise variance $\sigma^2$ in each mixture component. Then*

$$\Sigma_t = \gamma(t)^2\,\sigma^2 + \alpha(t)^2 = \sigma^2\cos^2\Big(\frac{\pi}{2}\,t\Big) + \sin^2\Big(\frac{\pi}{2}\,t\Big),$$

*and the closed-form drift of the Probability-Flow ODE is*

$$\boxed{\frac{dx_t}{dt} = -\frac{\pi}{2}\,\tan\Big(\frac{\pi}{2}\,t\Big)\,x_t + \frac{\frac{\pi}{2}\,\tan\big(\frac{\pi}{2}\,t\big)}{\Sigma_t}\Big[x_t - \cos\Big(\frac{\pi}{2}\,t\Big)\,m\,\tanh\Big(\frac{\cos(\frac{\pi}{2}t)\,m}{\Sigma_t}\,x_t\Big)\Big].}$$

*Proof.* We begin with the general drift in Cor. 2:

$$\frac{dx_t}{dt} = \frac{\gamma'}{\gamma}\,x_t + \Big[\alpha\,\alpha' - \frac{\gamma'}{\gamma}\,\alpha^2\Big]\frac{1}{\Sigma_t}\Big[x_t - \gamma\,m\,\tanh\Big(\frac{\gamma\,m}{\Sigma_t}\,x_t\Big)\Big].$$

For $\gamma(t) = \cos(\frac{\pi}{2}t),\ \alpha(t) = \sin(\frac{\pi}{2}t)$,

$$\gamma'(t) = -\tfrac{\pi}{2}\,\sin\Big(\frac{\pi}{2}\,t\Big) = -\tfrac{\pi}{2}\,\alpha(t), \quad \frac{\gamma'}{\gamma} = -\tfrac{\pi}{2}\,\tan\Big(\frac{\pi}{2}\,t\Big).$$

And

$$\alpha'(t) = \tfrac{\pi}{2}\,\cos\Big(\frac{\pi}{2}\,t\Big) = \tfrac{\pi}{2}\,\gamma(t),$$

so that

$$\alpha\,\alpha' - \frac{\gamma'}{\gamma}\,\alpha^2 = \frac{\pi}{2}\,\alpha\,\gamma + \frac{\pi}{2}\,\frac{\alpha^3}{\gamma} = \frac{\pi}{2}\,\frac{\alpha}{\gamma}(\alpha^2 + \gamma^2) = \frac{\pi}{2}\,\tan\Big(\frac{\pi}{2}\,t\Big).$$

Substituting into the general formula immediately yields the boxed drift. □

---

**Remark 5 (Linear schedule for the symmetric bimodal case).** *Specialize Cor. 2 to the "Linear" schedule*

$$\gamma(t) = 1 - t, \qquad \alpha(t) = t, \quad t \in [0,1].$$

*Then the marginal variance is*

$$\Sigma_t = \gamma(t)^2\,\sigma^2 + \alpha(t)^2 = (1-t)^2\,\sigma^2 + t^2,$$

*and one obtains the closed-form drift of the Probability-Flow ODE:*

$$\boxed{\frac{dx_t}{dt} = -\frac{x_t}{1-t} + \frac{t}{(1-t)\,\Sigma_t}\Big[x_t - m\,(1-t)\,\tanh\Big(\frac{m\,(1-t)}{\Sigma_t}\,x_t\Big)\Big].}$$

*Proof.* We begin with the general drift formula from Cor. 2:

$$\frac{dx_t}{dt} = \frac{\gamma'(t)}{\gamma(t)}\,x_t + \Big[\alpha(t)\,\alpha'(t) - \frac{\gamma'(t)}{\gamma(t)}\,\alpha(t)^2\Big]\frac{1}{\Sigma_t}\Big[x_t - \gamma(t)\,m\,\tanh\Big(\frac{\gamma(t)\,m}{\Sigma_t}\,x_t\Big)\Big].$$

We substitute $\gamma(t) = 1 - t$ and $\alpha(t) = t$ and compute each piece:

1. Derivative of $\gamma$:
$$\gamma'(t) = -1, \quad \implies \quad \frac{\gamma'(t)}{\gamma(t)} = -\frac{1}{1-t}.$$

2. Marginal variance:
$$\Sigma_t = (1-t)^2\,\sigma^2 + t^2.$$

3. Square of $\alpha$ and its derivative:
$$\alpha(t)^2 = t^2, \quad \frac{d}{dt}\big[\alpha(t)^2\big] = 2t \implies 2\,\alpha\,\alpha' = 2t \implies \alpha(t)\,\alpha'(t) = t.$$

4. Combination term:
$$\alpha\,\alpha' - \frac{\gamma'}{\gamma}\,\alpha^2 = t - \left(-\frac{1}{1-t}\right)t^2 = t + \frac{t^2}{1-t} = \frac{t}{1-t}.$$

Substituting these into the general drift gives
$$\frac{dx_t}{dt} = -\frac{x_t}{1-t} + \frac{t}{(1-t)\,\Sigma_t}\left[x_t - m\,(1-t)\,\tanh\!\left(\frac{m\,(1-t)}{\Sigma_t}\,x_t\right)\right],$$

which is the claimed closed-form Probability-Flow ODE. $\qquad\square$

---

**Remark 6 (OU-type schedule for the Hermite–Gaussian $n = 1$ case)**. *Apply Lem. 3 to the one-dimensional Hermite–Gaussian initial density*
$$p_1(x) \;\propto\; x\,e^{-x^2/2}, \quad x > 0,$$

*and the OU-type schedule*
$$\gamma(t) = e^{-st}, \qquad \alpha(t) = \sqrt{1 - e^{-2st}}.$$

*Then the Probability–Flow ODE (7) reduces to the scalar form*
$$\boxed{\frac{dx_t}{dt} \;=\; -\frac{s}{x_t}, \quad t \in [0,1],}$$

*and integrating from $t = 1$ (with $x(1) = x_1$) to any $t \in [0,1]$ yields the explicit solution*
$$\boxed{x_t \;=\; \sqrt{x_1^2 \,+\, 2\,s\,(1-t)}.}$$

---

*Proof.* By Lem. 3, the drift of the Probability–Flow ODE is
$$\frac{dx_t}{dt} = \frac{\gamma'(t)}{\gamma(t)}\,x_t \;-\; \Big[\alpha(t)\,\alpha'(t) - \tfrac{\gamma'(t)}{\gamma(t)}\,\alpha(t)^2\Big]\,\partial_{x_t}\ln p_t(x_t).$$

Under $\gamma(t) = e^{-st}$ and $\alpha(t) = \sqrt{1 - e^{-2st}}$ one computes
$$\frac{\gamma'}{\gamma} = -s, \quad 2\,\alpha\,\alpha' = 2s\,e^{-2st} \implies \alpha\,\alpha' = s\,e^{-2st}, \quad -\frac{\gamma'}{\gamma}\,\alpha^2 = s\,(1 - e^{-2st}),$$

hence
$$\alpha\,\alpha' \;-\; \frac{\gamma'}{\gamma}\,\alpha^2 = s\,e^{-2st} + s\,(1 - e^{-2st}) = s.$$

Moreover, one checks that the marginal density remains $p_t(x) \propto x\,e^{-x^2/2}$, so $\partial_x \ln p_t(x) = \frac{1}{x} - x$. Therefore
$$\frac{dx_t}{dt} = -\,s\,x_t \;-\; s\left(\tfrac{1}{x_t} - x_t\right) = -\frac{s}{x_t}.$$

Separating variables,
$$\frac{dx}{dt} = -\frac{s}{x} \quad \implies \quad \int_{x_1}^{x_t} x\,dx = -s\int_1^t ds \implies \frac{x_t^2 - x_1^2}{2} = -s\,(t-1),$$

whence
$$x_t^2 = x_1^2 + 2\,s\,(1-t), \quad x_t = \sqrt{x_1^2 + 2\,s\,(1-t)},$$

taking the positive root on $x > 0$. $\qquad\square$

**Lemma 6 (Picard–Lindelöf existence and uniqueness) .** *Let $v\colon \mathbb{R} \times [0,1] \to \mathbb{R}$ be continuous in $t$ and satisfy the* uniform Lipschitz *condition*

$$|v(x,t) - v(y,t)| \ \le\ L\,|x-y|, \quad \forall\, x,y \in \mathbb{R},\ t \in [0,1],$$

*for some constant $L < \infty$. Then for any $t_0 \in [0,1]$ and any initial value $x(t_0) = x_0$, there exists $\delta > 0$ and a unique function*

$$x \in C^1\big([t_0 - \delta, t_0 + \delta] \cap [0,1]\big)$$

*solving the ODE*

$$\frac{\mathrm{d}x}{\mathrm{d}t}(t) \ =\ v\big(x(t), t\big), \quad x(t_0) = x_0.$$

*Proof.* Fix $t_0 \in [0,1]$ and $x_0 \in \mathbb{R}$. Choose $\delta > 0$ so small that $(t_0 - \delta, t_0 + \delta) \subset [0,1]$ and $L\delta < 1$. Define the closed ball

$$B_R = \big\{ x \in C([t_0 - \delta, t_0 + \delta], \mathbb{R}) : \|x - x_0\|_\infty \le R \big\}$$

with $R > 0$ to be chosen. Consider the operator

$$(\Gamma x)(t) = x_0 + \int_{t_0}^t v\big(x(s), s\big)\, \mathrm{d}s.$$

Since $v$ is continuous on the compact set $B_R \times [t_0 - \delta, t_0 + \delta]$, it is bounded by some $M < \infty$. If we choose $R = M\delta$, then $\Gamma$ maps $B_R$ into itself:

$$\|\Gamma x - x_0\|_\infty \le \sup_t \int_{t_0}^t |v(x(s), s)|\, \mathrm{d}s \le M\,\delta = R.$$

Moreover, for any $x, y \in B_R$ and any $t$ in the interval,

$$|(\Gamma x)(t) - (\Gamma y)(t)| \le \int_{t_0}^t |v(x(s), s) - v(y(s), s)|\, \mathrm{d}s \le L\,\delta\,\|x - y\|_\infty < \|x - y\|_\infty,$$

so $\Gamma$ is a contraction. By the Banach fixed-point theorem, $\Gamma$ has a unique fixed point in $B_R$, which is precisely the unique $C^1$ solution of the ODE on $[t_0 - \delta, t_0 + \delta] \cap [0,1]$. $\qquad\square$

**Lemma 7 (Gronwall's inequality and no blow-up) .** *Let $x \in C^1([0,1])$ satisfy*

$$|x'(t)| \ \le\ K\big(1 + |x(t)|\big), \quad t \in [0,1],$$

*for some constant $K \ge 0$. Then*

$$|x(t)| \ \le\ \big(|x(1)| + 1\big)\, e^{K(1-t)} \ -\ 1, \quad \forall\, t \in [0,1],$$

*and in particular $x$ does not blow up in finite time on $[0,1]$.*

*Proof.* Define
$$y(t) \ =\ |x(t)| + 1 \ \ge\ 1.$$
Since $y(t)$ is Lipschitz, for almost every $t$ we have
$$y'(t) = \frac{\mathrm{d}}{\mathrm{d}t}\big(|x(t)| + 1\big) = \mathrm{sgn}(x(t))\, x'(t),$$
and hence
$$y'(t) \ \ge\ -\,|x'(t)| \ \ge\ -\,K\big(1 + |x(t)|\big) \ =\ -K\, y(t).$$
Equivalently,
$$y'(t) + K\, y(t) \ \ge\ 0.$$
Multiply both sides by the integrating factor $e^{Kt}$:
$$\frac{\mathrm{d}}{\mathrm{d}t}\big(e^{Kt} y(t)\big) = e^{Kt}\big(y'(t) + K\, y(t)\big) \ \ge\ 0.$$

Thus the function $t \mapsto e^{Kt} y(t)$ is non-decreasing on $[0, 1]$. For any $t \leq 1$ we then have

$$e^{Kt} y(t) \ \leq \ e^{K \cdot 1} y(1) \quad \implies \quad y(t) \ \leq \ y(1) e^{K(1-t)} = \big(|x(1)| + 1\big) e^{K(1-t)}.$$

Rewriting $y(t) = |x(t)| + 1$ gives

$$|x(t)| \ \leq \ \big(|x(1)| + 1\big) e^{K(1-t)} \ - \ 1,$$

as claimed. In particular $|x(t)| < \infty$ for all $t \in [0, 1]$, so no finite-time blow-up occurs. $\qquad \square$

---

**Lemma 8 (Gaussian convolution preserves linear-growth bound).** *Let $p_0 \in C^1(\mathbb{R})$ be a probability density satisfying*

$$\big|\partial_x \log p_0(x)\big| \ \leq \ A + B\,|x|, \quad A, B < \infty, \ \forall x \in \mathbb{R},$$

*and assume furthermore that $\|p_0\|_\infty \ = \ \sup_{x \in \mathbb{R}} p_0(x) \ \leq \ M \ < \ \infty$. For each $\sigma > 0$, define the Gaussian kernel $\phi_\sigma(u) \ = \ \frac{1}{\sqrt{2\pi}\,\sigma} \exp\!\big(-\frac{u^2}{2\sigma^2}\big)$, and set $p_\sigma(x) \ = \ (p_0 * \phi_\sigma)(x) \ = \ \int_{\mathbb{R}} p_0(y)\,\phi_\sigma(x - y)\,\mathrm{d}y$. Then $p_\sigma \in C^\infty(\mathbb{R})$ and there exist*

$$A(\sigma) = A + B\,M\,\sigma\sqrt{\tfrac{2}{\pi}}, \quad B(\sigma) = B,$$

*such that*

$$\big|\partial_x \log p_\sigma(x)\big| \ \leq \ A(\sigma) \ + \ B(\sigma)\,|x|, \quad \forall x \in \mathbb{R}.$$

---

*Proof. Smoothness and differentiation under the integral.* Since $\phi_\sigma \in C^\infty(\mathbb{R})$ decays rapidly and $p_0 \in L^\infty(\mathbb{R})$, by dominated convergence we may differentiate under the integral to get

$$p'_\sigma(x) = \int_{\mathbb{R}} p_0(y)\,\partial_x \phi_\sigma(x - y)\,\mathrm{d}y = \int_{\mathbb{R}} p_0(y)\,\phi'_\sigma(x - y)\,\mathrm{d}y.$$

Noting $\partial_y \phi_\sigma(x - y) = -\phi'_\sigma(x - y)$, we rewrite

$$p'_\sigma(x) = -\int_{\mathbb{R}} p_0(y)\,\partial_y \phi_\sigma(x - y)\,\mathrm{d}y.$$

*Integration by parts.* Integrating the above in $y$ and using that $p_0(y)\phi_\sigma(x - y) \to 0$ as $|y| \to \infty$, we obtain

$$p'_\sigma(x) = \int_{\mathbb{R}} p'_0(y)\,\phi_\sigma(x - y)\,\mathrm{d}y = \int_{\mathbb{R}} (\partial_y \log p_0)(y)\,p_0(y)\,\phi_\sigma(x - y)\,\mathrm{d}y.$$

*Bounding $\partial_x \log p_\sigma$.* Hence

$$\big|\partial_x \log p_\sigma(x)\big| = \frac{|p'_\sigma(x)|}{p_\sigma(x)} = \frac{\big|\int (\partial_y \log p_0)(y)\,p_0(y)\,\phi_\sigma(x - y)\,\mathrm{d}y\big|}{p_\sigma(x)}$$

$$\leq \frac{\int |\partial_y \log p_0(y)|\,p_0(y)\,\phi_\sigma(x - y)\,\mathrm{d}y}{p_\sigma(x)} \leq \frac{\int \big(A + B|y|\big)\,p_0(y)\,\phi_\sigma(x - y)\,\mathrm{d}y}{p_\sigma(x)}$$

$$= A + B\,\frac{\int |y|\,p_0(y)\,\phi_\sigma(x - y)\,\mathrm{d}y}{p_\sigma(x)}.$$

*Change of variables.* Set $u = y - x$. Then

$$\int |y|\,p_0(y)\,\phi_\sigma(x - y)\,\mathrm{d}y = \int |x + u|\,p_0(x + u)\,\phi_\sigma(u)\,\mathrm{d}u \leq |x|\,p_\sigma(x) + \int |u|\,p_0(x + u)\,\phi_\sigma(u)\,\mathrm{d}u.$$

Hence

$$\frac{\int |y|\,p_0(y)\,\phi_\sigma(x - y)\,\mathrm{d}y}{p_\sigma(x)} \leq |x| + \frac{\int |u|\,p_0(x + u)\,\phi_\sigma(u)\,\mathrm{d}u}{p_\sigma(x)}.$$

*Using the $L^\infty$-bound on $p_0$.* Since $p_0(x + u) \leq M$,

$$\int |u|\,p_0(x + u)\,\phi_\sigma(u)\,\mathrm{d}u \leq M \int |u|\,\phi_\sigma(u)\,\mathrm{d}u = M\,\sigma\sqrt{\tfrac{2}{\pi}}.$$

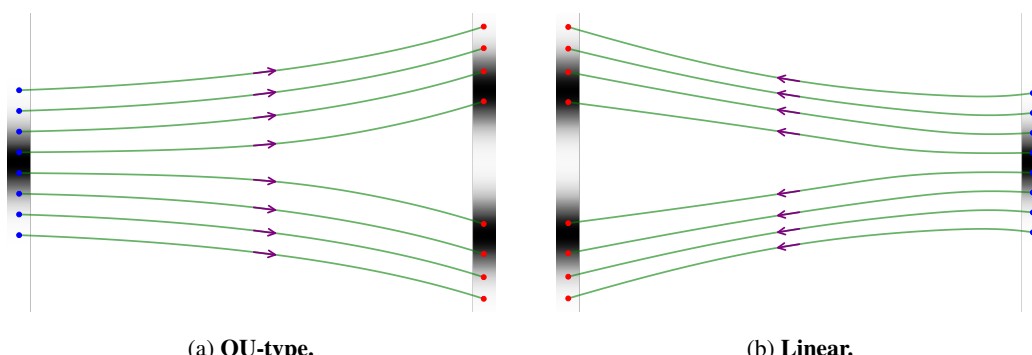

(a) **OU-type.**        (b) **Linear.**

Figure 10: **Comparison of two optimal Probability-Flow ODE trajectories on 1D data.** Starting from identical initial noise distributions and noise points, we apply two distinct transport types—OU-type and Linear—to analyze their trajectories. The results show that both types successfully converge to the same target distribution (a bimodal Gaussian) and accurately match the *same* target data points, despite following different ODE paths.

*Conclusion.* Combining the above estimates yields

$$\left|\partial_x \log p_\sigma(x)\right| \le A + B\left(|x| + M\,\sigma\sqrt{\tfrac{2}{\pi}}\right) = \left[A + B\,M\,\sigma\sqrt{\tfrac{2}{\pi}}\right] + B\,|x|.$$

Thus one may set

$$A(\sigma) = A + B\,M\,\sigma\sqrt{\tfrac{2}{\pi}}, \quad B(\sigma) = B,$$

and the lemma follows. $\qquad\square$

---

**Theorem 3 (Monotonicity and uniqueness of the 1D probability-flow map) .** *Let $p_0(x)$ be a probability density on $\mathbb{R}$ satisfying the linear-growth bound*

$$\left|\partial_x \log p_0(x)\right| \;\le\; A + B\,|x|, \qquad A, B < \infty, \quad \forall x \in \mathbb{R}.$$

*Let $z \sim \mathcal{N}(0,1)$ be independent of $x_0$, and let $\alpha, \gamma : [0,1] \to \mathbb{R}$ be $C^1$ functions with*

$$\alpha(0) = 0,\; \alpha(1) = 1, \quad \gamma(0) = 1,\; \gamma(1) = 0, \quad \gamma(t) \ne 0 \;\forall t \in (0,1).$$

*Define the forward process*

$$x_t \;=\; \alpha(t)\,z \;+\; \gamma(t)\,x_0, \qquad t \in [0,1],$$

*so that $x_0 \sim p_0$ and $x_1 \sim \mathcal{N}(0,1)$. Let $p_t$ denote the density of $x_t$. By Lem. 3, the velocity field:*

$$v(x,t) = \frac{\gamma'(t)}{\gamma(t)}\,x \;-\; \left[\alpha(t)\,\alpha'(t) \;-\; \frac{\gamma'(t)}{\gamma(t)}\,\alpha(t)^2\right]\partial_x \log p_t(x).$$

*Consider the backward ODE $\frac{\mathrm{d}}{\mathrm{d}t}\,x_t = v\big(x_t, t\big)$, Then for each $x_1 \in \mathbb{R}$ there is a unique $C^1$ solution $t \mapsto x_t(x_1)$ on $[0,1]$, and the map*

$$g(x_1) \;=\; x_0(x_1) \;=\; F_0^{-1}\big(F_1(x_1)\big)$$

*is strictly increasing on $\mathbb{R}$ and is the unique increasing transport pushing $p_1$ onto $p_0$.*

---

*Proof.* (1) Global existence and uniqueness. Since

$$x_t = \alpha(t)\,z + \gamma(t)\,x_0, \quad p_t = p_0 * \mathcal{N}\big(0, \alpha(t)^2\big),$$

standard Gaussian-convolution estimates imply $\left|\partial_x \log p_t(x)\right| \le A_t + B_t|x|$ for some continuous $A_t, B_t$ (cf., Lem. 8). Hence there exists $K < \infty$ such that

$$|v(x,t)| \;\le\; K\,(1 + |x|), \qquad \left|\partial_x v(x,t)\right| \;\le\; K, \quad \forall x \in \mathbb{R},\; t \in [0,1].$$

In particular $v$ is globally Lipschitz in $x$ (uniformly in $t$) and of linear growth. By the Lem. 6 together with Lem. 7 to prevent finite-time blow-up, the backward ODE admits for each $x_1$ a unique $C^1$ solution on $[0, 1]$.

(2) Conservation of the CDF. Let

$$F_t(x) = \int_{-\infty}^{x} p_t(u)\, \mathrm{d}u \quad \text{(the CDF of } p_t\text{)}.$$

Since $p_t$ satisfies the continuity equation $\partial_t p_t + \partial_x(v\, p_t) = 0$, along any characteristic $t \mapsto x_t$ one computes

$$\frac{\mathrm{d}}{\mathrm{d}t} F_t(x_t) = \int_{-\infty}^{x_t} \partial_t p_t(u)\, \mathrm{d}u + p_t(x_t)\, \frac{\mathrm{d}x_t}{\mathrm{d}t} = -\big[v\, p_t\big]_{-\infty}^{x_t} + p_t(x_t)\, v(x_t, t) = 0,$$

using $\lim_{u \to -\infty} p_t(u) = 0$. Hence $F_t(x_t) = F_1(x_1)$ for all $t \in [0, 1]$.

(3) Quantile representation. Evaluating at $t = 0$ gives

$$F_0\big(x_0(x_1)\big) = F_1(x_1).$$

Since $F_0 \colon \mathbb{R} \to (0, 1)$ is strictly increasing and onto, it has an inverse $F_0^{-1}$, and thus

$$x_0(x_1) = F_0^{-1}\big(F_1(x_1)\big).$$

(4) Monotonicity and uniqueness. If $x_1 < y_1$ then $F_1(x_1) < F_1(y_1)$, so

$$g(x_1) = F_0^{-1}\big(F_1(x_1)\big) < F_0^{-1}\big(F_1(y_1)\big) = g(y_1),$$

showing $g$ is strictly increasing. In one dimension the strictly increasing transport between two given laws is unique, so $g$ is the unique increasing map pushing $p_1$ onto $p_0$. A case study presented in Fig. 10 validates this theorem, considering the specific schedules discussed in Rem. 5 and Rem. 3. $\square$

> **Lemma 9 (Monotone transport from Gaussian to $P$).** *Let $Z \sim N(0, 1)$ be a standard normal random variable and let $X$ be a random variable with distribution $P$ on $\mathbb{R}$, having cumulative distribution function (CDF) $F_P$. Define*
>
> $$\Phi(z) = \Pr[Z \le z], \qquad F_P^{-1}(u) = \inf\{x : F_P(x) \ge u\},\ u \in (0, 1).$$
>
> *Then there exists a non-decreasing continuous function $g(z) = F_P^{-1}\big(\Phi(z)\big)$ such that $g(Z) \overset{d}{=} X$ if and only if $P$ has no atoms (i.e. $F_P$ is continuous). Moreover, if $F_P$ is strictly increasing then $g$ is unique.*

*Proof. Existence.* Since $\Phi : \mathbb{R} \to (0, 1)$ is continuous and strictly increasing, the random variable

$$U = \Phi(Z)$$

is distributed uniformly on $(0, 1)$. Hence for any $x \in \mathbb{R}$,

$$\Pr\big(F_P^{-1}(U) \le x\big) = \Pr\big(U \le F_P(x)\big) = F_P(x),$$

so $F_P^{-1}(U)$ has distribution $P$. The quantile function $F_P^{-1}$ is non-decreasing and, by standard results on generalized inverses (see e.g. Billingsley, *Probability and Measure*), is continuous on $(0, 1)$ if and only if $F_P$ is continuous. Therefore

$$g(z) = F_P^{-1}\big(\Phi(z)\big)$$

is non-decreasing and continuous exactly when $F_P$ is continuous, and in that case $g(Z) \overset{d}{=} X$.

*Necessity.* Suppose $P$ has an atom at $x_0$, i.e. $\Pr[X = x_0] = p > 0$. If there were a continuous non-decreasing $g$ with $g(Z) \overset{d}{=} X$, then to produce a point-mass $p$ at $x_0$ it would have to be constant on a set of positive $\Pr$-mass in the continuous law of $Z$. But continuity of $g$ then forces it to be constant on a strictly larger interval, yielding a mass $> p$ at $x_0$, a contradiction. Thus $F_P$ must be continuous.

*Uniqueness.* Let $g_1, g_2$ be two continuous non-decreasing functions with $g_i(Z) \overset{d}{=} P$. Define for $u \in (0, 1)$

$$h_i(u) = g_i\big(\Phi^{-1}(u)\big), \qquad i = 1, 2.$$

Each $h_i$ is continuous, non-decreasing, and pushes $\mathrm{Unif}(0, 1)$ onto $P$. When $F_P$ is strictly increasing, its quantile $F_P^{-1}$ is the unique such map (classical uniqueness of quantile functions for atomless laws). Hence $h_1 \equiv h_2 \equiv F_P^{-1}$ on $(0, 1)$, and therefore $g_1 \equiv g_2$ on $\mathbb{R}$. $\square$

### F.1.3 Learning Objective as $\lambda \to 1$

> **Lemma 10 ($L^p$-estimate for the difference of two absolutely continuous functions) .** *Let $I = [a, b]$ be a compact interval and $(E, \|\cdot\|)$ a Banach space. Suppose $f, g : I \to E$ are absolutely continuous with Bochner–integrable derivatives $f', g'$. Fix $1 \le p \le \infty$. Then*
>
> $$\|f - g\|_{L^p(I;E)} \le (b-a)^{1/p} \|f(a) - g(a)\| + \int_a^b (b-s)^{1/p} \|f'(s) - g'(s)\| \, ds,$$
>
> *where for $p = \infty$ one interprets $(b-s)^{1/p} = 1$. Moreover, if $1 < p < \infty$ and $p'$ denotes the conjugate exponent $1/p + 1/p' = 1$, then by Hölder's inequality one further deduces*
>
> $$\|f - g\|_{L^p(I;E)} \le (b-a)^{1/p} \|f(a) - g(a)\| + \left(\tfrac{p-1}{p}\right)^{1/p'} (b-a) \|f' - g'\|_{L^p(I;E)}.$$

*Proof.* Since $f$ and $g$ are absolutely continuous on $[a, b]$, the Fundamental Theorem of Calculus in the Bochner setting gives, for each $t \in [a, b]$,

$$f(t) - g(t) = \big(f(a) - g(a)\big) + \int_a^t \big(f'(s) - g'(s)\big) \, ds.$$

Set $X(s) = f'(s) - g'(s)$. Then for every $t \in [a, b]$,

$$\|f(t) - g(t)\| \le \|f(a) - g(a)\| + \left\|\int_a^t X(s) \, ds\right\|.$$

We now distinguish two cases.

*Case 1: $1 \le p < \infty$.* Taking the $L^p$–norm in the variable $t$ over $[a, b]$ and applying Minkowski's integral inequality for Bochner integrals yields

$$\|f - g\|_{L_t^p} \le \|f(a) - g(a)\| \, \|1\|_{L^p([a,b])} + \left\|\int_a^t X(s) \, ds\right\|_{L_t^p}$$

$$= (b-a)^{1/p} \|f(a) - g(a)\| + \left(\int_a^b \left\|\int_a^t X(s) \, ds\right\|^p dt\right)^{1/p}$$

$$\le (b-a)^{1/p} \|f(a) - g(a)\| + \int_a^b \left\|1_{[s,b]}(\cdot) \, X(s)\right\|_{L_t^p} ds.$$

Here we have written $\int_a^t X(s) \, ds = \int_a^b 1_{[a,t]}(s) \, X(s) \, ds$ and used the fact that

$$\left\|1_{[s,b]}(t)\right\|_{L_t^p} = \left(\int_a^b 1_{[s,b]}(t) \, dt\right)^{1/p} = (b-s)^{1/p}.$$

Hence

$$\|f - g\|_{L^p(I;E)} \le (b-a)^{1/p} \|f(a) - g(a)\| + \int_a^b (b-s)^{1/p} \|X(s)\| \, ds,$$

which is the claimed $L^p$–estimate.

*Case 2: $p = \infty$.* Taking the essential supremum in $t \in [a, b]$ in the pointwise bound $\|f(t) - g(t)\| \le \|f(a) - g(a)\| + \int_a^t \|X(s)\| \, ds$ gives immediately

$$\|f - g\|_{L^\infty(I;E)} \le \|f(a) - g(a)\| + \int_a^b \|X(s)\| \, ds,$$

which agrees with the above formula when $(b-s)^{1/p} = 1$.

*Refinement for $1 < p < \infty$.* Let $p'$ be the conjugate exponent, $1/p + 1/p' = 1$. Applying Hölder's inequality to the integral $\int_a^b (b-s)^{1/p} \|X(s)\| \, ds$ gives

$$\int_a^b (b-s)^{1/p} \|X(s)\| \, ds \le \left(\int_a^b (b-s)^{p'/p} \, ds\right)^{1/p'} \left(\int_a^b \|X(s)\|^p \, ds\right)^{1/p}.$$

Since $p'/p = 1/(p-1)$, a direct computation yields

$$\int_a^b (b-s)^{p'/p}\, ds = \int_0^{b-a} u^{1/(p-1)}\, du = \frac{p-1}{p}\,(b-a)^{p'}.$$

Hence

$$\left(\int_a^b (b-s)^{p'/p}\, ds\right)^{1/p'} = \left(\tfrac{p-1}{p}\right)^{1/p'} (b-a),$$

and we arrive at

$$\int_a^b (b-s)^{1/p}\, \|X(s)\|\, ds \le \left(\tfrac{p-1}{p}\right)^{1/p'} (b-a)\, \|X\|_{L^p(I;E)}.$$

Combining this with the previous display completes the proof of the refined estimate. $\qquad\square$

---

**Lemma 11 (Uniqueness of absolutely continuous functions) .** *Let $I = [a, b]$ be a compact interval and $(E, \|\cdot\|)$ a Banach space. Suppose $f, g : I \to E$ are absolutely continuous with Bochner–integrable derivatives $f', g'$. If*

$$f(a) = g(a) \quad \text{and} \quad f'(t) = g'(t) \quad \text{for almost every } t \in I,$$

*then $f(t) = g(t)$ for all $t \in I$.*

---

*Proof.* Apply Lem. 10 (the $L^p$–estimate for differences) in the case $p = \infty$. Since in this case one has

$$(b-s)^{1/p} = 1, \quad \|f(a) - g(a)\| = 0, \quad \|f'(s) - g'(s)\| = 0 \text{ a.e.,}$$

the conclusion of Lem. 10 reads

$$\| f - g \|_{L^\infty(I;E)} \le \|f(a) - g(a)\| + \int_a^b \|f'(s) - g'(s)\|\, ds = 0.$$

Hence $\|f - g\|_{L^\infty(I;E)} = 0$, which means

$$\sup_{t \in I} \|f(t) - g(t)\| = 0,$$

so $f(t) = g(t)$ for every $t \in I$. $\qquad\square$

---

**Theorem 4 (Pathwise consistency via zero total derivative) .** *Let $p(\mathbf{x})$ be a data distribution on $\mathbb{R}^d$, and let $\mathbf{z} \sim \mathcal{N}(\mathbf{0}, \mathbf{I}_d)$ be independent of $\mathbf{x}$. Let $\alpha, \gamma : [0, 1] \to \mathbb{R}$ be $C^1$ scalar functions satisfying*

$$\alpha(0) = 0,\ \alpha(1) = 1, \quad \gamma(0) = 1,\ \gamma(1) = 0, \quad \gamma(t) \ne 0\ \forall t \in (0, 1).$$

*Define the forward process*

$$\mathbf{x}_t = \alpha(t)\, \mathbf{z} + \gamma(t)\, \mathbf{x}, \quad t \in [0, 1],$$

*so that $\mathbf{x}_0 = \mathbf{x} \sim p(\mathbf{x})$ and $\mathbf{x}_1 = \mathbf{z} \sim \mathcal{N}(0, I)$. Let $p_t$ be the law of $\mathbf{x}_t$. By Lem. 3 the corresponding Probability Flow ODE is*

$$\mathbf{v}(\mathbf{x}_t, t) = \frac{\mathrm{d}}{\mathrm{d}t}\mathbf{x}_t = \frac{\gamma'(t)}{\gamma(t)}\mathbf{x}_t - \left[\alpha(t)\,\alpha'(t) - \frac{\gamma'(t)}{\gamma(t)}\alpha(t)^2\right] \nabla_{\mathbf{x}_t} \log p_t(\mathbf{x}_t).$$

*Given any point $\mathbf{x}_t$, define*

$$\mathbf{g}(\mathbf{x}_t, t) = \mathbf{x}_0 = \mathbf{x}_t + \int_t^0 \mathbf{v}(\mathbf{x}_u, u)\, \mathrm{d}u.$$

*Let $(\mathbf{z}, \mathbf{x}) \sim p(\mathbf{x}) \otimes \mathcal{N}(0, I)$ and $t \sim \mathrm{Unif}[0, 1]$ be all mutually independent. Write $\mathbb{E}_{(\mathbf{z}, \mathbf{x})}$ for expectation over $(\mathbf{z}, \mathbf{x})$ and $\mathbb{E}_{(\mathbf{z}, \mathbf{x}), t}$ for expectation over $(\mathbf{z}, \mathbf{x})$ and $t$. Suppose*

$$\mathbb{E}_{(\mathbf{z}, \mathbf{x})}\big\|\mathbf{f}(\mathbf{x}_0, 0) - \mathbf{g}(\mathbf{x}_0, 0)\big\| = 0, \quad \mathbb{E}_{(\mathbf{z}, \mathbf{x}), t}\left\|\frac{\mathrm{d}}{\mathrm{d}t}\,\mathbf{f}(\mathbf{x}_t, t)\right\| = 0.$$

*Then*

$$\mathbb{E}_{(\mathbf{z},\mathbf{x}),t}\big\|\boldsymbol{f}(\mathbf{x}_t,t) - \boldsymbol{g}(\mathbf{x}_t,t)\big\| = 0.$$

*Proof.* Fix a draw $(\mathbf{z},\mathbf{x})$. Along its forward trajectory $\mathbf{x}_t = \alpha(t)\mathbf{z} + \gamma(t)\mathbf{x}$, define the two curves

$$f(t) = \boldsymbol{f}\big(\mathbf{x}_t,t\big), \quad g(t) = \boldsymbol{g}\big(\mathbf{x}_t,t\big).$$

We check the hypotheses of Lem. 11 for $f, g : [0, 1] \to \mathbb{R}^d$.

*Absolute continuity.* Since $\boldsymbol{f}$ is $C^1$ in $(\mathbf{x}, t)$ and $t \mapsto \mathbf{x}_t$ is $C^1$, the composition $f(t) = \boldsymbol{f}(\mathbf{x}_t, t)$ is absolutely continuous, with

$$f'(t) = \frac{\mathrm{d}}{\mathrm{d}t}\,\boldsymbol{f}\big(\mathbf{x}_t,t\big), \quad \text{existing a.e.}$$

Also

$$g(t) = \mathbf{x}_t + \int_t^0 \mathbf{v}(\mathbf{x}_u, u)\,\mathrm{d}u = \mathbf{x}_0 - \int_0^t \mathbf{v}(\mathbf{x}_u, u)\,\mathrm{d}u$$

is the sum of a $C^1$ function and an absolutely continuous integral, hence itself absolutely continuous.

*Coincidence of initial values.* From $\mathbb{E}_{(\mathbf{z},\mathbf{x})}\|\boldsymbol{f}(\mathbf{x}_0, 0) - \boldsymbol{g}(\mathbf{x}_0, 0)\| = 0$ we get $\boldsymbol{f}(\mathbf{x}_0, 0) = \boldsymbol{g}(\mathbf{x}_0, 0)$ almost surely, so $f(0) = g(0)$ for almost every $(\mathbf{z}, \mathbf{x})$.

*Coincidence of derivatives a.e.* By Tonelli–Fubini,

$$0 = \mathbb{E}_{(\mathbf{z},\mathbf{x}),t}\Big\|\tfrac{\mathrm{d}}{\mathrm{d}t}\boldsymbol{f}(\mathbf{x}_t,t)\Big\| = \int \left(\int_0^1 \Big\|\tfrac{\mathrm{d}}{\mathrm{d}t}\boldsymbol{f}(\mathbf{x}_t,t)\Big\|\,\mathrm{d}t\right)\,\mathrm{d}\mathbb{P}(\mathbf{z},\mathbf{x}).$$

Hence for almost every $(\mathbf{z}, \mathbf{x})$, $\int_0^1 \|\partial_t \boldsymbol{f}(\mathbf{x}_t, t)\|\,\mathrm{d}t = 0$, which forces $\partial_t \boldsymbol{f}(\mathbf{x}_t, t) = 0$ for almost all $t$. Thus

$$f'(t) = 0 \quad \text{for a.e. } t \in [0, 1].$$

On the other hand

$$g'(t) = \frac{\mathrm{d}\mathbf{x}_t}{\mathrm{d}t} - \mathbf{v}(\mathbf{x}_t, t) = \mathbf{v}(\mathbf{x}_t, t) - \mathbf{v}(\mathbf{x}_t, t) = 0, \quad \forall t \in [0, 1].$$

*Conclusion by uniqueness.* We have shown $f, g$ are absolutely continuous, $f(0) = g(0)$, and $f'(t) = g'(t)$ for almost every $t$. By Lem. 11, $f(t) = g(t)$ for all $t \in [0, 1]$ (almost surely in $(\mathbf{z}, \mathbf{x})$). Hence $\boldsymbol{f}(\mathbf{x}_t, t) = \boldsymbol{g}(\mathbf{x}_t, t)$ a.s., and taking expectation yields $\mathbb{E}_{(\mathbf{z},\mathbf{x}),t}\big\|\boldsymbol{f}(\mathbf{x}_t,t) - \boldsymbol{g}(\mathbf{x}_t,t)\big\| = 0$. $\quad\square$

**Remark 7 (Consistency-training loss).** *By Thm. 4, to enforce $\boldsymbol{f}(\mathbf{x}_t, t) \approx \boldsymbol{g}(\mathbf{x}_t, t) = \mathbf{x}_0$ along the PF–ODE flow, we suggests two equivalent training objectives:*

*1.* Continuous PDE-residual loss

$$\mathcal{L}_{\mathrm{PDE}} = \mathbb{E}_{t,\mathbf{x}_t}\Big\|\partial_t \boldsymbol{f}(\mathbf{x}_t,t) + v(\mathbf{x}_t,t)\cdot\nabla_{\mathbf{x}_t}\boldsymbol{f}(\mathbf{x}_t,t)\Big\|^2.$$

*2.* Finite-difference consistency loss

$$\mathcal{L}_{\mathrm{cons}} = \mathbb{E}_{t,\mathbf{x}_0,\mathbf{z}}\Big\|\boldsymbol{f}\big(\mathbf{x}_{t+\Delta t},\, t + \Delta t\big) - \boldsymbol{f}\big(\mathbf{x}_t,t\big)\Big\|^2,$$

*where $\mathbf{x}_t = \alpha(t)\mathbf{z} + \gamma(t)\mathbf{x}_0$ and similarly for $\mathbf{x}_{t+\Delta t}$.*

*Proof.* We begin from the requirement that $\boldsymbol{f}(\mathbf{x}_t, t)$ remain constant along the flow:

$$\frac{\mathrm{d}}{\mathrm{d}t}\,\boldsymbol{f}(\mathbf{x}_t,t) = \big(\partial_t + \mathbf{v}(\mathbf{x}_t,t)\cdot\nabla_{\mathbf{x}_t}\big)\,\boldsymbol{f}(\mathbf{x}_t,t) = \partial_t\boldsymbol{f}(\mathbf{x}_t,t) + \underbrace{\frac{\mathrm{d}\mathbf{x}_t}{\mathrm{d}t}}_{= \mathbf{v}(\mathbf{x}_t,t)}\cdot\nabla_{\mathbf{x}_t}\boldsymbol{f}(\mathbf{x}_t,t) = 0.$$

This is exactly the linear transport PDE

$$(\partial_t + \mathbf{v}\cdot\nabla)\,\boldsymbol{f}(\mathbf{x},t) = 0.$$

To train a network $\boldsymbol{f}$ to satisfy it, one may minimize the $L^2$-residual over the joint law of $t$ and $\mathbf{x}_t$, yielding

$$\mathcal{L}_{\mathrm{PDE}} = \mathbb{E}_{t,\mathbf{x}_t}\left\|\partial_t \boldsymbol{f}(\mathbf{x}_t, t) + \mathbf{v}(\mathbf{x}_t, t)\cdot\nabla_{\mathbf{x}_t}\boldsymbol{f}(\mathbf{x}_t, t)\right\|^2.$$

In practice, computing the spatial gradient $\nabla_{\mathbf{x}_t}\boldsymbol{f}$ can be expensive. Instead, we use a small time increment $\Delta t$ and the finite-difference approximation

$$\boldsymbol{f}(\mathbf{x}_{t+\Delta t},\, t + \Delta t) - \boldsymbol{f}(\mathbf{x}_t, t) \;\approx\; \Delta t\left[\partial_t \boldsymbol{f} + \mathbf{v}\cdot\nabla\boldsymbol{f}\right](\mathbf{x}_t, t).$$

Squaring and taking expectations over $t, \mathbf{x}_0, \mathbf{z}$ then yields the discrete consistency loss

$$\mathcal{L}_{\mathrm{cons}} = \mathbb{E}_{t,\mathbf{x}_0,\mathbf{z}}\left\|\boldsymbol{f}\big(\mathbf{x}_{t+\Delta t},\, t + \Delta t\big) - \boldsymbol{f}\big(\mathbf{x}_t, t\big)\right\|^2.$$

This completes the derivation of both forms of the consistency-training objective. $\qquad\square$

Recall that $(\mathbf{z}, \mathbf{x}) \sim p(\mathbf{z}, \mathbf{x})$ is a pair of latent and data variables (typically independent), and let $t \in [0, 1]$. We have four differentiable scalar functions $\alpha, \gamma, \hat{\alpha}, \hat{\gamma}: [0, 1] \to \mathbb{R}$, the *noisy interpolant* $\mathbf{x}_t = \alpha(t)\,\mathbf{z} + \gamma(t)\,\mathbf{x}$ and $\boldsymbol{F}_t = \boldsymbol{F}_{\boldsymbol{\theta}}(\mathbf{x}_t, t)$. We define the $\mathbf{x}$- and $\mathbf{z}$-prediction functions by

$$\boldsymbol{f}^{\mathbf{x}}(\boldsymbol{F}_t, \mathbf{x}_t, t) = \frac{\alpha(t)\,\boldsymbol{F}_t \;-\; \hat{\alpha}(t)\,\mathbf{x}_t}{\alpha(t)\,\hat{\gamma}(t)\;-\;\hat{\alpha}(t)\,\gamma(t)}, \quad \text{and} \quad \boldsymbol{f}^{\mathbf{z}}(\boldsymbol{F}_t, \mathbf{x}_t, t) = \frac{\hat{\gamma}(t)\,\mathbf{x}_t\;-\;\gamma(t)\,\boldsymbol{F}_t}{\alpha(t)\,\hat{\gamma}(t)\;-\;\hat{\alpha}(t)\,\gamma(t)}.$$

Since

$$
\begin{aligned}
\boldsymbol{f}^{\mathbf{x}}(\boldsymbol{F}_0, \mathbf{x}_0, 0) &= \frac{\alpha(0)\cdot\boldsymbol{F}_{\boldsymbol{\theta}}(\mathbf{x}_0, 0) - \hat{\alpha}(0)\cdot\mathbf{x}_0}{\alpha(0)\cdot\hat{\gamma}(0) - \hat{\alpha}(0)\cdot\gamma(0)} \\
&= \frac{0\cdot\boldsymbol{F}_{\boldsymbol{\theta}}(\mathbf{x}_0, 0) - \hat{\alpha}(0)\cdot\mathbf{x}_0}{0\cdot\hat{\gamma}(0) - \hat{\alpha}(0)\cdot 1} \\
&= \frac{\mathbf{0} - \hat{\alpha}(0)\cdot\mathbf{x}_0}{0 - \hat{\alpha}(0)} \\
&= \mathbf{x}_0
\end{aligned}
$$

$\boldsymbol{f}^{\mathbf{x}}$ satisfies the boundary condition of consistency models (Song et al., 2023) and Thm. 4. To better understand the unified loss, let's analyze a bit further. For simplicity we use the notation $\boldsymbol{f}_{\boldsymbol{\theta}}(\mathbf{x}_t, t) := \boldsymbol{f}^{\mathbf{x}}(\boldsymbol{F}_{\boldsymbol{\theta}}(\mathbf{x}_t, t), \mathbf{x}_t, t)$, the training objective is then equal to

$$\mathcal{L}(\boldsymbol{\theta}) = \mathbb{E}_{t,(\mathbf{z},\mathbf{x})}\left[\frac{1}{\hat{\omega}(t)}\|\boldsymbol{f}_{\boldsymbol{\theta}}(\mathbf{x}_t, t) - \boldsymbol{f}_{\boldsymbol{\theta}^-}(\mathbf{x}_{\lambda t}, \lambda t)\|_2^2\right].$$

Let $\phi_t(\mathbf{x})$ be the solution of the PF-ODE determined by the velocity field $\mathbf{v}^*(\mathbf{x}_t, t) = \mathbb{E}_{(\mathbf{z},\mathbf{x})|\mathbf{x}_t}\left[\mathbf{v}^{(\mathbf{z},\mathbf{x})}(\mathbf{x}_t, t)|\mathbf{x}_t\right]$ (where $\mathbf{v}^{(\mathbf{z},\mathbf{x})}(\mathbf{y}, t) = \alpha'(t)\mathbf{z} + \gamma'(t)\mathbf{x}$) and an initial value $\mathbf{x}$ at time $t = 0$. Define $\boldsymbol{g}_{\boldsymbol{\theta}}(\mathbf{x}, t) := \boldsymbol{f}_{\boldsymbol{\theta}}(\phi_t(\mathbf{x}), t)$ that moves along the solution trajectory. When $\lambda \to 1$, the gradient of the loss tends to

$$
\begin{aligned}
\lim_{\lambda\to 1}\nabla_{\boldsymbol{\theta}}\frac{\mathcal{L}(\boldsymbol{\theta})}{2(1-\lambda)} &= \mathbb{E}_t\left[\frac{t}{\hat{\omega}(t)}\cdot\mathbb{E}_{(\mathbf{z},\mathbf{x})}\lim_{\lambda\to 1}\langle\frac{\boldsymbol{f}_{\boldsymbol{\theta}}(\mathbf{x}_t, t) - \boldsymbol{f}_{\boldsymbol{\theta}}(\mathbf{x}_{\lambda t}, \lambda t)}{t - \lambda t}, \nabla_{\boldsymbol{\theta}}\boldsymbol{f}_{\boldsymbol{\theta}}(\mathbf{x}_t, t)\rangle\right] \\
&= \mathbb{E}_t\left[\frac{t}{\hat{\omega}(t)}\cdot\mathbb{E}_{(\mathbf{z},\mathbf{x})}\langle\frac{\mathrm{d}\boldsymbol{f}_{\boldsymbol{\theta}}(\mathbf{x}_t, t)}{\mathrm{d}t}, \nabla_{\boldsymbol{\theta}}\boldsymbol{g}_{\boldsymbol{\theta}}(\phi_t^{-1}(\mathbf{x}_t), t)\rangle\right]
\end{aligned}
$$

The inner expectation can be computed as:

$$
\begin{aligned}
&\mathbb{E}_{(\mathbf{z},\mathbf{x}),\mathbf{x}_t}\langle\frac{\mathrm{d}\boldsymbol{f}_{\boldsymbol{\theta}}(\mathbf{x}_t, t)}{\mathrm{d}t}, \nabla_{\boldsymbol{\theta}}\boldsymbol{g}_{\boldsymbol{\theta}}(\phi_t^{-1}(\mathbf{x}_t), t)\rangle \\
&= \mathbb{E}_{(\mathbf{z},\mathbf{x}),\mathbf{x}_t}\langle\partial_1\boldsymbol{f}_{\boldsymbol{\theta}}(\mathbf{x}_t, t)\cdot\mathbf{v}^{(\mathbf{z},\mathbf{x})}(\mathbf{x}_t, t) + \partial_2\boldsymbol{f}_{\boldsymbol{\theta}}(\mathbf{x}_t, t), \nabla_{\boldsymbol{\theta}}\boldsymbol{g}_{\boldsymbol{\theta}}(\phi_t^{-1}(\mathbf{x}_t), t)\rangle \\
&= \mathbb{E}_{(\mathbf{z},\mathbf{x}),\mathbf{x}_t}\langle\partial_1\boldsymbol{f}_{\boldsymbol{\theta}}(\mathbf{x}_t, t)\cdot(\alpha'(t)\mathbf{z} + \gamma'(t)\mathbf{x}) + \partial_2\boldsymbol{f}_{\boldsymbol{\theta}}(\mathbf{x}_t, t), \nabla_{\boldsymbol{\theta}}\boldsymbol{g}_{\boldsymbol{\theta}}(\phi_t^{-1}(\mathbf{x}_t), t)\rangle \\
&= \mathbb{E}_{\mathbf{x}_t}\left[\mathbb{E}_{(\mathbf{z},\mathbf{x})|\mathbf{x}_t}\langle\partial_1\boldsymbol{f}_{\boldsymbol{\theta}}(\mathbf{x}_t, t)\cdot(\alpha'(t)\mathbf{z} + \gamma'(t)\mathbf{x}) + \partial_2\boldsymbol{f}_{\boldsymbol{\theta}}(\mathbf{x}_t, t), \nabla_{\boldsymbol{\theta}}\boldsymbol{g}_{\boldsymbol{\theta}}(\phi_t^{-1}(\mathbf{x}_t), t)\rangle\right] \\
&= \mathbb{E}_{\mathbf{x}_t}\langle\partial_1\boldsymbol{f}_{\boldsymbol{\theta}}(\mathbf{x}_t, t)\cdot\mathbb{E}_{(\mathbf{z},\mathbf{x})|\mathbf{x}_t}[\alpha'(t)\mathbf{z} + \gamma'(t)\mathbf{x}|\mathbf{x}_t] + \partial_2\boldsymbol{f}_{\boldsymbol{\theta}}(\mathbf{x}_t, t), \nabla_{\boldsymbol{\theta}}\boldsymbol{g}_{\boldsymbol{\theta}}(\phi_t^{-1}(\mathbf{x}_t), t)\rangle \\
&= \mathbb{E}_{\mathbf{x}_t}\langle\partial_1\boldsymbol{f}_{\boldsymbol{\theta}}(\mathbf{x}_t, t)\cdot\mathbf{v}^*(\mathbf{x}_t, t) + \partial_2\boldsymbol{f}_{\boldsymbol{\theta}}(\mathbf{x}_t, t), \nabla_{\boldsymbol{\theta}}\boldsymbol{g}_{\boldsymbol{\theta}}(\phi_t^{-1}(\mathbf{x}_t), t)\rangle \\
&= \mathbb{E}_{\mathbf{x}_t}\langle\partial_2\boldsymbol{g}_{\boldsymbol{\theta}}(\phi_t^{-1}(\mathbf{x}_t), t), \nabla_{\boldsymbol{\theta}}\boldsymbol{g}_{\boldsymbol{\theta}}(\phi_t^{-1}(\mathbf{x}_t), t)\rangle \\
&= \nabla_{\boldsymbol{\theta}}\mathbb{E}_{\phi_t^{-1}(\mathbf{x}_t)}\frac{1}{2}\|\boldsymbol{g}_{\boldsymbol{\theta}}(\phi_t^{-1}(\mathbf{x}_t), t) - \boldsymbol{g}_{\boldsymbol{\theta}^-}(\phi_t^{-1}(\mathbf{x}_t), t) + \partial_2\boldsymbol{g}_{\boldsymbol{\theta}}(\phi_t^{-1}(\mathbf{x}_t), t)\|_2^2
\end{aligned}
$$

Thus from the perspective of gradient, when $\lambda \to 1$ the training objective is equivalent to

$$\mathbb{E}_{\phi_t^{-1}(\mathbf{x}_t),t} \left[ \frac{t}{\hat{\omega}(t)} \cdot \|\boldsymbol{g_\theta}(\phi_t^{-1}(\mathbf{x}_t),t) - \boldsymbol{g_{\theta^-}}(\phi_t^{-1}(\mathbf{x}_t),t) + \partial_2 \boldsymbol{g_\theta}(\phi_t^{-1}(\mathbf{x}_t),t)\|_2^2 \right]$$

which naturally leads to the solution $\boldsymbol{g_\theta}(\mathbf{x},t) = \mathbf{x}$ (since $\boldsymbol{g_\theta}(\mathbf{x},0) \equiv \mathbf{x}$), or equivalently $\boldsymbol{f}^{\mathbf{x}}(\boldsymbol{F_{\theta^*}}(\mathbf{x}_t,t),\mathbf{x}_t,t) = \boldsymbol{f_{\theta^*}}(\mathbf{x}_t,t) = \phi_t^{-1}(\mathbf{x}_t)$, that is the definition of consistency function.

### F.1.4 ANALYSIS ON THE OPTIMAL SOLUTION FOR $\lambda \in [0,1]$

Below we provide some examples to illustrate the property of the optimal solution for the unified loss by considering some simple cases of data distribution.
(for simplicity define $\boldsymbol{f_\theta}(\mathbf{x}_t,t) = \boldsymbol{f}^{\mathbf{x}}(\boldsymbol{F_\theta}(\mathbf{x}_t,t),\mathbf{x}_t,t)$)
Assume $\mathbf{x} \sim \mathcal{N}(\boldsymbol{\mu},\Sigma)$. For $r < t$ the conditional mean

$$\mathbb{E}\left[\mathbf{x}_r | \mathbf{x}_t\right] = \gamma(r)\boldsymbol{\mu} + (\gamma(r)\gamma(t)\Sigma + \alpha(r)\alpha(t)\mathbf{I})\left(\gamma(t)^2\Sigma + \alpha(t)^2\mathbf{I}\right)^{-1}(\mathbf{x}_t - \gamma(t)\boldsymbol{\mu}),$$

denote

$$\mathbf{K}(r,t) := (\gamma(r)\gamma(t)\Sigma + \alpha(r)\alpha(t)\mathbf{I})\left(\gamma(t)^2\Sigma + \alpha(t)^2\mathbf{I}\right)^{-1},$$

using above equations we can get the optimal solution for diffusion model:

$$\boldsymbol{f}_{\boldsymbol{\theta^*}}^{\mathrm{DM}}(\mathbf{x}_t,t) = \mathbb{E}\left[\mathbf{x}|\mathbf{x}_t\right] = \boldsymbol{\mu} + \mathbf{K}(0,t)(\mathbf{x}_t - \gamma_t\boldsymbol{\mu}).$$

Now consider a series of $t$ together: $t = t_T > t_{T-1} > \ldots > t_1 > t_0 \approx 0$. This series could be obtained by $t_{j-1} = \lambda \cdot t_j, j = T, \ldots, 0$, for instance. With an abuse of notation, denote $\mathbf{x}_{t_j}$ as $\mathbf{x}_j$ and $\alpha(t_j)$ as $\alpha_j$, $\gamma(t_j)$ as $\gamma_j$. Since $t_0 \approx 0, \mathbf{x}_0 \approx \mathbf{x}$, we could conclude the trained model $\boldsymbol{f_{\theta^*}}(\mathbf{x}_1,t_1) = \mathbb{E}_{\mathbf{x}|\mathbf{x}_1}\left[\mathbf{x}|\mathbf{x}_1\right]$, and concequently

$$\boldsymbol{f_{\theta^*}}(\mathbf{x}_{j+1},t_{j+1}) = \mathbb{E}_{\mathbf{x}_j|\mathbf{x}_{j+1}}\left[\boldsymbol{f_{\theta^*}}(\mathbf{x}_j,t_j)|\mathbf{x}_{j+1}\right], j = 1, \ldots, T-1.$$

Using the property of the conditional expectation, we have $\mathbb{E}_{\mathbf{x}_j}\left[\boldsymbol{f_{\theta^*}}(\mathbf{x}_j,t_j)\right] = \mathbb{E}_{\mathbf{x}}\left[\mathbf{x}\right], \forall j$. Using the expressions above we have

$$\boldsymbol{f_{\theta^*}}(\mathbf{x}_1,t_1) = \boldsymbol{\mu} + \mathbf{K}(t_0,t_1)(\mathbf{x}_1 - \gamma_1\boldsymbol{\mu})$$

and

$$\boldsymbol{f_{\theta^*}}(\mathbf{x}_j,t_j) = \boldsymbol{\mu} + \left[\prod_{k=1}^{j} \mathbf{K}(t_{k-1},t_k)\right] \cdot (\mathbf{x}_t - \gamma_t\boldsymbol{\mu}), j = 2, \ldots, T$$

Further denote $c_j = \prod_{k=1}^{j} \alpha_{k-1}\alpha_k + \gamma_{k-1}\gamma_k$ and assume $\Sigma = \mathbf{I}, \alpha = \sin(t), \gamma(t) = \cos(t)$. For appropriate choice of the partition scheme (e.g. even or geometric), the coefficient $c_j$ can converge as $T$ grows. For instance, when evenly partitioning the interval $[0,t]$, we have:

$$\lim_{T\to\infty} c(t) = \lim_{T\to\infty} \prod_{k=1}^{T} \alpha_{k-1}\alpha_k + \gamma_{k-1}\gamma_k = \lim_{T\to\infty} (\cos(\frac{t}{T}))^T = 1.$$

Thus the trained model can be viewed as an interpolant between the consistency model($\lambda \to 1$ or $T \to \infty$) and the diffusion model($\lambda \to 0$ or $T \to 1$):

$$\boldsymbol{f_{\theta^*}}(\mathbf{x}_t,t) = \boldsymbol{\mu} + c(t)(\mathbf{x}_t - \gamma(t)\boldsymbol{\mu}),$$

$$\boldsymbol{f}_{\boldsymbol{\theta^*}}^{\mathrm{CM}}(\mathbf{x}_t,t) = \boldsymbol{\mu} + (\mathbf{x}_t - \gamma(t)\boldsymbol{\mu}),$$

$$\boldsymbol{f}_{\boldsymbol{\theta^*}}^{\mathrm{DM}}(\mathbf{x}_t,t) = \boldsymbol{\mu} + \gamma(t)(\mathbf{x}_t - \gamma(t)\boldsymbol{\mu}).$$

The expression of $\boldsymbol{f}_{\boldsymbol{\theta^*}}^{\mathrm{CM}}$ can be obtained by first compute the velocity field $\mathbf{v}^*(\mathbf{x}_t,t) = \mathbb{E}\left[\alpha'(t)\mathbf{z} + \gamma'(t)\mathbf{x}|\mathbf{x}_t\right] = \gamma'(t)\boldsymbol{\mu}$ then solve the initial value problem of ODE to get $\mathbf{x}(0)$.
The above optimal solution can be possibly obtained by training. For example if we set the parameterizition as $\boldsymbol{f_\theta}(\mathbf{x}_t,t) = (1 - \gamma_t c_t)\boldsymbol{\theta} + c_t\mathbf{x}_t$, the gradient of the loss can be computed as (let $r = \lambda \cdot t$):

$$\nabla_{\boldsymbol{\theta}}\|\boldsymbol{f_\theta}(\mathbf{x}_t,t) - \boldsymbol{f_{\theta^-}}(\mathbf{x}_r,r)\|_2^2 = 2(1 - \gamma_t c_t)\left[(\alpha_t\gamma_t - \alpha_r\gamma_r)\mathbf{z} + (\gamma_r c_r - \gamma_t c_t)(\boldsymbol{\theta} - \mathbf{x})\right],$$

$$\nabla_{\boldsymbol{\theta}}\mathbb{E}_{\mathbf{z},\mathbf{x}}\|\boldsymbol{f}_{\boldsymbol{\theta}}(\mathbf{x}_t,t)-\boldsymbol{f}_{\boldsymbol{\theta}^-}(\mathbf{x}_r,r)\|_2^2=2(1-\gamma_t c_t)(\gamma_r c_r-\gamma_t c_t)(\boldsymbol{\theta}-\boldsymbol{\mu}),$$

$$\nabla_{\boldsymbol{\theta}}\mathcal{L}(\boldsymbol{\theta})=\mathbb{E}_t\,\frac{2(1-\gamma_t c_t)(\gamma_r c_r-\gamma_t c_t)}{\hat{\omega}(t)}(\boldsymbol{\theta}-\boldsymbol{\mu})$$

$$=C(\boldsymbol{\theta}-\boldsymbol{\mu}),\,C=\mathbb{E}_t\,\frac{2(1-\gamma_t c_t)(\gamma_r c_r-\gamma_t c_t)}{\hat{\omega}(t)}\,.$$

Use gradient descent to update $\boldsymbol{\theta}$ during training:

$$\frac{d\boldsymbol{\theta}(s)}{ds}=-\nabla_{\boldsymbol{\theta}}\mathcal{L}(\boldsymbol{\theta})=-C(\boldsymbol{\theta}-\boldsymbol{\mu})\,.$$

The generalization loss thus evolves as:

$$\frac{d\|\boldsymbol{\theta}(s)-\boldsymbol{\mu}\|^2}{ds}=\langle\boldsymbol{\theta}(s)-\boldsymbol{\mu},\frac{d\boldsymbol{\theta}(s)}{ds}\rangle$$

$$=\langle\boldsymbol{\theta}(s)-\boldsymbol{\mu},-C(\boldsymbol{\theta}(s)-\boldsymbol{\mu})\rangle$$

$$=-C\|\boldsymbol{\theta}(s)-\boldsymbol{\mu}\|^2\,,$$

$$\implies\|\boldsymbol{\theta}(s)-\boldsymbol{\mu}\|^2=\|\boldsymbol{\theta}(0)-\boldsymbol{\mu}\|^2 e^{-Cs}\,.$$

### F.1.5 SURROGATE OBJECTIVE FOR UNIFIED OBJECTIVE

*Proof for Thm. 1.* For brevity, we omit the expectation operator $\mathbb{E}$ in the following derivation.

**Step 1. Omit the expectation operator.**

$$l(\boldsymbol{\theta})=\frac{1}{\hat{\omega}(t)}\big\|\boldsymbol{f}^{\mathbf{x}}(\boldsymbol{F}_{\boldsymbol{\theta}}(\mathbf{x}_t,t),\mathbf{x}_t,t)-\boldsymbol{f}^{\mathbf{x}}(\boldsymbol{F}_{\boldsymbol{\theta}^-}(\mathbf{x}_{\lambda t},\lambda t),\mathbf{x}_{\lambda t},\lambda t)\big\|_2^2.$$

**Step 2. Gradient of the loss.**

$$\nabla_\theta l(\theta)=\frac{1}{\hat{\omega}(t)}\big\langle\nabla_\theta\boldsymbol{f}^{\mathbf{x}}(\boldsymbol{F}_{\boldsymbol{\theta}}(\mathbf{x}_t,t),\mathbf{x}_t,t),\Delta f\big\rangle,\tag{10}$$

where

$$\Delta f=\boldsymbol{f}^{\mathbf{x}}(\boldsymbol{F}_{\boldsymbol{\theta}}(\mathbf{x}_t,t),\mathbf{x}_t,t)-\boldsymbol{f}^{\mathbf{x}}(\boldsymbol{F}_{\boldsymbol{\theta}^-}(\mathbf{x}_{\lambda t},\lambda t),\mathbf{x}_{\lambda t},\lambda t)$$

$$=[\mathbf{x}_t-t\cdot\boldsymbol{F}_{\boldsymbol{\theta}}(\mathbf{x}_t,t)]-[\mathbf{x}_{\lambda t}-\lambda t\cdot\boldsymbol{F}_{\boldsymbol{\theta}^-}(\mathbf{x}_{\lambda t},\lambda t)]$$

$$=(t-\lambda t)\big[(\mathbf{z}-\mathbf{x})-\boldsymbol{F}_{\boldsymbol{\theta}}(\mathbf{x}_t,t)\big]+\lambda t\cdot\big(\boldsymbol{F}_{\boldsymbol{\theta}^-}(\mathbf{x}_{\lambda t},\lambda t)-\boldsymbol{F}_{\boldsymbol{\theta}}(\mathbf{x}_t,t)\big).\tag{11}$$

Also,

$$\nabla_\theta\boldsymbol{f}^{\mathbf{x}}(\boldsymbol{F}_{\boldsymbol{\theta}}(\mathbf{x}_t,t),\mathbf{x}_t,t)=-t\cdot\nabla_\theta\boldsymbol{F}_{\boldsymbol{\theta}}(\mathbf{x}_t,t).\tag{12}$$

**Step 3. Substitute (11) and (12) into (10).**

$$\nabla_\theta l(\theta)=\frac{t(t-\lambda t)}{\hat{\omega}(t)}\big\langle\nabla_\theta\boldsymbol{F}_{\boldsymbol{\theta}}(\mathbf{x}_t,t),\boldsymbol{F}_{\boldsymbol{\theta}}(\mathbf{x}_t,t)-(\mathbf{z}-\mathbf{x})\big\rangle$$

$$+\frac{t\lambda t}{\hat{\omega}(t)}\big\langle\nabla_\theta\boldsymbol{F}_{\boldsymbol{\theta}}(\mathbf{x}_t,t),\boldsymbol{F}_{\boldsymbol{\theta}}(\mathbf{x}_t,t)-\boldsymbol{F}_{\boldsymbol{\theta}^-}(\mathbf{x}_{\lambda t},\lambda t)\big\rangle.$$

**Step 3. Use $\hat{\omega}(t)=t^2\cdot(1-\lambda)$.**

$$\nabla_\theta l(\theta)=\nabla_\theta\big\|\boldsymbol{F}_{\boldsymbol{\theta}}(\mathbf{x}_t,t)-(\mathbf{z}-\mathbf{x})\big\|_2^2+\frac{\lambda}{1-\lambda}\,\nabla_\theta\big\|\boldsymbol{F}_{\boldsymbol{\theta}}(\mathbf{x}_t,t)-\boldsymbol{F}_{\boldsymbol{\theta}^-}(\mathbf{x}_{\lambda t},\lambda t)\big\|_2^2.$$

This matches exactly the gradient of $\mathcal{G}(\theta)$. Hence,

$$\nabla_{\boldsymbol{\theta}}\mathcal{L}(\boldsymbol{\theta})=\nabla_{\boldsymbol{\theta}}\mathcal{G}(\boldsymbol{\theta}).$$

$\square$

**Theorem 5 (Surrogate Loss for Unified Objective of General Case) .** *Define*

$$A(t) := \frac{\alpha(t)}{D(t)} - \frac{\alpha(\lambda t)}{D(\lambda t)}, \quad B(t) := \frac{\alpha(t)}{D(t)}, \quad C(t) := \frac{\alpha(t)}{2D(t)}, \quad D(t) := \alpha(t)\hat{\gamma}(t) - \hat{\alpha}(t)\gamma(t).$$

*Let the* surrogate loss *be*

$$\mathcal{G}(\boldsymbol{\theta}) = \mathbb{E}_{\mathbf{z},\mathbf{x},t} \left[ \frac{C(t)}{\hat{\omega}(t)} \Big( A(t) \underbrace{\left\| \boldsymbol{F}_{\theta}(\mathbf{x}_t, t) - \mathbf{z}_t \right\|_2^2}_{\textit{Mean Velocity Alignment}} \right.$$

$$\left. + B(\lambda t) \underbrace{\left\| (\boldsymbol{F}_{\theta}(\mathbf{x}_t, t) - \boldsymbol{F}_{\theta^-}(\mathbf{x}_{\lambda t}, \lambda t)) - (\mathbf{z}_t - \mathbf{z}_{\lambda t}) \right\|_2^2}_{\textit{Velocity Difference Consistency}} \Big) \right] \tag{13}$$

*Then, for all* $\boldsymbol{\theta}$,

$$\nabla_{\boldsymbol{\theta}} \mathcal{L}(\boldsymbol{\theta}) = \nabla_{\boldsymbol{\theta}} \mathcal{G}(\boldsymbol{\theta}).$$

*Proof for Thm. 5.* For brevity, we omit the expectation operator $\mathbb{E}$ and weights $\hat{\omega}(t)$ in the following derivation.

$$l(\boldsymbol{\theta}) = \| \boldsymbol{f}^{\mathbf{x}}(\boldsymbol{F}_{\boldsymbol{\theta}}(\mathbf{x}_t, t), \mathbf{x}_t, t) - \boldsymbol{f}^{\mathbf{x}}(\boldsymbol{F}_{\boldsymbol{\theta}^-}(\mathbf{x}_{\lambda t}, \lambda t), \mathbf{x}_{\lambda t}, \lambda t) \|_2^2 \tag{14}$$

$$\nabla_{\boldsymbol{\theta}} l(\boldsymbol{\theta}) = \big\langle \nabla_{\theta} \boldsymbol{f}^{\mathbf{x}}(\boldsymbol{F}_{\boldsymbol{\theta}}(\mathbf{x}_t, t), \mathbf{x}_t, t), \Delta \boldsymbol{f}^{\mathbf{x}}(\boldsymbol{F}_{\boldsymbol{\theta}}(\mathbf{x}_t, t), \mathbf{x}_t, t) \big\rangle \tag{15}$$

In the following, we compute $\nabla_{\theta} \boldsymbol{f}^{\mathbf{x}}(\boldsymbol{F}_{\boldsymbol{\theta}}(\mathbf{x}_t, t), \mathbf{x}_t, t)$ and $\Delta \boldsymbol{f}^{\mathbf{x}}(\boldsymbol{F}_{\boldsymbol{\theta}}(\mathbf{x}_t, t), \mathbf{x}_t, t)$, respectively.

$$\nabla_{\theta} \boldsymbol{f}^{\mathbf{x}}(\boldsymbol{F}_{\boldsymbol{\theta}}(\mathbf{x}_t, t), \mathbf{x}_t, t) = \nabla_{\theta} \left( \frac{\alpha(t) \cdot \boldsymbol{F}_{\boldsymbol{\theta}}(\mathbf{x}_t, t) - \hat{\alpha}(t) \cdot \mathbf{x}_t}{\alpha(t) \cdot \hat{\gamma}(t) - \hat{\alpha}(t) \cdot \gamma(t)} \right)$$

$$= \frac{\alpha(t)}{\alpha(t) \cdot \hat{\gamma}(t) - \hat{\alpha}(t) \cdot \gamma(t)} \cdot \nabla_{\theta} \boldsymbol{F}_{\boldsymbol{\theta}}(\mathbf{x}_t, t)$$

$$= \frac{\alpha(t)}{D(t)} \cdot \nabla_{\theta} \mathbf{F}_{\boldsymbol{\theta}}(\mathbf{x}_t, t) \tag{16}$$

$$\Delta \boldsymbol{f}^{\mathbf{x}}(\boldsymbol{F}_{\boldsymbol{\theta}}(\mathbf{x}_t, t), \mathbf{x}_t, t) = \frac{\alpha(t) \cdot \boldsymbol{F}_{\boldsymbol{\theta}}(\mathbf{x}_t, t) - \hat{\alpha}(t) \cdot \mathbf{x}_t}{\alpha(t) \cdot \hat{\gamma}(t) - \hat{\alpha}(t) \cdot \gamma(t)} - \frac{\alpha(\lambda t) \cdot \boldsymbol{F}_{\boldsymbol{\theta}^-}(\mathbf{x}_{\lambda t}, \lambda t) - \hat{\alpha}(\lambda t) \cdot \mathbf{x}_{\lambda t}}{\alpha(\lambda t) \cdot \hat{\gamma}(\lambda t) - \hat{\alpha}(\lambda t) \cdot \gamma(\lambda t)}$$

$$= \frac{\alpha(t)}{D(t)} \cdot \mathbf{F}_{\boldsymbol{\theta}}(\mathbf{x}_t, t) - \frac{\hat{\alpha}(t)}{D(t)} \cdot \mathbf{x}_t - \frac{\alpha(\lambda t)}{D(\lambda t)} \cdot \mathbf{F}_{\boldsymbol{\theta}^-}(\mathbf{x}_{\lambda t}, \lambda t) + \frac{\hat{\alpha}(\lambda t)}{D(\lambda t)} \cdot \mathbf{x}_{\lambda t} \tag{17}$$

Now, we consider to replace $\mathbf{x}_t$ and $\mathbf{x}_{\lambda t}$ with $\mathbf{z}_t$ and $\mathbf{z}_{\lambda t}$. Let's consider a general term (Remind $\mathbf{z}_t = \hat{\alpha}(t) \cdot \mathbf{z} + \hat{\gamma}(t) \cdot \mathbf{x}$):

$$\frac{\hat{\alpha}(s) \cdot \mathbf{x}_s}{D(s)} = \frac{\hat{\alpha}(s) \cdot \alpha(s) \cdot \mathbf{z} + \hat{\alpha}(s) \cdot \gamma(s) \cdot \mathbf{x}}{\alpha(s) \cdot \hat{\gamma}(s) - \hat{\alpha}(s) \cdot \gamma(s)}$$

$$= \frac{\alpha(s) \cdot \mathbf{z}_s - (\alpha(s) \cdot \hat{\gamma}(s) - \hat{\alpha}(s) \cdot \gamma(s)) \cdot \mathbf{x}}{\alpha(s) \cdot \hat{\gamma}(s) - \hat{\alpha}(s) \cdot \gamma(s)}$$

$$= \frac{\alpha(s) \cdot \mathbf{z}_s}{\alpha(s) \cdot \hat{\gamma}(s) - \hat{\alpha}(s) \cdot \gamma(s)} - \mathbf{x}$$

$$= \frac{\alpha(s) \cdot \mathbf{z}_s}{D(s)} - \mathbf{x} \tag{18}$$

Therefore, by substituting (18) into (17), we get:

$$\Delta \boldsymbol{f}^{\mathbf{x}}(\boldsymbol{F_\theta}(\mathbf{x}_t, t), \mathbf{x}_t, t)$$

$$= \frac{\alpha(t)}{D(t)} \cdot \boldsymbol{F_\theta}(\mathbf{x}_t, t) - \frac{\hat{\alpha}(t)}{D(t)} \cdot \mathbf{x}_t - \frac{\alpha(\lambda t)}{D(\lambda t)} \cdot \boldsymbol{F_{\theta^-}}(\mathbf{x}_{\lambda t}, \lambda t) + \frac{\hat{\alpha}(\lambda t)}{D(\lambda t)} \cdot \mathbf{x}_{\lambda t}$$

$$= \frac{\alpha(t)}{D(t)} \cdot \mathbf{F_\theta}(\mathbf{x}_t, t) - \frac{\alpha(t)}{D(t)} \cdot \mathbf{z}_t - \frac{\alpha(\lambda t)}{D(\lambda t)} \cdot \mathbf{F_{\theta^-}}(\mathbf{x}_{\lambda t}, \lambda t) + \frac{\alpha(\lambda t)}{D(\lambda t)} \cdot \mathbf{z}_{\lambda t}$$

$$= \left( \frac{\alpha(t)}{D(t)} - \frac{\alpha(\lambda t)}{D(\lambda t)} \right) \cdot (\mathbf{F_\theta}(\mathbf{x}_t, t) - \mathbf{z}_t) + \frac{\alpha(\lambda t)}{D(\lambda t)} \cdot (\mathbf{F_\theta}(\mathbf{x}_t, t) - \mathbf{F_{\theta^-}}(\mathbf{x}_{\lambda t}, \lambda t)) + \frac{\alpha(\lambda t)}{D(\lambda t)} \cdot (\mathbf{z}_{\lambda t} - \mathbf{z}_t)$$

$$(19)$$

By substituting (19) and (16) into (15), we get:

$$\nabla_\theta l(\boldsymbol{\theta}) = \left\langle \nabla_{\boldsymbol{\theta}} \boldsymbol{f}^{\mathbf{x}}(\boldsymbol{F_\theta}(\mathbf{x}_t, t), \mathbf{x}_t, t), \Delta \boldsymbol{f}^{\mathbf{x}}(\boldsymbol{F_\theta}(\mathbf{x}_t, t), \mathbf{x}_t, t) \right\rangle$$

$$= \left\langle \frac{\alpha(t)}{D(t)} \cdot \nabla_{\boldsymbol{\theta}} \boldsymbol{F_\theta}(\mathbf{x}_t, t), \left( \frac{\alpha(t)}{D(t)} - \frac{\alpha(\lambda t)}{D(\lambda t)} \right) \cdot (\mathbf{F_\theta}(\mathbf{x}_t, t) - \mathbf{z}_t) \right\rangle$$

$$+ \left\langle \frac{\alpha(t)}{D(t)} \cdot \nabla_{\boldsymbol{\theta}} \boldsymbol{F_\theta}(\mathbf{x}_t, t), \frac{\alpha(\lambda t)}{D(\lambda t)} \cdot (\mathbf{F_\theta}(\mathbf{x}_t, t) - \mathbf{F_{\theta^-}}(\mathbf{x}_{\lambda t}, \lambda t)) \right\rangle$$

$$+ \left\langle \frac{\alpha(t)}{D(t)} \cdot \nabla_{\boldsymbol{\theta}} \mathbf{F_\theta}(\mathbf{x}_t, t), \frac{\alpha(\lambda t)}{D(\lambda t)} \cdot (\mathbf{z}_{\lambda t} - \mathbf{z}_t) \right\rangle$$

$$= \frac{\alpha(t)}{2D(t)} \cdot \left( \frac{\alpha(t)}{D(t)} - \frac{\alpha(\lambda t)}{D(\lambda t)} \right) \cdot \nabla_{\boldsymbol{\theta}} \| \boldsymbol{F_\theta}(\mathbf{x}_t, t) - \mathbf{z}_t \|_2^2$$

$$+ \frac{\alpha(t)}{2D(t)} \cdot \frac{\alpha(\lambda t)}{D(\lambda t)} \cdot \nabla_{\boldsymbol{\theta}} \| (\boldsymbol{F_\theta}(\mathbf{x}_t, t) - \boldsymbol{F_{\theta^-}}(\mathbf{x}_{\lambda t}, \lambda t)) - (\mathbf{z}_t - \mathbf{z}_{\lambda t}) \|_2^2$$

$$\square$$

### F.1.6 CLOSED-FORM SOLUTION ANALYSIS FOR $\lambda \in [0, 1]$

**Theorem 6 (optimal solution of surrogate objective in linear case ($\alpha(t) = t$, $\gamma(t) = 1 - t$)).**
*Under Assump. 1, let's consider a surrogate objective*

$$\mathcal{G}(\boldsymbol{\theta}) = \mathbb{E}_{\mathbf{z}, \mathbf{x}, t} \left[ \left\| \boldsymbol{F}_{\boldsymbol{\theta}}(\mathbf{x}_t, t) - \mathbf{z}_t \right\|_2^2 + \frac{\lambda}{1 - \lambda} \cdot \left\| \boldsymbol{F}_{\boldsymbol{\theta}}(\mathbf{x}_t, t) - \boldsymbol{F}_{\boldsymbol{\theta}^-}(\mathbf{x}_{\lambda t}, \lambda t) \right\|_2^2 \right], \quad (20)$$

*where $\mathbf{x}_t = t \cdot \mathbf{z} + (1 - t) \cdot \mathbf{x}$, $\mathbf{z}_t = \mathbf{z} - \mathbf{x}$, $0 < \lambda < 1$. Then the optimal solution of the surrogate objective is:*

$$\boldsymbol{F}_{\boldsymbol{\theta}^*}(\mathbf{x}_t, t) = \mathbb{E}_{\mathbf{z}, \mathbf{x}} \left[ \mathbf{z}_t + \frac{\lambda}{1 - \lambda} \cdot \boldsymbol{F}_{\boldsymbol{\theta}^-}(\mathbf{x}_{\lambda t}, \lambda t) \mid \mathbf{x}_t \right] \quad (21)$$

*Proof.* For any fixed $t$ and $\mathbf{x}_t$, the objective only depends on the value $\boldsymbol{F}_{\boldsymbol{\theta}}(\mathbf{x}_t, t)$ at this specific input. Therefore, we may optimize pointwise over $\boldsymbol{F}_{\boldsymbol{\theta}}(\mathbf{x}_t, t)$.
Define for shorthand:

$$\mathbf{f} := \boldsymbol{F}_{\boldsymbol{\theta}}(\mathbf{x}_t, t), \qquad \mathbf{a} := \mathbf{z}_t, \qquad \mathbf{b} := \boldsymbol{F}_{\boldsymbol{\theta}^-}(\mathbf{x}_{\lambda t}, \lambda t), \qquad \mathbf{f}^* := \mathbb{E}_{\mathbf{z}, \mathbf{x}} \left[ \mathbf{a} + \frac{\lambda}{1 - \lambda} \cdot \mathbf{b} \mid \mathbf{x}_t \right].$$

Then the surrogate objective specialized at $(\mathbf{x}_t, t)$ is

$$\begin{aligned}
\mathcal{G}_t(\mathbf{f}) &= \mathbb{E}_{\mathbf{z}, \mathbf{x}} \left[ \|\mathbf{f} - \mathbf{a}\|_2^2 + \frac{\lambda}{1 - \lambda} \|\mathbf{f} - \mathbf{b}\|_2^2 \,\Big|\, \mathbf{x}_t \right] \\
&= \mathbb{E}_{\mathbf{z}, \mathbf{x}} \left[ \|\mathbf{f} - \mathbf{f}^* + \mathbf{f}^* - \mathbf{a}\|_2^2 + \frac{\lambda}{1 - \lambda} \|\mathbf{f} - \mathbf{f}^* + \mathbf{f}^* - \mathbf{b}\|_2^2 \,\Big|\, \mathbf{x}_t \right] \\
&= \mathbb{E}_{\mathbf{z}, \mathbf{x}} \left[ \|\mathbf{f} - \mathbf{f}^*\|_2^2 + \|\mathbf{f}^* - \mathbf{a}\|_2^2 + 2\langle \mathbf{f} - \mathbf{f}^*, \mathbf{f}^* - \mathbf{a} \rangle \,\Big|\, \mathbf{x}_t \right] \\
&\quad + \frac{\lambda}{1 - \lambda} \cdot \mathbb{E}_{\mathbf{z}, \mathbf{x}} \left[ \|\mathbf{f} - \mathbf{f}^*\|_2^2 + \|\mathbf{f}^* - \mathbf{b}\|_2^2 + 2\langle \mathbf{f} - \mathbf{f}^*, \mathbf{f}^* - \mathbf{b} \rangle \,\Big|\, \mathbf{x}_t \right] \\
&\geq \mathbb{E}_{\mathbf{z}, \mathbf{x}} \left[ \|\mathbf{f} - \mathbf{f}^*\|_2^2 + \|\mathbf{f}^* - \mathbf{a}\|_2^2 \,\Big|\, \mathbf{x}_t \right] + \frac{\lambda}{1 - \lambda} \cdot \mathbb{E}_{\mathbf{z}, \mathbf{x}} \left[ \|\mathbf{f} - \mathbf{f}^*\|_2^2 + \|\mathbf{f}^* - \mathbf{b}\|_2^2 \,\Big|\, \mathbf{x}_t \right] \\
&= \mathbb{E}_{\mathbf{z}, \mathbf{x}} \left[ \|\mathbf{f} - \mathbf{f}^*\|_2^2 \,\Big|\, \mathbf{x}_t \right] + \mathcal{G}_t(\mathbf{f}^*)
\end{aligned}$$

In the following, we need to show that:

$$\mathbb{E}_{\mathbf{z}, \mathbf{x}} \left[ \langle \mathbf{f} - \mathbf{f}^*, \mathbf{f}^* - \mathbf{a} \rangle \,\Big|\, \mathbf{x}_t \right] + \frac{\lambda}{1 - \lambda} \cdot \mathbb{E}_{\mathbf{z}, \mathbf{x}} \left[ \langle \mathbf{f} - \mathbf{f}^*, \mathbf{f}^* - \mathbf{b} \rangle \,\Big|\, \mathbf{x}_t \right] = 0 \quad (22)$$

The (22) always holds because:

$$\begin{aligned}
&\mathbb{E}_{\mathbf{z}, \mathbf{x}} \left[ \langle \mathbf{f} - \mathbf{f}^*, \mathbf{f}^* - \mathbf{a} \rangle \,\Big|\, \mathbf{x}_t \right] + \frac{\lambda}{1 - \lambda} \cdot \mathbb{E}_{\mathbf{z}, \mathbf{x}} \left[ \langle \mathbf{f} - \mathbf{f}^*, \mathbf{f}^* - \mathbf{b} \rangle \,\Big|\, \mathbf{x}_t \right] \\
&= \langle \mathbf{f} - \mathbf{f}^*, \mathbf{f}^* - \mathbb{E}_{\mathbf{z}, \mathbf{x}}[\mathbf{a} \mid \mathbf{x}_t] \rangle + \langle \mathbf{f} - \mathbf{f}^*, \mathbf{f}^* - \frac{\lambda}{1 - \lambda} \cdot \mathbb{E}_{\mathbf{z}, \mathbf{x}}[\mathbf{b} \mid \mathbf{x}_t] \rangle \\
&= \langle \mathbf{f} - \mathbf{f}^*, \mathbf{f}^* - \mathbb{E}_{\mathbf{z}, \mathbf{x}}[\mathbf{a} + \frac{\lambda}{1 - \lambda} \cdot \mathbf{b} \mid \mathbf{x}_t] \rangle \\
&= 0
\end{aligned}$$

which completes the proof. In addition, the optimal solution of the surrogate objective also satisfies that the gradient of the surrogate objective is zero, i.e.

$$\nabla_{\boldsymbol{\theta}} \mathcal{G}(\boldsymbol{\theta}^*) = 0$$

$\square$

Because it is hard to directly analyze the behavior of the optimal solution of the surrogate objective, we analyze the gradient of the surrogate objective instead.

**Proposition 1 (Gradient of surrogate objective in linear case) .** *The gradient of the surrogate objective* (20) *is:*

$$\nabla_{\boldsymbol{\theta}}\mathcal{G}(\boldsymbol{\theta}) = 2\mathbb{E}[\langle \nabla_{\boldsymbol{\theta}}\boldsymbol{F}_{\boldsymbol{\theta}}(\mathbf{x}_t, t), (\boldsymbol{F}_{\boldsymbol{\theta}}(\mathbf{x}_t, t) - \mathbf{z}_t) + \frac{\lambda}{1-\lambda} \cdot (\boldsymbol{F}_{\boldsymbol{\theta}}(\mathbf{x}_t, t) - \boldsymbol{F}_{\boldsymbol{\theta}^-}(\mathbf{x}_{\lambda t}, \lambda t))\rangle] \quad (23)$$

*Proof.*

$$\nabla_{\boldsymbol{\theta}}\mathcal{G}(\boldsymbol{\theta}) = \nabla_{\boldsymbol{\theta}}\mathbb{E}\left[\left\|\boldsymbol{F}_{\boldsymbol{\theta}}(\mathbf{x}_t, t) - \mathbf{z}_t\right\|_2^2 + \frac{\lambda}{1-\lambda} \cdot \left\|\boldsymbol{F}_{\boldsymbol{\theta}}(\mathbf{x}_t, t) - \boldsymbol{F}_{\boldsymbol{\theta}^-}(\mathbf{x}_{\lambda t}, \lambda t)\right\|_2^2\right] \quad (24)$$

$$= 2\mathbb{E}\left[\langle \nabla_{\boldsymbol{\theta}}\boldsymbol{F}_{\boldsymbol{\theta}}(\mathbf{x}_t, t), \boldsymbol{F}_{\boldsymbol{\theta}}(\mathbf{x}_t, t) - \mathbf{z}_t\rangle + \frac{\lambda}{1-\lambda} \cdot \langle \nabla_{\boldsymbol{\theta}}\boldsymbol{F}_{\boldsymbol{\theta}}(\mathbf{x}_t, t), \boldsymbol{F}_{\boldsymbol{\theta}}(\mathbf{x}_t, t) - \boldsymbol{F}_{\boldsymbol{\theta}^-}(\mathbf{x}_{\lambda t}, \lambda t)\rangle\right] \quad (25)$$

$$= 2\mathbb{E}[\langle \nabla_{\boldsymbol{\theta}}\boldsymbol{F}_{\boldsymbol{\theta}}(\mathbf{x}_t, t), \frac{\boldsymbol{F}_{\boldsymbol{\theta}}(\mathbf{x}_t, t) - \lambda \cdot \boldsymbol{F}_{\boldsymbol{\theta}^-}(\mathbf{x}_{\lambda t}, \lambda t)}{1-\lambda} - \mathbf{z}_t\rangle] \quad (26)$$

$$= 2\mathbb{E}\left[\langle \nabla_{\boldsymbol{\theta}}\boldsymbol{F}_{\boldsymbol{\theta}}(\mathbf{x}_t, t), (\boldsymbol{F}_{\boldsymbol{\theta}}(\mathbf{x}_t, t) - \mathbf{z}_t) + \frac{\lambda}{1-\lambda} \cdot (\boldsymbol{F}_{\boldsymbol{\theta}}(\mathbf{x}_t, t) - \boldsymbol{F}_{\boldsymbol{\theta}^-}(\mathbf{x}_{\lambda t}, \lambda t))\rangle\right] \quad (27)$$

□

**Remark 8 (Behavior of the gradient) .** *The gradient term shows that $\boldsymbol{F}_{\boldsymbol{\theta}}(\mathbf{x}_t, t)$ is pushed towards the convex combination*

$$(\boldsymbol{F}_{\boldsymbol{\theta}}(\mathbf{x}_t, t) - \mathbf{z}_t) + \frac{\lambda}{1-\lambda} \cdot (\boldsymbol{F}_{\boldsymbol{\theta}}(\mathbf{x}_t, t) - \boldsymbol{F}_{\boldsymbol{\theta}^-}(\mathbf{x}_{\lambda t}, \lambda t))$$

*Therefore, $\lambda$ smoothly interpolates between two regimes:*

- *If $\lambda = 0$, the gradient reduces to standard flow matching gradient $\boldsymbol{F}_{\boldsymbol{\theta}}(\mathbf{x}_t, t) - \mathbf{z}_t$, which corresponds to the gradient of the multi-step objective.*
- *If $\lambda \to 1$, the gradient becomes $\boldsymbol{F}_{\boldsymbol{\theta}}(\mathbf{x}_t, t) - \mathbf{z}_t + t \cdot \frac{\mathrm{d}\boldsymbol{F}_{\boldsymbol{\theta}}(\mathbf{x}_t, t)}{\mathrm{d}t}$, where $\boldsymbol{F}_{\boldsymbol{\theta}}(\mathbf{x}_t, t)$ is the predicted velocity, $z_t$ is the target mean velocity, and $t \cdot \frac{\mathrm{d}\boldsymbol{F}_{\boldsymbol{\theta}}(\mathbf{x}_t, t)}{\mathrm{d}t}$ acts as a temporal consistency correction, encouraging the predicted velocity field to remain coherent across time. In other words, when $\boldsymbol{F}_{\boldsymbol{\theta}}(\mathbf{x}_t, t)$ becomes the mean velocity, there will need no correction term and the term $\frac{\mathrm{d}\boldsymbol{F}_{\boldsymbol{\theta}}(\mathbf{x}_t, t)}{\mathrm{d}t}$ constantly equals to zero.*
- *If $\lambda \in (0, 1)$, the gradient simultaneously enforces accuracy of the velocity prediction and strengthens the requirement that the velocity field be consistent across all time steps. As a result, the behavior increasingly resembles a few-step consistency model as $\lambda \to 1$ because the weights of the correction term becomes increasingly larger.*

**Theorem 7 (optimal solution of surrogate objective in General Case) .** *Under Assump. 1, let's define*

$$A(t) := \frac{\alpha(t)}{D(t)} - \frac{\alpha(\lambda t)}{D(\lambda t)}, \quad B(t) := \frac{\alpha(t)}{D(t)}, \quad C(t) := \frac{\alpha(t)}{2D(t)}, \quad D(t) := \alpha(t)\hat{\gamma}(t) - \hat{\alpha}(t)\gamma(t).$$

*Let the surrogate loss be*

$$\mathcal{G}(\boldsymbol{\theta}) = \mathbb{E}_{\mathbf{z}, \mathbf{x}, t}\left[\frac{C(t)}{\hat{\omega}(t)}\left(A(t)\left\|\boldsymbol{F}_{\theta}(\mathbf{x}_t, t) - \mathbf{z}_t\right\|_2^2\right.\right.$$

$$\left.\left. + B(\lambda t)\left\|(\boldsymbol{F}_{\theta}(\mathbf{x}_t, t) - \boldsymbol{F}_{\theta^-}(\mathbf{x}_{\lambda t}, \lambda t)) - (\mathbf{z}_t - \mathbf{z}_{\lambda t})\right\|_2^2\right)\right] \quad (28)$$

*where* $\mathbf{x}_t = \alpha(t) \cdot \mathbf{z} + \gamma(t) \cdot \mathbf{x}$, $\mathbf{z}_t = \hat{\alpha}(t) \cdot \mathbf{z} + \hat{\gamma}(t) \cdot \mathbf{x}$, $0 < \lambda < 1$. *Then, the optimal solution of the surrogate objective is:*

$$\boldsymbol{F}_{\boldsymbol{\theta}^*}(\mathbf{x}_t, t) \;=\; \mathbb{E}_{\mathbf{z}, \mathbf{x}}\left[\mathbf{z}_t + \frac{B(\lambda t)}{A(t) + B(\lambda t)}\left(\boldsymbol{F}_{\boldsymbol{\theta}^-}(\mathbf{x}_{\lambda t}, \lambda t) - \mathbf{z}_{\lambda t}\right) \,\Big|\, \mathbf{x}_t\right]. \tag{29}$$

*Proof.* The proof proceeds like the linear special case: for each fixed pair $(\mathbf{x}_t, t)$ we optimise pointwise over the vector value $\boldsymbol{F}_{\boldsymbol{\theta}}(\mathbf{x}_t, t)$.

For notational convenience define

$$\mathbf{f} := \boldsymbol{F}_{\boldsymbol{\theta}}(\mathbf{x}_t, t), \qquad \mathbf{a} := \mathbf{z}_t, \qquad \mathbf{a}_\lambda := \mathbf{z}_{\lambda t}, \qquad \mathbf{c} := \boldsymbol{F}_{\boldsymbol{\theta}^-}(\mathbf{x}_{\lambda t}, \lambda t).$$

Observe that

$$\left\|(\mathbf{f} - \mathbf{c}) - (\mathbf{a} - \mathbf{a}_\lambda)\right\|_2^2 = \left\|\mathbf{f} - (\mathbf{a} + \mathbf{c} - \mathbf{a}_\lambda)\right\|_2^2.$$

With this notation the pointwise contribution of the surrogate loss (omitting the common prefactor $1/\hat{\omega}(t)$) becomes

$$\mathcal{G}_t(\mathbf{f}) \;=\; C(t)A(t)\,\mathbb{E}\left[\|\mathbf{f} - \mathbf{a}\|_2^2 \mid \mathbf{x}_t\right] + C(t)B(\lambda t)\,\mathbb{E}\left[\|\mathbf{f} - (\mathbf{a} + \mathbf{c} - \mathbf{a}_\lambda)\|_2^2 \mid \mathbf{x}_t\right].$$

Define the weighted conditional mean

$$\mathbf{f}^* \;:=\; \mathbb{E}\left[\frac{A(t)\mathbf{a} + B(\lambda t)\left(\mathbf{a} + \mathbf{c} - \mathbf{a}_\lambda\right)}{A(t) + B(\lambda t)} \,\Bigg|\, \mathbf{x}_t\right]$$

which is well-defined under the assumptions on the coefficients. Expanding each squared term around $\mathbf{f}^*$ gives

$$\mathcal{G}_t(\mathbf{f}) = C(t)A(t)\,\mathbb{E}\left[\|\mathbf{f} - \mathbf{f}^*\|^2 + \|\mathbf{f}^* - \mathbf{a}\|^2 + 2\langle \mathbf{f} - \mathbf{f}^*, \mathbf{f}^* - \mathbf{a}\rangle \mid \mathbf{x}_t\right]$$
$$+ C(t)B(\lambda t)\,\mathbb{E}\big[\|\mathbf{f} - \mathbf{f}^*\|^2 + \|\mathbf{f}^* - (\mathbf{a} + \mathbf{c} - \mathbf{a}_\lambda)\|^2$$
$$+ 2\langle \mathbf{f} - \mathbf{f}^*, \mathbf{f}^* - (\mathbf{a} + \mathbf{c} - \mathbf{a}_\lambda)\rangle \mid \mathbf{x}_t\big].$$

Collecting terms yields

$$\mathcal{G}_t(\mathbf{f}) = C(t)\big(A(t) + B(\lambda t)\big)\mathbb{E}\left[\|\mathbf{f} - \mathbf{f}^*\|^2 \mid \mathbf{x}_t\right] + \mathcal{G}_t(\mathbf{f}^*) + 2\mathcal{I},$$

where the cross-term $\mathcal{I}$ equals

$$\mathcal{I} = \mathbb{E}\left[\langle \mathbf{f} - \mathbf{f}^*, \; C(t)A(t)\big(\mathbf{f}^* - \mathbf{a}\big) + C(t)B(\lambda t)\big(\mathbf{f}^* - (\mathbf{a} + \mathbf{c} - \mathbf{a}_\lambda)\big)\rangle \,\Big|\, \mathbf{x}_t\right].$$

By the definition of $\mathbf{f}^*$ as the conditional expectation of the weighted target

$$C(t)A(t)\mathbf{a} + C(t)B(\lambda t)\big(\mathbf{a} + \mathbf{c} - \mathbf{a}_\lambda\big),$$

it follows that the vector inside the inner product in $\mathcal{I}$ has zero conditional expectation:

$$C(t)A(t)\,\mathbb{E}[\mathbf{a} \mid \mathbf{x}_t] + C(t)B(\lambda t)\,\mathbb{E}[\mathbf{a} + \mathbf{c} - \mathbf{a}_\lambda \mid \mathbf{x}_t] = C(t)\big(A(t) + B(\lambda t)\big)\mathbf{f}^*.$$

Hence $\mathcal{I} = 0$, and we obtain the lower bound

$$\mathcal{G}_t(\mathbf{f}) \;\geq\; C(t)\big(A(t) + B(\lambda t)\big)\mathbb{E}\left[\|\mathbf{f} - \mathbf{f}^*\|^2 \mid \mathbf{x}_t\right] + \mathcal{G}_t(\mathbf{f}^*),$$

with equality at $\mathbf{f} = \mathbf{f}^*$. Therefore $\mathbf{f}^*$ minimises the pointwise objective, and we recover (29). $\square$

### F.1.7 UNIFIED TRAINING OBJECTIVE

**Theorem 8 .** *Let $\lambda \in (0,1)$ and define the scalar functions*

$$A(t) := B(t) - B(\lambda t), \qquad B(t) := \frac{\alpha(t)}{D(t)},$$

$$C(t) := \frac{\alpha(t)}{2D(t)}, \qquad D(t) := \alpha(t)\hat{\gamma}(t) - \hat{\alpha}(t)\gamma(t), \qquad \hat{\omega}(t) := C(t)\,A(t).$$

*For a pair $(\mathbf{z}, \mathbf{x}) \sim p(\mathbf{z}, \mathbf{x})$ and times $t, \lambda t$, we define the shorthand*

$$\Delta_{\boldsymbol{\theta}, \boldsymbol{\theta}^-}\, \boldsymbol{f}^{\mathbf{x}} := \boldsymbol{f}^{\mathbf{x}}\big(\boldsymbol{F}_{\boldsymbol{\theta}}(\mathbf{x}_t, t), \mathbf{x}_t, t\big) - \boldsymbol{f}^{\mathbf{x}}\big(\boldsymbol{F}_{\boldsymbol{\theta}^-}(\mathbf{x}_{\lambda t}, \lambda t), \mathbf{x}_{\lambda t}, \lambda t\big).$$

*Assume $\boldsymbol{\theta}$ and $\boldsymbol{\theta}^-$ are two different variables and equal in value. Now we define the three functionals:*

$$\mathcal{L}(\boldsymbol{\theta}) = \mathbb{E}_{\mathbf{z}, \mathbf{x}, t}\left[\frac{1}{\hat{\omega}(t)}\,\|\boldsymbol{f}^{\mathbf{x}}(\boldsymbol{F}_{\boldsymbol{\theta}}(\mathbf{x}_t, t), \mathbf{x}_t, t) - \boldsymbol{f}^{\mathbf{x}}(\boldsymbol{F}_{\boldsymbol{\theta}^-}(\mathbf{x}_{\lambda t}, \lambda t), \mathbf{x}_{\lambda t}, \lambda t)\|_2^2\right],$$

$$\mathcal{G}(\boldsymbol{\theta}) = \mathbb{E}_{\mathbf{z}, \mathbf{x}, t}\left[\left\|\boldsymbol{F}_{\boldsymbol{\theta}}(\mathbf{x}_t, t) - \mathbf{z}_t\right\|_2^2 + \frac{B(\lambda t)}{\hat{\omega}(t)}\left\|(\boldsymbol{F}_{\boldsymbol{\theta}}(\mathbf{x}_t, t) - \boldsymbol{F}_{\boldsymbol{\theta}^-}(\mathbf{x}_{\lambda t}, \lambda t)) - (\mathbf{z}_t - \mathbf{z}_{\lambda t})\right\|_2^2\right],$$

$$\mathcal{N}(\boldsymbol{\theta}) = \mathbb{E}_{\mathbf{z}, \mathbf{x}, t}\left[\frac{1}{2}\left\|\boldsymbol{F}_{\boldsymbol{\theta}}(\mathbf{x}_t, t) - \boldsymbol{F}_{\boldsymbol{\theta}^-}(\mathbf{x}_{\lambda t}, \lambda t) + 2 \cdot \frac{\Delta_{\boldsymbol{\theta}^-, \boldsymbol{\theta}^-}\, \boldsymbol{f}^{\mathbf{x}}}{A(t)}\right\|_2^2\right].$$

*Then, for all $\boldsymbol{\theta}$,*

$$\nabla_{\boldsymbol{\theta}}\mathcal{L}(\boldsymbol{\theta}) = \nabla_{\boldsymbol{\theta}}\mathcal{G}(\boldsymbol{\theta}) = \nabla_{\boldsymbol{\theta}}\mathcal{N}(\boldsymbol{\theta}).$$

*Proof.* The first equality $\nabla_{\boldsymbol{\theta}}\mathcal{L}(\boldsymbol{\theta}) = \nabla_{\boldsymbol{\theta}}\mathcal{G}(\boldsymbol{\theta})$ is straightforward by Thm. 5. Now, we prove the equality $\nabla_{\boldsymbol{\theta}}\mathcal{L}(\boldsymbol{\theta}) = \nabla_{\boldsymbol{\theta}}\mathcal{N}(\boldsymbol{\theta})$. For brevity, we omit the expectation operator $\mathbb{E}$ in the following derivation. Then, we compute the gradient of $\mathcal{L}(\boldsymbol{\theta})$ as follows:

$$\nabla_{\boldsymbol{\theta}}\mathcal{L}(\boldsymbol{\theta}) = \nabla_{\boldsymbol{\theta}}\left[\frac{1}{\hat{\omega}(t)}\big\|\boldsymbol{f}^{\mathbf{x}}(\boldsymbol{F}_{\boldsymbol{\theta}}(\mathbf{x}_t, t), \mathbf{x}_t, t) - \boldsymbol{f}^{\mathbf{x}}(\boldsymbol{F}_{\boldsymbol{\theta}^-}(\mathbf{x}_{\lambda t}, \lambda t), \mathbf{x}_{\lambda t}, \lambda t)\big\|_2^2\right]$$

$$= \frac{2}{\hat{\omega}(t)}\left\langle \nabla_{\boldsymbol{\theta}}\boldsymbol{f}^{\mathbf{x}}(\boldsymbol{F}_{\boldsymbol{\theta}}(\mathbf{x}_t, t), \mathbf{x}_t, t), \Delta_{\boldsymbol{\theta}, \boldsymbol{\theta}^-}\boldsymbol{f}^{\mathbf{x}}\right\rangle \qquad \text{(by chain rule)}$$

$$= \frac{2}{\hat{\omega}(t)}\left\langle \frac{\alpha(t)}{D(t)}\nabla_{\boldsymbol{\theta}}\boldsymbol{F}_{\boldsymbol{\theta}}(\mathbf{x}_t, t), \Delta_{\boldsymbol{\theta}, \boldsymbol{\theta}^-}\boldsymbol{f}^{\mathbf{x}}\right\rangle \qquad \text{(by definition of } \boldsymbol{f}^{\mathbf{x}}\text{)}$$

$$= \left\langle \nabla_{\boldsymbol{\theta}}\boldsymbol{F}_{\boldsymbol{\theta}}(\mathbf{x}_t, t), \frac{2\alpha(t)}{D(t)\hat{\omega}(t)}\Delta_{\boldsymbol{\theta}, \boldsymbol{\theta}^-}\boldsymbol{f}^{\mathbf{x}}\right\rangle.$$

Since $\boldsymbol{F}_{\boldsymbol{\theta}^-}$ does not depend on $\boldsymbol{\theta}$, we can equivalently write

$$\nabla_{\boldsymbol{\theta}}\mathcal{L}(\boldsymbol{\theta}) = \left\langle \nabla_{\boldsymbol{\theta}}\big(\boldsymbol{F}_{\boldsymbol{\theta}}(\mathbf{x}_t, t) - \boldsymbol{F}_{\boldsymbol{\theta}^-}(\mathbf{x}_t, t)\big), \frac{2\alpha(t)}{D(t)\hat{\omega}(t)}\Delta_{\boldsymbol{\theta}^-, \boldsymbol{\theta}^-}\boldsymbol{f}^{\mathbf{x}}\right\rangle.$$

Next, using the identity

$$\hat{\omega}(t) = \frac{\alpha(t)}{2D(t)}\,A(t) \implies \frac{2\alpha(t)}{D(t)\hat{\omega}(t)} = \frac{2}{A(t)},$$

we obtain

$$\nabla_{\boldsymbol{\theta}}\mathcal{L}(\boldsymbol{\theta}) = \left\langle \nabla_{\boldsymbol{\theta}}\Big(\boldsymbol{F}_{\boldsymbol{\theta}}(\mathbf{x}_t, t) - \boldsymbol{F}_{\boldsymbol{\theta}^-}(\mathbf{x}_t, t) + \frac{2}{A(t)}\Delta_{\boldsymbol{\theta}^-, \boldsymbol{\theta}^-}\boldsymbol{f}^{\mathbf{x}}\Big), \frac{2}{A(t)}\Delta_{\boldsymbol{\theta}^-, \boldsymbol{\theta}^-}\boldsymbol{f}^{\mathbf{x}}\right\rangle.$$

Finally, by the standard identity $\langle \nabla f(x), f(x)\rangle = \frac{1}{2}\nabla\|f(x)\|_2^2$, we conclude

$$\nabla_{\boldsymbol{\theta}}\mathcal{L}(\boldsymbol{\theta}) = \frac{1}{2}\nabla_{\boldsymbol{\theta}}\big\|\boldsymbol{F}_{\boldsymbol{\theta}}(\mathbf{x}_t, t) - \boldsymbol{F}_{\boldsymbol{\theta}^-}(\mathbf{x}_t, t) + \frac{2}{A(t)}\Delta_{\boldsymbol{\theta}^-, \boldsymbol{\theta}^-}\boldsymbol{f}^{\mathbf{x}}\big\|_2^2$$

$$= \nabla_{\boldsymbol{\theta}}\mathcal{N}(\boldsymbol{\theta}).$$

$\square$

#### F.1.8 ENHANCED TARGET SCORE FUNCTION

Training a model directly with objective in (6) fails to produce realistic samples without Classifier-Free Guidance (CFG) (Ho & Salimans, 2022). However, while enhancing semantic information, it introduces significant computational overhead by approximately doubling the required function evaluations.

A recent approach (Tang et al., 2025) proposes modifying the target score function. Instead of the standard conditional score (Song et al., 2020b), $\nabla_{\mathbf{x}_t} \log(p_t(\mathbf{x}_t|\mathbf{c}))$, they propose an enhanced version $\nabla_{\mathbf{x}_t} \log\left(p_t(\mathbf{x}_t|\mathbf{c})\left(p_{t,\theta}(\mathbf{x}_t|\mathbf{c})/p_{t,\theta}(\mathbf{x}_t)\right)^\zeta\right)$, where $\zeta \in (0,1)$ denotes the enhancement ratio. This modification eliminates the need for CFG, enabling high-fidelity sample generation at a significantly reduced inference cost.

Inspired by this, we propose enhancing the target score function in a manner compatible with our unified training objective in (6). Specifically, we introduce a time-dependent enhancement strategy:

(a) For $t \in [0, s]$, enhance $\mathbf{x}$ and $\mathbf{z}$ by applying $\mathbf{x}^\star = \mathbf{x} + \zeta \cdot \left(\boldsymbol{f}^{\mathbf{x}}(\boldsymbol{F}_t, \mathbf{x}_t, t) - \boldsymbol{f}^{\mathbf{x}}(\boldsymbol{F}_t^\varnothing, \mathbf{x}_t, t)\right)$, $\mathbf{z}^\star = \mathbf{z} + \zeta \cdot \left(\boldsymbol{f}^{\mathbf{z}}(\boldsymbol{F}_t, \mathbf{x}_t, t) - \boldsymbol{f}^{\mathbf{z}}(\boldsymbol{F}_t^\varnothing, \mathbf{x}_t, t)\right)$. Here, $\boldsymbol{F}_t^\varnothing = \boldsymbol{F}_{\boldsymbol{\theta}^-}(\mathbf{x}_t, t, \varnothing)$ and $\boldsymbol{F}_t = \boldsymbol{F}_{\boldsymbol{\theta}^-}(\mathbf{x}_t, t)$.

(b) For $t \in (s, 1]$, enhance $\mathbf{x}$ and $\mathbf{z}$ by applying $\mathbf{x}^\star = \mathbf{x} + \frac{1}{2}\left(\boldsymbol{f}^{\mathbf{x}}(\boldsymbol{F}_t, \mathbf{x}_t, t) - \mathbf{x}\right)$ and $\mathbf{z}^\star = \mathbf{z} + \frac{1}{2}\left(\boldsymbol{f}^{\mathbf{z}}(\boldsymbol{F}_t, \mathbf{x}_t, t) - \mathbf{z}\right)$. We consistently set $s = 0.75$ and see App. F.1.8 for more analysis.

An ablation study for this technique is shown in Sec. 4.4, and the pseudocode is shown in Alg. 1. Recall that CFG proposes to modify the sampling distribution as

$$\tilde{p}_\theta(\mathbf{x}_t|\mathbf{c}) \propto p_\theta(\mathbf{x}_t|\mathbf{c})p_\theta(\mathbf{c}|\mathbf{x}_t)^\zeta ,$$

Bayesian rule gives

$$p_\theta(\mathbf{c}|\mathbf{x}_t) = \frac{p_\theta(\mathbf{x}_t|\mathbf{c})p_\theta(\mathbf{c})}{p_\theta(\mathbf{x}_t)} ,$$

so we can futher deduce

$$\tilde{p}_\theta(\mathbf{x}_t|\mathbf{c}) \propto p_\theta(\mathbf{x}_t|\mathbf{c})p_\theta(\mathbf{c}|\mathbf{x}_t)^\zeta$$
$$= p_\theta(\mathbf{x}_t|\mathbf{c})\left(\frac{p_\theta(\mathbf{x}_t|\mathbf{c})p_\theta(\mathbf{c})}{p_\theta(\mathbf{x}_t)}\right)^\zeta$$
$$\propto p_\theta(\mathbf{x}_t|\mathbf{c})\left(\frac{p_\theta(\mathbf{x}_t|\mathbf{c})}{p_\theta(\mathbf{x}_t)}\right)^\zeta .$$

When $t \in [0, s]$ ($s = 0.75$), inspired by above expression and a recent work (Tang et al., 2025), we choose to use below as the target score function for training

$$\nabla_{\mathbf{x}_t} \log\left(p_t(\mathbf{x}_t|\mathbf{c})\left(\frac{p_{t,\boldsymbol{\theta}}(\mathbf{x}_t|\mathbf{c})}{p_{t,\boldsymbol{\theta}}(\mathbf{x}_t)}\right)^\zeta\right)$$

which equals to

$$\nabla_{\mathbf{x}_t} \log p_t(\mathbf{x}_t|\mathbf{c}) + \zeta\left(\nabla_{\mathbf{x}_t} \log p_{t,\boldsymbol{\theta}}(\mathbf{x}_t|\mathbf{c}) - \nabla_{\mathbf{x}_t} \log p_{t,\boldsymbol{\theta}}(\mathbf{x}_t)\right) .$$

In Thm. 2, we have shown that

$$\boldsymbol{f}^{\mathbf{z}}_\star(\boldsymbol{F}_t, \mathbf{x}_t, t) = -\alpha(t)\nabla_{\mathbf{x}_t} \log p_t(\mathbf{x}_t) , \text{ and } \boldsymbol{f}^{\mathbf{x}}_\star(\boldsymbol{F}_t, \mathbf{x}_t, t) = \frac{\mathbf{x}_t + \alpha^2(t)\nabla_{\mathbf{x}_t} \log p_t(\mathbf{x}_t)}{\gamma(t)} ,$$

so we can further deduce: For $\boldsymbol{f}^{\mathbf{z}}_\star$ we originally want to learn:

$$\boldsymbol{f}^{\mathbf{z}}_\star(\boldsymbol{F}_t, \mathbf{x}_t, t) = -\alpha(t)\nabla_{\mathbf{x}_t} \log p_t(\mathbf{x}_t) ,$$

now it turns to

$$\boldsymbol{f}^{\mathbf{z}}_\star(\boldsymbol{F}_t, \mathbf{x}_t, t) = -\alpha(t)\nabla_{\mathbf{x}_t} \log\left(p_t(\mathbf{x}_t|\mathbf{c})\left(\frac{p_{t,\boldsymbol{\theta}}(\mathbf{x}_t|\mathbf{c})}{p_{t,\boldsymbol{\theta}}(\mathbf{x}_t)}\right)^\zeta\right)$$
$$= -\alpha(t)\left[\nabla_{\mathbf{x}_t} \log p_t(\mathbf{x}_t|\mathbf{c}) + \zeta\left(\nabla_{\mathbf{x}_t} \log p_{t,\boldsymbol{\theta}}(\mathbf{x}_t|\mathbf{c}) - \nabla_{\mathbf{x}_t} \log p_{t,\boldsymbol{\theta}}(\mathbf{x}_t)\right)\right]$$
$$= -\alpha(t)\nabla_{\mathbf{x}_t} \log p_t(\mathbf{x}_t|\mathbf{c}) + \zeta\left(-\alpha(t)\nabla_{\mathbf{x}_t} \log p_{t,\boldsymbol{\theta}}(\mathbf{x}_t|\mathbf{c}) + \alpha(t)\nabla_{\mathbf{x}_t} \log p_{t,\boldsymbol{\theta}}(\mathbf{x}_t)\right)$$
$$= -\alpha(t)\nabla_{\mathbf{x}_t} \log p_t(\mathbf{x}_t|\mathbf{c}) + \zeta\left(\boldsymbol{f}^{\mathbf{z}}(\boldsymbol{F}_t, \mathbf{x}_t, t) - \boldsymbol{f}^{\mathbf{z}}(\boldsymbol{F}_t^\varnothing, \mathbf{x}_t, t)\right) ,$$

thus in training we set the objective for $f^{\mathbf{z}}$ as:

$$\mathbf{z}^{\star} \leftarrow \mathbf{z} + \zeta \cdot \left( f^{\mathbf{z}}(F_t, \mathbf{x}_t, t) - f^{\mathbf{z}}(F_t^{\varnothing}, \mathbf{x}_t, t) \right) .$$

Similarly, since $f_{\star}^{\mathbf{x}} = \frac{\mathbf{x}_t + \alpha^2(t)\nabla_{\mathbf{x}_t} \log p_t(\mathbf{x}_t)}{\gamma(t)}$ is also linear in the score function, we can use the same strategy to modify the training objective for $f^{\mathbf{x}}$:

$$\mathbf{x}^{\star} \leftarrow \mathbf{x} + \zeta \cdot \left( f^{\mathbf{x}}(F_t, \mathbf{x}_t, t) - f^{\mathbf{x}}(F_t^{\varnothing}, \mathbf{x}_t, t) \right) .$$

We can also derive that:

$$\mathbf{x}_t^{\star} = \alpha(t) \cdot \mathbf{z}^{\star} + \gamma(t) \cdot \mathbf{x}^{\star} = \mathbf{x}_t ,$$

When $t \in (s, 1]$ ($s = 0.75$), we further slightly modify the target score function to

$$\nabla_{\mathbf{x}_t} \log p_t(\mathbf{x}_t|\mathbf{c}) + \zeta \left( \nabla_{\mathbf{x}_t} \log p_{t,\boldsymbol{\theta}}(\mathbf{x}_t|\mathbf{c}) - \nabla_{\mathbf{x}_t} \log p_t(\mathbf{x}_t) \right), \zeta = 0.5$$

which corresponds to the following training objective:

$$\mathbf{x}^{\star} \leftarrow \mathbf{x} + \frac{1}{2} \left( f^{\mathbf{x}}(F_t, \mathbf{x}_t, t) - \mathbf{x} \right), \mathbf{z}^{\star} \leftarrow \mathbf{z} + \frac{1}{2} \left( f^{\mathbf{z}}(F_t, \mathbf{x}_t, t) - \mathbf{z} \right) .$$

After applying above enhanced target score matching, we can further deduce the training objective for $f^{\mathbf{x}}$ as:

$$\mathcal{L}(\boldsymbol{\theta}) = \mathbb{E}_{(\mathbf{z},\mathbf{x}) \sim p(\mathbf{z},\mathbf{x}),t} \left[ \frac{1}{\hat{\omega}(t)} \| f^{\mathbf{x}}(F_{\boldsymbol{\theta}}(\mathbf{x}_t, t), \mathbf{x}_t, t) - \mathbf{x}^{\star} \|_2^2 \right] .$$

Similarly, introducing the $\lambda$, we can decuce:

$$\mathcal{L}(\boldsymbol{\theta}) = \mathbb{E}_{(\mathbf{z},\mathbf{x}) \sim p(\mathbf{z},\mathbf{x}),t} \left[ \frac{1}{\hat{\omega}(t)} \| f^{\mathbf{x}}(F_{\boldsymbol{\theta}}(\mathbf{x}_t, t), \mathbf{x}_t, t) - f^{\mathbf{x}}(F_{\boldsymbol{\theta}}(\mathbf{x}_{\lambda t}, \lambda t), \mathbf{x}_{\lambda t}^{\star}, \lambda t) \|_2^2 \right] .$$

Using $\mathbf{x}_t^{\star} = \mathbf{x}_t$, we can further deduce:

$$\mathcal{L}(\boldsymbol{\theta}) = \mathbb{E}_{(\mathbf{z},\mathbf{x}) \sim p(\mathbf{z},\mathbf{x}),t} \left[ \frac{1}{\hat{\omega}(t)} \| f^{\mathbf{x}}(F_{\boldsymbol{\theta}}(\mathbf{x}_t, t), \mathbf{x}_t^{\star}, t) - f^{\mathbf{x}}(F_{\boldsymbol{\theta}}(\mathbf{x}_{\lambda t}, \lambda t), \mathbf{x}_{\lambda t}^{\star}, \lambda t) \|_2^2 \right] .$$

**Enhanced target score matching for training objective** (6). By following the derivation in Thm. 8, we can decuce:

$$\mathcal{N}(\boldsymbol{\theta}) = \mathbb{E}_{\mathbf{z},\mathbf{x},t} \left[ \frac{1}{2} \left\| F_{\boldsymbol{\theta}}(\mathbf{x}_t, t) - F_{\boldsymbol{\theta}^-}(\mathbf{x}_t, t) + 2 \cdot \frac{\Delta_{\boldsymbol{\theta}^-, \boldsymbol{\theta}^-} f^{\mathbf{x}}}{B(t) - B(\lambda t)} \right\|_2^2 \right].$$

where

$$\Delta_{\boldsymbol{\theta}^-, \boldsymbol{\theta}^-} f^{\mathbf{x}} = f^{\mathbf{x}} \left( F_{\boldsymbol{\theta}}(\mathbf{x}_t, t), \mathbf{x}_t^{\star}, t \right) - f^{\mathbf{x}} \left( F_{\boldsymbol{\theta}^-}(\mathbf{x}_{\lambda t}, \lambda t), \mathbf{x}_{\lambda t}^{\star}, \lambda t \right).$$

**Enhanced target score matching for training objective** (13). By following the derivation in Thm. 5, we can futher deduce:

$$\mathcal{G}(\boldsymbol{\theta}) = \mathbb{E}_{\mathbf{z},\mathbf{x},t} \left[ \left\| F_{\boldsymbol{\theta}}(\mathbf{x}_t, t) - \mathbf{z}_t^{\star} \right\|_2^2 + \frac{B(t)}{\hat{\omega}(t)} \left\| (F_{\boldsymbol{\theta}}(\mathbf{x}_t, t) - F_{\boldsymbol{\theta}^-}(\mathbf{x}_{\lambda t}, \lambda t)) - (\mathbf{z}_t^{\star} - \mathbf{z}_{\lambda t}^{\star}) \right\|_2^2 \right]$$

### F.1.9 UNIFIED SAMPLING PROCESS

**Deterministic sampling.** When the stochastic ratio $\rho = 0$, let's analyze a apecial case where the coefficients satisfying $\hat{\alpha}(t) = \frac{\mathrm{d}\alpha(t)}{\mathrm{d}t}, \hat{\gamma}(t) = \frac{\mathrm{d}\gamma(t)}{\mathrm{d}t}$. Let $\Delta t = t_{i+1} - t_i$, for the core updating rule we have:

$$\mathbf{x}' = \alpha(t_{i+1}) \cdot \hat{\mathbf{z}} + \gamma(t_{i+1}) \cdot \hat{\mathbf{x}}$$

$$= (\alpha(t_i) + \alpha'(t_i)\Delta t + o(\Delta t)) \cdot \hat{\mathbf{z}} + (\gamma(t_i) + \gamma'(t_i)\Delta t + o(\Delta t)) \cdot \hat{\mathbf{x}}$$

$$= (\alpha(t_i)\hat{\mathbf{z}} + \gamma(t_i)\hat{\mathbf{x}}) + (\hat{\alpha}(t_i)\hat{\mathbf{z}} + \hat{\gamma}(t_i)\hat{\mathbf{x}}) \cdot \Delta t + o(\Delta t)$$

$$= (\alpha(t_i)f^{\mathbf{z}}(F, \tilde{\mathbf{x}}, t_i) + \gamma(t_i)f^{\mathbf{x}}(F, \tilde{\mathbf{x}}, t_i)) + (\hat{\alpha}(t_i)f^{\mathbf{z}}(F, \tilde{\mathbf{x}}, t_i) + \hat{\gamma}(t_i)f^{\mathbf{x}}(F, \tilde{\mathbf{x}}, t_i)) \cdot \Delta t + o(\Delta t)$$

$$= (\alpha(t_i)\frac{\hat{\gamma}(t_i) \cdot \tilde{\mathbf{x}} - \gamma(t_i) \cdot F(\tilde{\mathbf{x}}, t_i)}{\alpha(t_i) \cdot \hat{\gamma}(t_i) - \hat{\alpha}(t_i) \cdot \gamma(t_i)} + \gamma(t_i)\frac{\alpha(t_i) \cdot F(\tilde{\mathbf{x}}, t_i) - \hat{\alpha}(t_i) \cdot \mathbf{x}_t}{\alpha(t_i) \cdot \hat{\gamma}(t_i) - \hat{\alpha}(t_i) \cdot \gamma(t_i)})$$

$$+ (\hat{\alpha}(t_i)\frac{\hat{\gamma}(t_i) \cdot \tilde{\mathbf{x}} - \gamma(t_i) \cdot F(\tilde{\mathbf{x}}, t_i)}{\alpha(t_i) \cdot \hat{\gamma}(t_i) - \hat{\alpha}(t_i) \cdot \gamma(t_i)} + \hat{\gamma}(t_i)\frac{\alpha(t_i) \cdot F(\tilde{\mathbf{x}}, t_i) - \hat{\alpha}(t_i) \cdot \mathbf{x}_t}{\alpha(t_i) \cdot \hat{\gamma}(t_i) - \hat{\alpha}(t_i) \cdot \gamma(t_i)}) \cdot \Delta t + o(\Delta t)$$

$$= \tilde{\mathbf{x}} + F(\tilde{\mathbf{x}}, t_i) \cdot \Delta t + o(\Delta t)$$

In this case $F(\cdot, \cdot)$ tries to predict the velocity field of the flow model, and we can see that the term $\tilde{\mathbf{x}} + F(\tilde{\mathbf{x}}, t_i) \cdot \Delta t$ corresponds to the sampling rule of the Euler ODE solver.

**Stochastic sampling.** As for case when the stochastic ratio $\rho \neq 0$, follow the Euler-Maruyama numerical methods of SDE, the noise injected should be a Gaussian with zero mean and variance proportional to $\Delta t$, so when the updating rule is $\mathbf{x}' = \alpha(t_{i+1}) \cdot (\sqrt{1-\rho} \cdot \hat{\mathbf{z}} + \sqrt{\rho} \cdot \mathbf{z}) + \gamma(t_{i+1}) \cdot \hat{\mathbf{x}}$, the coefficient of $\mathbf{z}$ should satisfy

$$\alpha(t_{i+1})\sqrt{\rho} \propto \sqrt{\Delta t}, \ \rho \propto \frac{\Delta t}{\alpha^2(t_{i+1})}$$

In practice, we set

$$\rho = \frac{2\Delta t \cdot \alpha(t_i)}{\alpha^2(t_{i+1})}.$$

which corresponds to $g(t) = \sqrt{2\alpha(t)}$ for the SDE $\mathrm{d}\mathbf{x} = \boldsymbol{f}(\mathbf{x}, t)\mathrm{d}t + g(t)\mathrm{d}\boldsymbol{w}$.

### F.1.10   EXTRAPOLATING ESTIMATION

**Theorem 9 (Local truncation error of the extrapolation estimation) .** *Assume the sampling process uses a uniform time step size $h = t_{i+1} - t_i = t_i - t_{i-1}$. Let $\boldsymbol{\Phi}(t)$ denote the virtual endpoint estimates (e.g., $\hat{\mathbf{x}}$ or $\hat{\mathbf{z}}$), assumed to be twice continuously differentiable. By setting the extrapolation coefficient $\kappa = 1$, the proposed predictor:*

$$\hat{\boldsymbol{\Phi}}_i^{ext} = \boldsymbol{\Phi}(t_i) + \kappa \cdot (\boldsymbol{\Phi}(t_i) - \boldsymbol{\Phi}(t_{i-1})) \tag{30}$$

*achieves a Local Truncation Error (LTE) of order $\mathcal{O}(h^2)$, effectively serving as a second-order approximation to the trajectory.*

*Proof.* We analyze the error by expanding the exact solution $\boldsymbol{\Phi}(t)$ around time $t_i$. Under the assumption of uniform steps, let $h$ be the step size. The Taylor expansion for the true target value at $t_{i+1}$ is:

$$\boldsymbol{\Phi}(t_{i+1}) = \boldsymbol{\Phi}(t_i) + \boldsymbol{\Phi}'(t_i)h + \frac{1}{2}\boldsymbol{\Phi}''(t_i)h^2 + \mathcal{O}(h^3). \tag{31}$$

Similarly, the historical value at $t_{i-1}$ (where $t_{i-1} = t_i - h$) is expanded as:

$$\boldsymbol{\Phi}(t_{i-1}) = \boldsymbol{\Phi}(t_i) - \boldsymbol{\Phi}'(t_i)h + \frac{1}{2}\boldsymbol{\Phi}''(t_i)h^2 + \mathcal{O}(h^3). \tag{32}$$

The algorithm computes the extrapolated estimate $\hat{\boldsymbol{\Phi}}_i^{ext}$ using the fixed constant $\kappa$. Substituting (32) into the extrapolation formula:

$$\hat{\boldsymbol{\Phi}}_i^{ext} = \boldsymbol{\Phi}(t_i) + \kappa\left(\boldsymbol{\Phi}(t_i) - \boldsymbol{\Phi}(t_{i-1})\right)$$

$$= \boldsymbol{\Phi}(t_i) + \kappa\left(\boldsymbol{\Phi}(t_i) - \left[\boldsymbol{\Phi}(t_i) - \boldsymbol{\Phi}'(t_i)h + \frac{1}{2}\boldsymbol{\Phi}''(t_i)h^2\right]\right) + \mathcal{O}(h^3)$$

$$= \boldsymbol{\Phi}(t_i) + \kappa h\boldsymbol{\Phi}'(t_i) - \frac{\kappa}{2}h^2\boldsymbol{\Phi}''(t_i) + \mathcal{O}(h^3). \tag{33}$$

To evaluate the local truncation error $\mathcal{E}_{loc} = \|\boldsymbol{\Phi}(t_{i+1}) - \hat{\boldsymbol{\Phi}}_i^{ext}\|_2$:

$$\mathcal{E}_{loc} = \|\boldsymbol{\Phi}(t_{i+1}) - \hat{\boldsymbol{\Phi}}_i^{ext}\|_2$$

$$= \|(1 - \kappa)h\boldsymbol{\Phi}'(t_i) + \frac{1}{2}(1 + \kappa)h^2\boldsymbol{\Phi}''(t_i)\|_2 + \mathcal{O}(h^3)$$

Thus, with uniform steps and $\kappa = 1$, the proposed extrapolation correctly captures the linear trend of the virtual endpoints, resulting in a local error of $\mathcal{O}(h^2)$ (set $\|\boldsymbol{\Phi}''(\cdot)\|_2 \leq C$):

$$\mathcal{E}_{loc} = \|\boldsymbol{\Phi}''(t_i)\|_2 \cdot h^2 + \mathcal{O}(h^3) \leq C \cdot h^2.$$

$\square$

**Theorem 10 (Global error of the extrapolation estimation) .** *Under the same assumptions as in Theorem 9, suppose the sampling process uses a uniform step size $h$, and let $\hat{\boldsymbol{\Phi}}_i$ denote the estimated virtual endpoints produced by the sampler at time $t_i$. When the extrapolation coefficient is set to $\kappa = 1$, the extrapolation-based update achieves a global error of order*

$$\left\|\boldsymbol{\Phi}(t_N) - \hat{\boldsymbol{\Phi}}_N\right\|_2 = \mathcal{O}(h), \tag{34}$$

> *where $t_N - t_0 = Nh$ is fixed. In other words, although the local truncation error is second-order, the accumulated global error over $N$ steps is of first order.*

*Proof.* Define the global error at step $i$ by

$$\mathbf{e}_i := \boldsymbol{\Phi}(t_i) - \hat{\boldsymbol{\Phi}}_i. \tag{35}$$

Following Algorithm 2, the extrapolated estimate used in the update is

$$\hat{\boldsymbol{\Phi}}^{\text{ext}} = \hat{\boldsymbol{\Phi}}_i + \kappa(\hat{\boldsymbol{\Phi}}_i - \hat{\boldsymbol{\Phi}}_{i-1}). \tag{36}$$

Consider the hypothetical extrapolation formed using the exact solution:

$$\boldsymbol{\Phi}_{\text{true}}^{\text{ext}} = \boldsymbol{\Phi}(t_i) + \kappa(\boldsymbol{\Phi}(t_i) - \boldsymbol{\Phi}(t_{i-1})). \tag{37}$$

The difference between the true and estimated extrapolations can be expressed directly using the global errors:

$$\boldsymbol{\Phi}_{\text{true}}^{\text{ext}} - \hat{\boldsymbol{\Phi}}^{\text{ext}} = (\boldsymbol{\Phi}(t_i) - \hat{\boldsymbol{\Phi}}_i) + \kappa\Big((\boldsymbol{\Phi}(t_i) - \hat{\boldsymbol{\Phi}}_i) - (\boldsymbol{\Phi}(t_{i-1}) - \hat{\boldsymbol{\Phi}}_{i-1})\Big) \tag{38}$$

$$= (1 + \kappa)\mathbf{e}_i - \kappa\mathbf{e}_{i-1}. \tag{39}$$

Next, the local truncation error established in Theorem 9 states that, for $\kappa = 1$,

$$\boldsymbol{\Phi}(t_{i+1}) - \boldsymbol{\Phi}_{\text{true}}^{\text{ext}} = \mathcal{O}(h^2). \tag{39}$$

Combining the two relations yields the recursion for the global error:

$$\mathbf{e}_{i+1} = \boldsymbol{\Phi}(t_{i+1}) - \hat{\boldsymbol{\Phi}}^{\text{ext}} \tag{40}$$

$$= \underbrace{\left(\boldsymbol{\Phi}(t_{i+1}) - \boldsymbol{\Phi}_{\text{true}}^{\text{ext}}\right)}_{\mathcal{O}(h^2)} + \left(\boldsymbol{\Phi}_{\text{true}}^{\text{ext}} - \hat{\boldsymbol{\Phi}}^{\text{ext}}\right)$$

$$= \mathcal{O}(h^2) + (1 + \kappa)\mathbf{e}_i - \kappa\mathbf{e}_{i-1}.$$

Setting $\kappa = 1$ gives the linear difference equation

$$\mathbf{e}_{i+1} = 2\mathbf{e}_i - \mathbf{e}_{i-1} + \mathcal{O}(h^2). \tag{41}$$

The corresponding homogeneous relation

$$\mathbf{e}_{i+1} - 2\mathbf{e}_i + \mathbf{e}_{i-1} = 0 \tag{42}$$

has characteristic polynomial $(r - 1)^2$, whose general solution is a linear function of $i$. Hence the homogeneous component introduces at most a linear growth factor in $i$ but no exponential amplification.

Unrolling the recursion over $N$ steps and noting that each step contributes an $\mathcal{O}(h^2)$ non-homogeneous term yields

$$\|\mathbf{e}_N\|_2 \leq C\,Nh^2 + \mathcal{O}(h^3). \tag{43}$$

Because the total integration time is fixed, $Nh = t_N - t_0 = \mathcal{O}(1)$, and thus

$$\|\boldsymbol{\Phi}(t_N) - \hat{\boldsymbol{\Phi}}_N\|_2 = \|\mathbf{e}_N\|_2 = \mathcal{O}(h). \tag{44}$$

Therefore, although the extrapolation step achieves a second-order local truncation error, the accumulated global error across $N$ uniform steps is of first order. $\square$

### F.1.11 INTERPRETATION OF UNIFIED SAMPLING PROCESS

**The validity of the decomposition:** The decomposition of $\tilde{\mathbf{x}}_t$ is guaranteed by the design of $\boldsymbol{f}^{\mathbf{x}}$ and $\boldsymbol{f}^{\mathbf{z}}$:

$$\alpha(t) \cdot \hat{\mathbf{z}}_t + \gamma(t) \cdot \hat{\mathbf{x}}_t = \alpha(t) \cdot \boldsymbol{f}^{\mathbf{z}}(\boldsymbol{F}_{\boldsymbol{\theta}^-}(\tilde{\mathbf{x}}_t, t), \tilde{\mathbf{x}}_t, t) + \gamma(t) \cdot \boldsymbol{f}^{\mathbf{x}}(\boldsymbol{F}_{\boldsymbol{\theta}^-}(\tilde{\mathbf{x}}_t, t), \tilde{\mathbf{x}}_t, t) \tag{45}$$

$$= \alpha(t) \cdot \frac{\hat{\gamma}(t) \cdot \tilde{\mathbf{x}}_t - \gamma(t) \cdot \boldsymbol{F}_t^{\boldsymbol{\theta}^-}}{\alpha(t) \cdot \hat{\gamma}(t) - \hat{\alpha}(t) \cdot \gamma(t)} + \gamma(t) \cdot \frac{\alpha(t) \cdot \boldsymbol{F}_t^{\boldsymbol{\theta}^-} - \hat{\alpha}(t) \cdot \tilde{\mathbf{x}}_t}{\alpha(t) \cdot \hat{\gamma}(t) - \hat{\alpha}(t) \cdot \gamma(t)} \tag{46}$$

$$= \frac{\alpha(t) \cdot \hat{\gamma}(t) \cdot \tilde{\mathbf{x}}_t - \alpha(t) \cdot \gamma(t) \cdot \boldsymbol{F}_t^{\boldsymbol{\theta}^-} + \gamma(t) \cdot \alpha(t) \cdot \boldsymbol{F}_t^{\boldsymbol{\theta}^-} - \gamma(t) \cdot \hat{\alpha}(t) \cdot \tilde{\mathbf{x}}_t}{\alpha(t) \cdot \hat{\gamma}(t) - \hat{\alpha}(t) \cdot \gamma(t)}$$

$$\tag{47}$$

$$= \frac{\alpha(t) \cdot \hat{\gamma}(t) \cdot \tilde{\mathbf{x}}_t - \gamma(t) \cdot \hat{\alpha}(t) \cdot \tilde{\mathbf{x}}_t}{\alpha(t) \cdot \hat{\gamma}(t) - \hat{\alpha}(t) \cdot \gamma(t)} \tag{48}$$

$$= \tilde{\mathbf{x}}_t \tag{49}$$

**The validity of the reconstruction:** Firstly, the decomposition and reconstruction forms one DDIM step. Specifically, the decomposition gives:

$$\mathbf{x}_t = \alpha(t) \cdot \boldsymbol{f}^{\mathbf{z}}(\boldsymbol{F}_{\boldsymbol{\theta}^-}(\mathbf{x}_t, t), \mathbf{x}_t, t) + \gamma(t) \cdot \boldsymbol{f}^{\mathbf{x}}(\boldsymbol{F}_{\boldsymbol{\theta}^-}(\mathbf{x}_t, t), \mathbf{x}_t, t)$$

and the reconstruction gives:

$$\begin{aligned}
\mathbf{x}_s &= \alpha(s) \cdot \boldsymbol{f}^{\mathbf{z}}(\boldsymbol{F}_{\boldsymbol{\theta}^-}(\mathbf{x}_t, t), \mathbf{x}_t, t) + \gamma(s) \cdot \boldsymbol{f}^{\mathbf{x}}(\boldsymbol{F}_{\boldsymbol{\theta}^-}(\mathbf{x}_t, t), \mathbf{x}_t, t) \\
&= \frac{\alpha(s)}{\alpha(t)} \cdot (\mathbf{x}_t - \gamma(t) \cdot \boldsymbol{f}^{\mathbf{x}}(\boldsymbol{F}_{\boldsymbol{\theta}^-}(\mathbf{x}_t, t), \mathbf{x}_t, t)) + \gamma(s) \cdot \boldsymbol{f}^{\mathbf{x}}(\boldsymbol{F}_{\boldsymbol{\theta}^-}(\mathbf{x}_t, t), \mathbf{x}_t, t) \quad \text{(by the decomposition)} \\
&= \frac{\alpha(s)}{\alpha(t)} \cdot \mathbf{x}_t + \left(\gamma(s) - \frac{\alpha(s)}{\alpha(t)} \cdot \gamma(t)\right) \cdot \boldsymbol{f}^{\mathbf{x}}(\boldsymbol{F}_{\boldsymbol{\theta}^-}(\mathbf{x}_t, t), \mathbf{x}_t, t)
\end{aligned}$$

This actually forms one DDIM step. In the formula (11) and appendix C.1 of (Zhou et al., 2025), they show that the DDIM interpolant is self-consistent and is marginal-preserving when $\boldsymbol{f}^{\mathbf{x}}(\boldsymbol{F}_{\boldsymbol{\theta}^\star}(\mathbf{x}_t, t), \mathbf{x}_t, t) = \mathbb{E}[\mathbf{x} \mid \mathbf{x}_t]$.

- For multi-step models ($\lambda = 0$), we have proved $\boldsymbol{f}^{\mathbf{x}}(\boldsymbol{F}_{\boldsymbol{\theta}^\star}(\mathbf{x}_t, t), \mathbf{x}_t, t) = \mathbb{E}[\mathbf{x} \mid \mathbf{x}_t]$ in Lem. 5 when $\rho = 0$ and $\kappa = 0$.

- For few-step consistency models ($\lambda \to 1$), we always set $\rho = 1$ in sampling process (see Table 6), which is the same as the sampling process of consistency model (Song et al., 2023).

> **Proposition 2 (Equivalence to multi-step consistency sampler when $\rho = 1$).** *Consider Algorithm 2 and assume the per-step predictor $\boldsymbol{f}^{\mathbf{x}}(\cdot)$ returns the same virtual-endpoint (denoised) estimate used by the multi-step consistency model. When the stochastic ratio is set to $\rho = 1$ and the sampler runs in first-order mode ($\nu = 1$), each update of Algorithm 2 is identical to the corresponding update of the multi-step consistency model sampler ((Song et al., 2023)).*

*Proof.* We compare a single step of Algorithm 2 (with $\nu = 1$) to the standard multi-step consistency sampler update. Fix a generic step index $i$ and use the algorithm's notation.

Under $\nu = 1$ the algorithm does not execute the second-order correction block; hence the core update used to produce the next state $\mathbf{x}'$ (line computing "estimated next time sample") is

$$\mathbf{x}' = \alpha(t_{i+1}) \cdot \left(\sqrt{1 - \rho}\,\hat{\mathbf{z}} + \sqrt{\rho}\,\mathbf{z}\right) + \gamma(t_{i+1}) \cdot \hat{\mathbf{x}}, \tag{50}$$

where $\hat{\mathbf{x}}$ (resp. $\hat{\mathbf{z}}$) denotes the extrapolated denoised (resp. latent) estimate computed from the current model output as in the algorithm, and $\mathbf{z} \sim \mathcal{N}(\mathbf{0}, \mathbf{I})$ is a freshly drawn Gaussian.

Set $\rho = 1$. Then $\sqrt{1 - \rho} = 0$ and $\sqrt{\rho} = 1$, so (50) reduces to

$$\mathbf{x}' = \alpha(t_{i+1})\,\mathbf{z} + \gamma(t_{i+1})\,\hat{\mathbf{x}}. \tag{51}$$

Now recall the common generative parameterization used by multi-step consistency models: a noisy state at time $t$ is written as

$$\mathbf{x}_t = \gamma(t)\,\mathbf{x}_0 + \alpha(t)\,\mathbf{z}, \qquad \mathbf{z} \sim \mathcal{N}(\mathbf{0}, \mathbf{I}),$$

and the consistency sampler constructs the next-step state by using the model's estimate of the denoised endpoint $\mathbf{x}_0$ (denote this estimate by the same symbol $\hat{\mathbf{x}}$) together with a fresh Gaussian $\mathbf{z}$ to form

$$\mathbf{x}_{t_{i+1}} = \gamma(t_{i+1})\,\hat{\mathbf{x}} + \alpha(t_{i+1})\,\mathbf{z}. \tag{52}$$

Comparing (51) and (52) we see they are algebraically identical: the next state is produced by the same deterministic combination of the denoised estimate $\hat{\mathbf{x}}$ and an independent Gaussian $\mathbf{z}$, scaled by the same schedule coefficients $\gamma(t_{i+1})$ and $\alpha(t_{i+1})$.

The equivalence in distribution follows because Algorithm 2 draws $\mathbf{z} \sim \mathcal{N}(\mathbf{0}, \mathbf{I})$ independently at each step (same as the consistency sampler), and by the proposition hypothesis $\boldsymbol{f}^{\mathbf{x}}$ returns the same denoised/endpoint estimate used by the consistency model. Therefore, for $\rho = 1$ and $\nu = 1$ each single-step mapping (and its randomness) produced by Algorithm 2 coincides exactly with the corresponding single-step mapping of the multi-step consistency sampler. Hence the first-order variant of Algorithm 2 with $\rho = 1$ is equivalent to the multi-step consistency model sampler. $\square$

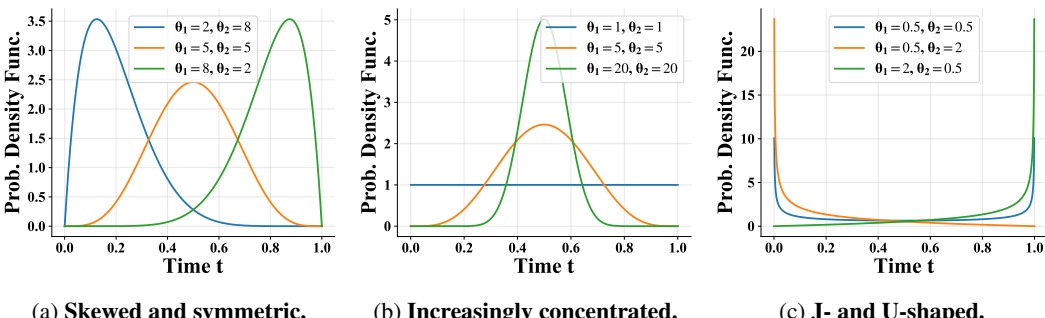

(a) **Skewed and symmetric.**     (b) **Increasingly concentrated.**     (c) **J- and U-shaped.**

Figure 11: **Probability density functions of the Beta distribution over the domain $t \in [0,1]$ for various shape-parameter $\theta_1, \theta_2$.**

### F.2 OTHER TECHNIQUES

#### F.2.1 BETA TRANSFORMATION

We utilize three representative cases to illustrate how the Beta transformation $f_{\text{Beta}}(t; \theta_1, \theta_2)$ generalizes time warping mechanisms for $t \in [0,1]$.

**Standard logit-normal time transformation (Yao et al., 2025; Esser et al., 2024).** For $t \sim \mathcal{U}(0,1)$, the logit-normal transformation $f_{\text{lognorm}}(t; 0, 1) = \frac{1}{1+\exp(-\Phi^{-1}(t))}$ generates a symmetric density profile peaked at $t = 0.5$, consistent with the central maximum of the logistic-normal distribution. Analogously, the Beta transformation $f_{\text{Beta}}(t; \theta_1, \theta_2)$ (with $\theta_1, \theta_2 > 1$) produces a density peak at $t = \frac{\theta_1 - 1}{\theta_1 + \theta_2 - 2}$. When $\theta_1 = \theta_2 > 1$, this reduces to $t = 0.5$, mirroring the logit-normal case. Both transformations concentrate sampling density around critical time regions, enabling importance sampling for accelerated training. Notably, this effect can be equivalently achieved by directly sampling $t \sim \text{Beta}(\theta_1, \theta_2)$.

**Uniform time distribution (Yao et al., 2025; Yu et al., 2024; Ma et al., 2024; Lipman et al., 2022).** The uniform limit case emerges when $\theta_1 = \theta_2 = 1$, reducing $f_{\text{Beta}}(t; 1, 1)$ to an identity transformation. This corresponds to a flat density $p(t) = 1$, reflecting no temporal preference—a baseline configuration widely adopted in diffusion and flow-based models.

**Approximately symmetrical time distribution (Song et al., 2023; Song & Dhariwal, 2023; Karras et al., 2022; 2024b).** For near-symmetric configurations where $\theta_1 \approx \theta_2 > 1$, the Beta transformation induces quasi-symmetrical densities with tunable central sharpness. For instance, setting $\theta_1 = \theta_2 = 2$ yields a parabolic density peaking at $t = 0.5$, while $\theta_1 = \theta_2 \to 1^+$ asymptotically approaches uniformity. This flexibility allows practitioners to interpolate between uniform sampling and strongly peaked distributions, adapting to varying requirements for temporal resolution in training. Such approximate symmetry is particularly useful in consistency models where balanced gradient propagation across time steps is critical.

Furthermore, Fig. 11 further demonstrates the flexibility of the beta distribution.

#### F.2.2 KUMARASWAMY TRANSFORMATION

> **Lemma 12 (Piecewise monotone error) .** *Suppose $f, g$ are continuous and nondecreasing on $[0,1]$, and agree at*
> $$0 = x_0 < x_1 < \cdots < x_n = 1,$$
> *i.e. $f(x_j) = g(x_j)$ for $j = 0, \dots, n$. Let $\Delta_j = g(x_j) - g(x_{j-1})$. Then for every $t \in [x_{j-1}, x_j]$,*
> $$|f(t) - g(t)| \le \Delta_j.$$
> *In particular, if each $\Delta_j \le \frac{1}{4}$, then $\|f - g\|_{L^\infty} \le \frac{1}{4}$.*

*Proof.* On $[x_{j-1}, x_j]$ monotonicity gives
$$f(t) - g(t) \le f(x_j) - g(x_{j-1}) = g(x_j) - g(x_{j-1}) = \Delta_j,$$
and similarly $g(t) - f(t) \le \Delta_j$. $\qquad\square$

**Theorem 11 ($L^2$ approximation bound of monotonic functions by generalized Kumaraswamy transformation)** . *Let $\mathcal{G} = \big\{ g \in C([0,1]) : g \text{ nondecreasing}, \ g(0) = 0, \ g(1) = 1 \big\}$, and define for $a, b, c > 0$, $f_{a,b,c}(t) = \big(1 - (1 - t^a)^b\big)^c$, $t \in [0,1]$. Then*

$$
\sup_{g \in \mathcal{G}} \ \inf_{a,b,c>0} \int_0^1 \big[ f_{a,b,c}(t) - g(t) \big]^2 \, \mathrm{d}t \ \le \ \frac{1}{16}.
$$

*Proof.* Let $g \in \mathcal{G}$. By continuity and the Intermediate-Value Theorem there exist

$$
0 < t_1 < t_0 < t_2 < 1, \quad g(t_1) = \tfrac{1}{4}, \ g(t_0) = \tfrac{1}{2}, \ g(t_2) = \tfrac{3}{4}.
$$

We will choose $(a, b, c) > 0$ so that

$$
f_{a,b,c}(t_j) = g(t_j) \quad (j = 1, 0, 2),
$$

and then apply the piecewise monotone Lem. 12 on the partition

$$
0, \ t_1, \ t_0, \ t_2, \ 1
$$

to conclude $\|f_{a,b,c} - g\|_{L^\infty} \le \tfrac{1}{4}$ and hence $\|f_{a,b,c} - g\|_{L^2}^2 \le \tfrac{1}{16}$.

**Existence via the implicit function theorem.**   Define

$$
F : \mathbb{R}_{>0}^3 \longrightarrow \mathbb{R}^3, \qquad F(a,b,c) = \begin{pmatrix} f_{a,b,c}(t_1) - \tfrac{1}{4} \\ f_{a,b,c}(t_0) - \tfrac{1}{2} \\ f_{a,b,c}(t_2) - \tfrac{3}{4} \end{pmatrix}.
$$

Then $F$ is $C^1$, and at the "base point" $(a,b,c) = (1,1,1)$ with $(t_1, t_0, t_2) = (\tfrac{1}{4}, \tfrac{1}{2}, \tfrac{3}{4})$ we have $f_{1,1,1}(t) = t$ so $F(1,1,1) = 0$, and the Jacobian $\partial F / \partial(a,b,c)$ there is invertible. By the Implicit Function Theorem, for each fixed $(t_1, t_0, t_2)$ near $(\tfrac{1}{4}, \tfrac{1}{2}, \tfrac{3}{4})$ there is a unique local solution $(a,b,c)$.

**Global non-degeneracy of the Jacobian.**   In order to continue this local solution to *all* triples $0 < t_1 < t_0 < t_2 < 1$, we show $\det\big(\partial_{(a,b,c)} F(a,b,c)\big)$ never vanishes.
Set

$$
u(t) = 1 - (1 - t^a)^b, \quad u_j = u(t_j) \in (0,1), \quad f_j = u_j^c.
$$

Then

$$
\partial_a f_j = c \, u_j^{c-1} \, \partial_a u_j, \quad \partial_b f_j = c \, u_j^{c-1} \, \partial_b u_j, \quad \partial_c f_j = u_j^c \ln u_j.
$$

Hence

$$
\det J = \det \begin{pmatrix} c \, u_1^{c-1} u_{1,a} & c \, u_1^{c-1} u_{1,b} & u_1^c \ln u_1 \\ c \, u_0^{c-1} u_{0,a} & c \, u_0^{c-1} u_{0,b} & u_0^c \ln u_0 \\ c \, u_2^{c-1} u_{2,a} & c \, u_2^{c-1} u_{2,b} & u_2^c \ln u_2 \end{pmatrix}.
$$

Factor $c$ from the first two columns and $u_j^{c-1}$ from each row:

$$
\det J = c^2 \, (u_1 u_0 u_2)^{c-1} \, \det \begin{pmatrix} u_{1,a} & u_{1,b} & u_1 \ln u_1 \\ u_{0,a} & u_{0,b} & u_0 \ln u_0 \\ u_{2,a} & u_{2,b} & u_2 \ln u_2 \end{pmatrix}.
$$

Now

$$
u_{j,b} = -(1 - t_j^a)^b \ln(1 - t_j^a) = -(1 - u_j) \ln(1 - t_j^a),
$$

$$
u_{j,a} = b \, (1 - t_j^a)^{b-1} t_j^a \ln t_j = -b \, (1 - u_j) \frac{t_j^a \ln t_j}{1 - t_j^a}.
$$

A direct—but straightforward—expansion shows

$$
\det \begin{pmatrix} u_{1,a} & u_{1,b} & u_1 \ln u_1 \\ u_{0,a} & u_{0,b} & u_0 \ln u_0 \\ u_{2,a} & u_{2,b} & u_2 \ln u_2 \end{pmatrix} = c^{-2} b \, \frac{u_1 u_0 u_2}{(1 - u_1)(1 - u_0)(1 - u_2)} (u_0 - u_1)(u_2 - u_1)(u_2 - u_0).
$$

Therefore

$$
\det J(a,b,c) = b \, (u_1 u_0 u_2)^c \, \frac{(u_0 - u_1)(u_2 - u_1)(u_2 - u_0)}{(1 - u_1)(1 - u_0)(1 - u_2)} > 0,
$$

since $0 < u_1 < u_0 < u_2 < 1$ and $a, b, c > 0$. Hence the Jacobian is everywhere non-zero, and the local solution by the Implicit Function Theorem extends along any path in the connected domain $\{0 < t_1 < t_0 < t_2 < 1\}$. We obtain a unique $(a, b, c) > 0$ solving

$$f_{a,b,c}(t_j) = g(t_j), \quad j = 1, 0, 2,$$

for *every* choice $0 < t_1 < t_0 < t_2 < 1$.

**Completing the error estimate.** By construction $f_{a,b,c}(0) = 0$, $f_{a,b,c}(1) = 1$, and $f_{a,b,c}(t_j) = g(t_j)$ for $j = 1, 0, 2$. On the partition

$$0, \ t_1, \ t_0, \ t_2, \ 1$$

the increments of $g$ are each $1/4$. The piecewise monotone error Lem. 12 yields $\|f_{a,b,c} - g\|_{L^\infty} \le \frac{1}{4}$, hence

$$\int_0^1 \left[ f_{a,b,c}(t) - g(t) \right]^2 \mathrm{d}t \le \|f - g\|_{L^\infty}^2 \le \frac{1}{16}.$$

Since $g$ was arbitrary in $\mathcal{G}$, we conclude

$$\sup_{g \in \mathcal{G}} \inf_{a,b,c>0} \int_0^1 \left[ f_{a,b,c}(t) - g(t) \right]^2 \mathrm{d}t \le \frac{1}{16}.$$

This completes the proof. $\qquad\square$

**Setting and notation.** Fix a positive real number $s > 0$ and consider the *shift function*

$$f_{\text{shift}}(t; s) = \frac{s\,t}{1 + (s-1)t}, \qquad t \in [0, 1].$$

For $a, b, c > 0$, define the *Kumaraswamy transform* as

$$f_{\text{Kuma}}(t; a, b, c) = \left( 1 - \left( 1 - t^a \right)^b \right)^c, \qquad t \in [0, 1].$$

Notice that when $a = b = c = 1$ one obtains

$$f_{\text{Kuma}}(t; 1, 1, 1) = 1 - \left( 1 - t^1 \right)^1 = t,$$

so that the identity function appears as a special case.

We work in the Hilbert space $L^2([0, 1])$ with the inner product

$$\langle f, g \rangle = \int_0^1 f(t)g(t)\,\mathrm{d}t.$$

Accordingly, we introduce the error functional

$$J(a, b, c) := \left\| f_{\text{Kuma}}(\cdot; a, b, c) - f_{\text{shift}}(\cdot; s) \right\|_2^2 \quad \text{and} \quad J_{\text{id}} := \left\| \text{id} - f_{\text{shift}}(\cdot; s) \right\|_2^2.$$

It is known that for $s \ne 1$ one has

$$\inf_{a,b,c} J(a, b, c) < J_{\text{id}}.$$

The goal is to quantify this improvement by optimally adjusting all three parameters $(a, b, c)$.

**Quadratic approximation around the identity.** Since the interesting behavior occurs near the identity $(a, b, c) = (1, 1, 1)$, we reparameterize as

$$\theta := \begin{pmatrix} \alpha \\ \beta \\ \gamma \end{pmatrix} := \begin{pmatrix} a - 1 \\ b - 1 \\ c - 1 \end{pmatrix}, \qquad \text{with } \|\theta\| \ll 1.$$

Thus, we study the function

$$f_{\text{Kuma}}(t; 1 + \alpha, 1 + \beta, 1 + \gamma)$$

in a small neighborhood of $(1, 1, 1)$. Writing

$$F(a, b, c; t) := f_{\text{Kuma}}(t; a, b, c) = \left(1 - (1 - t^a)^b\right)^c,$$

a second–order Taylor expansion around $(a, b, c) = (1, 1, 1)$ gives

$$f_{\text{Kuma}}(t; 1 + \alpha, 1 + \beta, 1 + \gamma) = t + \sum_{i=1}^{3} \theta_i\, g_i(t) + \frac{1}{2} \sum_{i,j=1}^{3} \theta_i \theta_j\, h_{ij}(t) + \mathcal{O}(\|\theta\|^3), \quad (53)$$

where

$$g_i(t) = \frac{\partial}{\partial \theta_i} f_{\text{Kuma}}(t; 1 + \theta)\Big|_{\theta=0} \quad \text{and} \quad h_{ij}(t) = \frac{\partial^2}{\partial \theta_i \partial \theta_j} f_{\text{Kuma}}(t; 1 + \theta)\Big|_{\theta=0}.$$

A short calculation yields:

(a) With respect to $a$ (noting that for $b = c = 1$ one has $f_{\text{Kuma}}(t; a, 1, 1) = t^a$):

$$g_1(t) = \frac{\partial f_{\text{Kuma}}}{\partial a}(t; 1, 1, 1) = \frac{\mathrm{d}}{\mathrm{d}a} t^a\Big|_{a=1} = t \ln t.$$

(b) With respect to $b$ (since for $a = 1$, $c = 1$ we have $f_{\text{Kuma}}(t; 1, b, 1) = 1 - (1 - t)^b$):

$$g_2(t) = \frac{\partial f_{\text{Kuma}}}{\partial b}(t; 1, 1, 1) = -(1 - t)\ln(1 - t).$$

(c) With respect to $c$ (noting that for $a = b = 1$ we have $f_{\text{Kuma}}(t; 1, 1, c) = t^c$):

$$g_3(t) = \frac{\partial f_{\text{Kuma}}}{\partial c}(t; 1, 1, 1) = t \ln t.$$

Thus, we observe that

$$g_1(t) = g_3(t),$$

which indicates an inherent redundancy in the three-parameter model. In consequence, the Gram matrix (defined below) will be of rank at most two.

Next, define the difference between the identity and the shift functions:

$$g(t) := \mathrm{id}(t) - f_{\text{shift}}(t; s) = t - \frac{s\, t}{1 + (s - 1)t} = (1 - s)\frac{t(1 - t)}{1 + (s - 1)t}.$$

Then, $J_{\text{id}} = \langle g, g \rangle$. Also, introduce the first-order moments and the Gram matrix:

$$v_i := \langle g, g_i \rangle, \qquad G_{ij} := \langle g_i, g_j \rangle, \quad i, j = 1, 2, 3.$$

Inserting the expansion (53) into the error functional gives

$$J(1 + \theta) = \left\| f_{\text{Kuma}}(\cdot; 1 + \theta) - f_{\text{shift}}(\cdot; s) \right\|_2^2 = J_{\text{id}} - 2 \sum_{i=1}^{3} \theta_i\, v_i + \sum_{i,j=1}^{3} \theta_i \theta_j\, G_{ij} + \mathcal{O}(\|\theta\|^3).$$

Thus, the quadratic approximation (or model) of the error is

$$\widehat{J}(\theta) := J_{\text{id}} - 2\,\theta^\top v + \theta^\top G\,\theta.$$

Since the Gram matrix $G$ is positive semidefinite (and has a nontrivial null-space due to $g_1 = g_3$), the minimizer is determined only up to the null-space. To select the unique (minimum–norm) minimizer, we choose

$$\theta^\star = G^\dagger v,$$

where $G^\dagger$ denotes the Moore-Penrose pseudoinverse. The quadratic model is then minimized at

$$\widehat{J}_{\min} = J_{\text{id}} - v^\top G^\dagger v.$$

A scaling argument now shows that for any sufficiently small $\varepsilon > 0$ one has

$$J(1 + \varepsilon\,\theta^\star) \leq \widehat{J}(\varepsilon\,\theta^\star) = J_{\text{id}} - \varepsilon^2\, v^\top G^\dagger v < J_{\text{id}},$$

so that the full nonlinear functional is improved by following the direction of $\theta^\star$.

For convenience we introduce the explicit improvement factor

$$\rho_3(s) := \frac{v^\top G^\dagger v}{J_{\text{id}}(s)} \in (0, 1), \qquad s \neq 1, \quad (54)$$

so that our main bound can be written succinctly as

$$\min_{a,b,c>0} J(a, b, c) \leq \left(1 - \rho_3(s)\right) J_{\text{id}}(s). \qquad (s > 0,\ s \neq 1) \quad (55)$$

**Computation of the Gram matrix $G$.** We now compute the inner products

$$G_{ij} = \langle g_i, g_j \rangle, \quad i, j = 1, 2, 3.$$

Since the functions $g_1$ and $g_3$ are identical, only two independent functions appear in the system. A standard fact from Beta-function calculus is that

$$\int_0^1 t^n \ln^2 t \, \mathrm{d}t = \frac{2}{(n+1)^3}, \quad n > -1.$$

Thus, one has

$$\langle g_1, g_1 \rangle = \int_0^1 t^2 \ln^2 t \, \mathrm{d}t = \frac{2}{3^3} = \frac{2}{27},$$

$$\langle g_2, g_2 \rangle = \int_0^1 (1-t)^2 \ln^2(1-t) \, \mathrm{d}t = \frac{2}{27},$$

since the change of variable $u = 1 - t$ yields the same result.

$$\langle g_1, g_2 \rangle = -\int_0^1 t(1-t) \ln t \, \ln(1-t) \, \mathrm{d}t = \frac{3\pi^2 - 37}{108}.$$

It is now convenient to express the Gram matrix with an overall factor:

$$G = \frac{2}{27} \begin{pmatrix} 1 & r & 1 \\ r & 1 & r \\ 1 & r & 1 \end{pmatrix}, r = \frac{3\pi^2 - 37}{8}.$$

Since $g_1 = g_3$, it is clear that the columns (and rows) corresponding to parameters $a$ and $c$ are identical, so that $\mathrm{rank}(G) = 2$. One can compute the Moore-Penrose pseudoinverse $G^\dagger$ by eliminating one of the redundant rows/columns, inverting the resulting $2 \times 2$ block, and then re-embedding into $\mathbb{R}^{3 \times 3}$. One obtains

$$G^\dagger = \frac{27}{8(1-r^2)} \begin{pmatrix} 1 & -2r & 1 \\ -2r & 4 & -2r \\ 1 & -2r & 1 \end{pmatrix}.$$

**Computation of the first-order moments $v_i$.** Recall that

$$g(t) = \mathrm{id}(t) - f_{\mathrm{shift}}(t; s) = t - \frac{s\,t}{1 + (s-1)t}.$$

This expression can be rewritten as

$$g(t) = (1-s)\,t(1-t)D_s(t), \quad \text{with} \quad D_s(t) := \frac{1}{1 + (s-1)t}.$$

Then, the first–order moments read

$$v_1 = v_3 = (1-s) \int_0^1 t(1-t)D_s(t)\, t \ln t \, \mathrm{d}t,$$

$$v_2 = -(1-s) \int_0^1 t(1-t)D_s(t)\, (1-t) \ln(1-t) \, \mathrm{d}t.$$

These integrals can be expressed in closed form (involving logarithms and powers of $(s-1)$); in the case $s \neq 1$ at least one of the $v_i$ is nonzero so that $\rho_3(s) > 0$.

**A universal numerical improvement.** Since projecting onto the three-dimensional subspace spanned by $\{g_1, g_2, g_3\}$ is at least as effective as projecting onto any one axis, we immediately deduce that

$$\rho_3(s) \geq \rho_1(s),$$

where the one-parameter improvement factor is defined by

$$\rho_1(s) := \frac{v_1(s)^2}{\langle g_1, g_1 \rangle \, J_{\mathrm{id}}(s)}.$$

By an elementary (albeit slightly tedious) estimate — for example, using the bounds $\frac{1}{2} \leq D_s(t) \leq 2$ valid for $|s - 1| \leq 1$ — one can show that

$$\rho_1(s) \geq \frac{49}{1536}.$$

Hence, one deduces that

$$\rho_3(s) \geq \frac{49}{1536} \approx 0.0319, \qquad \text{for } |s - 1| \leq 1.$$

In particular, for $s \in [0.5, 2] \setminus \{1\}$ the optimal three-parameter Kumaraswamy transform reduces the squared $L^2$ error by at least 3.19% compared with the identity mapping. Analogous bounds can be obtained on any compact subset of $(0, \infty) \setminus \{1\}$.

**Interpretation of the bound.** Inequality (55) strengthens the known qualitative result (namely, that the three-parameter model can outperform the identity mapping) in two important respects:

(a) Quantitative improvement: The explicit factor $\rho_3(s)$ is computable via one-dimensional integrals, providing a concrete measure of the error reduction.

(b) Utilization of all three parameters: Even though the redundancy (i.e. $g_1 = g_3$) implies that the Gram matrix is singular, the full three-parameter model still offers strict improvement; indeed, one has $\rho_3(s) \geq \rho_1(s) > 0$ for $s \neq 1$. (Equality would require, hypothetically, that $v_2(s) = 0$, which does not occur in practice.)

**Summary.** For every shift parameter $s > 0$ with $s \neq 1$ there exist parameters $(a, b, c)$ (in a neighborhood of $(1, 1, 1)$) such that

$$\left\| f_{\mathrm{Kuma}}(\cdot; a, b, c) - f_{\mathrm{shift}}(\cdot; s) \right\|_2^2 \leq \left(1 - \rho_3(s)\right) \left\| \mathrm{id} - f_{\mathrm{shift}}(\cdot; s) \right\|_2^2,$$

with the improvement factor $\rho_3(s)$ defined in (54) and satisfying

$$\rho_3(s) \geq 0.0319 \quad \text{on } s \in [0.5, 2] \setminus \{1\}.$$

Thus, the full three-parameter Kumaraswamy transform not only beats the identity mapping but does so by a quantifiable margin.

### F.2.3 DERIVATIVE ESTIMATION

**Proposition 3 (Error estimates for forward and central difference quotients) .** *Let $f \in C^3(I)$ where $I \subset \mathbb{R}$ is an open interval, and let $t \in I$. For $0 < \varepsilon$ small enough that $[t - \varepsilon, t + \varepsilon] \subset I$, define the forward and central difference quotients*

$$D_+ f(t) = \frac{f(t + \varepsilon) - f(t)}{\varepsilon}, \qquad D_0 f(t) = \frac{f(t + \varepsilon) - f(t - \varepsilon)}{2\varepsilon}.$$

*Then*

$$D_+ f(t) = f'(t) + \frac{\varepsilon}{2} f''(t) + \frac{\varepsilon^2}{6} f^{(3)}(t + \theta_1 \varepsilon), \qquad \text{for some } 0 < \theta_1 < 1,$$

$$D_0 f(t) = f'(t) + \frac{\varepsilon^2}{12} \left[ f^{(3)}(t + \theta_2 \varepsilon) + f^{(3)}(t - \theta_3 \varepsilon) \right], \qquad \text{for some } 0 < \theta_2, \theta_3 < 1.$$

*In particular,*

$$D_+ f(t) - f'(t) = O(\varepsilon), \qquad D_0 f(t) - f'(t) = O(\varepsilon^2),$$

*so for sufficiently small $\varepsilon$, the forward-difference error exceeds the central-difference error.*

*Proof.* By Taylor's theorem with Lagrange remainder, for some $0 < \theta_1 < 1$,

$$f(t + \varepsilon) = f(t) + f'(t)\,\varepsilon + \tfrac{1}{2}f''(t)\,\varepsilon^2 + \tfrac{1}{6}f^{(3)}(t + \theta_1\varepsilon)\,\varepsilon^3.$$

Dividing by $\varepsilon$ gives the formula for $D_+f(t)$. Hence

$$D_+f(t) - f'(t) = \frac{1}{2}f''(t)\,\varepsilon + \frac{1}{6}f^{(3)}(t + \theta_1\varepsilon)\,\varepsilon^2 = O(\varepsilon).$$

Similarly, applying Taylor's theorem at $t + \varepsilon$ and $t - \varepsilon$,

$$f(t + \varepsilon) = f(t) + f'(t)\,\varepsilon + \tfrac{1}{2}f''(t)\,\varepsilon^2 + \tfrac{1}{6}f^{(3)}(t + \theta_2\varepsilon)\,\varepsilon^3,$$

$$f(t - \varepsilon) = f(t) - f'(t)\,\varepsilon + \tfrac{1}{2}f''(t)\,\varepsilon^2 - \tfrac{1}{6}f^{(3)}(t - \theta_3\varepsilon)\,\varepsilon^3,$$

for some $0 < \theta_2, \theta_3 < 1$. Subtracting and dividing by $2\varepsilon$ yields the formula for $D_0f(t)$ and

$$D_0f(t) - f'(t) = \frac{\varepsilon^2}{12}\big[f^{(3)}(t + \theta_2\varepsilon) + f^{(3)}(t - \theta_3\varepsilon)\big] = O(\varepsilon^2).$$

This completes the proof. $\square$

---

**Proposition 4 .** *Let $f : \mathbb{R} \to \mathbb{R}$ be differentiable, let $t \in \mathbb{R}$ and $\varepsilon > 0$. In BF16 arithmetic (1-bit sign, 8-bit exponent, 7-bit significand) with unit roundoff $\eta = 2^{-7}$, define*

$$f_\pm = f(t \pm \varepsilon), \quad \Delta = f_+ - f_-,$$

$$E_1 = \frac{\mathrm{fl}(f_+) - \mathrm{fl}(f_-)}{2\,\varepsilon}, \qquad E_2 = \mathrm{fl}\Big(\frac{f_+}{2\varepsilon}\Big) - \mathrm{fl}\Big(\frac{f_-}{2\varepsilon}\Big).$$

*Suppose in addition that*
*(1) $|\Delta| < 2^{-126}$, so that $\Delta$ (and any nearby perturbation) lies in the BF16 subnormal range;*
*(2) writing $\mathrm{fl}(f_\pm) = f_\pm(1 + \delta_\pm)$ with $|\delta_\pm| \le \eta$, one has $\big|f_+\delta_+ - f_-\delta_-\big| < 2^{-126}$, so $\tilde{f}_+ - \tilde{f}_-$ remains subnormal;*
*(3) $\big|f_\pm/(2\varepsilon)\big| \ge 2^{-126}$, so each product $f_\pm/(2\varepsilon)$ lies in the normalized range;*
*(4) $|f_+| + |f_-| = O(|\Delta|)$, so that any rounding in the two multiplications is not amplified by a large subtraction.*
*Then the "subtract-then-scale" formula $E_1$ may incur a relative error of order $O(1)$, whereas the "scale-then-subtract" formula $E_2$ retains a relative error of order $O(\eta)$.*

*Proof.* We use two BF16 rounding models: (i) if $x \in [2^{-126}, 2^{128})$ then $\mathrm{fl}(x) = x(1 + \delta)$, $|\delta| \le \eta$; (ii) for any $x$ (including subnormals), $\big|\mathrm{fl}(x) - x\big| \le \tfrac{1}{2}\mathrm{ulp}(x)$, where $\mathrm{ulp}_{\mathrm{sub}} = 2^{-133}$ for subnormals. Set $\tilde{f}_\pm = \mathrm{fl}(f_\pm) = f_\pm(1 + \delta_\pm)$, $|\delta_\pm| \le \eta$.

**Error in $E_1$.** By (1) and (2), $\tilde{f}_+ - \tilde{f}_- = \Delta + (f_+\delta_+ - f_-\delta_-)$ lies in the subnormal range. Hence

$$d = \mathrm{fl}(\tilde{f}_+ - \tilde{f}_-) = (\tilde{f}_+ - \tilde{f}_-) + e_d, \quad |e_d| \le \tfrac{1}{2}\mathrm{ulp}_{\mathrm{sub}} = 2^{-134}.$$

Thus

$$d = \Delta + (f_+\delta_+ - f_-\delta_-) + e_d, \qquad |e_d|/|\Delta| = O(2^{-134}/|\Delta|)\mathbf{g}\eta.$$

Dividing by $2\varepsilon$ and rounding gives

$$E_1 = \mathrm{fl}\big(d/(2\varepsilon)\big) = \frac{d}{2\varepsilon}(1 + \delta_q), \quad |\delta_q| \le \eta,$$

so the relative error in $E_1$ can be $O(1)$.

**Error in $E_2$.** By (3), each $f_\pm/(2\varepsilon)$ is normalized, so

$$g_\pm = \mathrm{fl}\Big(\frac{f_\pm}{2\varepsilon}\Big) = \frac{f_\pm}{2\varepsilon}(1 + \delta'_\pm), \quad |\delta'_\pm| \le \eta.$$

Subtracting and rounding (still normalized) gives

$$E_2 = \mathrm{fl}(g_+ - g_-) = (g_+ - g_-)(1 + \delta'_d), \quad |\delta'_d| \le \eta.$$

Since

$$g_+ - g_- = \frac{\Delta}{2\varepsilon} + \frac{f_+\delta'_+ - f_-\delta'_-}{2\varepsilon},$$

we obtain

$$E_2 = \frac{\Delta}{2\varepsilon}(1 + \delta'_d) + \frac{f_+\delta'_+ - f_-\delta'_-}{2\varepsilon}(1 + \delta'_d).$$

The second term has magnitude $\leq \eta \frac{|f_+|+|f_-|}{2\varepsilon}(1 + \eta)$, and by (4) its relative size to $\Delta/(2\varepsilon)$ is $O\big(\eta \frac{|f_+|+|f_-|}{|\Delta|}\big) = O(\eta)$.

Hence $E_1$ may suffer $O(1)$ relative error, while $E_2$ attains $O(\eta)$ relative accuracy under (1)–(4). $\square$

### F.2.4 CALCULATION OF TRANSPORT

**Transport transformation from EDM to UCGM.** Take the formula (8) from EDM (Karras et al., 2022). With $\sigma_{\text{data}} = \frac{1}{2}$ and $\mathbf{n} = \sigma\mathbf{z}$, we can deduce:

$$\mathbb{E}_{\sigma,\mathbf{x},\mathbf{n}}\left[\lambda(\sigma)c_{\text{out}}(\sigma)^2 \left\| \boldsymbol{F}_\theta\big(c_{\text{in}}(\sigma)\cdot(\mathbf{x}+\mathbf{n});\, c_{\text{noise}}(\sigma)\big) - \frac{1}{c_{\text{out}}(\sigma)}(\mathbf{x} - c_{\text{skip}}(\sigma)\cdot(\mathbf{x}+\mathbf{n})) \right\|_2^2\right]$$

$$= \mathbb{E}_{\sigma,\mathbf{x},\mathbf{z}}\left[\left\| \boldsymbol{F}_\theta\Big(\frac{1}{\sqrt{\sigma^2+\sigma_{\text{data}}^2}}(\mathbf{x}+\sigma\mathbf{z});\, \tfrac{1}{4}\ln\sigma\Big) - \frac{\sqrt{\sigma_{\text{data}}^2+\sigma^2}}{\sigma\sigma_{\text{data}}}\Big(\mathbf{x} - \frac{\sigma_{\text{data}}^2}{\sigma^2+\sigma_{\text{data}}^2}(\mathbf{x}+\sigma\mathbf{z})\Big) \right\|_2^2\right]$$

$$= \mathbb{E}_{\sigma,\mathbf{x},\mathbf{z}}\left[\left\| \boldsymbol{F}_\theta\Big(\frac{1}{\sqrt{\sigma^2+\sigma_{\text{data}}^2}}(\mathbf{x}+\sigma\mathbf{z});\, \tfrac{1}{4}\ln\sigma\Big) - \frac{\sqrt{\sigma_{\text{data}}^2+\sigma^2}}{\sigma\sigma_{\text{data}}}\Big(\frac{\sigma^2}{\sigma^2+\sigma_{\text{data}}^2}\mathbf{x} - \frac{\sigma_{\text{data}}^2}{\sigma^2+\sigma_{\text{data}}^2}\sigma\mathbf{z}\Big) \right\|_2^2\right]$$

$$= \mathbb{E}_{\sigma,\mathbf{x},\mathbf{z}}\left[\left\| \boldsymbol{F}_\theta\Big(\frac{1}{\sqrt{\sigma^2+\sigma_{\text{data}}^2}}(\mathbf{x}+\sigma\mathbf{z});\, \tfrac{1}{4}\ln\sigma\Big) - \Big(\frac{\sigma}{\sigma_{\text{data}}\sqrt{\sigma^2+\sigma_{\text{data}}^2}}\mathbf{x} - \frac{\sigma_{\text{data}}}{\sqrt{\sigma^2+\sigma_{\text{data}}^2}}\mathbf{z}\Big) \right\|_2^2\right]$$

$$= \mathbb{E}_{\sigma,\mathbf{x},\mathbf{z}}\left[\left\| \boldsymbol{F}_\theta\Big(\frac{\sigma}{\sqrt{\sigma^2+\sigma_{\text{data}}^2}}\mathbf{z} + \frac{1}{\sqrt{\sigma^2+\sigma_{\text{data}}^2}}\mathbf{x};\, \tfrac{1}{4}\ln\sigma\Big) - \Big(-\frac{\sigma_{\text{data}}}{\sqrt{\sigma^2+\sigma_{\text{data}}^2}}\mathbf{z} + \frac{\sigma}{\sigma_{\text{data}}\sqrt{\sigma^2+\sigma_{\text{data}}^2}}\mathbf{x}\Big) \right\|_2^2\right]$$

$$= \mathbb{E}_{\sigma,\mathbf{x},\mathbf{z}}\left[\left\| \boldsymbol{F}_\theta\Big(\frac{\sigma}{\sqrt{\sigma^2+\frac{1}{4}}}\mathbf{z} + \frac{1}{\sqrt{\sigma^2+\frac{1}{4}}}\mathbf{x}\Big) - \Big(-\frac{1/2}{\sqrt{\sigma^2+\frac{1}{4}}}\mathbf{z} + \frac{2\sigma}{\sqrt{\sigma^2+\frac{1}{4}}}\mathbf{x}\Big) \right\|_2^2\right].$$

