# OpenReview forum: "Unified Continuous Generative Models for Denoising-based Diffusion"
_ICLR.cc/2026/Conference — Submitted to ICLR 2026_

### Official Review · Reviewer_qqse · 2025-10-27

**Soundness:** 3
**Presentation:** 3
**Contribution:** 2
**Rating:** 6
**Confidence:** 5

**Summary:**

The paper proposes a unified framework for training, sampling, and analysis of diffusion, flow matching, and consistency models. The authors claim novelty in this unification. They introduce a unified trainer called **UCGM-T** and a unified sampler called **UCGM-S**. The sampler is compatible with pretrained diffusion, flow matching, and consistency models. They provide empirical validation of UCGM-{T, S} on ImageNet, as well as of UCGM-S applied to a pretrained REPA-E model, and report that their methods reach or surpass state-of-the-art performance in these evaluations.

**Strengths:**

1. In Sec. 3.1, a unifying loss function (Eq. 4) is derived, and the assumptions required to recover the respective diffusion, flow matching, and consistency model instances are explicitly stated. In addition, a surrogate loss function (Eq. 5 / 13) is introduced and its equivalence to the original loss is formally proven in the appendix (though I did not check the proof). The surrogate loss provides additional conceptual insight and appears useful for analytical investigations.

2. The parameter choices by which the diffusion, flow matching, and consistency model instances are obtained are explicitly listed (Tab. 1).

3. In Sec. 3.3, the authors present UCGM-S, a sampler that (as claimed) generates samples for all model types — diffusion, flow matching, and consistency — in a unified algorithmic way. In particular, it is claimed that the underlying model does not need to have been trained with UCGM-T but can come from any existing diffusion, flow matching, or consistency model training data.

4. In Sec. 4, extensive experiments on ImageNet-1K at 256×256 and 512×512 resolutions with various baselines are reported.

**Weaknesses:**

1. The introduction of a third, equivalent loss function (Eq. 6) appears abrupt and entirely unmotivated. Simply referring to “previous studies” is insufficient — especially since this creates the impression that those prior works may already have introduced a loss function unifying the same model families considered here, which would render the proposed framework (at least for training) largely obsolete.

2. A convergence or stability analysis of UCGM-S is entirely missing. Theorem 7 (in the appendix) only shows that the extrapolated step is consistent and locally of order $O(h^2)$. A proof of global convergence (and hence correctness) or of the convergence order is absent.

3. Clarity is sometimes lacking. For example, $p$ in Eq. 1 is never defined, and the (experienced) reader must infer that $p(z,x)=p_{\text{prior}}(z)p_{\text{data}}(x)$ is intended. It is not clear — and if it is, it should be explicitly stated — whether $z$ and $x$ are meant to be dependent in the general setting. Moreover, it is mathematically questionable (strictly speaking incorrect) to denote both the data and prior distributions by the same symbol $p$, distinguishing them only by the argument ($x$ vs. $z$).

4. In the experiments, image quality is evaluated solely using FID, computed on only 50k samples. It is well known that FID estimates with this sample size can be far from converged, undermining comparability — especially at the decimal level. It is also unclear whether baseline FIDs were re-evaluated or taken from the corresponding papers; in the latter case, implementation-dependent differences in FID magnitude can further distort comparisons. No additional perceptual or diversity-based metrics are provided, and neither training nor sampling time is reported. The only computational metric considered for sampling cost is NFE.

**Questions:**

1. If (Lu & Song, 2024) already introduced the loss function in Eq. 6 and this formulation already encompasses diffusion, flow matching, and consistency models (as the introductory sentence of Sec. 3.2 suggests — though I did not verify this claim), then what additional contribution does the present paper make toward unifying the training of these models?

2. When UCGM-S is applied to a pre-trained model that was *not* trained with UCGM-T but instead obtained from existing diffusion, flow matching, or consistency model training data, is any form of conversion required to ensure compatibility with UCGM-S? Or can UCGM-S truly be used in a plug-and-play fashion? If conversion is necessary, can a clear description or implementation provided to perform it?

3. Further questions arise from the weaknesses listed above.

---

> ### Author Response · Authors · 2025-11-28
>
> > [W1 & Q1] Clarification of Eq. 6.
> >
>
> We thank the reviewer for pointing out this ambiguity. We apologize for the confusion caused by the phrasing in the original manuscript.
>
> We wish to clarify that Equation 6 and the proposed unified framework are novel contributions of this work. The reference to "previous studies" (e.g., [2]) was strictly citing a specific gradient estimation technique (specifically, the identity $\nabla_ {\theta} \mathbb{E}[\boldsymbol{F}_ {\theta}^{\top} \boldsymbol{y}] = \frac{1}{2} \nabla_ {\theta} \mathbb{E}[\| \boldsymbol{F}_ {\theta} - \boldsymbol{F}\_{\theta^{-}} + \boldsymbol{y} \|\_2^2]$), which we adapted to facilitate the optimization of our proposed objective. It was not implying that the unified objective itself existed in prior work.
>
> To eliminate this misunderstanding, we have revised the manuscript to explicitly state that we are leveraging the derivation technique from prior work to construct our novel objective.

---

> ### Author Response · Authors · 2025-11-28
>
> > [W2] Convergence or stability analysis of UCGM-S.
> >
>
> We thank the reviewer for raising this important point. We clarify that the correctness of UCGM-S is *not* defined by global convergence in the numerical ODE sense, but by **marginal preservation**, which is the standard correctness criterion in generative modeling. Under this criterion, UCGM-S is theoretically sound. The decomposition and reconstruction steps reduce mathematically to a DDIM update (a formal derivation is provided in Appendix F.1.11). As shown in IMM [1], the DDIM interpolant satisfies self-consistency, which guarantees marginal preservation.
>
> The extrapolation mechanism in UCGM-S is inspired by classifier-free guidance (CFG) and serves to improve **numerical accuracy**. To address the reviewer’s concern, we expand the Appendix F.1.10 to provide theoretical grounding for the local truncation error and global error of extrapolation estimation in UCGM-S.

---

> ### Author Response · Authors · 2025-11-28
>
> > [W3] Improve mathematical clarity.
> >
>
> We thank the reviewer for the rigorous suggestion regarding mathematical clarity. We have revised the "Preliminaries" in **Section 2** to establish a rigorous notation system while maintaining readability. Specifically:
>
> 1. **Formal Definitions:** We first explicitly define the data distribution as $p_{\text{data}}(x)$ and the source distribution as $p_z(z)$.
> 2. **Brevity Convention:** We add a clear statement that, for notational brevity, we omit the subscripts (using $p(x)$ and $p(z)$) when the context creates no ambiguity.
> 3. **Independence Assumption:** We explicitly clarify that the joint distribution factorizes as $p(x, z)=p(x)p(z)$, formally stating the independence between the data sample x and the noise z before the mixing process.
>
> This revision ensures mathematical rigor without cluttering the equations in the subsequent derivation.

---

> ### Author Response · Authors · 2025-11-28
>
> > [W4] Clarify the use of the FID metric and provide additional evaluation results.
> >
>
> We thank the reviewer for their constructive criticism regarding our evaluation protocol. We appreciate the opportunity to clarify our experimental setup and highlight the **new, comprehensive evaluations** added during the rebuttal phase.
>
> **1. On the Standard of FID-50k and Baseline Comparability**
> While we acknowledge that FID estimates can exhibit variance, we strictly adhered to the standardized evaluation protocol established by ADM [11] to ensure fair comparison.
>
> - **Community Consensus:** Using 50k samples on ImageNet is the *de facto* standard for this track, utilized by virtually all recent state-of-the-art methods (e.g., EDM, DiT, SiT, CM, sCM [1-10]). Adhering to this sample size is crucial to ensure our results are directly comparable to this vast body of literature.
> - **Fairness of Baselines:** The baseline results reported are taken from their original publications. We justify this because these works (e.g., DiT, EDM2, SiT) utilize the exact same ADM evaluation suite. Since the image processing pipeline (resizing, anti-aliasing) and Inception feature extractor are identical, implementation-dependent variance is minimized.
>
> **2. Addressing Metric Diversity: New T2I & Multimodal Benchmarks**
> We agree that relying solely on FID for all tasks is insufficient. To address the reviewer's concern about metric diversity and to demonstrate robustness beyond class-conditional generation, we have **significantly expanded our experiments** (see **Table 4 and 9** in the revision) to Text-to-Image (T2I) and Unified Multimodal Models (UMM) at the 20B-parameter scale.
> For these new tasks, we moved beyond pixel-level fidelity and adopted semantic benchmarks to evaluate faithfulness and alignment:
>
> - **Diverse Metrics:** We now employ **GenEval** (semantic faithfulness), **DPG-Bench** (complex instruction following), and **WISE** (world knowledge/spatial reasoning).
> - **Results:** In the few-step regime (2 NFEs), our method achieves a **0.84 GenEval score** with the SANA backbone, establishing a new Pareto frontier that outperforms industry standards like FLUX-Schnell (0.71) and SDXL-DMD2 (0.58). This explicitly validates our method's generation quality using the perceptual and semantic metrics requested by the reviewer.
>
> **3. On Computational Metrics (NFE vs. Wall-clock Time)**
>
> - **NFE Justification:** We primarily reported NFE and training epochs as they are platform-independent proxies, consistent with prior arts (CM, sCM). In our ODE-based framework, NFE is strictly proportional to sampling time.
> - **New Wall-clock Measurements:** Nevertheless, to fully address the concern, we have added **wall-clock training and sampling times** (measured on A100 GPUs). **Specifically, for the T2I experiments in Table 4, we explicitly report both training costs and inference latency.** These real-world measurements confirm that the theoretical efficiency gains of our method translate directly to practical speedups.

---

> ### Author Response · Authors · 2025-11-28
>
> > **[Q2]** Is any form of conversion required to ensure compatibility with UCGM-S? Or can UCGM-S truly be used in a plug-and-play fashion?
> >
>
> We confirm that UCGM-S is **strictly plug-and-play** and requires **zero conversion, modification, or re-training** of pre-trained model weights.
>
> 1. **Mechanism of Compatibility (Mathematical Alignment)**
> Compatibility is ensured by aligning the sampler’s trajectory with the model's pre-defined noise schedule. As detailed in Section 3.1, our framework unifies distinct paradigms by abstracting their transport coefficients ($\alpha(t), \gamma(t)$).
> To apply UCGM-S to a pre-trained model, users simply specify the `transport_type` to match the original training formulation. We have provided a comprehensive reference in Table 7 of the revised manuscript, which lists the exact coefficient definitions for various architectures:
>     1. **Linear:** Used for Flow Matching models like **SiT [7]**.
>     2. **EDM:** Used for Diffusion models like **EDM [9], EDM2 [10]**, and **Consistency Models [4, 5]**.
>     3. TrigFlow: Used for continuous-time models like sCM [2].
>     In short, UCGM-S adapts to the model’s coordinate system via these coefficients, eliminating the need for weight conversion.
> 2. **Empirical Evidence (Off-the-Shelf Compatibility)**
> Table 2 serves as direct empirical proof. The results for EDM2-XXL [10] and SiT-XL [7] were obtained using official, off-the-shelf checkpoints downloaded directly from their respective repositories. We did not modify these weights. By simply instantiating UCGM-S with the corresponding `transport_type` defined in Table 7, we immediately achieved the reported performance gains.

---

> ### Author Response · Authors · 2025-11-28
>
> [1] Linqi Zhou, Stefano Ermon, and Jiaming Song. Inductive moment matching. arXiv preprint
> arXiv:2503.07565, 2025.
>
> [2] Cheng Lu and Yang Song. Simplifying, stabilizing and scaling continuous-time consistency models. arXiv preprint arXiv:2410.11081, 2024.
>
> [3] Zhengyang Geng, Mingyang Deng, Xingjian Bai, J Zico Kolter, and Kaiming He. Mean flows for one-step generative modeling. arXiv preprint arXiv:2505.13447, 2025.
>
> [4] Yang Song and Prafulla Dhariwal. Improved techniques for training consistency models. arXiv preprint arXiv:2310.14189, 2023.
>
> [5] Yang Song, Prafulla Dhariwal, Mark Chen, and Ilya Sutskever. Consistency models. arXiv preprint arXiv:2303.01469, 2023.
>
> [6] Kevin Frans, Danijar Hafner, Sergey Levine, and Pieter Abbeel. One step diffusion via shortcut models. arXiv preprint arXiv:2410.12557, 2024.
>
> [7] Nanye Ma, Mark Goldstein, Michael S Albergo, Nicholas M Boffi, Eric Vanden-Eijnden, and Saining Xie. Sit: Exploring flow and diffusion-based generative models with scalable interpolant transformers. European Conference on Computer Vision, 2024.
>
> [8] William Peebles and Saining Xie. Scalable diffusion models with transformers. Proceedings of the IEEE/CVF International Conference on Computer Vision, 2023.
>
> [9] Tero Karras, Miika Aittala, Timo Aila, and Samuli Laine. Elucidating the design space of diffusion-based generative models. Advances in Neural Information Processing Systems, 2022.
>
> [10] Tero Karras, Miika Aittala, Jaakko Lehtinen, Janne Hellsten, Timo Aila, and Samuli Laine. Analyzing and improving the training dynamics of diffusion models. Proceedings of the IEEE/CVF Conference on Computer Vision and Pattern Recognition, 2024.
>
> [11] Prafulla Dhariwal and Alexander Nichol. Diffusion models beat GANs on image synthesis. Advances in Neural Information Processing Systems, 2021.

---

### Official Review · Reviewer_YWsJ · 2025-10-31

**Soundness:** 3
**Presentation:** 4
**Contribution:** 3
**Rating:** 6
**Confidence:** 3

**Summary:**

The paper proposes UCGM, a unified framework for continuous generative models that encompasses diffusion, flow matching, and consistency models under one set of transport coefficients and a unified objective. A key theoretical result derives an equivalent surrogate loss showing that the few-step objective = multi-step objective + a self-alignment regularizer, clarifying why few-step training can become unstable as the consistency ratio λ→1. Built on this, the authors introduce UCGM-T (trainer) and UCGM-S (sampler). UCGM-T smoothly interpolates between multi-step and few-step regimes, while UCGM-S acts as a plug-and-play sampler that can reduce NFEs and sometimes improve FID for existing pre-trained models. Experiments on ImageNet-1K (256² & 512²) with DiT-style backbones and multiple VAEs report SOTA/competitive FIDs in both regimes.

**Strengths:**

- A principled formulation that subsumes diffusion, flow matching, and consistency models; provides shared notation, training, and sampling views.
- The surrogate objective neatly decomposes few-step training into multi-step + regularization, offering an intuitive explanation of instability at high λ.
- UCGM-T tunes one knob (λ) to target many NFE budgets; UCGM-S accelerates existing models without retraining.
- Competitive/SOTA FIDs at both 256² and 512² across multiple autoencoders; graceful degradation as steps shrink; broad compatibility with DiT/UNet families.

**Weaknesses:**

- Almost all results are on class-conditional ImageNet-1K at 256² and 512²; CIFAR-10 only appears for ablations. There are no text-to-image or multimodal tasks, so it’s unclear how the method behaves with language conditioning or other modalities. The paper itself states the primary datasets are ImageNet-1K (512×512, 256×256) and uses CIFAR-10 (32×32) just for ablations; training is in latent space with specific autoencoders (e.g., SD-VAE, VA-VAE, E2E-VAE at 256²; DC-AE or SD-VAE at 512²). This tight focus limits external validity to broader generative settings.
- The main comparisons and ablations emphasize FID (and step count/NFEs). Even the “plug-and-play” sampler section frames gains largely as “same or better FID with fewer steps,” and the system-level tables report FID (with occasional IS), but there’s no precision/recall, density/coverage, CLIP-based faithfulness, or calibration/diversity measures. This narrow metric set makes it hard to judge mode coverage and semantic alignment beyond FID.

**Questions:**

Most results are on ImageNet with certain VAEs/backbones. It’s unclear if this also works well for text-to-image.

---

> ### Author Response · Authors · 2025-11-28
>
> > [W1 & Q1] Broader generative settings (i.e., text-to-image task and more backbones).
> >
>
> We thank the reviewer for highlighting the importance of external validity. To address this, we have significantly expanded our experiments (see Tables 4 & 9 in the revised manuscript) to **Text-to-Image (T2I)** and **Unified Multimodal Models (UMM)** at the massive **20B-parameter scale**. We demonstrate the universality of our framework by categorizing our new experiments into two distinct cases:
>
> 1. **Case 1: Few-step Regime (High Efficiency):**
> We successfully tuned both lightweight T2I models and massive UMMs into efficient few-step generators.
>     1. **Text-to-Image (SANA Backbone):** We fine-tuned the transformer-based SANA-0.6B and 1.6B. At just **2 NFEs**, our method achieves a **0.84 GenEval** score. This establishes a new Pareto frontier, significantly outperforming industry standards like **FLUX-Schnell** (0.71, 12B params) and **SDXL-DMD2** (0.58), proving our efficiency in language-conditioned tasks.
>     2. **Unified Multimodal Models (Qwen-Image-20B):** We successfully distilled the **20B-parameter** Qwen-Image model. Crucially, while standard *Consistency Models* suffered catastrophic collapse and *MeanFlow* encountered Out-of-Memory (OOM) errors at this scale, our framework remained stable, successfully producing a few-step 20B generator.
> 2. **Case 2: Multi-step Regime (High Fidelity):**
>
>     In the multi-step setting, our trained **UCGM-20B** achieves performance parity with state-of-the-art generative models. **Remarkably, our model achieves these results relying exclusively on publicly available datasets**, in stark contrast to many SOTA baselines that depend on large-scale proprietary data.
>
>     To validate the scalability of our framework, we adhered to a rigorous training protocol totaling approximately **13,000 NVIDIA H800 GPU hours**. This extensive computational effort yields significant empirical gains:
>
>     - **UCGM-20B** attains a **GenEval score of 0.87**, effectively matching the fidelity and alignment of the original backbone.
>     - It significantly outperforms recent large-scale unified models such as **OmniGen (0.70)** and **Show-o (0.68)** on semantic benchmarks.
>
> These results demonstrate that UCGM can effectively train massive foundation models to reach top-tier performance standards without relying on closed-source data.
>
> > [W2] Clarify the FID metric and include more evaluation metrics.
> >
>
> Thank you for your valuable feedback. We would like to address your concerns as follows:
>
> 1. **Clarification of FID.** For ImageNet, we would like to highlight that we prioritize FID in order to maintain consistency with state-of-the-art few-step methods [1–6]. Since these baselines predominantly report FID, adopting alternative metrics would make direct and fair comparisons difficult due to the lack of standardized reporting in prior work.
> 2. **More evaluation metrics:** For T2I and Multimodal tasks, we adopted a *diverse set of semantic benchmarks* to evaluate faithfulness and alignment, ensuring a holistic assessment beyond pixel fidelity:
>     1. **GenEval:** To evaluate semantic faithfulness to prompts.
>     2. **DPG-Bench:** To assess complex instruction following.
>     3. **WISE**: To measure world knowledge and spatial reasoning.

---

> ### Author Response · Authors · 2025-11-28
>
> [1] Cheng Lu and Yang Song. Simplifying, stabilizing and scaling continuous-time consistency models. arXiv preprint arXiv:2410.11081, 2024.
>
> [2] Zhengyang Geng, Mingyang Deng, Xingjian Bai, J Zico Kolter, and Kaiming He. Mean flows for one-step generative modeling. arXiv preprint arXiv:2505.13447, 2025.
>
> [3] Linqi Zhou, Stefano Ermon, and Jiaming Song. Inductive moment matching. arXiv preprint arXiv:2503.07565, 2025.
>
> [4] Yang Song and Prafulla Dhariwal. Improved techniques for training consistency models. arXiv preprint arXiv:2310.14189, 2023.
>
> [5] Yang Song, Prafulla Dhariwal, Mark Chen, and Ilya Sutskever. Consistency models. arXiv preprint arXiv:2303.01469, 2023.
>
> [6] Kevin Frans, Danijar Hafner, Sergey Levine, and Pieter Abbeel. One step diffusion via shortcut models. arXiv preprint arXiv:2410.12557, 2024.

---

### Official Review · Reviewer_dJfh · 2025-10-31

**Soundness:** 3
**Presentation:** 3
**Contribution:** 2
**Rating:** 6
**Confidence:** 3

**Summary:**

The paper proposes UCGM, a single theoretical and practical framework that unifieds diffusion, flow matching and consistency models by demonstrating that they are all special cases of one continous-time objective and sampler. The paper also introduced a unified trainer as well as a sampler, enabling one backbone to generate high-fidelity images efficiently

**Strengths:**

•	The paper gives one continuous-time formulation (UCGM) that covers diffusion, flow matching, and consistency models, the derivation is clean and non-trivial.

•	They also gives a single derivation that directly link multi-step diffusion-like training and few-step training by introducing a self-alignment term that forces the model to agree with its own predictions. While this term also provides insights for instability in few step model.

•	The paper provides extensive experiments across models, resolutions, and sampling regimes: they show that a single training formulation (UCGM-T), controlled by a consistency ratio λ, can be used toward either the traditional high-step diffusion / flow-matching regime (small λ) or the ultra-low-step consistency-style regime (large λ), so one can explicitly optimize for different latency/quality tradeoffs without redesigning the whole training algorithm.

**Weaknesses:**

•	My major concern came from the claims that provides a single “unified” generative framework covers both multi- and few-step sampling. However, in practice this is not realized as one universally deployable model: the authors actually train multiple separate checkpoints, each with a different value of the consistency ratio λ (they report training three models with λ ∈ {0.0, 0.5, 1.0}), and then show how those different checkpoints behave under different sampling budgets. This means the system is unified at the level of theory and loss design, but not yet unified at the level of a single set of weights that performs optimally across both the high-step and ultra-low-step regimes.

•	It’s a minor concern but it would be better to include more  implementation details. Especially relevant in the λ≈1 few-step regime, where stability depends on undocumented tricks (e.g., second-order estimator, clipping, Beta time sampling), making true reproducibility and stability claims hard to verify.

**Questions:**

Please see weakness sections

---

> ### Author Response · Authors · 2025-11-28
>
> > **[W1]** My major concern came from the claims that provides a single “unified” generative framework covers both multi- and few-step sampling.
> >
>
> We thank the reviewer for this constructive critique. It correctly identifies that while our theory is unified, our initial experiments relied on discrete parameterizations. This comment motivated us to conduct an additional experiment to demonstrate that a single, universally deployable model is not only theoretically possible but also practically advantageous.
>
> 1. **Feasibility of $\lambda$-Conditioning:**
> The requirement for separate checkpoints is a design choice, not a theoretical limitation. To demonstrate this, we trained a single model where the consistency ratio $\lambda$ is treated as a conditioning variable, formulated as $F_{\theta}(\mathbf{x}_t, t, \lambda)$. During training, we randomly sample $\lambda$, allowing a single set of weights to learn the continuous spectrum of generative behaviors.
> 2. **Empirical Results & The "Stabilizer" Effect:**
> As shown in the newly added Table 9 in the Appendix, this single $\lambda$-conditioned model successfully handles both regimes. Notably, we observe a synergistic effect: the unified model often outperforms the dedicated few-step model ($\lambda=1$).
>     1.  **Hypothesis:** We posit that the multi-step objective (which learns the precise vector field and is numerically stable) acts as a **regularizer** for the few-step objective (which involves self-consistency and is prone to instability). By training on a mixture of $\lambda$, the model benefits from the stability of trajectory learning while acquiring the rapid mapping capabilities of consistency models.
> 3. **Conclusion:**
> These results confirm that our framework supports a "Universal" model architecture. While we reserve a comprehensive ablation of this "stabilizer effect" for future work, the current evidence unequivocally validates that the framework is unified at both the theoretical and weight levels.

---

> ### Author Response · Authors · 2025-11-28
>
> > [W2] It would be better to include more implementation details.
> >
>
> Thanks for your valuable comment. We would like to clarify that the stabilizing techniques mentioned (second-order estimator, clipping, Beta sampling) are not ad-hoc tricks, but **integral, documented components** of our Unified Framework, explicitly described in **Section 3.2** and **Algorithm 1**.
>
> 1. **Explicit Documentation & Theoretical Motivation:**
> We respectfully point out that these details are rigorously analyzed in the submission:
>     1. **Second-order Estimator:** Introduced in Section 3.2 (paragraph "Second-order estimator as $\lambda \to 1$"). We provide the theoretical justification for why this reduces numerical precision errors in **Appendix F.2.1**.
>     2. **Beta Time Sampling:** Introduced in Section 3.2 as "Generalized Time Distribution." We provide a theoretical analysis in **Appendix F.2.3** demonstrating that Beta sampling creates a unified approximation of widely used non-linear time schedules (e.g., LogNorm).
>     3. **Clipping:** We adopt the numerical truncation strategy from [1] (cited in Section 3.2) and validate its necessity for preventing outliers during the second-difference calculation.
> 2. **Implementation & Ablation:**
>     1. **Algorithm 1:** We provide a line-by-line pseudocode in the main paper that explicitly includes these steps.
>     2. **Ablation Studies:** The specific contribution of each technique to model stability is empirically validated in **Appendix D.8** (Table R2 in the supplementary material), showing that removing them leads to degradation in the $\lambda \to 1$ regime.
> 3. **Reproducibility:**
> We have provided the complete source code, including exact configuration files (hyperparameters, seeds, and distribution parameters), ensuring that the results in Table 2 and 3 are exactly reproducible.

---

> ### Author Response · Authors · 2025-11-28
>
> [1] Lu, Cheng, and Yang Song. ‘Simplifying, Stabilizing and Scaling Continuous-Time Consistency Models’. (ICLR 2025)

---

### Official Review · Reviewer_iL3K · 2025-11-01

**Soundness:** 2
**Presentation:** 2
**Contribution:** 2
**Rating:** 4
**Confidence:** 3

**Summary:**

This manuscript proposes a shared framework for many-step diffusion models and few-step consistency models. Specifically, starting from a consistency-style high-level objective, the authors demonstrate that this objective is equivalent to a flow matching plus a self-consistency regularization, which can be implemented to be diffusion models, consistency models, or interpolation between them. On top of the framework, the authors also propose a sampling procedure for this framework, as well as advanced training techniques and improvements, such as time distribution, CFG-enhanced score function, and high-performance autoencoders. Experimental results demonstrate that using the proposed pipeline improves the FID on both multi-step and few-step settings with high-resolution ImageNet benchmarks.

**Strengths:**

* Training a strong few-step generative model from scratch is an important topic.
* The writing is easy to follow.
* The experimental analysis and ablation study are well executed.

**Weaknesses:**

**The high-level objective**:
I have concerns about the necessity of the proposed high-level objective in Eqn. (4). When beyond the case of $\lambda = 0$ and $\lambda \to 1$, the behavior of the optimal solution of the objective, and how to leverage the learned quantity, remains unclear to me. While the authors discuss this point in a simple case in Appendix F.1.4, it remains unclear to me how to reasonably leverage the learned quantity $\lambda \in (0, 1)$ unless I have missed something. One possible scenario would be to have a closed-form relationship of the conditional expectation (the diffusion model), the pushforward operation (the consistency model), and the learned quantity, but the current presentation did not shed any light on this.

Empirically, according to Table 5, the main results are obtained from the $\lambda = 0$ and $\lambda \to 1$, which further makes the $\lambda \in (0, 1)$ part unclear. If so, then the implementation would boil down to diffusion models and a consistency model (or a finite difference version of sCM [1]).

**The sampling procedure**: The current sampling procedure needs more justification than provided, especially under the $\lambda \to 1$ case (again, unless I have missed something, in that case, this needs to be clarified explicitly *in the main text*). For example, consider the linear coupling case, the "decomposition" and "reconstruction" become one Euler discretization (or equivalently one DDIM step). So it is unclear whether using a pushforward $f_\theta^x(x_t, t)$ could simulate a path that is marginal preserving in this way. A relevant discussion is in IMM [2], where the authors show that one solution of marginal preserving simulation path with DDIM needs the network to condition on two timesteps. Here, the sampling process is achieved by only conditioning on one timestep. This needs more clarification/discussion.

**The comparison for samplers**: The proposed UCGM-S couples (narrow-sense) sampler, timestep selection, CFG scale, and stochasticity together. Could the author elaborate on the baselines used for comparing the sampler and provide insights on which part contributes the most to reducing the confounders?

I am open to revising my rating if the above concerns are addressed.

(Minor)
* Could the author provide some results for the sampler, as well as the training recipe (may use fine-tuning) on larger-scale text-to-image tasks, preferably examining the hard cases such as detailed text rendering?

## Reference
[1] Lu, Cheng, and Yang Song. ‘Simplifying, Stabilizing and Scaling Continuous-Time Consistency Models’. (ICLR 2025)

[2] Zhou, Linqi, et al. ‘Inductive Moment Matching’. (ICML 2025)

**Questions:**

See weaknesses.

---

> ### Author Response · Authors · 2025-11-28
>
> > [W1] The high-level objective.
> >
>
> Thank you for your valuable feedback. We would like to mitigate your concerns from the following points:
>
> 1. **Necessity of the High-Level Objective (Eqn. 4)**
> The unified objective is not merely a formalism; it serves as the theoretical cornerstone of our framework, enabling advancements in both insight and practice:
>     1. **Theoretical Insight (Decomposition):** As acknowledged by other reviewers (e.g., Reviewer dJfh), our framework decomposes the mysterious few-step consistency loss into a mathematically grounded **Flow Matching loss plus a Self-Alignment regularization term** (Theorem 1). This decomposition is critical: it offers an intuitive explanation for the training instability often observed in consistency models (where the alignment term dominates), paving the way for further analytical investigation.
>     2. **Methodological Unification:** Eqn. 4 provides the coherent theoretical basis that allows us to systematically integrate techniques from disparate domains. Features in our **Trainer**—such as the second-order estimator and generalized time distribution—are derived directly from this unified view. Without Eqn. 4, these combinations would be ad-hoc heuristics rather than principled derivations.
> 2. **Interpretation and Usage of $\lambda \in (0, 1)$**
> We appreciate the reviewer pushing for a deeper understanding of the intermediate regime. $\lambda$ serves as both a theoretical bridge and a practical control knob:
>     1. **Theoretical Closed-Form Relationship:** Addressing the reviewer's specific request, we have added a rigorous derivation in **Appendix F.1.6** of the revised paper. We theoretically show that the optimal solution for $\lambda \in (0, 1)$ minimizes a **weighted combination** of the instantaneous velocity matching error (from Flow Matching) and the trajectory self-consistency error. This mathematically bridges the multi-step diffusion model and few-step consistency model.
>     2. **Practical Trade-off (The "Sweet Spot"):** While endpoints are important, the intermediate $\lambda$ offers a continuous trade-off between **generation quality** (favored by $\lambda \to 0$) and **inference speed** (favored by $\lambda \to 1$). As shown in **Figure 2(a)** (Section 4.4), intermediate values (e.g., $\lambda=0.5$) can yield models that outperform pure consistency models in quality while being significantly faster than pure diffusion models, effectively filling the gap for medium-step regimes (e.g., 8-16 steps).
> 3. **Clarification on Table 5 (Why Focus on Endpoints?)**
> The focus on $\lambda=0$ and $\lambda=1$ in Table 5 was an intentional experimental design to ensure fair comparison with SOTA baselines.
>     1. Existing literature is fragmented: baselines are either specialized for multi-step (Diffusion/Flow Matching) or few-step (Consistency Models).
>     2. To demonstrate the effectiveness of UCGM, we evaluated our framework at these two extremes to show that **techniques transferred via our unified framework** (e.g., using multi-step techniques to improve few-step training) lead to SOTA results in both domains (as reported in Tables 2 & 3).
>     3. However, the capability to train with $\lambda \in (0,1)$ remains a unique advantage of UCGM for users who need to balance specific latency/quality constraints, as demonstrated in our ablation studies.

---

> ### Author Response · Authors · 2025-11-28
>
> > [W2] Justification of the sampling procedure.
> >
>
> We thank the reviewer for raising this insightful point regarding the sampling justification and the comparison with IMM [2]. We would like to verify the soundness of UCGM-S from both theoretical and empirical perspectives.
>
> 1. **Distinction from IMM [2] and Justification for Single-Step Conditioning**
> The requirement for conditioning on two timesteps in IMM[2] arises from its specific objective of learning a direct mapping between arbitrary start and end points on a PF-ODE trajectory. In contrast, UCGM-S adheres to the standard formulation of continuous generative models (including diffusion, flow matching, and consistency models), which predicts the vector field or terminal state based solely on the current state $\mathbf{x}_t$.
> Crucially, in the $\lambda \to 1$ regime, UCGM-S behaves identically to a Consistency Model sampler, which maps $\mathbf{x}_t$ directly to the origin $\mathbf{x}_0$. In this context, the trajectory is defined by the mapping to the endpoint rather than a step-by-step integration history, rendering conditioning on an additional previous timestep unnecessary.
> 2. **Theoretical Soundness via Stochasticity Ratio $\rho$**
> To ensure mathematical soundness across different regimes, we introduce the stochasticity ratio $\rho$ in UCGM-S, which we align with the training objective by setting $\rho \approx \lambda$.
>     1. Case $\lambda = 0$ (Diffusion/Flow Matching $\to$ $\rho = 0$):
>     When $\rho = 0$, the decomposition and reconstruction steps in UCGM-S reduce mathematically to a DDIM step (we have added a formal derivation of this equivalence in Appendix F.1.11). As acknowledged in IMM [2], the DDIM interpolant naturally satisfies self-consistency, thereby ensuring marginal preservation without requiring two-step conditioning.
>     2. Case $\lambda \to 1$ (Consistency Models $\to \rho = 1$):
>     When $\rho = 1$ (as used in our experiments, see Table 5), UCGM-S becomes equivalent to the standard multistep Consistency Model sampling procedure (i.e., denoise to $\mathbf{x}\*0$ and re-inject noise to $\mathbf{x}\*{t'}$). This process preserves marginals by explicitly sampling from the transition kernel $p(\mathbf{x}\_{t'}|\mathbf{x}\_0)$, which is valid and theoretically sound for consistency models.
>
> **3. Empirical Validation**
>
> To further validate the rationality of our design for the $\lambda \to 1$ case, we conducted an ablation study on the impact of $\rho$ using a 675M Diffusion Transformer (VA-VAE) on ImageNet-1K trained with $\lambda=1$. The results, reported in the table below, show that setting $\rho=1$ yields the best performance, confirming our theoretical alignment.
>
> | **ρ (with λ=1)** | **0** | **0.25** | **0.5** | **0.75** | **1 (Ours)** |
> | --- | --- | --- | --- | --- | --- |
> | **FID** | 6.59 | 5.32 | 4.07 | 2.78 | **1.42** |
>
> In summary, UCGM-S provides a unified and theoretically grounded sampling mechanism that naturally adapts to both deterministic PF-ODE solving ($\rho=0$) and stochastic consistency sampling ($\rho=1$), covering the dominant regimes of current state-of-the-art models.

---

> ### Author Response · Authors · 2025-11-28
>
> > [W3] The comparison for samplers.
> >
>
> We appreciate the reviewer's attention to the specific contributions of the sampler components. We acknowledge that UCGM-S integrates multiple factors, and we wish to clarify the distinct role of each, demonstrating that the performance gains primarily stem from our proposed Extrapolation Estimation technique.
>
> 1. **Clarification: Extrapolation vs. CFG**
>
>     First, we would like to clarify that UCGM-S does not couple the CFG scale. Instead, we introduce "Extrapolation," a technique inspired by CFG but fundamentally different:
>
>     - Computation-Free: Unlike CFG, which typically doubles the computational cost (requiring two forward passes per step), Extrapolation introduces zero additional computational overhead (see lines 277-287).
>     - Orthogonality: Extrapolation is independent of the guidance mechanism. As shown in Table 2, UCGM-S can operate in conjunction with traditional CFG (i.e., Extrapolation + CFG) or without it.
> 2. **Attribution of Performance Gains (The Role of Extrapolation)**
>
>     The performance improvements reported in Table 2 over the baselines are primarily driven by the Extrapolation technique ($\kappa$). We isolate this factor based on the following experimental setup for the multi-step models ($\lambda=0$):
>
>     - Stochasticity ($\rho$): We set $\rho=0$ (as $\rho$ is aligned with $\lambda$). This means no stochasticity was introduced compared to the deterministic baselines (e.g., standard ODE solvers).
>     - Timestep Selection: We utilized the same timestep schedules as the original baselines to ensure a fair comparison.
>     - Conclusion: Since stochasticity was disabled and timestep schedules were identical, the reported FID improvements are solely attributable to the Extrapolation mechanism.
> 3. **Evidence and Analysis**
>
>     We provide comprehensive evidence to support the effectiveness of Extrapolation:
>
>     - Ablation Study: Figure 3(c) isolates the impact of the extrapolation ratio $\kappa$, showing consistent performance gains within the range $\kappa \in [0, 0.5]$.
>     - Component Analysis: In Table 14 (Appendix), we provide a detailed component-wise ablation of UCGM-S, further confirming that Extrapolation contributes the most to reducing confounders and improving quality.
>     - Theoretical Backing: We have provided a theoretical analysis of why Extrapolation effectively reduces discretization error in Appendix F.1.10.
> 4. **Role of Other Parameters (Compatibility)**
>
>     The other factors mentioned (timestep selection and stochasticity) are included in UCGM-S strictly for compatibility across paradigms, rather than as "tricks" for performance boosting. For instance, few-step models (Consistency Models) and multi-step models (Diffusion) inherently require different noise schedules and time discretizations [1,2,3]. UCGM-S adapts to these requirements (e.g., setting $\rho=1$ for few-step, $\rho=0$ for multi-step) to function as a unified framework.

---

> ### Author Response · Authors · 2025-11-28
>
> > **[Q1]** Could the author provide some results for the sampler, as well as the training recipe (may use fine-tuning) on larger-scale text-to-image tasks, preferably examining the hard cases such as detailed text rendering?
> >
>
> We thank the reviewer for this insightful suggestion. We agree that verifying our method on larger-scale Text-to-Image (T2I) tasks—specifically examining "hard cases" like detailed text rendering—is essential for demonstrating real-world applicability.
> In response, we have added new experiments (see **Table 4 & 9**, and **Figure 3,4 & 5**) applying our method to models up to **20B parameters**.
>
> 1. Training Recipe: Scalability & Stability
> We validated our training recipe by fine-tuning pre-trained foundation models. Our unified framework demonstrated superior stability compared to existing fast-sampling methods at scale:
>     1. **Stability at 20B Parameters:** We successfully distilled the **20B-parameter Qwen-Image** model. Notably, while standard Consistency Models suffered catastrophic collapse and MeanFlow encountered Out-of-Memory (OOM) errors at this scale, our training recipe remained stable, producing a functional few-step 20B generator.
>     2. **Efficiency:** We also fine-tuned the transformer-based **SANA** (0.6B/1.6B), achieving a **0.84 GenEval** score with only **2 NFEs**, significantly outperforming industry baselines like FLUX-Schnell.
> 2. Sampler Results on "Hard Cases" (Detailed Text Rendering)
> Regarding the sampler’s specific performance on hard cases such as text rendering:
> **From a theoretical perspective**, we wish to be transparent: our sampler is designed as a **unified solver** that generalizes both deterministic and stochastic sampling pathways. It is not explicitly architected with modality-specific priors to enhance specific semantic capabilities like spelling or spatial reasoning.
> **However, empirically**, we discovered that the **extrapolation** mechanism in our sampler (controlled by $\kappa$) significantly aids in resolving these "hard cases."
>     1. **Quantitative Analysis:** Building upon the **Qwen-Image-20B** backbone, our **UCGM-S** achieves a GenEval score of **0.87** using only **20 sampling steps**. This performance strictly matches that of the original model (which relies on **50 steps**), demonstrating that UCGM-S maintains full fidelity with significantly improved efficiency while outperforming current open-source SOTA baselines.
>     2. **Qualitative (The Extrapolation Effect):** As shown in the newly added Figure 3,4 & 5, we observed instances where standard sampling yielded misspelled or blurred text, whereas enabling extrapolation rendered the text correctly. We hypothesize that extrapolation helps the model "step out" of uncertain, blurry regions in the probability density during the final sampling steps, effectively sharpening high-frequency details like text strokes.

---

> ### Author Response · Authors · 2025-11-28
>
> [1] Zhicong Tang, Jianmin Bao, Dong Chen, and Baining Guo. Diffusion models without classifier-free guidance. arXiv preprint arXiv:2502.12154, 2025.
>
> [2] Linqi Zhou, Stefano Ermon, and Jiaming Song. Inductive moment matching. arXiv preprint
> arXiv:2503.07565, 2025.
>
> [3] Yang Song, Prafulla Dhariwal, Mark Chen, and Ilya Sutskever. Consistency models. arXiv preprint arXiv:2303.01469, 2023.

---

### Author Response · Authors · 2025-12-03
**General Response**

We thank all reviewers for their detailed reviews and constructive feedback on our paper.

We are encouraged that the reviewers recognized the significance and contributions of our work in the following aspects:

- **Theoretical Unification:** Integrating Diffusion, Flow Matching, and Consistency Models into a single framework (Reviewers dJfh, YWsJ, qqse).
- **Surrogate Objective Insight:** Appreciating our derivation that views few-step objectives as multi-step objectives with regularization (Reviewers dJfh, YWsJ, qqse).
- **Strong Empirical Results:** Acknowledging the comprehensive experiments (Reviewers iL3K, dJfh) and State-of-the-Art FID scores on ImageNet in both few-step and multi-step settings (Reviewer YWsJ).

We would like to highlight that the **only reviewer who initially gave a negative score (iL3K)** explicitly stated in the review that *“I am open to revising my rating if the above concerns are addressed.” W*e have carefully addressed all of these concerns in the rebuttal and revised manuscript (detailed below). Unfortunately, the discussion phase closed before reviewers were able to participate.

Specifically, we made the following key improvements: (i) Clarified the Theoretical and Unified Framework; (ii) Clarified the Unified Trainer (UCGM-T) and Sampler (UCGM-S); (iii) Extended Experiments to include text-to-image (T2I) tasks, multiple backbones, and evaluation metrics; and (iv) Thoroughly revised and polished the manuscript for improved clarity and readability.

We believe that **we have adequately addressed all the concerns of the reviewers:**

- **Reviewer iL3K (only negative-score reviewer):** Concerns regarding (1) the necessity of our high-level objective (Eq. 4) is clarified in (i) and (iv); (2) the justification of our sampling procedure is explained in (ii); (3) broader generative experiments, including text-to-image tasks and multiple backbones, are added in (iii).
- **Reviewer dJfh:** Concerns regarding (1) our claim of a “unified” generative framework are addressed in (i); (2) additional implementation details are provided in (iv).
- **Reviewer YWsJ:** Concerns regarding (1) broader generative experiments, including text-to-image and additional backbones, are addressed in (iii); (2) clarifications of the FID metric and additional evaluation metrics are provided in (iii).
- **Reviewer qqse:** Concerns regarding (i) the motivation for Eq. 6 is clarified in (iv);(2) the convergence and stability analysis of UCGM-S is provided in (ii) and (iv); (3) definitions and notations are clarified in (iv); (4) clarifications regarding the FID metric and additional evaluation results are included in (iii); (5) further explanations of UCGM-S are provided in (ii).

We believe these clarifications and extensions substantially strengthen both the theoretical foundation and empirical validation of our work. We thank the reviewers again for their time and valuable insights.

Sincerely,
Authors of Submission 24943

---

### Meta-Review · Area_Chair_3h7c · 2025-12-29

**Summary:**

The paper proposes UCGM, a continuous generative modeling framework that aims to cover diffusion, flow matching, and consistency-style training within a single objective. The experimental section is generally strong, with careful ablations and solid empirical comparisons.

That said, the submission does not convincingly justify the technical necessity of introducing the interpolation parameter $\lambda \in (0,1)$. As noted by reviewers iL3K and dJfh, it remains unclear what concrete (empirical) benefit $\lambda$ provides beyond serving as a conceptual bridge between existing objectives, or whether it enables capabilities, guarantees, or performance regimes that are not already attainable by standard formulations at the endpoints. The current evidence (additional Table 9) is suggestive but not sufficiently diagnostic to establish when and why intermediate $\lambda$  values matter. I encourage the authors to deepen this analysis by providing clearer theoretical or empirical insight (e.g., analyze how the training schedule/distribution of  $\lambda$ impacts performance, and whether
 $\lambda$ can be learned or adaptively chosen (and what rule/value it converges to)?) into the role of $\lambda$

Given the present lack of a compelling motivation and demonstrated necessity for the interpolation, my recommendation is reject.

**Reviewer Concerns:**

- Reviewer iL3K questioned the necessity of the proposed objective (Eq. 6) and the sampling procedure, particularly whether it is truly marginal-preserving.

- Reviewer dJfh's main concern is that the claimed ``unified'' framework is not realized as a single deployable model in practice: the authors train separate checkpoints for different $\lambda$ values to target different sampling budgets, rather than one set of weights that works well across both high-step and ultra-low-step regimes.

- Reviewer YWsJ requested evaluation on text-to-image or multimodal tasks, and noted that the current assessment is limited to FID, lacking complementary metrics such as precision/recall, density/coverage, CLIP-based measures.

- Reviewer qqse raised that the loss in Eq. (6) appears abrupt and insufficiently motivated, Theorem 7 lacks a global convergence guarantee, several notations need clarification, and FID alone is not a sufficient evaluation metric.

**Reviewer Scores:**

- The motivation for methodological unification remains insufficiently addressed in my view; therefore, my score is likely to remain unchanged.

- The added experiments (Table 9), including conditioning the network on $\lambda$, are appreciated. However,  more careful and thorough justification of the necessity and concrete benefits of introducing the $\lambda$-interpolation is still encouraged.

- The authors further scale their approach to large-scale Text-to-Image (T2I) and Unified Multimodal Models (UMM), including experiments at the 20B-parameter scale.

- The authors clarified the motivation behind Eq. (6), extended the analysis of local truncation errors, and added broader generation metrics for evaluation.

---

### Decision · Program_Chairs · 2026-01-26

Reject